# CLEAR: CALIBRATED LEARNING FOR EPISTEMIC AND ALEATORIC RISK

**Ilia Azizi**[1,4*]   **Juraj Bodik**[1,2*]   **Jakob Heiss**[2*]   **Bin Yu**[2,3]

[1] Department of Operations, HEC, University of Lausanne, Switzerland
[2] Department of Statistics, University of California, Berkeley, USA
[3] Department of Electrical Engineering and Computer Science, University of California, Berkeley
[4] BegooAI, Switzerland
[*] Equal contribution
`first.last@unil.ch, jakob.heiss@berkeley.edu, binyu@berkeley.edu`

## ABSTRACT

Accurate uncertainty quantification is critical for reliable predictive modeling. Existing methods typically address either aleatoric uncertainty due to measurement noise or epistemic uncertainty resulting from limited data, but not both in a balanced manner. We propose CLEAR, a calibration method with two distinct parameters, $\gamma_1$ and $\gamma_2$, to combine the two uncertainty components and improve the conditional coverage of predictive intervals for regression tasks. CLEAR is compatible with any pair of aleatoric and epistemic estimators; we show how it can be used with (i) quantile regression for aleatoric uncertainty and (ii) ensembles drawn from the Predictability–Computability–Stability (PCS) framework for epistemic uncertainty. Across 17 diverse real-world datasets, CLEAR achieves an average improvement of 28.3% and 17.5% in the interval width compared to the two individually calibrated baselines while maintaining nominal coverage. Similar improvements are observed when applying CLEAR to Deep Ensembles (epistemic) and Simultaneous Quantile Regression (aleatoric). The benefits are especially evident in scenarios dominated by high aleatoric or epistemic uncertainty. Project page: `https://unco3892.github.io/clear/`

## 1 INTRODUCTION

Uncertainty quantification (UQ) is essential for building reliable machine learning systems (Abdar et al., 2021; Gawlikowski et al., 2023). Despite their impressive capabilities, modern machine learning methods can give a false sense of reliability; therefore, producing sharp valid prediction intervals remains an open problem. Calibration (Kuleshov et al., 2018) and conformal methods (Vovk et al., 2005; Vovk, 2012; Angelopoulos et al., 2024) adjust prediction intervals to obtain marginal coverage (that is, covering a certain percentage of the data on average). However, they may suffer from poor conditional coverage, meaning well-calibrated coverage at the individual or subgroup level (Gibbs et al., 2024). In particular, under distribution shift or model misspecification, conditional coverage can degrade substantially, especially in extrapolation regions. Most conformal methods, such as conformalized quantile regression (CQR) (Romano et al., 2019), only capture aleatoric uncertainty while ignoring epistemic uncertainty.

It is important to distinguish between the two main sources of uncertainty, namely epistemic and aleatoric. Epistemic uncertainty (Hüllermeier & Waegeman, 2021) arises from our limited understanding of the data generation process and the model, encompassing issues related to data collection, preprocessing, transformation, and model specification. Notably, this uncertainty is typically large in extrapolation regions where training data is sparse. In contrast, aleatoric uncertainty (Kirchhof et al., 2025) reflects the inherent variability within the data (stemming from measurement errors, missing covariates, randomness, or intrinsic noise) that cannot be reduced simply by gathering more observations or refining the model unless the data acquisition process itself is improved where more features are measured. Separating the two sources can be beneficial for various applications (Tagasovska &

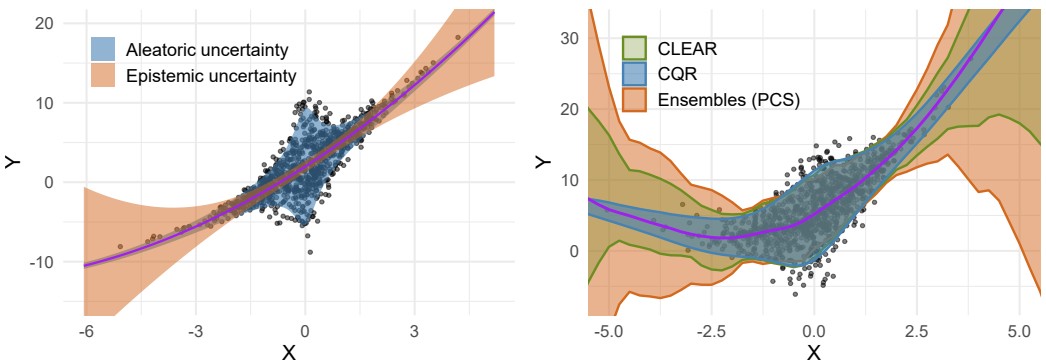

Figure 1: **Left:** Blue represents aleatoric uncertainty, which reflects randomness inherent in the data such as measurement noise. Red represents epistemic uncertainty, which arises from limited sample size. **Right:** Estimated prediction sets using the CLEAR method, which combines both sources of uncertainty in a data-driven manner.

Lopez-Paz, 2019; Laves et al., 2021). For instance, in active learning, epistemic uncertainty helps in selecting which samples to label, while aleatoric uncertainty is less relevant (Settles, 2012). However, for prediction tasks, appropriately combining both epistemic and aleatoric parts is necessary to account for the overall uncertainty in the model's predictions (Marques & Berenson, 2024).

In this work, we contribute to the existing literature by combining aleatoric and epistemic uncertainties in a data-driven manner. We consider prediction intervals $C(x)$ of the form

$$C(x) = \left[ \hat{f}(x) \pm \left( \gamma_1 \times \text{aleatoric}_\pm(x) + \gamma_2 \times \text{epistemic}_\pm(x) \right) \right], \quad \text{with } \gamma_2 = \lambda \gamma_1, \quad (1)$$

where $\hat{f}$ is a point estimate, and $\gamma_1, \gamma_2 \in [0, \infty)$ are coefficients selected to 1) calibrate marginal coverage and 2) optimally balance the two types of uncertainty (optimality is defined in terms of a quantile loss metric introduced later). The parameter $\lambda$ controls the trade-off between the two uncertainty types: when $\lambda = 0$, the interval reflects only aleatoric uncertainty, while as $\gamma_1 \to 0, \lambda = \frac{\gamma_2}{\gamma_1} \to \infty$, it reflects only epistemic uncertainty. By allowing an adaptively chosen $\lambda$, we ensure a more flexible and data-driven trade-off to the two components, leading to prediction intervals that are both well-calibrated and more informative (Figure 1). Moreover, estimating $\lambda$ can help practitioners better understand which source of uncertainty is the dominant contributor to the overall uncertainty.

The combination of epistemic and aleatoric uncertainty is not new: Bayesian methods have long incorporated both components (Kendall & Gal, 2017; Depeweg et al., 2018), and several recent approaches have also explored this decomposition in the context of conformal prediction (Rossellini et al., 2024; Hofman et al., 2024; Cabezas et al., 2025). However, to the best of our knowledge, existing methods either do not explicitly distinguish between the two types of uncertainty or implicitly fix the combination ratio, for instance, setting $\lambda = 1$ (Lakshminarayanan et al., 2017; Kendall & Gal, 2017; Depeweg et al., 2018), or fixing $\gamma_1 = 1$ (Rossellini et al., 2024). This fixed choice may be suboptimal, as the relative importance of each uncertainty type varies with the data distribution and prediction task (see Appendix A for more details on related work).

## 1.1 CONTRIBUTIONS

1. We are the first to introduce two calibration parameters $\gamma_1$ and $\gamma_2$, to balance the scales of aleatoric and epistemic uncertainty on the validation dataset.

2. We demonstrate that fitting quantiles on the residuals provides much more sensible estimators of aleatoric uncertainty than fitting the quantiles directly on the targets.

3. We are the first to combine the ensemble perturbation intervals of the PCS framework (Yu & Kumbier, 2020) with the CQR aleatoric uncertainty estimator, and empirically show the strengths of this combination.

4. We conduct large-scale UQ benchmarking for several models on 17 regression datasets.

## 2 METHOD

### 2.1 PROBLEM SCENARIO

Consider a classical setting, where an i.i.d. sample $(X_i, Y_i), i = 1, \ldots, n$ is drawn from distribution $P_X \times P_{Y|X}$. The goal of conformal inference is to construct a prediction set $C(X_{n+1}) \subseteq \text{supp}(Y)$ for a new data-point $(X_{n+1}, Y_{n+1})$ satisfying marginal coverage

$$\mathbb{P}\big(Y_{n+1} \in C(X_{n+1})\big) \geq 1 - \alpha, \tag{2}$$

where $\alpha \in (0, 1)$ is for instance $\alpha = 0.05$. In order to construct $C$, data $\mathcal{D} = \{(X_i, Y_i), i = 1, \ldots, n\}$ can be split into train and calibration subsets $\mathcal{D}_{\text{train}}, \mathcal{D}_{\text{cal}}$. On the training data, a first estimate of $C$ can be constructed, and then we can use data from $\mathcal{D}_{\text{cal}}$ to calibrate $C$ such that (2) is satisfied.

In case of CQR, we first estimate conditional quantiles $\hat{q}_{\alpha/2}(x), \hat{q}_{1-\alpha/2}(x)$ using $\mathcal{D}_{\text{train}}$, and then construct $C(x) = [\hat{q}_{\alpha/2}(x) - \gamma, \hat{q}_{1-\alpha/2}(x) + \gamma]$, where the calibration parameter $\gamma$ is chosen as the smallest value such that the prediction interval $C(X_i)$ contains $Y_i$ for at least $\lceil (1-\alpha)(|\mathcal{D}_{\text{cal}}| + 1) \rceil$ points in the calibration set $\mathcal{D}_{\text{cal}}$.

While this procedure guarantees finite-sample distribution-free marginal coverage (Angelopoulos et al., 2024), conditional coverage

$$\mathbb{P}\big(Y_{n+1} \in C(X_{n+1}) \mid X_{n+1} = x\big) \geq 1 - \alpha$$

does not need to hold. As pointed out in Lei & Wasserman (2014); Barber et al. (2020), any algorithm with finite-sample distribution-free conditional coverage guarantees for all $x$ must be trivial $C(x) = (-\infty, \infty)$. However, we aim to design estimators such that conditional coverage holds approximately under reasonable real-world scenarios, even if exact finite-sample guarantees are impossible in general.

### 2.2 EPISTEMIC UNCERTAINTY

The traditional machine learning approach trains a predictive algorithm on a single version of the cleaned/preprocessed dataset and uses the best-performing algorithm (compared using the validation set) for future predictions. While theoretically sound in the infinite-sample limit, this approach ignores the uncertainty stemming from finite sample size and model choice (epistemic uncertainty). Various methods have been proposed to estimate this uncertainty, including Deep Ensembles (Lakshminarayanan et al., 2017), MC dropout in NN (Gal & Ghahramani, 2016), Orthonormal Certificates (Tagasovska & Lopez-Paz, 2019), NOMU (Heiss et al., 2022b), BNNs (MacKay, 1992) and Laplace Approximation (Ritter et al., 2018), among others.

Estimating epistemic uncertainty via PCS: In practice, additional sources of uncertainty arise from subjective choices in data cleaning, imputation, and dataset construction, which we also consider as extended epistemic uncertainty. The Predictability, Computability, and Stability (PCS) framework (Yu & Kumbier, 2020) offers a holistic point of view on the data-science-life-cycle, without explicitly modeling aleatoric uncertainty. One can obtain an ensemble of $m$ estimators $\hat{f}_1, \ldots, \hat{f}_m$ as follows:

1. Split the data into $\mathcal{D}_{\text{train}}, \mathcal{D}_{\text{val}}$.
2. Create $N_1$ differently preprocessed versions of the data and define $N_2$ different models (e.g., linear regression, random forest, neural networks). Then, train models for all $N_1 \times N_2$ combinations on $\mathcal{D}_{\text{train}}$ and pick the top-$k$ based on their performance on $\mathcal{D}_{\text{val}}$.
3. Refit each of the top-$k$ models on $b$ bootstrap samples $\mathcal{D}_{\text{train}}^1, \ldots, \mathcal{D}_{\text{train}}^b$ of $\mathcal{D}_{\text{train}}$ to obtain an ensemble of $m = k \times b$ estimators $\hat{f}_1, \ldots, \hat{f}_m$.

Taking the point-wise median of $\hat{f}_1, \ldots, \hat{f}_m$ yields the final PCS estimate $\hat{f}$, and the point-wise $\alpha/2$ and $1 - \alpha/2$ quantiles (denoted $\hat{f}_{\alpha/2}$ and $\hat{f}_{1-\alpha/2}$, respectively) define the uncalibrated uncertainty band. The widths of this interval, denoted as $\hat{q}_{1-\alpha/2}^{\text{epi}}(x) := \hat{f}_{1-\alpha/2}(x) - \hat{f}(x)$ and $\hat{q}_{\alpha/2}^{\text{epi}}(x) := \hat{f}(x) - \hat{f}_{\alpha/2}(x)$, quantify the uncalibrated epistemic uncertainty. Agarwal et al. (2025) extends this by using the combined data set $\mathcal{D}_{\text{train}} \cup \mathcal{D}_{\text{val}}$ for training and calibrating uncertainty based on

out-of-bag data. Our main experiments in Sections 4.1 and 4.2 focus only on the modeling step of the PCS framework. However, in Section 4.3, we show through a case study that CLEAR can also be applied successfully in the $N_1 > 1$ setting, incorporating uncertainty from data cleaning and pre-processing.

## 2.3 ALEATORIC UNCERTAINTY

Aleatoric uncertainty can be estimated by modeling the conditional distribution of the outcome given the inputs. Common approaches include direct conditional quantile regression (Koenker & Bassett, 1978), either parametric or nonparametric, such as smooth quantile regression (Fasiolo et al., 2020) and quantile random forests (QRF) (Meinshausen, 2006); heteroskedastic models that estimate input-dependent noise levels $\sigma(x)$ under Gaussian assumptions (Nix & Weigend, 1994); and distributional regression techniques such as simultaneous quantile regression (Tagasovska & Lopez-Paz, 2019). More flexible alternatives include conditional density estimation and deep generative models such as conditional generative adversarial networks (Oberdiek et al., 2022), conditional variational autoencoders (Han et al., 2020), and diffusion models (Chang et al., 2023).

In this work, we estimate aleatoric uncertainty using quantile regression models $\hat{r}_{\alpha/2}, \hat{r}_{0.5}, \hat{r}_{1-\alpha/2}$ (selected in Line 2 from PCS) trained on the residuals $Y_i - \hat{f}(X_i)$. This approach offers improved stability: underfitting quantile regression directly on the $y$-values can severely distort aleatoric uncertainty estimates. In contrast, extreme underfitting on residuals, at worst, corresponds to assuming homoskedastic noise, which can be an acceptable bias. This improves the stability with respect to hyperparameters. To further improve the stability, we apply a PCS-inspired bagging strategy by taking the empirical median

$$\hat{q}^{\text{ale}}_{1-\alpha/2}(x) := \text{Median}\left[\left(\hat{r}_{1-\alpha/2}(x) - \hat{r}_{0.5}(x)\right)^+\right] \text{ and } \hat{q}^{\text{ale}}_{\alpha/2}(x) := \text{Median}\left[\left(\hat{r}_{0.5}(x) - \hat{r}_{\alpha/2}(x)\right)^+\right]$$

over the ensemble members. Overall, it is computationally efficient and straightforward to implement.

## 2.4 CLEAR: COMBINING ALEATORIC & EPISTEMIC UNCERTAINTY

To combine both aleatoric and epistemic uncertainties, we use a weighted scheme as in Equation (1). Specifically, using a PCS-type estimator $\hat{q}^{\text{epi}}_\alpha$ and a quantile regression estimator $\hat{q}^{\text{ale}}_\alpha$ trained on the residuals $Y_i - \hat{f}(X_i)$, we define the prediction interval:

$$C = \left[\hat{f} - \gamma_1 \hat{q}^{\text{ale}}_{\alpha/2} - \gamma_2 \hat{q}^{\text{epi}}_{\alpha/2}, \quad \hat{f} + \gamma_1 \hat{q}^{\text{ale}}_{1-\alpha/2} + \gamma_2 \hat{q}^{\text{epi}}_{1-\alpha/2}\right]. \tag{3}$$

Given a fixed ratio $\lambda = \frac{\gamma_2}{\gamma_1}$, we compute $\gamma_1$ on a held-out calibration set using the standard split conformal prediction procedure. While the natural choice $\gamma_1 = \gamma_2$ may seem appealing, it is often suboptimal. The relative contribution of aleatoric and epistemic uncertainty can vary across datasets, and the corresponding estimators may differ substantially in scale and precision when $\gamma_1 = \gamma_2$. Additionally, we adopt this global linear form for simplicity (minimizing overfitting via only two parameters), interpretability ($\lambda$ captures the epistemic/aleatoric ratio), and stability, consistent with standard conformal scaling (Angelopoulos et al., 2024).

To choose $\lambda$ from data, we evaluate a grid of positive values $\Lambda$. For each candidate $\lambda \in \Lambda$, we construct the (calibrated) interval $C_\lambda$. To ensure the best trade-off between uncertainty sources, we select $\lambda^\star$ such that $C_{\lambda^\star}$ performs best under the chosen metric on $\mathcal{D}_{\text{val}}$. We have chosen quantile loss (Koenker & Bassett, 1978) (defined in Algorithm 1, which is equivalent to the pinball loss or interval score loss, see Appendix B.3.4) as a simple metric to balance both coverage and width. However, any other metric can also be used. As a proper scoring rule, quantile loss incentivizes truthfulness from a theoretical perspective (see Appendix B.3). This procedure is summarized in Algorithm 1.

Parameter $\lambda^\star$ balances aleatoric and epistemic uncertainties: if one estimator fails, $\lambda^\star$ compensates by re-weighting the other. When both estimators $\hat{q}^{\text{epi}}$ and $\hat{q}^{\text{ale}}$ are reliable (up to scaling), $\lambda^\star$ is interpretable. A large ratio

$$\lambda^\star \frac{\hat{q}^{\text{epi}}_{1-\alpha/2}(x) + \hat{q}^{\text{epi}}_{\alpha/2}(x)}{\hat{q}^{\text{ale}}_{1-\alpha/2}(x) + \hat{q}^{\text{ale}}_{\alpha/2}(x)} \gg 1$$

indicates that epistemic uncertainty dominates at $x$ (reducible with more training observations or stronger assumptions), while a small ratio $\ll 1$ indicates aleatoric uncertainty dominates (not reducible by adding more training observations, though sometimes reducible by adding covariates).

---

**Algorithm 1** CLEAR: Calibrated Learning for Epistemic and Aleatoric Risk

---

1: **Input:** Data $(X_i, Y_i)$ for $i = 1, \ldots, n$, split into training $\mathcal{D}_{\text{train}}$, calibration $\mathcal{D}_{\text{cal}}$, and validation $\mathcal{D}_{\text{val}}$ (we consider $\mathcal{D}_{\text{cal}} = \mathcal{D}_{\text{val}}$); grid of $\lambda$ values $\Lambda$; significance level $\alpha$.

2: **Step 1: Estimate epistemic uncertainty on $\mathcal{D}_{\text{train}}$.**

    Example: Estimate stable point predictor $\hat{f}$ and epistemic quantiles $\hat{q}^{\text{epi}}_{\alpha/2}, \hat{q}^{\text{epi}}_{1-\alpha/2}$ using PCS ensembles across data perturbations.

3: **Step 2: Estimate aleatoric uncertainty on $\mathcal{D}_{\text{train}}$.**

    Example: train a quantile regression model on the residuals $Y_i - \hat{f}(X_i)$ to estimate conditional quantiles $\hat{q}^{\text{ale}}_{\alpha/2}, \hat{q}^{\text{ale}}_{1-\alpha/2}$.

4: **Step 3: Define prediction intervals for each $\lambda \in \Lambda$.**

    Define $C_\lambda$ by selecting the smallest value $\gamma_1$ such that the prediction set

$$C_\lambda = \left[\hat{f} - \gamma_1 \hat{q}^{\text{ale}}_{\alpha/2} - \lambda\gamma_1 \hat{q}^{\text{epi}}_{\alpha/2}, \ \hat{f} + \gamma_1 \hat{q}^{\text{ale}}_{1-\alpha/2} + \lambda\gamma_1 \hat{q}^{\text{epi}}_{1-\alpha/2}\right]$$

    contains at least $\lceil (1-\alpha)(|\mathcal{D}_{\text{cal}}| + 1) \rceil$ of points in $\mathcal{D}_{\text{cal}}$. See Appendix B.5 for implementation.

5: **Step 4: Select $\lambda^\star$ by minimizing the quantile loss on $\mathcal{D}_{\text{val}}$.**

    Evaluate the quantile loss of $C_\lambda$ on $\mathcal{D}_{\text{val}}$ and set

$$\lambda^\star = \arg\min_{\lambda \in \Lambda} \text{QuantileLoss}(\mathcal{D}_{\text{val}}, C_\lambda),$$

    where $\text{QuantileLoss}(\mathcal{D}_{\text{val}}, C_\lambda) := \frac{1}{2|\mathcal{D}_{\text{val}}|} \sum_{i \in \mathcal{D}_{\text{val}}} \left[QL_{\alpha/2}\big(Y_i, l(X_i)\big) + QL_{1-\alpha/2}\big(Y_i, u(X_i)\big)\right]$,
with $l(x), u(x)$ denoting the bounds of $C_\lambda(x)$, and $QL_\tau(y, q) = (y - q)\big(\tau - \mathbb{1}_{(-\infty, q]}(y)\big)$.

6: **Output:** $\lambda^\star$ and calibrated prediction interval $C_{\lambda^\star}(x)$.

---

**Lemma 2.1.** *Let $\Lambda$ be compact. Suppose that at least $k$ of the base models used in the PCS ensemble are consistent for the true function $f(x)$, and the quantile regression estimators $\hat{q}^{ale}_\tau$ are consistent for both $\tau \in \{\alpha/2, 1 - \alpha/2\}$. Then we obtain **asymptotic conditional validity**: for any fixed $x \in \mathcal{X}$, it holds that*

$$\liminf_{|\mathcal{D}_{train}|, |\mathcal{D}_{val}|, |\mathcal{D}_{cal}| \to \infty} \mathbb{P}\big(Y_{n+1} \in C(X_{n+1}) \mid X_{n+1} = x\big) \geq 1 - \alpha.$$

The proof is given in Appendix B.2, building on (Angelopoulos et al., 2024, Section 5).

Note that these assumptions are satisfied by the finite grid $\Lambda$ used in our implementation and by many base models, including tree-based methods and neural networks. While the epistemic component vanishes asymptotically, it plays a crucial role in finite samples by preventing under-coverage in data-sparse regions (see Section 4.1 and Appendix B.4). With an infinitely large calibration set, joint calibration is no worse than single-parameter baselines (see Lemma B.9). Appendix B provides further theoretical discussion, including properties of the quantile loss, intuitive motivation for CLEAR, and conformal marginal-coverage guarantees for a modified variant of CLEAR (Lemma B.2).

## 3 EXPERIMENTAL SETUP

### 3.1 DATA

We conduct experiments using both simulations and real-world data. The synthetic experiments demonstrate the main intuition behind our approach. We sample $X \sim \mathcal{N}(0_d, I_d)$ and compute the response $Y = \mu(X) + \sigma(X) \cdot \varepsilon$, where $\varepsilon \sim \mathcal{N}(0, 1)$. The mean function $\mu(X)$ introduces non-linearity through transformations of input features (involving absolute values and fractional powers with random coefficients); its explicit form, alongside $\sigma(X)$, is detailed in Appendix C. The sample size is fixed at $n = 5000$ and divided into 70-30% training and validation splits. In the univariate case, we generate 100 datasets for $d = 1$, and in multivariate, 100 datasets randomly sampled for $d \in \{2, 3, 20\}$. We then assess the conditional performance as a function of distance from $\mathbb{E}[X]$, where test points are randomly generated on the surfaces of spheres with varying radii.

For the real-world scenarios, we use 17 regression datasets curated by Agarwal et al. (2025), forming one of the largest benchmarks for UQ (see Appendix D.1 for details). Categorical features are one-hot

encoded, with no further preprocessing. To ensure robustness, each dataset is evaluated over 10 random train-validation-test splits (60%-20%-20%). We conduct experiments in two configurations: **standard** (using $\mathcal{D}_{\text{val}}$ also as $\mathcal{D}_{\text{cal}}$) and **conformalized** (splitting the 20% validation into 10% $\mathcal{D}_{\text{val}}$ + 10% $\mathcal{D}_{\text{cal}}$ for stronger theoretical guarantees). Note that while the body of the paper only focuses on the standard experiments, the conformalized results are provided in Appendix G.

## 3.2 BASELINES

We compare CLEAR against its core components, CQR and PCS ensemble (for details, including PCS implementation and further baselines that are relevant only to the appendices, see Appendix D). Notably, the underlying uncertainty estimation models from CQR and PCS are reused within the CLEAR framework. Our CQR follows the classical implementation (Romano et al., 2019) with the difference that in our experiments, we primarily utilize an enhanced variant, termed **ALEATORIC**, which uses bootstrapping ($b = 100$) on $Y_i$ and the model selection from PCS. We further improve ALEATORIC in a new baseline called **ALEATORIC-R**, which models the residuals $Y_i - \hat{f}(X_i)$ instead, using $\hat{f}$ from the corresponding PCS ensemble. The CLEAR method combines the uncalibrated aleatoric uncertainty from ALEATORIC-R and the uncalibrated epistemic uncertainty from PCS.

We perform ablation studies by exploring three different variants of base models for PCS (epistemic) and CQR (aleatoric). Our main approach, which we refer to as **variant (a)**, uses a **quantile PCS** for estimating epistemic uncertainty. We employ a diverse set of models designed to estimate conditional quantiles, namely quantile random forests (QRF) (Meinshausen, 2006), quantile XGBoost (QXGB) (Chen & Guestrin, 2016), and Expectile GAM (Servén & Brummitt, 2018). Then, the top-performing model ($k = 1$), in terms of the RMSE of $\hat{f}$ on $\mathcal{D}_{\text{val}}$, is selected and bootstrapped ($b = 100$) to generate the epistemic uncertainty estimate $\hat{q}^{\text{epi}}$ and the median $\hat{f}$. CLEAR then combines this bootstrapped $\hat{q}^{\text{epi}}$ with the bootstrapped aleatoric estimate from ALEATORIC-R, ensuring ALEATORIC-R also uses the same selected quantile model type. The other two variants are explained in Appendix D.4, both using the same $b$ and $k$ as above and only modifying the models. **Variant (b)** restricts quantile PCS as well as CQR baselines to **only QXGB** to remove any impact on CLEAR's evaluation due to the choice of the base models. In contrast, **variant (c)** uses the standard PCS models to estimate the **conditional mean**. In all cases, PCS intervals are calibrated using the multiplicative method.

To further validate the generalizability of CLEAR, we also evaluate its application to other uncertainty estimators. This is a separate setup, where for epistemic uncertainty, we employ **Deep Ensembles (DE)** (Lakshminarayanan et al., 2017)—an ensemble of neural networks trained with diverse initializations—and for aleatoric uncertainty, we use **Simultaneous Quantile Regression (SQR)** (Tagasovska & Lopez-Paz, 2019), which directly models multiple conditional quantiles (more detail in Appendix D.3). These represent state-of-the-art deep learning approaches for uncertainty quantification, complementing our primary PCS and CQR baselines.

In all cases, CLEAR parameters $(\lambda, \gamma_1)$ are optimized via quantile loss on the validation set. $\lambda$ is chosen from a dense grid $\Lambda$ combining linearly spaced values from 0 to 0.09 and logarithmically spaced values from 0.1 to 100 (totaling over 4000 points), and $\gamma_1$ is determined via conformal calibration for the chosen $\lambda$. All intervals are evaluated at 95% nominal coverage. In our benchmarks, we skip the data pre-processing steps of PCS (i.e., set $N_1 = 1$), as in Agarwal et al. (2025), addressing only the model perturbation step of PCS. However, we show an example in Section 4.3 of how CLEAR can be applied in the $N_1 > 1$ setting from Yu & Barter (2024, Chapter 13).

## 3.3 METRICS

To evaluate the quality of our prediction intervals, we employ interval coverage (PICP), normalized interval width (NIW), average interval score loss (AISL) and the quantile loss that are common in the interval prediction literature (Pearce et al., 2018; V'yugin & Trunov, 2019; Azizi et al., 2025). We also evaluate our work on Normalized Calibrated Interval Width (NCIW), defined as:

$$\text{NCIW}(\hat{f}, l, u) = \text{NIW}\left(\hat{f} - c_{\text{test-cal}}l, \hat{f} + c_{\text{test-cal}}u\right),$$

with a calibration constant

$$c_{\text{test-cal}} := \arg\min_{c \geq 0} \left\{ \text{PICP}\left(\hat{f} - cl, \hat{f} + cu\right) \geq 1 - \alpha \right\}.$$

In the standard configuration, all methods are calibrated on the validation set; in the conformalized configuration, calibration uses a separate 10% calibration split. In both cases, the methods are already calibrated before evaluation on the test set, resulting in $c_{\text{test-cal}} \approx 1$. We primarily discuss NCIW and quantile loss, but provide the results for all the available metrics.

## 4 RESULTS

### 4.1 SIMULATIONS

Figure 2 shows conditional coverage and interval width for the univariate homoskedastic case. ALEATORIC-R achieves good coverage in high-density regions but undercovers in low-density or extrapolation regions. Conversely, calibrated PCS intervals slightly undercover in data-rich regions but widen too much in extrapolation regions. Only CLEAR maintains approximate conditional coverage throughout, adapting interval width as needed. Additional results for heteroskedastic and multivariate settings in Appendix C.2.2 show similar findings.

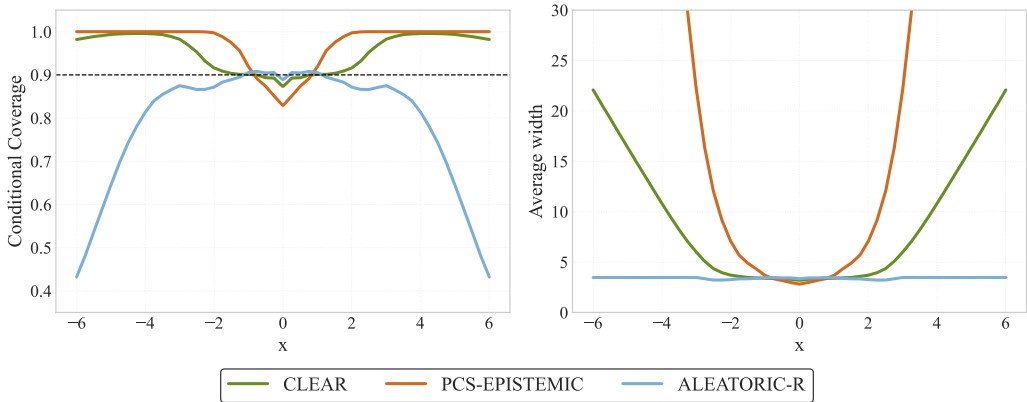

Figure 2: Results for univariate homoskedastic case averaged over 100 simulations: On the left, conditional coverage, and on the right, mean width, for $X = x$. It compares CLEAR, PCS, and ALEATORIC-R (bootstrapped CQR trained on residuals $Y_i - \hat{f}(X_i)$). The dashed horizontal line is the target coverage level of 0.9. CLEAR adapts to maintain target coverage across the input space.

### 4.2 REAL-WORLD DATA

Figure 3 shows the Normalized Calibrated Interval Width (NCIW) and Quantile Loss for 95% prediction intervals across all datasets for our main approach, CLEAR (variant a), compared to several baselines. CLEAR consistently demonstrates superior performance, achieving better or comparable interval width and loss metrics while consistently maintaining nominal coverage. The inset boxplots show that CLEAR (a) compared to PCS, ALEATORIC, and ALEATORIC-R has an improved quantile loss of 15.8%, 34.4%, and 9.4%, respectively. Similar relative increases are observed for NCIW, with PCS, ALEATORIC, and ALEATORIC-R exhibiting increases of 17.5%, 28.3%, and 3%. Moreover, CLEAR (a) was, in fact, the top-performing method on 15 of the 17 datasets, while remaining the most stable compared to the baselines. These trends hold across our other model variants as well (Appendix F for standard and Appendix G for conformalized results): both CLEAR (b) and CLEAR (c) exhibit very similar relative improvements (Figures 7 and 8 in Appendices F.2 and F.3), with variant (a) remaining the strongest and most robust configuration.

While we observed that setting $\lambda = 1$ or $\gamma_1 = 1$ could marginally improve results if one had prior knowledge of uncertainty components—an unlikely scenario in practice—fully optimizing both parameters offers greater robustness. A limited size of the validation dataset can lead to

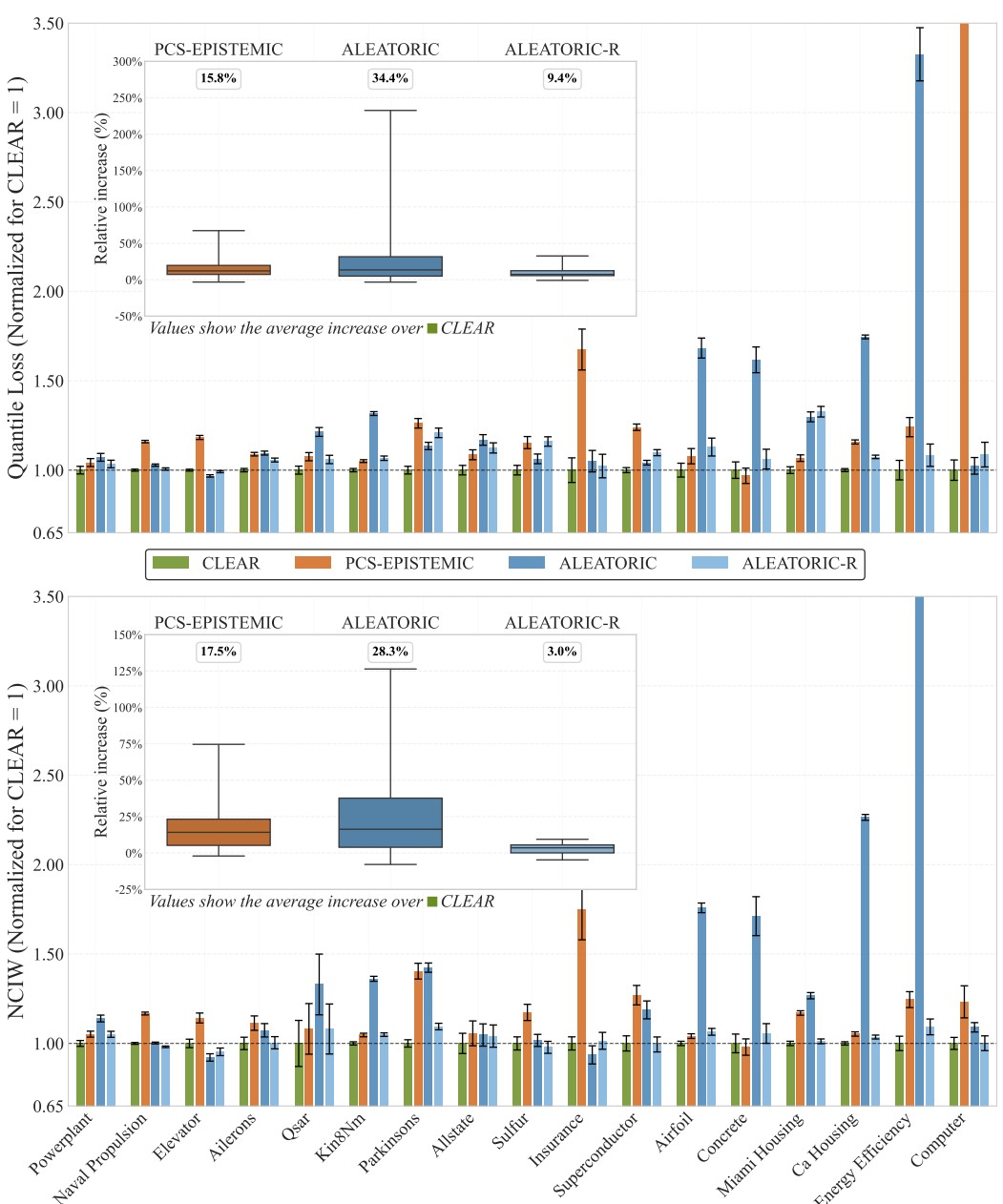

Figure 3: Results for real-world data: Quantile loss and NCIW performance of different methods over 10 seeds normalized relative to CLEAR (baseline = 1.0) with error bars are $\pm 1\ \sigma$. Lower values are better. The inset boxplot shows the average (%) relative increase of the metric over CLEAR. EPISTEMIC is PCS-UQ, ALEATORIC is bootstrapped CQR, and ALEATORIC-R uses residuals.

overfitting of the two parameters, and incorporating some prior on them could further improve the results. Importantly, the absolute value of $\lambda$ is relative to the pre-calibrated scales of the uncertainty estimators and dataset noise; its interpretation is best contextualized by observing its behavior when systematically varying dataset characteristics, such as the number of features or observations. While fixed parameters ($\lambda = 1$ or $\gamma_1 = 1$) could marginally improve results with prior knowledge, fully optimizing both parameters offers greater robustness. CLEAR's dual-parameter calibration enhances stability by adaptively re-weighting potentially unreliable uncertainty components, as evidenced in datasets like `energy_efficiency`, where baselines show markedly larger NCIW. The dataset-

dependent variability in optimal $\lambda$ underscores the need for adaptive selection over fixed heuristics. The calibration runtime (grid-search) is also extremely negligible in practice (Appendix F.5).

**Empirical comparison with UACQR (Rossellini et al., 2024).** The comparison is summarized in Table 1, showing the percentage improvement of (standard) CLEAR over both UACQR variants across all 17 datasets and three metrics (detailed metric-specific tables for the conformalized version are provided in Tables 43 to 47). The performance of CLEAR is much more reliable across the considered datasets and metrics. For example, on the `airfoil`, `energy_efficiency`, and `naval_propulsion` datasets, CLEAR is significantly better (40–70%) in metrics such as NCIW, AISL, and average width, while still exceeding 94.5% coverage. In contrast, UACQR outperforms CLEAR across these metrics on only one dataset (`insurance`). In some instances, UACQR-P can output infinitely wide predictive intervals (Rossellini et al., 2024, p. 5), which we observed for `energy_efficiency` in our experiments. CLEAR conclusively outperforms both versions of UACQR on 14 out of 17 datasets. These large differences in performance can partially be explained by our approach for fitting aleatoric uncertainty to the residuals. We hypothesize that CLEAR is more stable and robust because it can more easily compensate for the shortcomings of the base models. If aleatoric uncertainty is over- or underestimated, CLEAR can correct its scale by adjusting $\gamma_1$.

Table 1: Improvement (%) of standard CLEAR variant (c) over UACQR-S and UACQR-P at 95% coverage across 17 datasets. **Bold values with +** indicate CLEAR outperforms the baseline. $+\infty_v^{1/10}$ denotes that UACQR-P produced infinitely wide intervals for 1 out of the 10 seeds, and the mean improvement on the remaining 9 seeds was $v$.

| Dataset | UACQR-S | | | UACQR-P | | |
|---|---|---|---|---|---|---|
| | NCIW | AISL | Width | NCIW | AISL | Width |
| ailerons | -1.5% | **+3.2%** | -0.4% | -9.3% | **+2.7%** | -18.3% |
| airfoil | **+40.3%** | **+44.5%** | **+49.4%** | **+40.7%** | **+43.3%** | **+45.7%** |
| allstate | **+6.8%** | **+1.2%** | **+8.8%** | **+0.4%** | -3.2% | **+1.9%** |
| ca_housing | **+23.4%** | **+8.8%** | **+22.4%** | **+15.0%** | **+5.7%** | **+13.9%** |
| computer | **+20.4%** | **+9.5%** | **+21.1%** | **+6.7%** | **+6.6%** | **+3.4%** |
| concrete | **+21.3%** | **+21.0%** | **+31.4%** | **+22.0%** | **+18.1%** | **+26.4%** |
| elevator | **+36.5%** | **+29.1%** | **+36.1%** | **+24.5%** | **+27.6%** | **+19.2%** |
| energy_efficiency | **+69.6%** | **+62.4%** | **+72.4%** | $+\infty_{+70.9\%}^{1/10}$ | $+\infty_{+63.1\%}^{1/10}$ | $+\infty_{+72.8\%}^{1/10}$ |
| insurance | -15.5% | -19.5% | -29.1% | -18.6% | -23.5% | -32.5% |
| kin8nm | **+27.1%** | **+26.7%** | **+28.2%** | **+25.8%** | **+24.1%** | **+26.8%** |
| miami_housing | **+19.6%** | **+20.0%** | **+23.0%** | **+13.4%** | **+15.2%** | **+15.5%** |
| naval_propulsion | **+55.8%** | **+52.0%** | **+55.4%** | **+7.4%** | **+59.2%** | **+6.4%** |
| parkinsons | **+20.5%** | **+5.1%** | **+20.6%** | **+8.8%** | -4.6% | **+7.7%** |
| powerplant | **+19.9%** | **+13.5%** | **+22.7%** | **+15.6%** | **+10.1%** | **+18.8%** |
| qsar | **+21.5%** | **+10.3%** | **+23.1%** | **+15.0%** | **+5.5%** | **+16.1%** |
| sulfur | **+13.5%** | **+11.3%** | **+11.8%** | **+5.0%** | **+7.0%** | **+3.2%** |
| superconductor | **+17.7%** | **+6.4%** | **+16.6%** | **+12.6%** | **+3.8%** | **+11.1%** |

**Empirical comparison with DE and SQR.** Beyond PCS and CQR, CLEAR demonstrates substantial improvements when applied to DE and SQR (Table 2). When using DE for epistemic uncertainty and SQR for aleatoric uncertainty at 95% nominal coverage, CLEAR achieves average width reductions (NCIW) of 28.6% and 13.4%, respectively, with similar improvements in quantile loss (24.0% and 13.7%). The gains persist even after conformal calibration of the baselines, particularly relevant to the aleatoric component (SQR). The results underscore CLEAR's ability to balance the two uncertainty sources well, regardless of the underlying estimators. This consistency across both PCS and neural approaches validates the generality of CLEAR (full results in Appendix E).

## 4.3 CASE STUDY: ACCOUNTING FOR DATA UNCERTAINTY

We consider a case study on the Ames Housing dataset, detailed in Appendix H, where we demonstrate the full PCS pipeline and vary both the number of predictor variables and training sample size to explore how aleatoric and epistemic uncertainty respond to different data characteristics. We induce changes in aleatoric uncertainty by restricting features (80 vs. 2) and epistemic uncertainty by subsampling training data (100%, 50%, 20%). Tables 3 and 4 show the results. CLEAR correctly

Table 2: Mean (%) improvement of CLEAR over DE & SQR across all datasets at 95% coverage (higher is better). **Bold** values indicate CLEAR outperforms the baseline.

| Metric | DE | SQR | DE-conformal | SQR-conformal |
|---|---|---|---|---|
| PICP | **+0.05%** | -0.66% | -0.09% | -0.15% |
| NIW | **+28.81%** | **+17.38%** | **+29.55%** | **+14.07%** |
| NCIW | **+28.57%** | **+13.36%** | **+27.90%** | **+13.23%** |
| QuantileLoss | **+23.98%** | **+13.66%** | **+24.08%** | **+10.12%** |

identifies the dominant uncertainty source: in the low-information regime (2 features), it prioritizes aleatoric uncertainty ($\lambda = 0.64$), whereas in the high-feature but low-data regime (20% samples), it drastically shifts weight to epistemic uncertainty ($\lambda = 100$, calibrated E/A ratio $\approx 250$). This adaptive weighting allows CLEAR to outperform baselines or perform comparably to them.

Table 3: Ames Housing results with varying the number of predictors (90% coverage target).

| Experiment | Method | NCIW | Quantile Loss | Average Width ($) | Coverage |
|---|---|---|---|---|---|
| 2 features | PCS | 0.214 | 3,818 | 107,880 | 0.87 |
| | CQR | 0.186 | 3,448 | 104,741 | **0.90** |
| | CLEAR | **0.171** | **3,131** | **95,177** | 0.89 |
| All features | PCS | 0.105 | **1,922** | 57,594 | **0.89** |
| | CQR | 0.117 | 2,194 | 62,398 | 0.88 |
| | CLEAR | **0.103** | 1,923 | **55,910** | 0.88 |

Table 4: CLEAR calibration parameters across experimental scenarios (90% coverage target).

| Experiment | $\lambda$ | $\gamma_1$ | Epistemic/Aleatoric Ratio |
|---|---|---|---|
| 2 features | 0.64 | 0.99 | 0.03 |
| All features | 14.45 | 0.13 | 7.72 |
| All features, 50% data | 17.97 | 0.09 | 14.09 |
| All features, 20% data | 100 | 0.01 | 250.5 |

## 5 CONCLUSION, LIMITATIONS & FUTURE WORK

This paper introduces CLEAR, a novel framework for constructing prediction intervals by adaptively balancing epistemic and aleatoric uncertainty. Through a calibration process involving two distinct parameters, $\gamma_1$ and $\gamma_2$, CLEAR offers an improvement over classical methods that often address these uncertainty types in isolation or rely on their fixed, non-adaptive combination. Our evaluations using CQR (and SQR) for aleatoric uncertainty and PCS (and DE) for epistemic uncertainty show that interval width and quantile loss improve on both simulated and the 17 real-world datasets. CLEAR consistently achieves improved conditional coverage, notably by adapting interval widths appropriately in extrapolation regions, while yielding narrower interval widths.

Limitations remain despite CLEAR's significant advantages, and these warrant further discussion. When scaling CLEAR to significantly larger datasets and models, specific hyperparameters (e.g., number of bootstraps) can be adjusted to save computational costs. The accuracy of CLEAR is intrinsically linked to the quality of the base estimators for epistemic and aleatoric uncertainty. All our calibration datasets contained at least 150 data points, but for smaller calibration datasets, overfitting $\gamma_1$ and $\lambda$ could be a problem. Epistemic uncertainty, as demonstrated in the case study, must be addressed through careful judgment calls, in line with the principles of the PCS framework. The additive combination of scaled uncertainties, while powerful, is also a specific structural choice.

Future work could explore extension to classification tasks (Heiss et al., 2026), alternative $\lambda$ selection techniques, integration with active learning, and more scalable epistemic UQ approaches (Tagasovska et al., 2023). Extending CLEAR to time series settings could also be valuable, particularly for capturing the dynamics of temporal uncertainty. Finally, a deeper study of the interpretability of learned parameters could provide insights into the dominance of the two uncertainty sources.

## ACKNOWLEDGMENTS

We are grateful to Abhineet Agarwal and Michael Xiao for their valuable feedback, helpful discussions, and for sharing early previews and detailed explanations of their code. We also thank Natasa Tagasovska, João Vitor Romano, Erez Buchweitz, Omer Ronen, Jakob Weissteiner, Hanna Wutte, Florian Krach, Markus Chardonnet, Josef Teichmann, Bertrand Charpentier, Janis Postels, Alexander Immer, James-A. Goulet, and Hans Bühler for insightful and stimulating discussions. We are also grateful to Valérie Chavez-Demoulin and Marc-Olivier Boldi for their continuous support.

Ilia Azizi acknowledges support from the HEC Research Fund for research that enabled and facilitated computations on DCSR clusters at the University of Lausanne.

Juraj Bodik acknowledges support from the Hasler Foundation [grant number 24009]. Part of this work was conducted during his research visit to the UC Berkeley, supported by the Hasler Foundation. He thanks the Department of Statistics at UC Berkeley for their kind hospitality.

Jakob Heiss gratefully acknowledges support from the Swiss National Science Foundation (SNSF) Postdoc.Mobility fellowship [grant number P500PT_225356] and the Department of Statistics at UC Berkeley. He also wishes to thank the Department of Mathematics at ETH Zürich where the project's early ideas originated.

Bin Yu gratefully acknowledges partial support from NSF grant DMS-2413265, NSF grant DMS 2209975, NSF grant 2023505 on Collaborative Research: Foundations of Data Science Institute (FODSI), the NSF and the Simons Foundation for the Collaboration on the Theoretical Foundations of Deep Learning through awards DMS-2031883 and 814639, NSF grant MC2378 to the Institute for Artificial CyberThreat Intelligence and OperatioN (ACTION), and NIH (DMS/NIGMS) grant R01GM152718.

## REPRODUCIBILITY STATEMENT AND USAGE OF LARGE LANGUAGE MODELS

All code and datasets used in this work are provided at `https://github.com/Unco3892/clear` to ensure full reproducibility of our results. We declare that we used a large language model for grammar and language polishing, as well as for limited coding assistance (e.g., boilerplate code and debugging). All conceptual and theoretical contributions, experimental designs, and conclusions are our own.

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

Table 5: List of notable abbreviations used in the main paper.

| Abbreviation | Full Term | Description / Citation |
|---|---|---|
| PCS | Predictability, Computability, and Stability | Framework for veridical data science (Yu & Kumbier, 2020; Yu & Barter, 2024). |
| CQR | Conformalized Quantile Regression | Distribution-free prediction intervals (Romano et al., 2019) . |
| ALEATORIC | Bootstrapped CQR | CQR variant, where we compute CQR on bootstrapped data and take the median as final estimate. Follows from PCS framework (Yu & Barter, 2024) |
| ALEATORIC-R | Residual-based Bootstrapped CQR | ALEATORIC applied on residuals $Y_i - \hat{f}(X_i)$, where $\hat{f}$ is obtained from PCS. |
| UACQR | Uncertainty aware CQR | Method from (Rossellini et al., 2024) with two variants UACQR-S and UACQR-P |
| QRF | Quantile Random Forests | Tree-based model for quantile estimation (Meinshausen, 2006). |
| QXGB | Quantile XGBoost | Gradient boosting for quantile estimation. |
| GAM | Generalized Additive Model | Used for expectile or quantile regression (Servén & Brummitt, 2018). |
| DE | Deep Ensembles | Ensemble of neural networks for epistemic uncertainty (Lakshminarayanan et al., 2017). |
| SQR | Simultaneous Quantile Regression | Neural network approach for multiple quantile estimation (Tagasovska & Lopez-Paz, 2019). |
| AISL | Average Interval Score Loss | Computed from (Rossellini et al., 2024) and as defined in Appendix B.3.4. |
| PICP | Prediction Interval Coverage Probability | Fraction of targets within predicted intervals. |
| NIW | Normalized Interval Width | Average interval width normalized by target range. |
| NCIW | Normalized Calibrated Interval Width | NIW after test-time calibration. |
| RMSE | Root Mean Squared Error | Square root of the average squared prediction errors. |
| MAE | Mean Absolute Error | Average of absolute prediction errors. |
| $\mathcal{N}(0,1)$ | Standard Normal Distribution | Gaussian distribution with mean 0 and variance 1. |
| supp | support | Support of a distribution or function. |

LIST OF APPENDICES

# A  RELATED LITERATURE

In the main paper, Section 1 provides a high-level overview of uncertainty quantification in ML. Section 2.2 contains an overview of epistemic uncertainty, and Section 2.3 discusses the aleatoric uncertainty. In this appendix, we discuss these concepts in more depth. For an introduction to epistemic and aleatoric uncertainty, see Hüllermeier & Waegeman (2021), which provides a comprehensive overview.

## A.1  LITERATURE THAT IMPLICITLY ASSUMES $\gamma_1 = 1$

UACQR, introduced by Rossellini et al. (2024), is conceptually closest to CLEAR. Conceptually on a high level, this method corresponds to a variant of CLEAR where $\gamma_1 = 1$ is fixed, and only $\lambda$ is tuned. This is justified when aleatoric uncertainty is well-calibrated, which may often hold asymptotically. When the validation dataset is very small, setting $\gamma_1 = 1$ can even bring small advantages compared to tuning $\gamma_1$. In practice, however, aleatoric uncertainty is often miscalibrated due to over- or under-regularization. Tuning $\gamma_1$ helps correct its scale. Furthermore, it can happen in practice that the estimator of the aleatoric uncertainty has a much lower or much higher quality than the epistemic uncertainty. In this case, optimizing both parameters $\gamma_1$ and $\lambda$ allows us to compensate for the failure of one of the two uncertainties to some extent by putting more weight on the other type of uncertainty without changing the empirical marginal coverage. This results in a higher robustness and stability of CLEAR.

## A.2  LITERATURE THAT IMPLICITLY ASSUMES $\lambda = 1$

Most of the literature that combines epistemic and aleatoric uncertainty implicitly assumes that $\lambda = 1$, when they simply combine epistemic and aleatoric uncertainty in the ratio 1:1. On top of the resulting uncertainty, one can use (conformal) calibration, which corresponds to CLEAR's calibration of $\gamma_1$. However, in contrast to CLEAR, they do not rebalance the ratio of epistemic and aleatoric uncertainty with $\lambda$.

In practice, it is possible to underestimate the aleatoric uncertainty and overestimate epistemic uncertainty. This results in a one-dimensional calibration via $\gamma_1$, and consequently, narrow intervals in regions of dominating aleatoric uncertainty or in wide intervals in regions of dominating epistemic uncertainty. $\gamma_1$ alone cannot solve this. CLEAR can deal well with such situations by compensating for such an imbalance via $\lambda$.

### A.2.1  DEEP ENSEMBLES

While it is quite common to refer to *deep ensembles* (DE), whenever one uses an ensemble of neural networks (NNs) for uncertainty estimation, it is important to note that Lakshminarayanan et al. (2017) introduced DE as a method that both estimates epistemic and aleatoric uncertainty. For regression, they train each NN with 2 outputs estimating $\mu$ and $\sigma$ via a Gaussian Maximum-Likelihood-loss (as in Nix & Weigend (1994)), where $\sigma$ is responsible for the aleatoric uncertainty, which they refer to as "ambiguity in targets y for a given x". Moreover, they use the ensemble diversity to estimate epistemic uncertainty, which they refer to as "model uncertainty". Although DE is mainly known for its ensembling approach, (Lakshminarayanan et al., 2017, Table 2 in Appendix A.1) clearly shows that both the aleatoric and epistemic parts are crucial in terms of empirical performance. In their paper, the authors do not apply any calibration on top, i.e., as it is implicitly assumed that $\gamma_2 = \gamma_1 = 1 = \lambda$. However, it is common to apply (conformal) calibration on top. The commonly used calibration techniques only calibrate $\gamma_1$, while keeping $\lambda = 1$ fixed, in contrast to CLEAR. In particular, for DE, both the diversity of the ensemble and the bias on aleatoric are very sensitive to various hyperparameters. One type of uncertainty may be strongly underestimated. In contrast, the other type is strongly overestimated, which motivates the need to explicitly calibrate $\lambda$ in a data-driven way.

**Technical details on adversarial attacks for deep ensembles.** The original paper Lakshminarayanan et al. (2017) is written as if applying adversarial attacks is an integral part of DEs and one of the paper's main contributions. However, to the best of our knowledge, many practitioners refer to "DEs" without implying adversarial attacks during training, and adversarial attacks during training

are seen as an optional add-on to DEs, but not an important part of DEs. Furthermore, the regression results in (Lakshminarayanan et al., 2017, Table 2 in Appendix A.1) do not suggest that adversarial attacks during training are particularly beneficial to the performance.

**Technical details on measuring deep ensemble's diversity.** Intuitively, one should estimate high epistemic uncertainty if there is a significant disagreement among the ensemble's predictions and small epistemic uncertainty if they agree. While (Yu & Barter, 2024, Chapter 13) and Agarwal et al. (2025) suggest using quantiles, Lakshminarayanan et al. (2017) suggest using the empirical standard deviation to estimate DE's disagreement among ensemble members' predictions. It seems plausible that in Lakshminarayanan et al. (2017)'s case of 5 ensemble members, using the standard deviation and some Gaussian assumptions can be more appropriate while in (Yu & Barter, 2024, Chapter 13)'s and Agarwal et al. (2025)'s case of 100 or more ensemble members, quantiles can be more accurate.

**Variations of deep ensembles.** There exist several modifications of DE that, for example, promote the ensemble's diversity on the function space via an additional loss term during training (Wang et al., 2019), ensemble over multiple different hyperparameters (Wenzel et al., 2020b), or reduce the computational training cost (Kendall & Gal, 2017; Gal & Ghahramani, 2016; Wen et al., 2020; Havasi et al., 2021; Rossellini et al., 2024; Chan et al., 2025; Agarwal et al., 2025).

### A.2.2 MONTE CARLO DROPOUT

Gal & Ghahramani (2016) originally studied Monte Carlo Dropout (MC Dropout) without explicitly modeling aleatoric uncertainty, which was then extended by Kendall & Gal (2017) to also explicitly model aleatoric uncertainty. While Gal & Ghahramani (2016) see MC Dropout as an approximation of a Bayesian neural network, one can also see it as another ensemble method. The main difference to DE is that MC Dropout only trains one NN with dropout and obtains an ensemble after training by randomly setting weights of the model to zero at inference time. Analogously to DE, MC Dropout also adds epistemic and aleatoric uncertainty in the ratio 1:1 with the same disadvantages as described before in Appendix A.2.1.

### A.2.3 "A DEEPER LOOK INTO ALEATORIC AND EPISTEMIC UNCERTAINTY DISENTANGLEMENT"

Valdenegro-Toro & Saromo (2022) empirically concludes that ensembles have the best uncertainty and disentangling behavior of epistemic and aleatoric uncertainty. In their paper, the authors do not use any form of calibration. This would correspond to $\gamma_2 = \gamma_1 = 1 = \lambda$ in our notation. Their paper suggests a different loss function, which they call $\beta$-NLL, for training to mitigate the underestimation of aleatoric uncertainty to some extent in their experimental setting without comparing this approach to calibration. In (Valdenegro-Toro & Saromo, 2022, Figure 6), one can observe that even without the $\beta$-NLL, both epistemic and aleatoric uncertainty already have a good shape (that is, good relative uncertainty, see Appendix I) for deep ensembles (DE). The main problem of DE in (Valdenegro-Toro & Saromo, 2022, Figure 6) is that the epistemic uncertainty is too small by a very large factor (that is, poor absolute scale of uncertainty, see Appendix I), while aleatoric uncertainty has already almost the correct scaling. From our perspective, applying CLEAR on their DE would probably largely fix their problem of DE if CLEAR chooses $\gamma_1 \approx 1$ and $\lambda \gg 1$. However, they show that for this specific experiment, $\beta$-NLL also fixes the problem. In general, where one suspects that the aleatoric or the epistemic uncertainty might be too small or too large across the domain, we strongly recommend simply applying CLEAR on top of the already trained ensemble instead of retraining all the models with a new training pipeline. The concept of CLEAR can be implemented in a few minutes and calibrates an already trained ensemble in a few seconds. We considered an interesting open problem to study if $\beta$-NLL can improve the relative epistemic and aleatoric uncertainty (see Appendix I).

### A.2.4 "RECALIBRATION OF ALEATORIC AND EPISTEMIC REGRESSION UNCERTAINTY IN MEDICAL IMAGING"

Laves et al. (2021) also combines epistemic and aleatoric uncertainty in the ratio 1:1 and applies a single constant (corresponding to $\gamma_1$ in our notation) to scale the total predictive uncertainty, which corresponds to fixing $\lambda = 1$, resulting in the same potential for problems as mentioned before.

### A.3 BAYESIAN MODELS

If one had access to a perfect prior, perfectly computed Bayesian inference, it would provide well-calibrated epistemic and aleatoric uncertainty, at least in theory. However, in practice, the ratio of estimated epistemic and aleatoric uncertainty can be very wrong, and the uncertainty can be miscalibrated.

#### A.3.1 GAUSSIAN PROCESS REGRESSION (GPR)

Gaussian Process regression (Edward, 2003) provides a closed form for exact posterior epistemic uncertainty for a given Gaussian process as a prior and for a given known Gaussian noise distribution. If the (scale of the) prior or the scale (of the noise) is misspecified, the ratio of estimated epistemic and aleatoric uncertainty can be arbitrarily bad, and the uncertainty can be miscalibrated. Therefore, it is common to optimize hyperparameters like the noise scale and the prior scale (typically in a non-Bayesian way). The main differences to CLEAR are that 1) for GPR, one has to refit the model for every considered possibility of hyperparameters, while CLEAR optimizes $\gamma_1$ and $\lambda$ after fitting the model, resulting in much lower computational costs; 2) CLEAR can be applied to other base models as well such as tree-based models which are more popular in many applications; 3) standard-implementations of GPR fit the hyperparameter on the training data rather than on the validation data.

#### A.3.2 BAYESIAN NEURAL NETWORKS (BNNS)

Bayesian neural networks MacKay (1992); Neal (1996) offer a principled Bayesian framework for quantifying both epistemic and aleatoric uncertainty through the placement of a prior distribution on network weights. As for GPR, the ratio of estimated epistemic and aleatoric uncertainty in BNNs is highly sensitive to the choice of prior. Consequently, we advocate applying CLEAR to an already trained BNN, calibrating both uncertainty types via scaling factors $\gamma_1$ and $\gamma_2$ with negligible additional computational overhead. While exact Bayesian inference in large BNNs is computationally intractable, numerous approximation techniques have been proposed (Graves, 2011; Blundell et al., 2015; Hernández-Lobato & Adams, 2015; Gal & Ghahramani, 2016; Lakshminarayanan et al., 2017; Ritter et al., 2018; Daxberger et al., 2021; Heiss et al., 2022b; Wenzel et al., 2020a; Nguyen & Goulet, 2022b;a; Cong et al., 2024; Shen et al., 2024). Interestingly, theoretical (Heiss et al., 2022a; Heiss, 2024) and empirical (Wenzel et al., 2020a) studies suggest that some of these approximations can actually provide superior estimates compared to their exact counterparts, due to poor choices of priors, such as i.i.d. Gaussian priors, in certain settings.

#### A.3.3 EPICSCORE

The recent work by Cabezas et al. (2025) introduces EPICSCORE. Similar to our work, this method addresses the limitations of standard conformal prediction in capturing epistemic uncertainty. EPIC-SCORE focuses on enhancing existing conformal scores with Bayesian epistemic uncertainty. They also compute the average interval score for 95% predictive intervals on the datasets `airfoil` (where their best method out of 6 variants performs more than 2 times worse than the worst of the 3 variants of CLEAR), `concrete` (where their best method performs more than 1.5 times worse than the worst variant of CLEAR) and `Superconductivity` (where their best method performs approximately 1.3 times worse than the worst variant of CLEAR).

#### A.3.4 TABPFN

*TabPFN* (Müller et al., 2021; Hollmann et al., 2025) is a transformer trained to emulate Bayesian inference over a diverse prior of realistic tabular problems, achieving strong predictive uncertainty estimates with a single forward pass. Its prior spans diverse noise structures and function classes, yielding uncertainty estimates that inherently mix aleatoric and epistemic components. Although recent variants such as TabPFN-TS (Hoo et al., 2025) extend its applicability to time series, the method is limited to tabular datasets of limited size and does not disentangle uncertainty types. In contrast, our approach explicitly separates and calibrates epistemic and aleatoric uncertainty and scales to arbitrary modalities and data sizes.

### A.4 CONFORMAL LITERATURE THAT DOES NOT ACCOUNT FOR EPISTEMIC AND ALEATORIC UNCERTAINTY

The classical conformal prediction literature (Vovk et al., 2005; 2009; Vovk, 2012; 2013; Lei & Wasserman, 2014; Romano et al., 2019; Kivaranovic et al., 2020; Angelopoulos et al., 2024) primarily focuses on achieving marginal coverage, often with attention to asymptotic conditional coverage. However, it often overlooks epistemic uncertainty, which is especially critical in finite-sample settings. While Barber et al. (2020) established the impossibility of achieving exact distribution-free conditional coverage in finite samples, several recent works attempt to improve coverage guarantees in more restricted settings under certain assumptions. For instance, Gibbs et al. (2024) propose coverage guarantees over a subclass of distribution shifts, effectively interpolating between marginal and conditional coverage. Others, such as Guan (2022) and Dwivedi et al. (2020), provide guarantees over a finite set of prespecified subgroups. We argue that to obtain reliable conditional coverage, one must model the uncertainty arising from each stage of the data science life cycle (Yu & Barter, 2024), and appropriately integrate these uncertainties to achieve meaningful coverage guarantees.

### A.5 OTHER CALIBRATION METHODS

Predictive uncertainty can be expressed either as a predictive set for a given level $\alpha$, as a predictive distribution, or as a numerical value quantifying the level of uncertainty (e.g., the entropy of the predictive distribution). Our implementation of CLEAR yields predictive intervals, making it compatible with both base models that output intervals (such as QR) and with base models that output predictive distributions from which one can easily obtain predictive intervals. Conceptually, CLEAR could also be extended to predictive distributions or numerical values.

In the applied literature on **predictive intervals**, it is quite common to calibrate a constant factor to rescale the width of predictive intervals (Laves et al., 2021; Heiss et al., 2022b; Yu & Barter, 2024; Agarwal et al., 2025). Independently, the conformal community came up with almost identical methods based on theoretical considerations, as already discussed in Appendix A.4.

In the literature on **predictive distributions**, early works suggested more sophisticated non-linear transformations for the predictive distributions (Kuleshov et al., 2018; Song et al., 2019; Kuleshov & Deshpande, 2022). However, Levi et al. (2022) demonstrated that simply calibrating a constant scaling factor performs on par with these more sophisticated calibration methods.

To summarize, there are multiple (slightly) different methods to calibrate uncertainty with similar empirical performance. These methods only monotonically transform the uncertainty without changing the ranking of the uncertainties (i.e., if one was more uncertain on $x$ than on $\tilde{x}$ before the calibration step, one will also be more uncertain on $x$ than on $\tilde{x}$ afterwards; see Appendix I). In contrast, CLEAR and UACQR can also change the ranking of the uncertainties (e.g., in the situation of Figure 1, a small value of $\lambda$ would assign more uncertainty to the center around $x = 0$ compared to the region around $x = -6$, whereas a large value of $\lambda$ would assign more uncertainty to $x = -6$ than to $x = 0$).

### A.6 METHODS THAT MAINLY FOCUS ON DISTRIBUTIONAL ALEATORIC UNCERTAINTY

In this section, we provide more details on the literature overview on aleatoric uncertainty from Section 2.3.

One can directly estimate **conditional quantiles** for a given level $\alpha$ via *quantile regression* (QR) (Koenker & Bassett, 1978), either parametric or nonparametric, such as smooth quantile regression (Fasiolo et al., 2020) and quantile random forests (QRF) (Meinshausen, 2006). This is computationally cheap if you a priori know which level $\alpha$ is of interest for your predictive intervals.

Alternatively, one can also try to estimate the **conditional distribution**. This can initially be computationally more expensive, but once the conditional distributions are estimated, one can easily obtain conditional quantiles for multiple different levels $\alpha$. One of the computationally cheapest ways to estimate conditional distributions is to estimate parameters of a specific distribution (e.g., $\mu(x)$ and $\sigma(x)$ of a Gaussian distribution) (Nix & Weigend, 1994). There are semi-parametric extensions of this with universal approximation properties (Kratsios, 2023). There are many similar approaches to estimate conditional densities (Rothfuss et al., 2019). Another way to obtain a conditional distribution is to estimate the conditional quantiles for all levels $\alpha \in [0, 1]$ simultaneously, as in *simultaneous*

*quantile regression* (SQR) (Tagasovska & Lopez-Paz, 2019) and its extension MAQR (Chung et al., 2021). Especially for applications where the model output is high-dimensional (e.g., models that output images), more modern deep generative models have become popular, e.g., conditional generative adversarial networks (Oberdiek et al., 2022), conditional variational autoencoders (Han et al., 2020), and diffusion models (Chang et al., 2023; Chan et al., 2025).

**Limitations and empirical comparison.** The majority of these works fail to properly account for epistemic uncertainty[1], and don't include a post-hoc calibration step. Each of these methods could be incorporated into CLEAR as an alternative to our recommended ALEATORIC-R module. For example, in some preliminary experiments combining SQR (Tagasovska & Lopez-Paz, 2019) with ensemble-diversity (Lakshminarayanan et al., 2017) via CLEAR's calibration step results in significant improvements over the (calibrated versions) of the individual models. This shows the applicability of CLEAR's calibration step on pure deep-learning pipelines. Furthermore, when we compare the results of our entire CLEAR pipeline reported in our paper with the results of the entire SQR-pipeline and MAQR-pipeline reported in (Tagasovska & Lopez-Paz, 2019; Chung et al., 2021), we can see that CLEAR massively outperforms their pipelines. This shows that our PCS-based approach and our novel variant of QR are already highly recommendable choices, resulting in state-of-the-art performance of the CLEAR-pipeline directly out of the box.

## A.7 FURTHER RELATED LITERATURE

### A.7.1 "BEYOND PINBALL LOSS: QUANTILE METHODS FOR CALIBRATED UNCERTAINTY QUANTIFICATION"

Chung et al. (2021) explores the limitations of the pinball loss (which we call Quantile Loss), criticizing that its direct minimization does not guarantee correct marginal coverage. In CLEAR, we address this by framing the optimization of $\gamma_1$ and $\lambda$ as a constrained optimization problem (see Equation (4) in Appendix B.3): we minimize the pinball loss on a calibration dataset $\mathcal{D}_{cal}$ (see Line 5 in Algorithm 1) while constraining the marginal coverage on $\mathcal{D}_{cal}$ (see Line 4 in Algorithm 1). This constrained approach directly mitigates the criticism of Chung et al. (2021).

### A.7.2 UNCERTAINTY QUANTIFICATION FOR CONDITIONAL IMAGE GENERATION

Chan et al. (2025) suggests combining a diffusion model for aleatoric uncertainty with a hyper-network-generated ensemble for epistemic uncertainty to quantify the uncertainty in conditional image generation. Diffusion models are particularly well-suited for conditional image generation, and using a hyper-network can be a computationally more efficient way to obtain an ensemble. However, they combine these two sources of uncertainty simply in the ratio 1:1 without any form of calibration (corresponding to hard-coding $\lambda = \gamma_1 = \gamma_2 = 1$ in our notation). It would be interesting future work to apply calibration in the spirit of CLEAR on top of Chan et al. (2025) to extend the idea of CLEAR to conditional image generation.

### A.7.3 UNCERTAINTY QUANTIFICATION FOR PRETRAINED MODELS

Wang & Ji (2024) estimates the epistemic uncertainty for pre-trained classification models, while CLEAR focuses on both epistemic and aleatoric components for regression. It would be interesting future work to extend CLEAR to also be applicable to already pre-trained models using techniques presented in Wang & Ji (2024).

### A.7.4 UNCERTAINTY QUANTIFICATION AS A BINARY CLASSIFICATION PROBLEM

Altieri et al. (2024) do not provide predictive intervals or distributions. Instead, they partition test data into "good" (more certain) and "bad" (less certain) points. As future work, one could derive a binary classifier from CLEAR by thresholding the predictive interval width and then evaluating it under their proposed metrics.

---

[1]Chan et al. (2025) include epistemic uncertainty in the ratio 1:1; Chung et al. (2021) conduct limited experiments on including epistemic uncertainty in their appendix; Tagasovska & Lopez-Paz (2019) proposes *Orthonormal Certificates* for estimating epistemic uncertainty, but explicitly leave the combination for future work.

### A.7.5 CREDAL SETS

Hofman et al. (2024) describe epistemic uncertainty via credal sets on the space of distributions. This approach is quite natural for classification, where the space of distributions over $K$ classes is $(K-1)$-dimensional. However, they do not provide any concrete algorithms for regression, where the space of all distributions over a continuous set is infinite-dimensional. Hofman et al. (2024) suggest multiple ideas on how one can translate credal sets into two real-valued numbers describing the magnitude of the epistemic and the aleatoric uncertainty. Javanmardi et al. (2025) proposes a method that can provide conformal predictive sets for classification with guaranteed input-conditional coverage under the assumption that one already has access to credal sets that guaranteeably cover the true input/conditional distribution. They also briefly discuss that for real-world applications, this assumption is not satisfied.

### A.7.6 IN-SAMPLE CALIBRATION YIELDS CONFORMAL CALIBRATION GUARANTEES

For the distributional regression, Allen et al. (2025) suggest using conformal binning or conformal isotonic distributional regression and prove theoretical guarantees under the exchangeability assumption.

### A.7.7 ENGRESSION AND EXTRAPOLATION

Shen & Meinshausen (2024) uses generative methods for uncertainty estimation that, in particular, take advantage of pre-additive noise (i.e., noise that is directly added to $x$). Bodik (2024) uses extreme value theory for dimension reduction in the extrapolating region in a causal setting.

### A.7.8 UNCERTAINTY QUANTIFICATION FOR TIME SERIES

Adapting CLEAR to temporal settings offers a promising avenue for capturing complex uncertainty dynamics. Existing approaches tackle temporal uncertainty from various angles: Xu & Xie (2021) apply conformal methods, while Bodik & Pasche (2024) study uncertainty in time series stemming from extreme events from a causal perspective. For irregularly observed data, Herrera et al. (2021); Krach et al. (2022); Andersson et al. (2023); Krach & Teichmann (2024); Heiss et al. (2024); Krach (2025); Crowell et al. (2025) utilize neural jump ordinary differential equations (NJODEs) to forecast trajectories and estimate the conditional variance, providing natural estimators of aleatoric uncertainty. However, existing NJODE literature does not combine this aleatoric component with epistemic uncertainty estimation. An extension of CLEAR could address this gap by using NJODE-based variance estimates as the aleatoric component and combining them with PCS-inspired ensemble-based epistemic uncertainty estimates via CLEAR's dual-parameter calibration. For example, Adamov et al. (2025) uses NJODE's uncertainty estimation to detect outliers—a setting where disentangling epistemic and aleatoric sources could further improve detection. Similarly, recent works leverage foundation models for time series forecasting (Hoo et al., 2025; Moroshan et al., 2026). None of these works explicitly decompose the uncertainty into its epistemic and aleatoric components, suggesting that CLEAR's calibration framework could complement them as well.

## B  THEORY: COVERAGE GUARANTEES AND THEORETICAL JUSTIFICATIONS

### B.1  FINITE-SAMPLE MARGINAL COVERAGE FOR CLEAR

While conformal methods typically offer finite-sample marginal coverage guarantees, our implementation of CLEAR does not strictly adhere to these guarantees for two reasons:

1. We reuse the validation dataset $\mathcal{D}_{\text{val}}$ as the calibration dataset, i.e., $\mathcal{D}_{\text{cal}} = \mathcal{D}_{\text{val}}$.

2. We optimize two parameters $\gamma_1$ and $\lambda$ on the calibration dataset $\mathcal{D}_{\text{cal}}$.

However, our empirical results in Tables 14, 19 and 24 show that for reasonably sized datasets, this theoretical discrepancy does not visibly impact our practical marginal coverage. From a theoretical perspective, CLEAR still achieves asymptotic marginal coverage (as a consequence of asymptotic conditional coverage, see Lemma 2.1).

By slightly modifying the CLEAR procedure, we can obtain a theoretical finite-sample guarantee for marginal coverage under the standard exchangeability assumption, as in conformal inference, without sacrificing asymptotic conditional coverage. The key idea is to optimize the parameter $\lambda$ using only a validation dataset $\mathcal{D}_{\text{val}}$, and then calibrate $\gamma_1$ using a separate, previously unseen calibration dataset $\mathcal{D}_{\text{cal}}$.

**Definition B.1** (Conformalized CLEAR). Let the available data be split into a training set $\mathcal{D}_{\text{train}}$ and an old validation set $\mathcal{D}_{\text{val}}^{\text{old}}$. We first split $\mathcal{D}_{\text{val}}^{\text{old}}$ into two disjoint sets: a new validation set $\mathcal{D}_{\text{val}}$ and a calibration set $\mathcal{D}_{\text{cal}}$. The procedure is as follows:

1. **Train and Optimize $\lambda$ by running Algorithm 1 on $\mathcal{D}_{\text{train}}$ and $\mathcal{D}_{\text{val}}$**: The model $\hat{f}$ and the uncertainty estimators are trained on $\mathcal{D}_{\text{train}}$ (Step 2 and 3 of Algorithm 1). The optimal value $\lambda^*$ is found by optimizing the QuantileLoss on $\mathcal{D}_{\text{val}}$, without using any data from $\mathcal{D}_{\text{cal}}$ (Step 4 and 5 of Algorithm 1).

2. **Compute Conformity Scores:** For each data point $(X_i, Y_i) \in \mathcal{D}_{\text{cal}}$, compute the conformity score $S_i^{\lambda^*}$:

$$S_i^{\lambda^*} = \max\left\{ \frac{\hat{f}(X_i) - Y_i}{\hat{f}(X_i) - \tilde{l}_{\lambda^*}(X_i)}, \frac{Y_i - \hat{f}(X_i)}{\tilde{u}_{\lambda^*}(X_i) - \hat{f}(X_i)} \right\}$$

where $\tilde{l}_{\lambda^*} := \hat{f} - \hat{q}_{\alpha/2}^{\text{ale}} - \lambda^* \hat{q}_{\alpha/2}^{\text{epi}}$ and $\tilde{u}_{\lambda^*} := \hat{f} + \hat{q}_{1-\alpha/2}^{\text{ale}} + \lambda^* \hat{q}_{1-\alpha/2}^{\text{epi}}$.

3. **Calibrate the Prediction Interval:** Set the calibration parameter $\gamma_1^*$ to be the $\lceil (1 - \alpha)(|\mathcal{D}_{\text{cal}}| + 1) \rceil$-th smallest value among the conformity scores $S_i^{\lambda^*}$ from $\mathcal{D}_{\text{cal}}$. If $\lceil (1 - \alpha)(|\mathcal{D}_{\text{cal}}| + 1) \rceil > |\mathcal{D}_{\text{cal}}|$, set $\gamma_1^* = \infty$.

4. **Form the Final Prediction Interval:** For a new test point $x_{\text{new}}$, the final $(1 - \alpha)$-prediction interval is given by:

$$C(x_{\text{new}}) = \left[ \hat{f}(x_{\text{new}}) - \gamma_1^* \left( \hat{q}_{\alpha/2}^{\text{ale}} + \lambda^* \hat{q}_{\alpha/2}^{\text{epi}} \right), \quad \hat{f}(x_{\text{new}}) + \gamma_1^* \left( \hat{q}_{1-\alpha/2}^{\text{ale}} + \lambda^* \hat{q}_{1-\alpha/2}^{\text{epi}} \right) \right]$$

This modified version of CLEAR, which we call Conformalized CLEAR, satisfies the standard finite-sample marginal coverage guarantee of conformal prediction.

**Lemma B.2.** *Under the assumption that the data points in the calibration set $\mathcal{D}_{cal}$ and the test point $(X_{new}, Y_{new})$ are exchangeable, the prediction interval $C(X_{new})$ generated by the Conformalized CLEAR procedure satisfies:*

$$\mathbb{P}(Y_{new} \in C(X_{new})) \geq 1 - \alpha.$$

*Proof.* The proof is a direct application of the standard theoretical guarantees for split conformal prediction. Since $\lambda^*$ is determined using only data from $\mathcal{D}_{\text{val}}$, it is fixed with respect to the calibration set $\mathcal{D}_{\text{cal}}$. The conformity scores $S_i^{\lambda^*}$ are therefore exchangeable for all $(X_i, Y_i) \in \mathcal{D}_{\text{cal}} \cup \{(X_{\text{new}}, Y_{\text{new}})\}$. The choice of $\gamma_1^*$ as the empirical $(1 - \alpha)(1 + 1/|\mathcal{D}_{\text{cal}}|)$-quantile of the calibration scores ensures that the resulting prediction interval achieves at least $1 - \alpha$ marginal coverage, as established by (Angelopoulos et al., 2024, Theorem 1.4). □

However, in practice, we recommend using our default implementation of CLEAR, where the validation data is reused for calibration ($\mathcal{D}_{\text{cal}} = \mathcal{D}_{\text{val}}$). This improves data efficiency while still achieving strong approximate marginal coverage in our experiments (see Tables 14, 19 and 24). Avoiding a split allows more data for both validation and calibration, which improves model selection and stabilizes marginal and conditional coverage. Even when marginal coverage $\mathbb{P}(Y_{\text{new}} \in C(X_{\text{new}})) \geq 1 - \alpha$ is satisfied, the actual coverage $\mathbb{P}(Y_{\text{new}} \in C(X_{\text{new}})|\mathcal{D}_{\text{train}}, \mathcal{D}_{\text{val}}, \mathcal{D}_{\text{cal}})$ can be significantly lower than the target coverage $1 - \alpha$ for a fixed calibration dataset $\mathcal{D}_{\text{cal}}$, due to the inherent variation of $\mathcal{D}_{\text{cal}}$. Only with sufficiently large calibration datasets $\mathcal{D}_{\text{cal}}$ can we expect $\mathbb{P}(Y_{\text{new}} \in C(X_{\text{new}})|\mathcal{D}_{\text{train}}, \mathcal{D}_{\text{val}}, \mathcal{D}_{\text{cal}})$ to be reliably close to $\mathbb{P}(Y_{\text{new}} \in C(X_{\text{new}}))$.[2]

### B.1.1 LIMITATIONS OF CONFORMAL MARGINAL COVERAGE GUARANTEES

The conformal theory heavily relies on the assumptions of exchangeability. Exchangeability means that the joint distribution of calibration and test observations is invariant to permutations (e.g., iid observations satisfy this assumption). While exchangeability is theoretically convenient, it is unrealistic in many real-world settings. Models are typically trained on past data and deployed in the future, where the distribution of $X_{\text{new}}$ usually shifts, i.e., $\mathbb{P}[X_{\text{new}}] \neq \mathbb{P}[X]$. Even if $\mathbb{P}[Y_{\text{new}}|X_{\text{new}}] = \mathbb{P}[Y|X]$ remains fixed, such marginal shifts in $X_{\text{new}}$ can cause conformal methods to catastrophically fail to provide valid marginal coverage. In Section 4.1 and Appendix C (Figures 2 and 4 to 6), CLEAR empirically remains robust, while CQR and Naive-Conformal fail under distribution shifts in $X_{\text{new}}$. E.g., Figure 4, suggests $\mathbb{P}[Y_{\text{new}} \in C_{\text{Naive}}(X_{\text{new}}) \mid |X_{\text{new}}| \geq 2] \leq 70\% \ll 90\% = 1 - \alpha$, thus a marginal distribution shift of $X_{\text{new}}$ that strongly increases the probability of $|X_{\text{new}}| > 2$, would lead to a large drop of marginal coverage for $(X_{\text{new}}, Y_{\text{new}})$. CLEAR likewise lacks formal guarantees under extreme shifts, but consistently performs more reliably across our experiments. In the case study (Section 4.3), a realistic temporal split (see (Yu & Barter, 2024, Section 8.4.3)) also violates exchangeability, and CLEAR outperforms conformal baselines in marginal coverage. Caution is required when trusting conformal guarantees, as the assumption of exchangeability is often not met in practice, and some conformal methods catastrophically fail for slight deviations from the exchangeability assumption.

Even under the assumption of exchangeability, conformal guarantees have further weaknesses:

1. The conformal marginal coverage guarantee

$$\mathbb{P}[Y_{\text{new}} \in C(X_{\text{new}})] = \mathbb{E}_{\mathcal{D}_{\text{train}}, \mathcal{D}_{\text{cal}}}[\mathbb{P}[Y_{\text{new}} \in C(X_{\text{new}})|\mathcal{D}_{\text{train}}, \mathcal{D}_{\text{cal}}]] \geq 1 - \alpha$$

   does not imply that $\mathbb{P}[Y_{\text{new}} \in C(X_{\text{new}})|\mathcal{D}_{\text{train}}, \mathcal{D}_{\text{cal}}] \geq 1 - \alpha$ for a fixed realization of the calibration set $\mathcal{D}_{\text{cal}}$, as already discussed before. If the calibration residuals are small by chance, conformal intervals may be too narrow, especially with tiny calibration datasets. Guaranteed reliable calibration is generally unattainable with tiny calibration datasets: Even if the exchangeability assumption is satisfied, even methods with conformal guarantees often strongly undercover, i.e., $\mathbb{P}_{\mathcal{D}_{\text{train}}, \mathcal{D}_{\text{cal}}}[\mathbb{P}[Y_{\text{new}} \in C_{\text{conformal}}(X_{\text{new}})|\mathcal{D}_{\text{train}}, \mathcal{D}_{\text{cal}}] \ll 1 - \alpha] \gg 0$. Therefore, caution is required when communicating conformal guarantees to practitioners. Kivaranovic et al. (2020) also discuss this problem and suggest a method to mitigate this via Probably Approximately Valid prediction intervals (Kivaranovic et al., 2020, Def. 2).

2. Beyond marginal coverage, CLEAR is designed to improve *conditional calibration*: $\mathbb{P}[Y_{\text{new}} \in C(X_{\text{new}})|X_{\text{new}}] \approx 1 - \alpha$. This is crucial in human-in-the-loop settings, where interventions are prioritized based on an accurate *ranking* of predictive uncertainty across data points (see Appendix I). Marginal coverage guarantees offer no guarantees for such rankings nor for conditional coverage. A method could have perfect marginal coverage but rank uncertainties arbitrarily. In other words, marginal coverage guarantees only address one specific performance metric (marginal coverage), while ignoring many other metrics that are often more important in practice.

To summarize, conformal marginal coverage guarantees (such as Lemma B.2) say very little about the overall quality of an uncertainty quantification method. Conformal marginal coverage guarantees only

---

[2]Conformal theory provides theoretical guarantees for $\mathbb{P}(Y_{\text{new}} \in C(X_{\text{new}})) \geq 1 - \alpha$ and $\mathbb{P}(Y_{\text{new}} \in C(X_{\text{new}})|\mathcal{D}_{\text{train}}, \mathcal{D}_{\text{val}}) \geq 1 - \alpha$ under the standard exchangeability assumption . However, it does *not* provide finite-sample guarantees for $\mathbb{P}(Y_{\text{new}} \in C(X_{\text{new}})|\mathcal{D}_{\text{train}}, \mathcal{D}_{\text{val}}, \mathcal{D}_{\text{cal}})$ and $\mathbb{P}(Y_{\text{new}} \in C(X_{\text{new}})|\mathcal{D}_{\text{cal}})$ for a fixed calibration dataset, even under the standard exchangeability assumption.

shed light on a very specific aspect of uncertainty quantification and only under the quite unrealistic assumption of exchangeability.

## B.2 ASYMPTOTIC CONDITIONAL COVERAGE FOR CLEAR: LEMMA 2.1 PROOF & ASSUMPTIONS

*Proof of Lemma 2.1.* $|\mathcal{D}_{\text{train}}|, |\mathcal{D}_{\text{val}}| \to \infty$ ensures that model selection on $\mathcal{D}_{\text{val}}$ identifies $k$ consistent base models among the candidates, since the validation loss converges to the population loss. The consistency of the predictors implies $\hat{q}_\tau^{\text{epi}}(x) \to 0$, and the consistency of the quantile estimators implies $\hat{q}_\tau^{\text{ale}}(x) \to q_\tau^{\text{ale}}(x)$ point-wise. Hence, the estimated pre-calibrated intervals converge to their population analogues. Since $\sup_{\lambda \in \Lambda} \lambda \hat{q}_\tau^{\text{epi}}(x) \to 0$ converges to 0 uniformly over $\lambda \in \Lambda$ (due to compactness of $\Lambda$), therefore $C(x)$ is asymptotically equal to true conditional quantiles $[q_{\alpha/2}(x), q_{1-\alpha/2}(x)]$; and necessarily $\lim_{|\mathcal{D}_{\text{cal}}| \to \infty} \gamma_1 = 1$, as is the case in classical CQR (see e.g. (Angelopoulos et al., 2024, Section 5)). □

The epistemic component vanishes asymptotically in this proof. However, for finite samples, the epistemic uncertainty plays a crucial role in preventing under-coverage in out-of-sample regions as we intuitively explain in Appendix B.4 and empirically demonstrate in Section 4.1 and Appendix C. While these synthetic datasets provide the most direct and intuitive empirical evidence of the importance of combining epistemic and aleatoric uncertainty, all our experiments (Sections 3 and 4 and Appendices C to H) support this hypothesis.

The assumptions underlying Lemma 2.1 are generally mild and often satisfied in practice. The assumption that $\Lambda$ is compact is particularly mild. In our experiments, we use a finite grid; for example, a combination of linearly spaced values from 0 to 0.09 and logarithmically spaced values from 0.1 to 100, resulting in approximately 4010 points. We conjecture that even if $\Lambda$ were unbounded, the result of Lemma 2.1 would still hold, since QuantileLoss is a strictly proper scoring rule and thus would not lead to excessively large values of $\lambda^*$ in the limit.

Lemma 2.1 also assumes consistency of the estimators. Many estimators satisfy this condition for both regression and quantile regression tasks:

1. Quantile Random Forests have been shown to consistently estimate quantiles under reasonable conditions, such as by regularizing the minimum number of samples per leaf (Meinshausen, 2006). Similarly, classical Random Forests are known to be consistent under standard assumptions (Scornet et al., 2015). Both QRF and Random Forests are available in our implementation of CLEAR.

2. Boosting methods trained on general loss functions can be made consistent under certain conditions, especially when regularized through early stopping (Zhang & Yu, 2005). This suggests that XGBoost and its quantile version, QXGB, can be consistent for suitable choices of hyperparameters. Both are included in our implementation of CLEAR.

3. More generally, regularized minimization of QuantileLoss over a sufficiently expressive function class on $\mathcal{D}_{\text{train}}$ yields consistent quantile estimators under broad assumptions (Steinwart & Christmann, 2011).

In our experiments, we set $k = 1$, so Lemma 2.1 requires only one of the base models to be consistent.

Finally, note that Lemma 2.1 implicitly assumes i.i.d. data, as described in Section 2.1. For real-world datasets, this i.i.d. assumption is often the most difficult to justify in practice and may pose a greater challenge than the other assumptions in the lemma.

## B.3 PROPERTIES OF THE QUANTILE LOSS

We define the $\text{QuantileLoss}(\mathcal{D}, C) := \frac{1}{|\mathcal{D}|} \sum_{(x,y) \in \mathcal{D}} \left[ QL_{\alpha/2}(y, l(x)) + QL_{1-\alpha/2}(y, u(x)) \right] / 2$ (also denoted as "pinball loss", "quantile loss", "asymmetric piecewise linear loss", "linlin loss", "hinge loss", "tick loss", or "newsvendor loss" (Gneiting, 2011)), with $l(x), u(x)$ denoting the bounds of $C(x)$, and $QL_\tau(y, q) = (y - q)\left(\tau - \mathbb{1}_{(-\infty, q]}(y)\right)$. Note that the majority of the literature defines the quantile loss as $QL_\tau$, whereas our QuantileLoss already aggregates $QL_\tau$ over both the upper and

the lower bound of the intervals $C$ and over the data points in $\mathcal{D}$, similarly to the interval score loss (see Appendix B.3.4).

We use the QuantileLoss for three different purposes in this paper:

1. Some QR-methods in Step 2 of Algorithm 1 use the QuantileLoss to train their models.

2. In Step 4 of Algorithm 1 we minimize the QuantileLoss to determine $\lambda^*$. In other words, Step 3 and 4 of Algorithm 1 together (approximately) solve

$$(\gamma_1^\star, \lambda^\star) = \underset{\substack{(\gamma_1, \lambda) \in (0,\infty) \times \Lambda \\ \text{s.t. } \left|\left\{(x,y) \in \mathcal{D}_{\text{cal}} : y \in C_{\gamma_1, \lambda}(x)\right\}\right| \geq \lceil (1-\alpha)(|\mathcal{D}_{\text{cal}}|+1)\rceil}}{\arg\min} \text{QuantileLoss}(\mathcal{D}_{\text{val}}, C_{\gamma_1, \lambda}), \quad (4)$$

where $C_{\gamma_1, \lambda} = \left[\hat{f} - \gamma_1 \hat{q}_{\alpha/2}^{\text{ale}} - \lambda\gamma_1 \hat{q}_{\alpha/2}^{\text{epi}}, \ \hat{f} + \gamma_1 \hat{q}_{1-\alpha/2}^{\text{ale}} + \lambda\gamma_1 \hat{q}_{1-\alpha/2}^{\text{epi}}\right]$.

3. We use the QuantileLoss as an evaluation metric on the test dataset $\mathcal{D}_{\text{test}}$.

In the following, we will discuss multiple favorable properties of the QuantileLoss (which equivalently also hold for the Interval Score Loss, see Appendix B.3.4).

### B.3.1 INTUITION BEHIND THE QUANTILE LOSS

In many real-world applications, the severity of a prediction error often depends on the magnitude of the deviation from the predictive interval. For example, in financial portfolio management, a massive drop below the predicted lower bound can be significantly more damaging than a slight deviation. Similarly, for flood protection systems where dam heights are based on an upper predictive bound, a large amount of overflow causes substantially more damage than a small overflow. Traditional metrics like PICP treat coverage as a binary outcome, distinguishing only between points that are covered and those that are not. The QuantileLoss, however, offers a more nuanced evaluation by penalizing the magnitude of a data point's distance from the predictive interval when it is not covered.

Intuitively, the QuantileLoss strongly penalizes the distance to the predictive intervals for data points outside the predictive interval, and gently penalizes the width of the predictive intervals that do not cover the data points. This incentivizes the predictive intervals to widen in regions of large uncertainty and to adaptively narrow down in regions of low uncertainty, incentivizing good relative uncertainty (see Appendix I). Simultaneously, the QuantileLoss incentivizes good absolute uncertainty (i.e., marginal calibration, see Appendix I): If the proportion of data points below the upper bounds is less than $1 - \frac{\alpha}{2}$, then increasing the upper bounds by a constant improves the QuantileLoss until the proportion reaches $1 - \frac{\alpha}{2}$, i.e., for all $c > 0$,

$$\left|\{(x, y) \in \mathcal{D}_{\text{cal}} : y \leq u(x) + c\}\right| < (1 - \frac{\alpha}{2})|\mathcal{D}_{\text{cal}}|$$
$$\implies \text{QuantileLoss}(\mathcal{D}, [l, u]) > \text{QuantileLoss}(\mathcal{D}, [l, u + c]),$$

and vice versa if the proportion of data points below the upper bounds is more than $1 - \frac{\alpha}{2}$, i.e., for all $c > 0$,

$$\left|\{(x, y) \in \mathcal{D}_{\text{cal}} : y \leq u(x)\}\right| > (1 - \frac{\alpha}{2})|\mathcal{D}_{\text{cal}}|$$
$$\implies \text{QuantileLoss}(\mathcal{D}, [l, u]) < \text{QuantileLoss}(\mathcal{D}, [l, u + c]),$$

and analogously, the QuantileLoss incentivizes the lower bound to be above $\frac{\alpha}{2}|\mathcal{D}_{\text{cal}}|$ data points and below $(1 - \frac{\alpha}{2})|\mathcal{D}_{\text{cal}}|$ data points. The QuantileLoss simultaneously evaluates multiple different properties of predictive intervals and cannot be easily tricked, in contrast to other metrics: PICP can be easily maximized by infinitely wide intervals, which are completely useless in practice; NIW can be minimized by zero-width intervals, which are useless in practice; NCIW only measures the relative uncertainty while completely ignoring the marginal coverage.

### B.3.2 THE QUANTILELOSS MEASURES INPUT-CONDITIONAL CALIBRATION

In the following, we will mathematically argue why the QuantileLoss measures input-conditional calibration. Within this paper, we denote input conditional calibration $\mathbb{P}[Y_{\text{new}} \in C(X_{\text{new}})|X_{\text{new}}] = 1 - \alpha$ simply as conditional calibration.

The true input conditional quantiles minimize the expected QuantileLoss, i.e.,

$$(q_{\alpha/2}, q_{1-\alpha/2}) \in \underset{(l,u) \in \mathcal{Y}^{\mathcal{X}} \times \mathcal{Y}^{\mathcal{X}}}{\arg \min} \mathbb{E}_{(X_{\text{new}}, Y_{\text{new}})} \left[ \text{QuantileLoss}\big(\{(X_{\text{new}}, Y_{\text{new}})\}, [l, u]\big) \right]. \tag{5}$$

Any minimizer of the QuantileLoss satisfies input-conditional coverage almost surely (a.s.)[3], i.e., any solution $(l^*, u^*)$ of the minimization problem (5) satisfies that $l^*(X)$ is an input-conditional $\alpha/2$-quantile a.s. and $u^*(X)$ is an input-conditional $1 - \alpha/2$-quantile a.s., thus, $\mathbb{P}[Y_{\text{new}} \in [l(X_{\text{new}}), u(X_{\text{new}})]|X_{\text{new}}] \overset{\text{a.s.}}{\geq} 1 - \alpha$. In other words, any deviation from the true quantiles gets penalized by the expected QuantileLoss, as it is a strictly proper scoring rule (Koenker, 2005). If the true conditional CDF is a.s. continuous, then any solution $(l^*, u^*)$ of (5) satisfies input-conditional calibration $\mathbb{P}[Y_{\text{new}} \in [l(X_{\text{new}}), u(X_{\text{new}})]|X_{\text{new}}] \overset{\text{a.s.}}{=} 1 - \alpha$. In other words, any deviation from input-conditional coverage gets penalized. Applying the QuantileLoss on a finite dataset $\mathcal{D}$ can be seen as a Monte-Carlo approximation of the expected QuantileLoss.

Another common evaluation method is to compare the average interval width NIW (or NCIW) among methods that approximately obtain the targeted marginal coverage. However, this evaluation method can be exploited: Even if you perfectly know the true distribution, reporting intervals that do not satisfy input-conditional coverage would be optimal. This evaluation method prefers intervals that over-cover in regions with low uncertainty and under-cover in regions of high uncertainty, as demonstrated in the following examples (Examples B.3 and B.4 are easier to derive, but Examples B.5 and B.6 are slightly more insightful).

*Example* B.3. Let $X \in \{1, 2\}$, with $\mathbb{P}[X = 1] = 0.5$ and $\mathbb{P}[X = 2] = 0.5$. Let $Y|X = 1 \sim \mathcal{U}(-1, 1)$ and $Y|X = 2 \sim \mathcal{U}(-2, 2)$. For a target coverage of $1 - \alpha = 0.9$, the true conditional intervals are:

- $[-0.9, 0.9]$ when $X = 1$ (width=1.8, coverage=0.9)

- $[-1.8, 1.8]$ when $X = 2$ (width=3.6, coverage=0.9)

The average width of the true intervals is $\mathbb{E}[\text{width}] = 0.5 \times 2 \cdot 0.9 + 0.5 \times 2 \cdot 1.8 = 2.7$. The marginal coverage is exactly 0.9. Now, consider an alternative method that sacrifices conditional calibration to minimize average width. This method could report the following intervals:

- $[-1, 1]$ when $X = 1$ (width=2, coverage=1)

- $[-1.6, 1.6]$ when $X = 2$ (width=3.2, coverage=0.8)

The marginal coverage of this method is $\mathbb{P}[\text{covered}] = 0.5 \times 1 + 0.5 \times 0.8 = 0.9$. It still achieves the target marginal coverage of $90\%$, but its average width is $\mathbb{E}[\text{width}] = 0.5 \times 2 + 0.5 \times 2 \cdot 1.6 = 2.6$. Since $2.6 < 2.7$, this method would be preferred, demonstrating that NIW can incentivize deviations from input-conditional calibration. $\diamond$

*Example* B.4. Under the setting of Example B.3, the situation gets even more extreme if we change to $\alpha = 0.5$: For a target coverage of $1 - \alpha = 0.5$, the true conditional intervals are:

- $[-0.5, 0.5]$ when $X = 1$ (width=1, coverage=0.5)

- $[-1, 1]$ when $X = 2$ (width=2, coverage=0.5)

The average width of the true intervals is $\mathbb{E}[\text{width}] = 0.5 \times 1 + 0.5 \times 2 = 1.5$. The marginal coverage is exactly 0.5. Now, consider an alternative method that sacrifices conditional calibration to minimize average width. This method could report the following intervals:

- $[-1, 1]$ when $X = 1$ (width=2, coverage=1)

- $[127, 127]$ when $X = 2$ (width=0, coverage=0)

The marginal coverage of this untruthful method is $\mathbb{P}[\text{covered}] = 0.5 \times 1 + 0.5 \times 0 = 0.5$. It still achieves the target marginal coverage, but its average width is $\mathbb{E}[\text{width}] = 0.5 \times 2 + 0.5 \times 0 = 1$.

---

[3]A statement holds *almost surely* if it holds with $100\%$ probability. E.g., a standard normally distributed random variable $X \sim \mathcal{N}(0, 1)$ is a.s. not exactly equal to $\sqrt{2}$, i.e., $X \overset{\text{a.s.}}{\neq} \sqrt{2}$.

Since $1 < 1.5$, this untruthful method would be preferred, demonstrating that NIW can incentivize strong deviations from input-conditional calibration. Here, the interval that minimizes NIW (and NCIW) under the constraint of maintaining marginal coverage, outputs a much wider interval for $X = 1$ than for $X = 2$, while there is obviously more uncertainty for $X = 2$ than for $X = 1$. ◇

*Example* B.5. Let $\mathbb{P}[X = 1] = 0.5 = \mathbb{P}[X = 2]$, and $Y|X = x \sim \mathcal{N}(\mu = 0, \sigma = x)$. Then for $\alpha = 5\%$, the true conditional quantiles would be $q_{0.975}(x) = \Phi^{-1}(0.975)x \approx 1.96x$ and $q_{0.025}(x) = \Phi^{-1}(0.025)x \approx -1.96x$, thus $[q_{0.025}, q_{0.975}]$ exactly satisfy input-conditional calibration $\mathbb{P}\left[Y_{\text{new}} \in [q_{0.025}(X_{\text{new}}), q_{0.975}(X_{\text{new}})] \mid X_{\text{new}}\right] \stackrel{\text{a.s.}}{=} 95\%$. However, the on average narrowest intervals satisfying marginal coverage would be approximately

- $[-2.16357, 2.16357]$ when $X = 1$ (width $= 2 \cdot 2.16357$, coverage $\approx 0.969502$)

- $[-1.81514 \cdot 2, 1.81514 \cdot 2]$ when $X = 2$ (width $= 2 \cdot 1.81514 \cdot 2$, coverage $\approx 0.930498$).

These intervals fail input-conditional coverage while maintaining marginal coverage and result in a narrower average width $5.79385 < 5.88$ than the true intervals $[q_{0.025}, q_{0.975}]$. ◇

*Example* B.6. Let $\mathbb{P}[X = 1] = \frac{9}{19} = \mathbb{P}[X = 2]$, $\mathbb{P}[X = 11] = \frac{1}{19}$, and $Y|X = x \sim \mathcal{N}(\mu = 0, \sigma = x)$. Then for $\alpha = 10\%$, the true conditional quantiles would be $q_{0.95}(x) = \Phi^{-1}(0.95)x \approx 1.645x$ and $q_{0.05}(x) = \Phi^{-1}(0.05)x \approx -1.645x$, thus $[q_{0.05}, q_{0.95}]$ exactly satisfy input-conditional calibration $\mathbb{P}\left[Y_{\text{new}} \in [q_{0.05}(X_{\text{new}}), q_{0.95}(X_{\text{new}})] \mid X_{\text{new}}\right] \stackrel{\text{a.s.}}{=} 90\%$. However, the on average narrowest intervals satisfying marginal coverage would be approximately

- $[-2.16357, 2.16357]$ when $X = 1$ (width $= 2 \cdot 2.16357$, coverage $\approx 0.969502$)

- $[-1.81514 \cdot 2, 1.81514 \cdot 2]$ when $X = 2$ (width $= 2 \cdot 1.81514 \cdot 2$, coverage $\approx 0.930498$).

- $[12345, 12345]$ when $X = 11$ (width $= 0$, coverage $= 0$).

These intervals fail input-conditional coverage while maintaining marginal coverage and result in a narrower average width $5.48891 < 6.579415$ than the true intervals $[q_{0.05}, q_{0.95}]$. ◇

**Interpretation of Examples B.3 to B.6.** While QuantileLoss is only minimized for intervals consisting of the true quantiles $[q_{\frac{\alpha}{2}}, q_{1-\frac{\alpha}{2}}]$, the Examples B.3 to B.6 showed that metrics like PICP, NIW, NCIW, and their combinations can be misaligned with input-conditional coverage. This misalignment can be very severe in Examples B.4 and B.6, whereas it is less severe in Examples B.3 and B.5. Intuitively, the misalignment can be particularly extreme for large values of $\alpha$ (as in Example B.4) or for inputs that are much more uncertain than the average uncertainty (as for $x = 11$ in Example B.5). In our experiments on real-world data in Sections 4.2 and 4.3 and Appendix F, compared to these extreme examples, the different metrics are more aligned, in the sense that CLEAR shows superior performance across all considered metrics. Reasons for this empirically observed alignment could be that 1st, none of the compared methods actively tries to exploit the weaknesses of PICP, NIW, NCIW; 2nd, in our experiments, we use only small values of $\alpha = 5\%$ (and $\alpha = 10\%$); 3rd, there might not be many regions of extremely high uncertainty in these datasets.

Now we have concluded that under the QuantileLoss, it is optimal always to report the true quantiles if the true underlying distributions are known. In the next subsection, we will discuss how this translates to situations where the true distributions are not known.

### B.3.3 THE QUANTILELOSS INCENTIVIZES TRUTHFULNESS EVEN UNDER INCOMPLETE INFORMATION (FROM A BAYESIAN POINT OF VIEW)

Proper scoring rules are mathematically guaranteed to incentivize reporting the true distribution, but their application to uncertainty quantification requires care in interpreting what kind of uncertainty they incentivize. Buchweitz et al. (2025) show that some proper scoring rules impose asymmetric penalties for over- versus under-estimation. While this may appear to induce bias, we argue that such asymmetry faithfully captures epistemic uncertainty rather than distorting the estimate of *total* predictive uncertainty.

In our case, the relevant scoring rule is the QuantileLoss. For a given quantile, e.g., $q_{0.975}$, the loss penalizes underestimation more heavily than overestimation. This asymmetry is not a flaw; rather, it

incentivizes appropriately wider predictive intervals in the presence of epistemic uncertainty. Specifically, this asymmetry encourages intervals to widen more in regions of high epistemic uncertainty than in regions of low epistemic uncertainty.

This phenomenon can be described more quantitatively from a Bayesian perspective. The posterior predictive distribution,

$$\mathbb{P}[Y_{\text{new}} \mid X_{\text{new}}, \mathcal{D}_{\text{train}}, \pi] = \mathbb{E}\left[\mathbb{P}[Y_{\text{new}} \mid X_{\text{new}}, \theta] \mid \mathcal{D}_{\text{train}}, \pi\right]$$

optimally reflects both aleatoric uncertainty $\mathbb{P}[Y_{\text{new}} \mid X_{\text{new}}, \theta]$ (e.g., noise) and epistemic uncertainty $\mathbb{P}[\theta \mid \mathcal{D}_{\text{train}}, \pi]$ (e.g., over parameters) given a prior $\pi$.[4] Marginalizing over the posterior naturally yields a wider distribution than relying on a single point estimate, with a greater widening effect in regions of high epistemic uncertainty. Let $q_{\tau, P}$ denote the $\tau$-quantile of a distribution $P$. The posterior predictive quantiles $(q_{\alpha/2, \mathbb{P}[Y_{\text{new}}|X_{\text{new}}, \mathcal{D}_{\text{train}}, \pi]}, q_{1-\alpha/2, \mathbb{P}[Y_{\text{new}}|X_{\text{new}}, \mathcal{D}_{\text{train}}, \pi]})$ minimize the expected quantile loss:

$$\mathbb{E}\left[\text{QuantileLoss}\big(\{(X_{\text{new}}, Y_{\text{new}})\}, [l, u]\big) \mid \mathcal{D}_{\text{train}}, \pi\right].$$

Since the posterior predictive distribution includes epistemic uncertainty, its quantiles also appropriately include this component. This stands in contrast to other estimators, such as $q_{1-\alpha/2, \mathbb{P}[Y_{\text{new}}|X_{\text{new}}, \hat{\theta}]}$, $\mathbb{E}\left[q_{1-\alpha/2, \mathbb{P}[Y_{\text{new}}|X_{\text{new}}, \theta]} \mid \mathcal{D}_{\text{train}}, \pi\right]$, or $\text{Median}\left[q_{1-\alpha/2, \mathbb{P}[Y_{\text{new}}|X_{\text{new}}, \theta]} \mid \mathcal{D}_{\text{train}}, \pi\right]$ (similar to how we estimate the pure aleatoric uncertainty in Section 2.3), which ignore epistemic uncertainty.

*Example* B.7. Let $Y|X = x \sim \mathcal{N}(\theta(x), \sigma = 1)$ with an unknown mean $\theta(x) \in \mathbb{R}$ and a known standard deviation of 1. Further, assume that the posterior distribution of $\theta(x)$ is $\mathcal{N}(0, \sigma = 10)$ due to large epistemic uncertainty. Then, the predictive posterior distribution of $Y|X = x$ is $\mathcal{N}\left(0, \sigma = \sqrt{10^2 + 1}\right)$, which includes both aleatoric and epistemic uncertainty.

- $q_{0.975, \mathbb{P}[Y_{\text{new}}|X_{\text{new}}, \mathcal{D}_{\text{train}}, \pi]}) = q_{0.975, \mathcal{N}\left(0, \sigma = \sqrt{10^2 + 1}\right)} \approx 1.96 \cdot \sqrt{10^2 + 1}$ appropriately takes epistemic uncertainty into account.

- $q_{0.975, \mathbb{P}[Y_{\text{new}}|X_{\text{new}}, \hat{\theta}]} = q_{0.975, \mathcal{N}\left(\hat{\theta}(x), \sigma = 1\right)} \approx \hat{\theta}(x) + 1.96$ ignores the large epistemic uncertainty (with an interval width of $2 \cdot 1.96$ as $q_{0.025, \mathbb{P}[Y_{\text{new}}|X_{\text{new}}, \hat{\theta}]} \approx \hat{\theta}(x) - 1.96$).

- $\mathbb{E}\left[q_{0.975, \mathbb{P}[Y_{\text{new}}|X_{\text{new}}, \theta]} \mid \mathcal{D}_{\text{train}}, \pi\right] = \mathbb{E}\left[q_{0.975, \mathcal{N}(\theta(x), \sigma = 1)} \mid \mathcal{D}_{\text{train}}, \pi\right] \approx \mathbb{E}\left[\theta(x) + 1.96 \mid \mathcal{D}_{\text{train}}, \pi\right] = 1.96$ ignores the large epistemic uncertainty.

- $\text{Median}\left[q_{0.975, \mathbb{P}[Y_{\text{new}}|X_{\text{new}}, \theta]} \mid \mathcal{D}_{\text{train}}, \pi\right] = \text{Median}\left[q_{0.975, \mathcal{N}(\theta(x), \sigma = 1)} \mid \mathcal{D}_{\text{train}}, \pi\right] \approx \text{Median}\left[\theta(x) + 1.96 \mid \mathcal{D}_{\text{train}}, \pi\right] = 1.96$ ignores the large epistemic uncertainty.

$\diamond$

We fully agree with Buchweitz et al. (2025), that minimizing the QuantileLoss leads to a biased estimator of *aleatoric* uncertainty alone (as in Example B.7, the aleatoric uncertainty corresponds to an interval width of $2 \cdot 1.96$). However, minimizing the QuantileLoss yields a principled estimator for *total predictive uncertainty*, as it is optimal from a Bayesian point of view, taking epistemic uncertainty into account. An unbiased estimator for total predictive uncertainty must also incorporate the epistemic uncertainty and is therefore naturally biased for estimating aleatoric uncertainty.

Thus, the asymmetry of the QuantileLoss is precisely what makes it appropriate for evaluating models that estimate total predictive uncertainty. This justifies its use both in Step 4 of our Algorithm 1 and as an evaluation metric on the unseen test dataset $\mathcal{D}_{\text{test}}$.

### B.3.4 The interval score loss is equivalent to the quantile loss up to a multiplicative factor.

Our definition of the quantile loss

$$\text{QuantileLoss}(\mathcal{D}, (l, u)) := \frac{1}{|\mathcal{D}|} \sum_{(x,y) \in \mathcal{D}} \left[QL_{\alpha/2}\big(y, l(x)\big) + QL_{1-\alpha/2}\big(y, u(x)\big)\right]/2$$

---

[4] Mathematicians (with a background in measure-theory) should read "$\mathbb{P}[Y \mid X]$" as "$\mathbb{P}[Y \in (\cdot) \mid X]$".

only differs from the average interval score loss (Dunsmore, 1968; Winkler, 1972; Gneiting & Raftery, 2007; Brehmer & Gneiting, 2021)

$$\mathrm{AISL}(\mathcal{D}, (l, u)) := \frac{1}{|\mathcal{D}|} \sum_{(x,y)\in\mathcal{D}} \left[ (u(x) - l(x)) + \begin{cases} \frac{2}{\alpha}(l(x) - y) & \text{, if } y < l(x) \\ \frac{2}{\alpha}(y - u(x)) & \text{, if } y > u(x) \\ 0 & \text{, else} \end{cases} \right],$$

by the constant factor of $\alpha/4$, i.e., $\mathrm{QuantileLoss}(\mathcal{D}, (l, u)) = \frac{\alpha}{4}\mathrm{AISL}(\mathcal{D}, (l, u))$.

Traditionally the terms "quantile loss", "pinball loss", "asymmetric piecewise linear loss", "linlin loss", "hinge loss", "tick loss", or "newsvendor loss" refer to the loss $QL_\tau(y, q) = (y - q)\big(\tau - \mathbb{1}_{(-\infty, q]}(y)\big)$ that only evaluates a single approximate quantile $q$ (Koenker & Bassett, 1978; Gneiting, 2011), while the interval score always evaluates two approximate quantiles $(l, u)$ simultaneously (Dunsmore, 1968; Winkler, 1972; Gneiting & Raftery, 2007; Brehmer & Gneiting, 2021). However, within this work, we also denote the average $\mathrm{QuantileLoss}(\mathcal{D}, (l, u))$ as quantile loss. This shouldn't create any confusion, as we never evaluate single quantiles within this work.

## B.4 Intuitive theoretical motivation of CLEAR

PCS achieves empirically good results and nicely captures common sense, with limited theoretical guarantees, while CQR satisfies theoretical guarantees, but misses crucial common sense. In our work, we combine practical experience, theoretical insights, common sense, and empirical results, always with the goal in mind to obtain a useful, veridical, reliable, stable, high-performing method for practical real-world applications.

Note that the original version of PCS did not have any theoretical guarantees for uncertainty quantification Yu & Barter (2024), and Agarwal et al. (2025) mentioned a modified version of PCS-UQ that satisfies conformal marginal coverage guarantees. However, no version of PCS offered a theoretical guarantee for asymptotic input-conditional coverage.

Via CLEAR, we are the first to propose an extension of PCS-UQ that provably satisfies asymptotic input-conditional coverage guarantees (see Lemma 2.1). All previous versions of PCS did not only lack a proof for input-conditional coverage guarantees, but actually do not satisfy input-conditional coverage guarantees. Pure PCS-UQ has the systematic bias of under-coverage in regions with many observations and large noise. This was a central motivation for CLEAR. While this theoretical bias is quite obvious from a theoretical common sense point of view, we conducted our large-scale experimental evaluation to see that this bias can actually have a significant impact on the performance, which can be substantially mitigated by CLEAR.

On the other hand, CQR (or QR) has major problems based on common sense. If one looks closely into the proof of the asymptotic input-conditional coverage of CQR (or QR), one can see that the main idea of the proof is that, asymptotically, you will have infinitely many training data points in close proximity to every point $x$, resulting in accurate quantile predictions. However, for finitely many data points, there will always be regions in the input space with too few data points. In these regions (C)QR cannot accurately estimate the quantiles. However, (C)QR will sometimes output very narrow intervals in these regions, which are fundamentally unreliable and will undercover substantially in these regions. We can see this phenomenon across all different synthetic data-generating processes that we considered (see Sections 3.1 and 4.1 and Appendix C). When we sample $X \sim \mathcal{N}(0, I_d)$ standard Gaussian, we can see that even for $n = 5000$ training data points, we still heavily under-cover in low-density regions, i.e., $\mathbb{P}\left[Y \in C_{\mathrm{CQR}} \mid \|X\|_2 > 4\right] \ll 1 - \alpha$. Based on common sense, it is strongly expected that for every sample size $n$, there exists a radius $r(n)$ beyond which $\mathbb{P}\left[Y \in C_{\mathrm{CQR}} \mid \|X\|_2 > r(n)\right] \ll 1 - \alpha$, since the intervals of (C)QR do not have any reason to become wider out-of-sample while their predictions become more unreliable the farther you move away from the training data.

**Hypothesis B.8.** *We hypothesize that under some fairly general and mild assumptions, for every "non-trivial" data-generating process,*

$$\exists c > 0 : \forall n \in \mathbb{N} : \exists R(n) \in (0, \infty) : \forall r > R(n)\mathbb{P}\left[Y \in C_{CQR} \mid \|X\|_2 > r\right] < 1 - \alpha - c$$

*holds. We think that this claim strongly agrees with basic intuition, and all our empirical results strongly support this hypothesis. However, we leave the proof for future work. The main technical challenge is probably to define "non-trivial" appropriately.*

This does not contradict *asymptotic* input-conditional coverage guarantees as $\lim_{n\to\infty} r(n) = \infty$. However, for any finite sample size $n$, this is a serious problem both from a common-sense perspective and from what you can see from our empirical results. In other words, in regions where the ensemble members heavily disagree on the predictions (usually in regions with few training data), (C)QR would be substantially over-confident. CQR tries to compensate for this overconfidence in the calibration step, which results in CQR slightly over-covering in regions with many training data points while still considerably under-covering in regions with few training data points (see Figures 1, 2 and 4 to 6).

This is the reason why PCS-UQ and CQR form a particularly effective symbiosis. In regions with many training data points, the ensemble predictions coming from different bootstraps of the training data will be close to each other. PCS-UQ has a systematic bias to under-cover in regions with many training data points and to over-cover in regions with few training data points, whereas for CQR, it is exactly the other way around. If you apply any monotonic transformation of the uncertainty of PCS (multiplicative (conformal) calibration, additive (conformal) calibration, or the calibration methods suggested by Kuleshov et al. (2018); Song et al. (2019); Kuleshov & Deshpande (2022); Levi et al. (2022)), you will always have the systematic bias that any monotonically calibrated version of PCS-UQ will under-cover in regions with many training data points and will over-cover in regions with few training data points, whereas it is exactly the other way around for any monotonically calibrated version of CQR. By combining the two of them, we can mitigate this bias (see Figures 1, 2 and 4 to 6). This aligns very well with classical statistical results, telling us that for predictive intervals, one needs to combine the estimated noise structure with epistemic uncertainty on the parameters (confidence intervals).

## B.5 CLEAR ALGORITHM: DETAILS

---

**Algorithm 2** Fast Conformal Implementation of Step 3 from Algorithm 1

---

1: **Input:** Data $(X_i, Y_i)$ for $i = 1, \ldots, n$, split into training $\mathcal{D}_{\text{train}}$, calibration $\mathcal{D}_{\text{cal}}$, and validation $\mathcal{D}_{\text{val}}$ (we consider $\mathcal{D}_{\text{cal}} = \mathcal{D}_{\text{val}}$); grid of $\lambda$ values $\Lambda$; significance level $\alpha$.

2: **Step 3: Define prediction intervals for each $\lambda \in \Lambda$.**
First, define a preliminary (non-calibrated) interval:

$$\tilde{C}_\lambda = \left[ \hat{f} - \hat{q}^{\text{ale}}_{\alpha/2} - \lambda \hat{q}^{\text{epi}}_{\alpha/2}, \ \hat{f} + \hat{q}^{\text{ale}}_{1-\alpha/2} + \lambda \hat{q}^{\text{epi}}_{1-\alpha/2} \right]$$

Then, compute conformity scores $S_i^\lambda = \max\left\{ \frac{\hat{f}(X_i)-Y_i}{\hat{f}(X_i)-\tilde{l}_\lambda(X_i)}, \frac{Y_i-\hat{f}(X_i)}{\tilde{u}_\lambda(X_i)-\hat{f}(X_i)} \right\}$, where $\tilde{l}_\lambda(x), \tilde{u}_\lambda(x)$ are the lower and upper bounds of $\tilde{C}_\lambda(x)$. Let $\gamma_1$ be the $\lceil (1-\alpha)(|\mathcal{D}_{\text{cal}}| + 1) \rceil$-th smallest score among $\{S_i^\lambda\}$ (if $\lceil (1-\alpha)(|\mathcal{D}_{\text{cal}}| + 1) \rceil > |\mathcal{D}_{\text{cal}}|$, take the largest score). Define the calibrated interval:

$$C_\lambda = \left[ \hat{f} - \gamma_1 \hat{q}^{\text{ale}}_{\alpha/2} - \lambda\gamma_1 \hat{q}^{\text{epi}}_{\alpha/2}, \ \hat{f} + \gamma_1 \hat{q}^{\text{ale}}_{1-\alpha/2} + \lambda\gamma_1 \hat{q}^{\text{epi}}_{1-\alpha/2} \right]$$

3: **Output:** calibrated prediction intervals $C_\lambda$ for each $\lambda$ from the grid $\Lambda$.

---

## B.6 ASYMPTOTIC OPTIMALITY OF JOINT CALIBRATION RELATIVE TO SINGLE-PARAMETER BASELINES

In this subsection, we formalize the intuition that CLEAR's joint calibration spans a strictly richer solution space than single-parameter baselines. We analyze the methods at the population level, which corresponds to the limit of infinitely large calibration and validation datasets.

Let $(X, Y) \sim P$ be the true data distribution. For any non-negative calibration parameters $\gamma_1, \gamma_2 \geq 0$, define the prediction interval at input $x$ as:

$$C_{\gamma_1,\gamma_2}(x) = \left[ \hat{f}(x) - \gamma_1 \hat{q}^{\text{ale}}_{\alpha/2}(x) - \gamma_2 \hat{q}^{\text{epi}}_{\alpha/2}(x), \ \hat{f}(x) + \gamma_1 \hat{q}^{\text{ale}}_{1-\alpha/2}(x) + \gamma_2 \hat{q}^{\text{epi}}_{1-\alpha/2}(x) \right] \quad (6)$$

Let $\text{Cov}(\gamma_1, \gamma_2) = \mathbb{P}(Y \in C_{\gamma_1,\gamma_2}(X))$ be the population marginal coverage. We assume $\text{Cov}(\gamma_1, \gamma_2)$ is continuous and strictly increasing in both arguments until it saturates at 1.

Let $\Lambda \subset [0, \infty)$ be the search space of ratios $\lambda = \gamma_2/\gamma_1$ evaluated by CLEAR. For any $\lambda \in \Lambda$, let $\gamma_1(\lambda) = \inf\{\gamma > 0 : \text{Cov}(\gamma, \lambda\gamma) \geq 1 - \alpha\}$ be the minimal population scaling factor required to achieve $1 - \alpha$ marginal coverage. Let $L(\lambda) = \mathbb{E}[\text{QuantileLoss}(C_{\gamma_1(\lambda), \lambda\gamma_1(\lambda)}(X, Y))]$ denote the population expected quantile loss of the resulting interval.

We define the population-level intervals produced by the three methods as follows:

1. **CLEAR** ($C_{\textbf{CLEAR}}$): Selects $\lambda^* = \arg\min_{\lambda \in \Lambda} L(\lambda)$ and outputs $C_{\gamma_1(\lambda^*), \lambda^*\gamma_1(\lambda^*)}$.

2. **Fixed-$\lambda$ Baseline** ($C_{\textbf{base-}\lambda}$): Fixes a hyperparameter $\lambda_0 \in \Lambda$ and outputs $C_{\gamma_1(\lambda_0), \lambda_0\gamma_1(\lambda_0)}$. E.g., $\lambda_0 = 0$ for methods that do not explicitly model epistemic uncertainty (see Appendix A.6), or $\lambda_0 = 1$ for methods that model both but do not rebalance them (see Appendix A.2).

3. **Fixed-$\gamma_1$ Baseline** ($C_{\textbf{base-}\gamma_1}$): Fixes a hyperparameter $\gamma_{1,0} > 0$ (e.g., $\gamma_{1,0} = 1$ as in Appendix A.1) and calibrates $\gamma_2$ to achieve exact coverage, finding $\gamma_{2,\text{base}} = \inf\{\gamma > 0 : \text{Cov}(\gamma_{1,0}, \gamma) \geq 1 - \alpha\}$. It outputs $C_{\gamma_{1,0}, \gamma_{2,\text{base}}}$. We assume its implicit ratio $\lambda' = \frac{\gamma_{2,\text{base}}}{\gamma_{1,0}}$ is contained in $\Lambda$.

**Lemma B.9** (Population Optimality of CLEAR). *Assume that the population marginal coverage function $Cov(\gamma_1, \gamma_2)$ is continuous and strictly increasing in both arguments. Furthermore, assume that the predefined search grid $\Lambda$ contains both the hyperparameter $\lambda_0$ of the Fixed-$\lambda$ baseline, and the implicit optimal ratio $\lambda' = \frac{\gamma_{2,base}}{\gamma_{1,0}}$ of the Fixed-$\gamma_1$ baseline. Then, the expected quantile loss of CLEAR is upper-bounded by the expected quantile loss of both single-parameter baselines:*

$$\mathbb{E}[QuantileLoss(C_{CLEAR})] \leq \mathbb{E}[QuantileLoss(C_{base\text{-}\lambda})] \tag{7}$$

$$\mathbb{E}[QuantileLoss(C_{CLEAR})] \leq \mathbb{E}[QuantileLoss(C_{base\text{-}\gamma_1})] \tag{8}$$

*Proof.* By definition, CLEAR outputs the interval corresponding to $\lambda^*$, which minimizes the population quantile loss $L(\lambda)$ over the set $\Lambda$. Thus, $\mathbb{E}[\text{QuantileLoss}(C_{\text{CLEAR}})] = L(\lambda^*)$.

**Part 1: fixed-$\lambda$ baseline.** The Fixed-$\lambda$ baseline evaluates the interval corresponding to a pre-specified $\lambda_0$. Since $\lambda_0 \in \Lambda$, it trivially holds that the minimum over $\Lambda$ is bounded by the value at $\lambda_0$:

$$L(\lambda^*) = \min_{\lambda \in \Lambda} L(\lambda) \leq L(\lambda_0) = \mathbb{E}[\text{QuantileLoss}(C_{\text{base-}\lambda})] \tag{9}$$

**Part 2: fixed-$\gamma_1$ baseline.** The Fixed-$\gamma_1$ baseline enforces $\gamma_1 = \gamma_{1,0}$ and sets $\gamma_2 = \gamma_{2,\text{base}}$ such that exact marginal coverage is achieved: $\text{Cov}(\gamma_{1,0}, \gamma_{2,\text{base}}) = 1 - \alpha$. Let the implied ratio be $\lambda' = \frac{\gamma_{2,\text{base}}}{\gamma_{1,0}}$.

By definition, evaluating CLEAR's population subroutine at $\lambda'$ requires finding the minimal $\gamma$ such that $\text{Cov}(\gamma, \lambda'\gamma) \geq 1 - \alpha$. Because Cov is strictly increasing, substituting $\gamma = \gamma_{1,0}$ yields exactly the required coverage:

$$\text{Cov}(\gamma_{1,0}, \lambda'\gamma_{1,0}) = \text{Cov}\left(\gamma_{1,0}, \frac{\gamma_{2,\text{base}}}{\gamma_{1,0}}\gamma_{1,0}\right) = \text{Cov}(\gamma_{1,0}, \gamma_{2,\text{base}}) = 1 - \alpha \tag{10}$$

Because $\gamma_{1,0}$ achieves the $1 - \alpha$ coverage constraint exactly, the minimal scaling factor $\gamma_1(\lambda')$ evaluated by CLEAR at $\lambda'$ is exactly $\gamma_{1,0}$.

Since $\lambda' \in \Lambda$ by assumption, it follows that:

$$L(\lambda^*) = \min_{\lambda \in \Lambda} L(\lambda) \leq L(\lambda') = \mathbb{E}[\text{QuantileLoss}(C_{\text{base-}\gamma_1})] \tag{11}$$

This concludes the proof. $\square$

*Remark* B.10 (Finite Sample Convergence). Lemma B.9 establishes that CLEAR is optimal among this class of methods at the population level. In practice, Algorithm 1 operates on finite calibration and validation sets $\mathcal{D}_{\text{cal}}$ and $\mathcal{D}_{\text{val}}$. Because the quantile loss is Lipschitz continuous and the bounds of the intervals are linear combinations of fixed heuristic functions, the interval class has bounded Rademacher complexity. Standard results in statistical learning theory guarantee that the empirical loss $\hat{L}_{|\mathcal{D}_{\text{val}}|}(\lambda)$ converges uniformly to the population loss $L(\lambda)$ almost surely as $|\mathcal{D}_{\text{cal}}|, |\mathcal{D}_{\text{val}}| \to \infty$. Continuous mapping theorems for M-estimators therefore ensure that the limits of the empirical minimum and empirical minimizer converge to their true population counterparts, translating Lemma B.9's guarantees to the asymptotic limit of the finite-sample algorithm. $\diamond$

*Remark* B.11 (Reuse of the Validation Dataset). In finite-sample regimes, reusing the same dataset $\mathcal{D}_{\text{cal}} = \mathcal{D}_{\text{val}}$ can lead to a violation of the theoretical marginal coverage guarantees, see Appendix B.1. However, in the limit of an infinitely large dataset ($|\mathcal{D}_{\text{cal}}| = |\mathcal{D}_{\text{val}}| \to \infty$), the empirical distribution perfectly converges to the true population distribution $P$. Consequently, empirical coverage and empirical expected quantile loss evaluated on this dataset become exactly identical to their population counterparts $\text{Cov}(\gamma_1, \gamma_2)$ and $L(\lambda)$. Because the finite-sample noise vanishes entirely, "overfitting" is mathematically impossible in this asymptotic regime. Thus, whether CLEAR uses independent infinite datasets for validation and calibration, or reuses the exact same infinite dataset ($\mathcal{D}_{\text{cal}} = \mathcal{D}_{\text{val}}$), the resulting parameters $(\gamma_1, \lambda)$ converge to the identical optimal population parameters. ⋄

## C  SIMULATIONS: DETAILS

For each simulation run, the coefficients $\beta_1, \ldots, \beta_d$ are drawn independently from a Gaussian distribution $\mathcal{N}(1, 0.5^2)$. The mean function $\mu(X)$ is then defined as $\mu(X) = 5.0 + \sum_{i=1}^{d} (-1)^{i+1} \beta_i |X_i|^{e_i}$, where the exponent $e_i = 1.5$ if $i$ is odd, and $e_i = 1.25$ if $i$ is even.

### C.1  HETEROSKEDASTIC CASE

In Section 3.1, we focused on the univariate homoskedastic setting ($d = 1$ and $\sigma(x) = 1$). Here, we briefly report results for heteroskedastic data, which show similar patterns. Figures 4 and 5 illustrate the conditional coverage and interval width of the algorithms under two heteroskedastic noise structures: $\sigma_2(x) = 1 + |x|$ and $\sigma_3(x) = 1 + \frac{1}{1+x^2}$. As mentioned, the results are analogous to the homoskedastic setting in Section 3.1.

### C.2  MULTIVARIATE CASE

#### C.2.1  TEST POINT GENERATION

Following setup in (Bodik & Chavez-Demoulin, 2025), let $r_1 < r_2 < \cdots < r_K$ denote a set of predetermined distances. For each radius $r_k$, we sample points uniformly from the surface of the unit $d$-dimensional sphere as follows. First, we draw a vector $\mathbf{v} \in \mathbb{R}^d$ whose entries are independent standard normal random variables. We then normalize $\mathbf{v}$ to obtain a unit vector $\mathbf{u} = \mathbf{v}/\|\mathbf{v}\|_2$. Finally, we scale $\mathbf{u}$ by $r_k$ to obtain the test point $\mathbf{x} = r_k \cdot \mathbf{u}$. In the one-dimensional case, this procedure reduces to selecting $x = r_k$ or $x = -r_k$ with equal probability. This mechanism ensures that, for each $r_k$, the generated points are uniformly distributed on the surface of the sphere of radius $r_k$, thereby allowing a precise evaluation of prediction intervals as a function of distance from the origin. Finally, we generate $Y$ with the same data-generating mechanism as in the train set.

#### C.2.2  RESULTS

Figure 6 illustrates the conditional coverage and interval width of the evaluated algorithms in the multivariate setting, where $X$ is drawn from an independent multivariate Gaussian distribution and $Y = \mu(X) + \varepsilon$, with $\varepsilon \sim \mathcal{N}(0, 1)$. The regression function $\mu(X)$, as previously defined, involves a sum of randomly weighted power transformations of the absolute values of the input features.

Consistent with the univariate homoskedastic results in Section 3.1, Figure 6 shows that both CQR and naive conformal prediction achieve reliable coverage in high-density regions but tend to under-cover in low-density or extrapolation areas. In contrast, CLEAR maintains valid conditional coverage across the entire input space by appropriately adjusting interval widths.

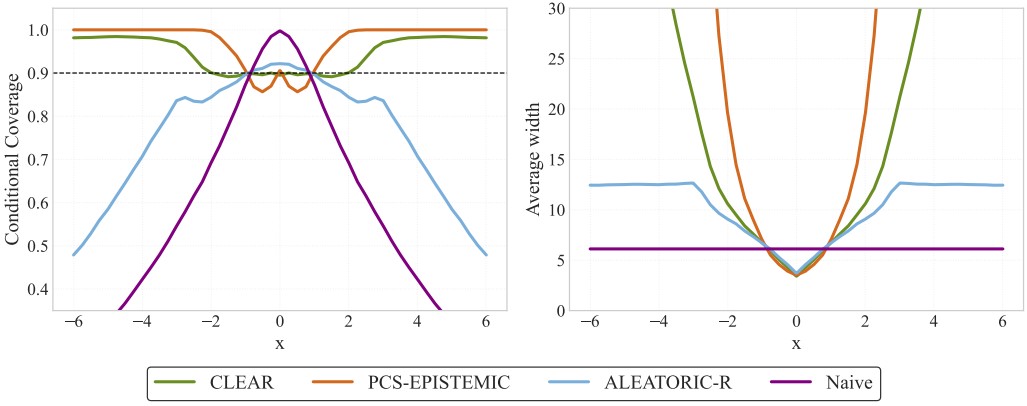

Figure 4: Univariate conditional coverage and average width of the prediction intervals for a heteroskedastic case where $\sigma_2(x) = 1 + |x|$.

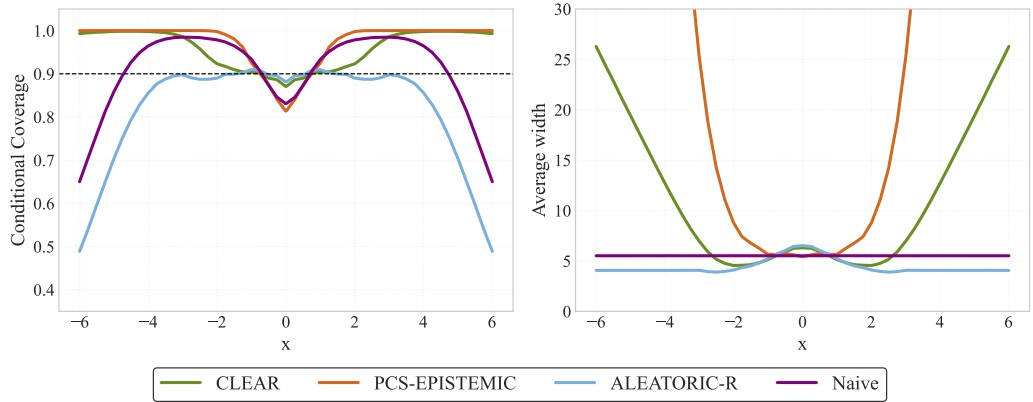

Figure 5: Univariate conditional coverage and average width of the prediction intervals for a heteroskedastic case where $\sigma_3(x) = 1 + \frac{1}{1+x^2}$.

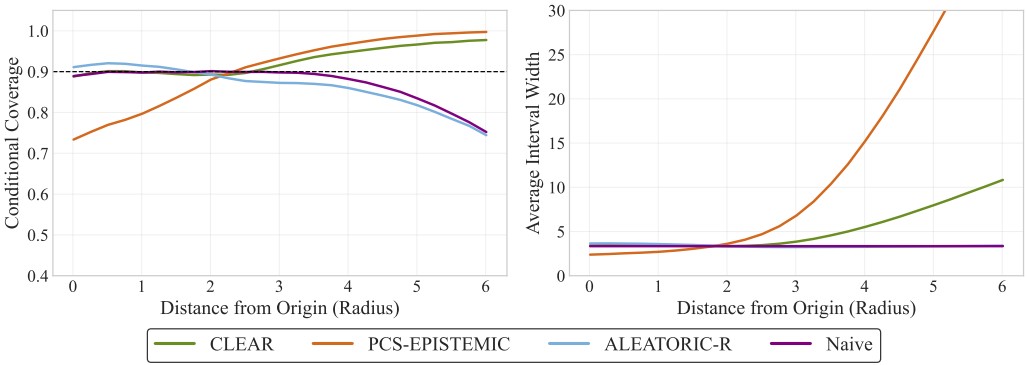

Figure 6: Multivariate conditional coverage and average width (distance from the origin is shown) of the prediction intervals for a homoskedastic case where $d \in \{1, 2, 3, 20\}$ was randomly selected.

# D  EXPERIMENTAL SETUP: DETAILS

## D.1  REAL-WORLD DATA

To evaluate our method in real-world scenarios, we apply it to 17 publicly available regression datasets curated by Agarwal et al. (2025) that includes various domains such as housing, energy, materials science, and healthcare Kelley Pace & Barry (1997); Mayer et al. (2022); Tüfekci (2014); Tsanas & Xifara (2012); Hamidieh (2018); Camacho & Torgo (2014); Tsanas et al. (2009); Coraddu et al. (2016); Yeh (1998); Cassotti et al. (2015); Brooks et al. (1989); PyCaret (n.d.); Torgo (2014); Computer Activity; Allstate Claims Severity; Grinsztajn et al. (2022); Ghahramani (1996); kin8nm dataset; sulfur dataset. Description of the datasets can be found in Table 13.

## D.2  BASELINES FOR VARIANTS (A), (B), (C)

The following baselines provide further context and comparisons, omitted from the main paper for brevity. All compared methods (except UACQR) use the same median $\hat{f}(x) = \hat{f}_{\text{PCS}}(x)$ obtained from PCS. The results from Appendix F.1 use PCS variant (a), Appendix F.2 uses PCS variant (b), and Appendix F.3 uses PCS variant (c). This allows us to isolate the effect of different UQ methods.

1. **Conformalized PCS median (Naive):** For the PCS ensemble, we compute the median prediction $\hat{f}(x) = \hat{f}_{\text{PCS}}(x)$ for each $x$. On the calibration set, absolute residuals $a_i = |y_i - \hat{f}(x_i)|$ are computed. The $(1 - \alpha)$-th quantile $\gamma_{\text{naive}}$ of these absolute residuals is then used to define the interval:

$$C_{\text{Naive}}(x) = [\hat{f}(x) - \gamma_{\text{naive}}, \ \hat{f}(x) + \gamma_{\text{naive}}]$$

   This method applies a constant, symmetric width adjustment to the point predictions. The result of this baseline has been presented for the simulations. We omit it from the real-world data for brevity. However, all the baselines can be found in the supplementary code.

2. **CLEAR with fixed $\lambda = 1$:** This is a variant of the main CLEAR methodology where the ratio $\lambda = \gamma_2/\gamma_1$ is fixed to 1. This implies $\gamma_1 = \gamma_2$. The prediction interval, based on Equation (3), becomes:

$$C_{\lambda=1}(x) = \left[ \hat{f}(x) - \gamma_1 \left( \hat{q}^{\text{ale}}_{\alpha/2}(x) + \hat{q}^{\text{epi}}_{\alpha/2}(x) \right), \quad \hat{f}(x) + \gamma_1 \left( \hat{q}^{\text{ale}}_{1-\alpha/2}(x) + \hat{q}^{\text{epi}}_{1-\alpha/2}(x) \right) \right].$$

   In this configuration, the single parameter $\gamma_1$ is calibrated using the standard split conformal procedure on the validation set. It effectively learns a single scaling factor for the sum of the pre-calibrated aleatoric and epistemic uncertainty widths, without adjusting their ratio.

3. **CLEAR with fixed $\gamma_1 = 1$:** Another variant of CLEAR where $\gamma_1$ is fixed to 1. With $\gamma_1 = 1$, then $\gamma_2 = \lambda$ and the prediction interval from Equation (3) reads as:

$$C_{\gamma_1=1}(x) = \left[ \hat{f}(x) - \hat{q}^{\text{ale}}_{\alpha/2}(x) - \lambda \hat{q}^{\text{epi}}_{\alpha/2}(x), \quad \hat{f}(x) + \hat{q}^{\text{ale}}_{1-\alpha/2}(x) + \lambda \hat{q}^{\text{epi}}_{1-\alpha/2}(x) \right].$$

   Here, $\lambda$ (or equivalently $\gamma_2$) is the parameter calibrated on the calibration set through a coverage-based adjustment. This approach fixes the contribution of the (pre-calibrated) aleatoric uncertainty component and adaptively scales the epistemic uncertainty component.

4. **UACQR-S and UACQR-P** For variant (c), instead of computing $C_{\gamma_1=1}(x)$ and $C_{\lambda=1}(x)$ with CLEAR, we directly use the implementation from Rossellini et al. (2024) to assess our performance against this alternative. Since this is only relevant for variant (c), we use the exact same configuration as the aleatoric part for QRF (see Appendix D.4 and Table 6).

## D.3  DEEP ENSEMBLES AND SIMULTANEOUS QUANTILE REGRESSION IMPLEMENTATION

To demonstrate CLEAR's versatility beyond tree-based methods, we implement state-of-the-art deep learning approaches for uncertainty quantification.

### D.3.1 Deep ensembles for epistemic uncertainty

Our deep ensemble implementation follows Lakshminarayanan et al. (2017) with several enhancements for improved diversity and calibration. Each ensemble consists of $M = 5$ neural networks (adaptive based on dataset size), with the following specifications:

- **Architecture**: Each network employs a fully-connected architecture with hidden layers of sizes (256, 128), ReLU activations, batch normalization, and dropout (rate=0.1). Skip connections are incorporated to stabilize training.
- **Diversity strategies**: To promote ensemble diversity, we employ: (i) different random initializations, (ii) bootstrap sampling where each member trains on different subsamples, (iii) varied learning rates, (iv) different learning rate schedules, and (v) small input noise augmentation.
- **Training**: Each network is trained for up to 1500 epochs using Adam optimizer with learning rate $10^{-3}$, early stopping (patience=50), weight decay ($10^{-5}$), and batch size 64.
- **Calibration**: Following training, we apply a multiplicative calibration factor computed on the validation set using a grid search over c-values to ensure the ensemble achieves target coverage.

The epistemic uncertainty is quantified through the empirical quantiles of the ensemble predictions: $\hat{q}^{\text{epi}}_{\alpha/2}(x) = \hat{f}(x) - Q_{\alpha/2}(\{\hat{f}_m(x)\}_{m=1}^M)$ and $\hat{q}^{\text{epi}}_{1-\alpha/2}(x) = Q_{1-\alpha/2}(\{\hat{f}_m(x)\}_{m=1}^M) - \hat{f}(x)$, where $\hat{f}(x)$ is the ensemble median and $Q_\tau$ denotes the empirical quantile function.

### D.3.2 Simultaneous quantile regression for aleatoric uncertainty

For aleatoric uncertainty, we implement simultaneous quantile regression (SQR) following Tagasovska & Lopez-Paz (2019), which directly models multiple conditional quantiles through a single neural network with specialized architecture:

- **Architecture**: The network uses hidden layers of sizes (256, 256, 128) with LeakyReLU activations (negative slope=0.01), layer normalization, and dropout (rate=0.2). The output layer produces three values corresponding to the $\alpha/2$, 0.5, and $1 - \alpha/2$ quantiles.
- **Loss function**: We minimize a combined loss consisting of: (i) the pinball loss for each quantile, and (ii) a crossing penalty term that encourages monotonicity of quantiles: $\mathcal{L}_{\text{crossing}} = \text{ReLU}(q_{\alpha/2} - q_{0.5}) + \text{ReLU}(q_{0.5} - q_{1-\alpha/2})$.
- **Training**: The model is trained for up to 3000 epochs using Adam optimizer with learning rate $5 \times 10^{-4}$, cosine annealing schedule, and early stopping (patience=200). Gradient clipping (max norm=1.0) prevents training instabilities.
- **Ensemble averaging**: To improve stability, we train an ensemble of SQR models with different random seeds and average their quantile predictions.

The aleatoric uncertainty estimates are computed as $\hat{q}^{\text{ale}}_{\alpha/2}(x) = q_{0.5}(x) - q_{\alpha/2}(x)$ and $\hat{q}^{\text{ale}}_{1-\alpha/2}(x) = q_{1-\alpha/2}(x) - q_{0.5}(x)$, ensuring consistency with the median prediction.

### D.3.3 Integration with CLEAR

The DE and SQR components are integrated into CLEAR using the same calibration procedure as our PCS-UQ (model selection and tree-based methods). The epistemic estimates from DE and aleatoric estimates from SQR are combined using the optimized parameters $\lambda$ and $\gamma_1$ according to Equation (3). The conformal calibration step ensures approximate marginal coverage while the adaptive $\lambda$ selection balances the relative contributions of epistemic and aleatoric uncertainties, accounting for potential scale differences between neural network and tree-based estimators.

### D.4 PCS implementation: details

This section outlines the specific implementation details for generating the PCS ensembles, which provide the point predictor $\hat{f}$ and the raw epistemic uncertainty estimates $\hat{q}^{\text{epi}}$ used as input for the

CLEAR method (Section 2.4), and also form the basis for the standalone calibrated PCS baseline intervals. We explicitly opted for experimenting with three variants of CLEAR to assess the framework's performance and robustness across different modeling choices for epistemic and aleatoric uncertainty components, as discussed in the main text. The core methodology, involving model selection and model perturbations via bootstrapping to capture epistemic uncertainty, follows the principles described in Section 2.

The process begins with data partitioning and bootstrapping. For each dataset and unique random seed, the data is divided into training (60%), validation (20%), and test (20%) sets. Subsequently, $n_{boot} = 100$ bootstrap resamples are drawn from this designated training set to construct the PCS ensemble. Then, for our variants, two distinct pools of base models were developed to generate these PCS ensembles, catering to the different CLEAR variants. Table 6 details the quantile estimators used for CLEAR variant (a). Variant (b) only uses QXGB from Table 6. Table 7 describes the mean estimators utilized for CLEAR variant (c).

Table 6: Base models and key hyperparameters for the quantile models used in CLEAR variants (a). Variant (b) uses only QXGB from this table. All models target the conditional median ($\tau = 0.5$).

| Model | Key Hyperparameters | Ref. |
|---|---|---|
| QRF | 100 trees, min. leaf size: 10 | Meinshausen (2006) |
| QXGB | 100 trees, tree method: histogram, min. child weight: 10 | Chen & Guestrin (2016) |
| Expectile GAM | 10 P-splines (order 3), smoothing parameter optimized via CV (5 min. timeout)* | Servén & Brummitt (2018) |

*Included only if hyperparameter optimization converged successfully, otherwise using the default values.

Table 7: Base models and key hyperparameters for the mean estimators (used in CLEAR variant c). All models target the conditional mean and are from Scikit-learn (Buitinck et al., 2013), except XGBoost, which is from (Chen & Guestrin, 2016). All unspecified hyperparameters use the Scikit-learn defaults.

| Category | Model | Key Hyperparameters |
|---|---|---|
| Linear Models | Ordinary Least Squares | Default |
| | Ridge | Default alphas (CV) |
| | Lasso | 3-fold CV |
| | ElasticNet | 3-fold CV |
| Tree Ensembles | Random Forest | 100 trees, min. leaf size: 5, max. features: 0.33 |
| | Extra Trees | Same hyperparameters as random forest |
| | AdaBoost | Default |
| | XGBoost | Default |
| Neural Network | MLP | Single hidden layer (64 neurons) |

**Ensemble construction and model selection.** For each of the $n_{boot} = 100$ bootstrap samples, all models within the relevant pool (median estimators for variants a/b, mean estimators for variant c) were trained. The single best-performing model type ($k = 1$) was then identified based on the lowest RMSE achieved on the held-out validation set. Consequently, the final PCS ensemble for each random seed comprised 100 instances of this selected top-performing model type. Specifically, CLEAR variant (a) considered all models from Table 6 for this selection process, variant (b) was restricted to selecting always QXGB from this pool, and variant (c) considered all the models from Table 7 instead. All models used default parameters from their respective libraries unless otherwise specified in the tables, and random states were fixed to ensure reproducibility.

**Derivation of $\hat{f}$ and $\hat{q}^{\mathbf{epi}}$ for CLEAR.** The point predictor $\hat{f}(x)$ and the raw epistemic uncertainty contributions $\hat{q}_{\alpha/2}^{\mathrm{epi}}(x)$ and $\hat{q}_{1-\alpha/2}^{\mathrm{epi}}(x)$ supplied to the CLEAR method are derived from this final ensemble of 100 model instances. As detailed in Section 2.2, $\hat{f}(x)$ is the pointwise empirical median of the ensemble's predictions. The epistemic uncertainty terms represent the pointwise distances from this median to the ensemble's empirical $\alpha/2$ and $1 - \alpha/2$ quantiles, respectively.

**Calibration of the standalone PCS baseline.** The standalone PCS baseline method involves a distinct calibration process. From the ensemble's raw pointwise $\alpha/2$ and $1-\alpha/2$ quantile predictions ($\hat{f}_{\alpha/2}(x)$ and $\hat{f}_{1-\alpha/2}(x)$), a single, global multiplicative calibration factor, $\gamma_{\mathrm{PCS}}$, is computed. This $\gamma_{\mathrm{PCS}}$ is the smallest value ensuring that prediction intervals, formed by scaling the raw epistemic uncertainty around $\hat{f}(x)$ (i.e., $[\hat{f}(x) - \gamma_{\mathrm{PCS}}(\hat{f}(x) - \hat{f}_{\alpha/2}(x)), \hat{f}(x) + \gamma_{\mathrm{PCS}}(\hat{f}_{1-\alpha/2}(x) - \hat{f}(x))]$), achieve the target $1 - \alpha$ coverage on the validation set, incorporating the standard finite-sample correction. This $\gamma_{\mathrm{PCS}}$ is then applied to generate the PCS baseline intervals on the test set. It is important to reiterate that this $\gamma_{\mathrm{PCS}}$ is separate and computed independently from $\gamma_1$ and $\lambda$ parameters optimized within the CLEAR framework.

## D.5 METRICS

This appendix provides detailed definitions for the evaluation metrics used in the main paper. We consider test data $(X_i, Y_i)_{i=1}^N$ and prediction intervals $[L_i, U_i]$.

- **Prediction Interval Coverage Probability (PICP)**: Measures the proportion of true values falling within the predicted intervals, calculated as:

$$\mathrm{PICP}(L, U) = \frac{1}{N} \sum_{i=1}^N \mathbb{1}_{[L_i, U_i]}(Y_i)$$

where $\mathbb{1}_{[L_i, U_i]}(Y_i)$ is the indicator function; it equals 1 if $Y_i \in [L_i, U_i]$ and 0 otherwise.

- **Normalized Interval Width (NIW):** Quantifies the average width of prediction intervals normalized by the range of the target variable:

$$\mathrm{NIW}(L, U) = \frac{\frac{1}{N} \sum_{i=1}^N (U_i - L_i)}{\max(Y) - \min(Y)}$$

- **Quantile Loss (also known as pinball loss)**: Evaluates the accuracy of predicted quantiles by penalizing both under- and overestimation. It reflects a trade-off between coverage (PICP) and interval width (NIW), rewarding narrow intervals that still maintain proper coverage and penalizing data points that are far outside the intervals (see Appendix B.3 for more details). For a given quantile level $\tau$, the quantile loss function is:

$$QL_\tau(y, q) = (y - q)\big(\tau - \mathbb{1}_{(-\infty, q]}(y)\big),$$

where $q$ is the predicted $\tau$-quantile. For prediction intervals at level $1 - \alpha$, we evaluate this at both $\tau = \alpha/2$ and $\tau = 1 - \alpha/2$ using

$$\mathrm{QuantileLoss}(L, U) = \sum_{i=1}^N \big[ QL_{\alpha/2}(Y_i, L(X_i)) + QL_{1-\alpha/2}(Y_i, U(X_i)) \big] / 2.$$

- **Average Interval Score Loss (AISL)** (Gneiting & Raftery, 2007): This score balances interval width with coverage penalties (with the same properties as the quantile loss explained in Appendix B.3), defined as

$$\mathrm{AISL}(L, U) = \frac{1}{N} \sum_{i=1}^N \left[ (U_i - L_i) + \frac{2}{\alpha}(L_i - Y_i)\mathbb{1}\{Y_i < L_i\} + \frac{2}{\alpha}(Y_i - U_i)\mathbb{1}\{Y_i > U_i\} \right],$$

where $\mathbb{1}\{\cdot\}$ is the indicator function.

# E CLEAR WITH DE AND SQR: RESULTS ON REAL-WORLD DATA

This section presents the comprehensive experimental results of CLEAR when combined with deep learning-based uncertainty estimators: deep ensembles (DE) for epistemic uncertainty and simultaneous quantile regression (SQR) for aleatoric uncertainty. The experiments further validate CLEAR's generality beyond the PCS and CQR methods that are presented in the body of our paper. The results demonstrate that CLEAR remains effective across different modeling paradigms. We evaluate both the neural baselines individually and their integration through CLEAR, comparing against conformalized versions to assess the added value of our dual-parameter calibration approach. Results are reported across all 17 datasets with 95% nominal coverage, using up to 10 random seeds for robustness.

Table 8: DE-SQR PICP at 95% prediction intervals, aggregated across 10 seeds. CLEAR consists of the deep ensemble (DE) and the simultaneous quantile regressor (SQR). DE-conformal is the conformalized (calibrated) deep ensemble and SQR-conformal is the conformalized simultaneous quantile regressor.

| Dataset | CLEAR | DE | SQR | DE-conformal | SQR-conformal |
|---|---|---|---|---|---|
| ailerons | $0.95 \pm 0.00$ | $0.95 \pm 0.00$ | $0.94 \pm 0.01$ | $0.95 \pm 0.00$ | $0.95 \pm 0.00$ |
| airfoil | $0.96 \pm 0.02$ | $0.95 \pm 0.02$ | $0.98 \pm 0.01$ | $0.95 \pm 0.02$ | $0.96 \pm 0.01$ |
| allstate | $0.95 \pm 0.01$ | $0.95 \pm 0.01$ | $0.91 \pm 0.01$ | $0.95 \pm 0.01$ | $0.94 \pm 0.01$ |
| ca_housing | $0.95 \pm 0.01$ | $0.95 \pm 0.00$ | $0.95 \pm 0.00$ | $0.95 \pm 0.00$ | $0.95 \pm 0.00$ |
| computer | $0.95 \pm 0.01$ | $0.95 \pm 0.01$ | $0.94 \pm 0.01$ | $0.95 \pm 0.01$ | $0.95 \pm 0.01$ |
| concrete | $0.94 \pm 0.02$ | $0.95 \pm 0.02$ | $0.98 \pm 0.01$ | $0.96 \pm 0.02$ | $0.95 \pm 0.03$ |
| elevator | $0.95 \pm 0.00$ | $0.95 \pm 0.00$ | $0.94 \pm 0.01$ | $0.95 \pm 0.00$ | $0.95 \pm 0.00$ |
| energy_efficiency | $0.96 \pm 0.01$ | $0.95 \pm 0.02$ | $0.99 \pm 0.01$ | $0.95 \pm 0.02$ | $0.96 \pm 0.03$ |
| insurance | $0.96 \pm 0.01$ | $0.95 \pm 0.01$ | $0.96 \pm 0.01$ | $0.96 \pm 0.01$ | $0.96 \pm 0.02$ |
| kin8nm | $0.95 \pm 0.01$ | $0.95 \pm 0.01$ | $0.98 \pm 0.00$ | $0.95 \pm 0.01$ | $0.95 \pm 0.01$ |
| miami_housing | $0.95 \pm 0.00$ | $0.95 \pm 0.01$ | $0.95 \pm 0.01$ | $0.95 \pm 0.01$ | $0.95 \pm 0.00$ |
| naval_propulsion | $0.95 \pm 0.01$ | $0.95 \pm 0.01$ | $1.00 \pm 0.00$ | $0.95 \pm 0.01$ | $0.95 \pm 0.01$ |
| parkinsons | $0.95 \pm 0.01$ | $0.95 \pm 0.01$ | $0.96 \pm 0.01$ | $0.95 \pm 0.01$ | $0.95 \pm 0.01$ |
| powerplant | $0.95 \pm 0.01$ | $0.95 \pm 0.00$ | $0.95 \pm 0.00$ | $0.95 \pm 0.00$ | $0.95 \pm 0.00$ |
| qsar | $0.95 \pm 0.01$ | $0.96 \pm 0.01$ | $0.93 \pm 0.01$ | $0.96 \pm 0.01$ | $0.95 \pm 0.01$ |
| sulfur | $0.95 \pm 0.01$ | $0.95 \pm 0.01$ | $0.95 \pm 0.00$ | $0.95 \pm 0.01$ | $0.95 \pm 0.01$ |
| superconductor | $0.95 \pm 0.00$ | $0.95 \pm 0.00$ | $0.96 \pm 0.00$ | $0.95 \pm 0.00$ | $0.95 \pm 0.00$ |

Table 9: DE-SQR NIW at 95% prediction intervals, aggregated across 10 seeds. CLEAR consists of the deep ensemble (DE) and the simultaneous quantile regressor (SQR). DE-conformal is the conformalized (calibrated) deep ensemble and SQR-conformal is the conformalized simultaneous quantile regressor. Values $\geq 100$ or $< 0.01$ are presented in scientific notation with 1 decimal place. **Bold** values are the minimum (best) for that dataset, while underlined values indicate the second-best result.

| Dataset | CLEAR | DE | SQR | DE-conformal | SQR-conformal |
|---|---|---|---|---|---|
| ailerons | $\underline{0.201 \pm 0.017}$ | $0.284 \pm 0.025$ | $\mathbf{0.196 \pm 0.014}$ | $0.285 \pm 0.025$ | $0.203 \pm 0.015$ |
| airfoil | $\mathbf{0.185 \pm 0.023}$ | $\underline{0.223 \pm 0.023}$ | $0.349 \pm 0.025$ | $0.226 \pm 0.023$ | $0.311 \pm 0.025$ |
| allstate | $0.333 \pm 0.061$ | $0.380 \pm 0.073$ | $\mathbf{0.283 \pm 0.043}$ | $0.383 \pm 0.075$ | $\underline{0.324 \pm 0.058}$ |
| ca_housing | $\mathbf{0.357 \pm 9.1e\text{-}03}$ | $0.502 \pm 0.020$ | $0.378 \pm 8.0e\text{-}03$ | $0.502 \pm 0.020$ | $\underline{0.374 \pm 4.0e\text{-}03}$ |
| computer | $\mathbf{0.087 \pm 3.2e\text{-}03}$ | $0.121 \pm 6.3e\text{-}03$ | $\underline{0.104 \pm 8.9e\text{-}03}$ | $0.122 \pm 6.1e\text{-}03$ | $0.106 \pm 5.7e\text{-}03$ |
| concrete | $\mathbf{0.264 \pm 0.033}$ | $\underline{0.365 \pm 0.068}$ | $0.441 \pm 0.041$ | $0.369 \pm 0.067$ | $0.392 \pm 0.069$ |
| elevator | $\underline{0.117 \pm 6.6e\text{-}03}$ | $0.174 \pm 0.011$ | $\mathbf{0.114 \pm 0.011}$ | $0.174 \pm 0.011$ | $0.117 \pm 8.8e\text{-}03$ |
| energy_efficiency | $\mathbf{0.085 \pm 0.012}$ | $\underline{0.115 \pm 0.020}$ | $0.480 \pm 0.059$ | $0.120 \pm 0.020$ | $0.368 \pm 0.036$ |
| insurance | $\mathbf{0.393 \pm 0.044}$ | $0.523 \pm 0.135$ | $\underline{0.397 \pm 0.073}$ | $0.558 \pm 0.141$ | $0.406 \pm 0.085$ |
| kin8nm | $\mathbf{0.197 \pm 8.2e\text{-}03}$ | $0.260 \pm 9.5e\text{-}03$ | $0.246 \pm 8.7e\text{-}03$ | $0.261 \pm 9.6e\text{-}03$ | $\underline{0.208 \pm 8.8e\text{-}03}$ |
| miami_housing | $\mathbf{0.093 \pm 2.9e\text{-}03}$ | $0.131 \pm 5.6e\text{-}03$ | $0.101 \pm 2.8e\text{-}03$ | $0.131 \pm 5.7e\text{-}03$ | $\underline{0.100 \pm 1.8e\text{-}03}$ |
| naval_propulsion | $1.5e\text{-}03 \pm 4.2e\text{-}04$ | $5.4e\text{-}03 \pm 1.1e\text{-}03$ | $3.7e\text{-}03 \pm 1.3e\text{-}04$ | $5.4e\text{-}03 \pm 1.1e\text{-}03$ | $\underline{1.7e\text{-}03 \pm 1.5e\text{-}04}$ |
| parkinsons | $\mathbf{0.378 \pm 0.014}$ | $0.476 \pm 0.027$ | $0.419 \pm 0.021$ | $0.479 \pm 0.027$ | $\underline{0.403 \pm 0.017}$ |
| powerplant | $\mathbf{0.186 \pm 6.7e\text{-}03}$ | $0.266 \pm 0.012$ | $0.204 \pm 6.8e\text{-}03$ | $0.267 \pm 0.013$ | $\underline{0.202 \pm 7.3e\text{-}03}$ |
| qsar | $\underline{0.410 \pm 0.136}$ | $0.508 \pm 0.178$ | $\mathbf{0.402 \pm 0.135}$ | $0.509 \pm 0.179$ | $0.456 \pm 0.153$ |
| sulfur | $\mathbf{0.105 \pm 8.5e\text{-}03}$ | $0.199 \pm 0.015$ | $0.120 \pm 7.4e\text{-}03$ | $0.200 \pm 0.015$ | $\underline{0.120 \pm 8.3e\text{-}03}$ |
| superconductor | $\mathbf{0.208 \pm 0.024}$ | $0.295 \pm 0.038$ | $0.247 \pm 0.027$ | $0.295 \pm 0.038$ | $\underline{0.245 \pm 0.028}$ |

Table 10: DE-SQR Quantile Loss at 95% prediction intervals, aggregated across 10 seeds. CLEAR consists of the deep ensemble (DE) and the simultaneous quantile regressor (SQR). DE-conformal is the conformalized (calibrated) deep ensemble and SQR-conformal is the conformalized simultaneous quantile regressor. Values $\geq 100$ or $< 0.01$ are presented in scientific notation with 1 decimal place. **Bold** values are the minimum (best) for that dataset, while underlined values indicate the second-best result.

| Dataset | CLEAR | DE | SQR | DE-conformal | SQR-conformal |
|---|---|---|---|---|---|
| ailerons | 9.6e-06 $\pm$ 3.1e-07 | 1.3e-05 $\pm$ 3.5e-07 | 9.6e-06 $\pm$ 4.0e-07 | 1.3e-05 $\pm$ 3.5e-07 | **9.5e-06 $\pm$ 3.7e-07** |
| airfoil | **0.097 $\pm$ 6.8e-03** | 0.117 $\pm$ 0.010 | 0.162 $\pm$ 0.017 | 0.117 $\pm$ 0.010 | 0.153 $\pm$ 0.018 |
| allstate | **1.4e+02 $\pm$ 9.329** | 1.5e+02 $\pm$ 8.112 | 1.5e+02 $\pm$ 9.210 | 1.5e+02 $\pm$ 8.106 | 1.4e+02 $\pm$ 8.368 |
| ca_housing | 3.0e+03 $\pm$ 63.719 | 3.9e+03 $\pm$ 1.2e+02 | 3.0e+03 $\pm$ 54.590 | 3.9e+03 $\pm$ 1.2e+02 | **3.0e+03 $\pm$ 53.118** |
| computer | **0.146 $\pm$ 5.5e-03** | 0.183 $\pm$ 6.8e-03 | 0.162 $\pm$ 8.3e-03 | 0.183 $\pm$ 6.8e-03 | 0.162 $\pm$ 7.9e-03 |
| concrete | **0.348 $\pm$ 0.049** | 0.427 $\pm$ 0.056 | 0.467 $\pm$ 0.031 | 0.428 $\pm$ 0.057 | 0.458 $\pm$ 0.051 |
| elevator | 1.2e-04 $\pm$ 1.9e-06 | 1.6e-04 $\pm$ 3.5e-06 | 1.1e-04 $\pm$ 2.5e-06 | 1.6e-04 $\pm$ 3.5e-06 | **1.1e-04 $\pm$ 2.4e-06** |
| energy_efficiency | **0.052 $\pm$ 9.5e-03** | 0.067 $\pm$ 0.012 | 0.221 $\pm$ 0.023 | 0.067 $\pm$ 0.012 | 0.188 $\pm$ 0.020 |
| insurance | 3.9e+02 $\pm$ 42.155 | 5.3e+02 $\pm$ 97.010 | **3.9e+02 $\pm$ 37.867** | 5.4e+02 $\pm$ 93.793 | 3.9e+02 $\pm$ 45.372 |
| kin8nm | **4.1e-03 $\pm$ 4.4e-05** | 5.2e-03 $\pm$ 8.2e-05 | 4.5e-03 $\pm$ 5.7e-05 | 5.2e-03 $\pm$ 8.2e-05 | 4.2e-03 $\pm$ 7.1e-05 |
| miami_housing | 4.5e+03 $\pm$ 2.1e+02 | 5.4e+03 $\pm$ 1.7e+02 | 4.4e+03 $\pm$ 1.2e+02 | 5.4e+03 $\pm$ 1.7e+02 | **4.4e+03 $\pm$ 1.2e+02** |
| naval_propulsion | **3.6e-05 $\pm$ 9.4e-06** | 1.3e-04 $\pm$ 2.7e-05 | 8.2e-05 $\pm$ 2.9e-06 | 1.3e-04 $\pm$ 2.7e-05 | 4.4e-05 $\pm$ 4.3e-06 |
| parkinsons | 0.298 $\pm$ 0.014 | 0.343 $\pm$ 9.8e-03 | 0.291 $\pm$ 5.7e-03 | 0.343 $\pm$ 9.8e-03 | **0.289 $\pm$ 5.8e-03** |
| powerplant | **0.227 $\pm$ 0.013** | 0.301 $\pm$ 0.014 | 0.234 $\pm$ 0.012 | 0.301 $\pm$ 0.014 | 0.234 $\pm$ 0.013 |
| qsar | **0.055 $\pm$ 3.2e-03** | 0.064 $\pm$ 3.5e-03 | 0.061 $\pm$ 3.7e-03 | 0.064 $\pm$ 3.5e-03 | 0.060 $\pm$ 3.6e-03 |
| sulfur | **1.7e-03 $\pm$ 1.0e-04** | 2.7e-03 $\pm$ 1.9e-04 | 1.8e-03 $\pm$ 1.3e-04 | 2.7e-03 $\pm$ 1.9e-04 | 1.8e-03 $\pm$ 1.3e-04 |
| superconductor | **0.545 $\pm$ 0.024** | 0.707 $\pm$ 0.024 | 0.560 $\pm$ 0.017 | 0.707 $\pm$ 0.024 | 0.559 $\pm$ 0.018 |

Table 11: DE-SQR NCIW at 95% prediction intervals, aggregated across 10 seeds. CLEAR consists of the deep ensemble (DE) and the simultaneous quantile regressor (SQR). DE-conformal is the conformalized (calibrated) deep ensemble and SQR-conformal is the conformalized simultaneous quantile regressor. Values $\geq 100$ or $< 0.01$ are presented in scientific notation with 1 decimal place. **Bold** values are the minimum (best) for that dataset, while underlined values indicate the second-best result.

| Dataset | CLEAR | DE | SQR | DE-conformal | SQR-conformal |
|---|---|---|---|---|---|
| ailerons | 0.205 $\pm$ 0.017 | 0.292 $\pm$ 0.027 | 0.205 $\pm$ 0.017 | 0.291 $\pm$ 0.026 | **0.204 $\pm$ 0.017** |
| airfoil | **0.173 $\pm$ 0.016** | 0.220 $\pm$ 0.025 | 0.290 $\pm$ 0.026 | 0.217 $\pm$ 0.025 | 0.293 $\pm$ 0.026 |
| allstate | **0.328 $\pm$ 0.048** | 0.384 $\pm$ 0.062 | 0.342 $\pm$ 0.051 | 0.381 $\pm$ 0.060 | 0.330 $\pm$ 0.045 |
| ca_housing | **0.354 $\pm$ 8.1e-03** | 0.501 $\pm$ 0.021 | 0.372 $\pm$ 7.1e-03 | 0.501 $\pm$ 0.021 | 0.373 $\pm$ 8.3e-03 |
| computer | **0.088 $\pm$ 1.7e-03** | 0.120 $\pm$ 5.0e-03 | 0.106 $\pm$ 6.5e-03 | 0.120 $\pm$ 5.1e-03 | 0.106 $\pm$ 7.1e-03 |
| concrete | **0.274 $\pm$ 0.026** | 0.345 $\pm$ 0.053 | 0.371 $\pm$ 0.036 | 0.339 $\pm$ 0.051 | 0.376 $\pm$ 0.040 |
| elevator | 0.117 $\pm$ 7.0e-03 | 0.172 $\pm$ 0.010 | 0.117 $\pm$ 9.0e-03 | 0.172 $\pm$ 0.010 | **0.116 $\pm$ 9.4e-03** |
| energy_efficiency | **0.079 $\pm$ 7.4e-03** | 0.114 $\pm$ 0.017 | 0.351 $\pm$ 0.049 | 0.106 $\pm$ 0.014 | 0.346 $\pm$ 0.046 |
| insurance | 0.364 $\pm$ 0.060 | 0.465 $\pm$ 0.125 | 0.351 $\pm$ 0.064 | 0.460 $\pm$ 0.122 | **0.350 $\pm$ 0.061** |
| kin8nm | **0.194 $\pm$ 6.6e-03** | 0.255 $\pm$ 5.6e-03 | 0.205 $\pm$ 6.3e-03 | 0.254 $\pm$ 5.3e-03 | 0.208 $\pm$ 6.3e-03 |
| miami_housing | **0.092 $\pm$ 3.5e-03** | 0.129 $\pm$ 7.8e-03 | 0.100 $\pm$ 2.6e-03 | 0.129 $\pm$ 7.7e-03 | 0.100 $\pm$ 2.9e-03 |
| naval_propulsion | **1.5e-03 $\pm$ 4.2e-04** | 5.4e-03 $\pm$ 1.2e-03 | 1.7e-03 $\pm$ 2.0e-04 | 5.4e-03 $\pm$ 1.2e-03 | 1.6e-03 $\pm$ 1.7e-04 |
| parkinsons | **0.382 $\pm$ 0.015** | 0.473 $\pm$ 0.025 | 0.398 $\pm$ 0.013 | 0.470 $\pm$ 0.022 | 0.398 $\pm$ 0.013 |
| powerplant | **0.190 $\pm$ 6.0e-03** | 0.272 $\pm$ 9.5e-03 | 0.202 $\pm$ 8.1e-03 | 0.272 $\pm$ 9.5e-03 | 0.202 $\pm$ 8.1e-03 |
| qsar | **0.407 $\pm$ 0.134** | 0.477 $\pm$ 0.150 | 0.450 $\pm$ 0.151 | 0.476 $\pm$ 0.150 | 0.441 $\pm$ 0.146 |
| sulfur | **0.104 $\pm$ 6.2e-03** | 0.198 $\pm$ 0.016 | 0.120 $\pm$ 7.1e-03 | 0.197 $\pm$ 0.015 | 0.120 $\pm$ 7.2e-03 |
| superconductor | **0.209 $\pm$ 0.024** | 0.303 $\pm$ 0.036 | 0.242 $\pm$ 0.027 | 0.302 $\pm$ 0.036 | 0.245 $\pm$ 0.027 |

Table 12: DE-SQR Interval Score Loss at 95% prediction intervals, aggregated across 10 seeds. CLEAR consists of the deep ensemble (DE) and the simultaneous quantile regressor (SQR). DE-conformal is the conformalized (calibrated) deep ensemble and SQR-conformal is the conformalized simultaneous quantile regressor. Values $\geq 100$ or $< 0.01$ are presented in scientific notation with 1 decimal place. **Bold** values are the minimum (best) for that dataset, while underlined values indicate the second-best result.

| Dataset | CLEAR | DE | SQR | DE-conformal | SQR-conformal |
|---|---|---|---|---|---|
| ailerons | 7.7e-04 ± 2.5e-05 | 1.0e-03 ± 2.8e-05 | 7.7e-04 ± 3.2e-05 | 1.0e-03 ± 2.8e-05 | **7.6e-04 ± 2.9e-05** |
| airfoil | **7.761 ± 0.546** | 9.368 ± 0.826 | 12.937 ± 1.386 | 9.370 ± 0.813 | 12.279 ± 1.418 |
| allstate | **1.1e+04 ± 7.5e+02** | 1.2e+04 ± 6.5e+02 | 1.2e+04 ± 7.4e+02 | 1.2e+04 ± 6.5e+02 | 1.2e+04 ± 6.7e+02 |
| ca_housing | 2.4e+05 ± 5.1e+03 | 3.1e+05 ± 9.2e+03 | 2.4e+05 ± 4.4e+03 | 3.1e+05 ± 9.2e+03 | **2.4e+05 ± 4.2e+03** |
| computer | **11.669 ± 0.436** | 14.633 ± 0.543 | 12.938 ± 0.662 | 14.634 ± 0.545 | 12.926 ± 0.634 |
| concrete | **27.850 ± 3.946** | 34.179 ± 4.507 | 37.343 ± 2.464 | 34.203 ± 4.564 | 36.678 ± 4.051 |
| elevator | 9.2e-03 ± 1.5e-04 | 0.013 ± 2.8e-04 | 9.0e-03 ± 2.0e-04 | 0.013 ± 2.8e-04 | **9.0e-03 ± 1.9e-04** |
| energy_efficiency | **4.154 ± 0.758** | 5.369 ± 0.965 | 17.719 ± 1.871 | 5.387 ± 0.974 | 15.020 ± 1.595 |
| insurance | 3.1e+04 ± 3.4e+03 | 4.2e+04 ± 7.8e+03 | **3.1e+04 ± 3.0e+03** | 4.3e+04 ± 7.5e+03 | 3.1e+04 ± 3.6e+03 |
| kin8nm | **0.327 ± 3.5e-03** | 0.417 ± 6.5e-03 | 0.358 ± 4.6e-03 | 0.417 ± 6.6e-03 | 0.340 ± 5.7e-03 |
| miami_housing | 3.6e+05 ± 1.7e+04 | 4.3e+05 ± 1.4e+04 | 3.5e+05 ± 9.5e+03 | 4.3e+05 ± 1.4e+04 | **3.5e+05 ± 9.5e+03** |
| naval_propulsion | **2.9e-03 ± 7.5e-04** | 0.011 ± 2.2e-03 | 6.5e-03 ± 2.3e-04 | 0.011 ± 2.2e-03 | 3.5e-03 ± 3.5e-04 |
| parkinsons | 23.860 ± 1.124 | 27.437 ± 0.787 | 23.297 ± 0.456 | 27.449 ± 0.784 | **23.159 ± 0.461** |
| powerplant | **18.145 ± 1.005** | 24.110 ± 1.141 | 18.714 ± 0.990 | 24.106 ± 1.140 | 18.706 ± 1.002 |
| qsar | **4.419 ± 0.255** | 5.152 ± 0.278 | 4.875 ± 0.294 | 5.154 ± 0.278 | 4.800 ± 0.286 |
| sulfur | **0.134 ± 8.1e-03** | 0.213 ± 0.015 | 0.144 ± 0.010 | 0.213 ± 0.015 | 0.144 ± 0.010 |
| superconductor | **43.610 ± 1.937** | 56.594 ± 1.943 | 44.763 ± 1.388 | 56.591 ± 1.943 | 44.731 ± 1.406 |

## F  CLEAR WITH PCS AND CQR: RESULTS ON REAL-WORLD DATA

This appendix presents the detailed quantitative results from the benchmark experiments conducted on the 17 real-world datasets, complementing the summary findings in the main paper (Section 4.2). As a reminder, we evaluate three distinct configurations (variants) of our proposed CLEAR method alongside the baseline approaches (PCS, ALEATORIC, ALEATORIC-R). The differences among these 3 variants are considered base models. For each variant, CLEAR is applied on the already trained PCS and uncalibrated ALEATORIC-R. To recap these variants:

- **Variant (a):** Employs a quantile PCS approach using a diverse pool of quantile estimators (QRF, QXGB, Expectile GAM[5]) for estimating the conditional medians based on the bootstraps. The top-performing model type ($k = 1$) on the validation set is selected. 1) Based on the selected model type, we compute the epistemic component $\hat{q}^{\text{epi}}$ (via empirical quantiles over the estimated medians from the $b = m$ bootstraps) and the median $\hat{f}$ (as empirical median over the estimated medians). 2) Using the same selected quantile model type, the aleatoric component $\hat{q}^{\text{ale}}$ is derived from a bootstrapped ALEATORIC-R model.

- **Variant (b):** Similar to (a), but restricts the model pool for both PCS and the paired ALEATORIC-R exclusively to QXGB.

- **Variant (c):** Uses a standard PCS approach with mean-based regression models (e.g., Random Forest, XGBoost) for the epistemic component (via empirical quantiles over the estimated medians) and median prediction $\hat{f}$ (as empirical median over the estimated means), selecting the top model ($k = 1$). The aleatoric component is derived from a bootstrapped ALEATORIC-R model using QRF.

The subsequent subsections present the comprehensive results for each variant (a, b, c). All experiments were run across 10 different random seeds, and the results on the test set are presented. The subsequent tables report the average performance metric across these 10 seeds, along with the standard deviation ($\pm\sigma$), to indicate the variability associated with data splitting. For each variant, we provide tables detailing the metrics presented in Appendix D.5 for all methods across all datasets at a 95% nominal coverage level. Lower values are preferable for all metrics except PICP, which should ideally be close to the target of 0.95. Additionally, summary plots (Figures 3, 7 and 8) visualize the relative NCIW and Quantile Loss performance normalized against the respective CLEAR variant baseline. Additionally, only for variant (c), we provide Table 29 with the values of $\lambda$ and $\gamma_1$ to provide some insights into the aleatoric/epistemic allocations of CLEAR. Finally, Figure 9 (Appendix F.4) provides a direct comparison of the NCIW and Quantile Loss between the three CLEAR variants themselves, using variant (a) as the reference baseline.

Appendix F.4 shows that the primary configuration of CLEAR presented in the main text, variant (a), leverages a diverse set of quantile models for robust epistemic uncertainty estimation and generally delivers the most consistent and superior performance. As detailed in Figure 9 in the appendix, CLEAR in variant (a) tends to marginally outperform the model in variant (b), which is restricted to QXGB, and variant (c), which utilizes standard mean-based PCS models. For instance, in variant (a), QXGB was selected in approximately 64% of cases, QRF in 24%, and ExpectileGAM in the remainder of cases. In variant (c), XGBoost was chosen about 70% of the time, with other models (excluding Ridge) sharing the rest. The importance of this dynamic model selection is evident in datasets like `naval_propulsion`, where variants (b) and (c) can struggle due to their fixed model choices (QXGB and QRF for the aleatoric part, respectively), whereas variant (a) can adapt by selecting, for example, ExpectileGAM. This highlights the stability advantage of CLEAR (a) and the benefit of PCS's dynamic model choice, especially when the aleatoric component can also adapt. Nevertheless, the CLEAR framework overall demonstrates considerable robustness even with these alternative model choices. For instance, variant (b) still provides NCIW reductions of approximately 17.5% and 4.9% over its corresponding PCS and ALEATORIC-R baselines, respectively (Figure 7). A similar advantage is observed for variant (c) (Figure 8), which reduces NCIW by about 7.9% against ALEATORIC-R, the best-performing variant of CQR that uses our novel residual-based technique.

---

[5]Strictly speaking, Expectile GAM estimates expectiles rather than quantiles. While it is not a consistent estimator for quantiles in theory, mixing expectiles and quantiles may still be practical in applications.

**Detailed motivation behind variant (a).** Overall, variant (a) achieves the best results (see Figure 9). Therefore, we recommend to use CLEAR variant (a). The experiments in variant (a) are fair, since each of the competing methods uses the same base model, which is selected per data set based on the RMSE on the validation data set. See Appendix F.1 for the results.

**Detailed motivation behind variant (b).** Variant (b) is the simplest to understand, easiest to implement, and computationally cheapest variant and provides maximal fairness, making it particularly scientifically sound. Each method simply uses QXGB as base model without any model selection step. Strictly speaking, variant (b) is the only variant where `ALEATORIC` and `ALEATORIC-R` are fully conformal, since variant (b) uses the calibration data set only for calibration, while variant (a) and (c) reuse the validation set used for model selection as calibration data set. However, in practice, we observe this re-usage does not hurt calibration in any significant way (Table 14 and Table 24). See Appendix F.2 for the results of variant (b).

**Detailed motivation behind variant (c).** Variant (c) uses the same set of base models as suggested by the authors of PCS uncertainty quantification as Agarwal et al. (2025). They conducted extensive experiments showing that PCS uncertainty quantification substantially outperforms popular conformal baselines such as split conformal regression (Lei et al., 2018), Studentized conformal regression (Lei et al., 2018), and Majority Vote (Gasparin & Ramdas, 2024) (by more than 20% on average in terms of interval width). Our experiments showing that CLEAR (a) outperforms CLEAR (c) (Figure 9) and CLEAR (c) outperforms PCS uncertainty quantification (c) (Appendix F.3), together with the experiments by Agarwal et al. (2025), strongly suggests that CLEAR clearly outperforms these popular conformal baselines. See Appendix F.3 for the results of variant (c).

Overall, CLEAR shows the strongest performance across all variants (a), (b) and (c), demonstrating CLEAR's stability across different settings, data sets, and metrics.

Table 13: Dataset statistics where $d$ represents the number of variables, $n$ represents the number of observations, followed by the minimum, maximum, and range values for $y$.

| Dataset | $n$ | $d$ | $y_{min}$ | $y_{max}$ | $y_{range}$ |
|---|---|---|---|---|---|
| ailerons | 13,750 | 40 | -0.0036 | 0.00e+00 | 0.0036 |
| airfoil | 1,503 | 5 | 104.2040 | 140.1580 | 35.9540 |
| allstate | 5,000 | 1037 | 200.0000 | 3.31e+04 | 3.29e+04 |
| ca_housing | 20,640 | 8 | 1.50e+04 | 5.00e+05 | 4.85e+05 |
| computer | 8,192 | 21 | 0.00e+00 | 99.0000 | 99.0000 |
| concrete | 1,030 | 8 | 2.3300 | 81.7500 | 79.4200 |
| elevator | 16,599 | 18 | 0.0120 | 0.0780 | 0.0660 |
| energy_efficiency | 768 | 10 | 6.0100 | 43.1000 | 37.0900 |
| insurance | 1,338 | 8 | 1121.8739 | 6.26e+04 | 6.15e+04 |
| kin8nm | 8,192 | 8 | 0.0632 | 1.4585 | 1.3953 |
| miami_housing | 13,932 | 28 | 7.20e+04 | 2.65e+06 | 2.58e+06 |
| naval_propulsion | 11,934 | 24 | 0.0690 | 1.8320 | 1.7630 |
| parkinsons | 5,875 | 18 | 7.0000 | 54.9920 | 47.9920 |
| powerplant | 9,568 | 4 | 420.2600 | 495.7600 | 75.5000 |
| qsar | 5,742 | 500 | -6.2400 | 11.0000 | 17.2400 |
| sulfur | 10,081 | 5 | 0.00e+00 | 1.0000 | 1.0000 |
| superconductor | 21,263 | 79 | 3.25e-04 | 185.0000 | 184.9997 |

## F.1 VARIANT (A)

Table 14: Variant (a) PICP at 95% prediction intervals, aggregated across 10 seeds.

| Dataset | CLEAR | ALEATORIC | ALEATORIC-R | PCS-EPISTEMIC | Naive | $\gamma_1 = 1$ | $\lambda = 1$ |
|---|---|---|---|---|---|---|---|
| ailerons | 0.95 ± 0.00 | 0.95 ± 0.00 | 0.95 ± 0.00 | 0.95 ± 0.01 | 0.95 ± 0.01 | 0.95 ± 0.00 | 0.95 ± 0.00 |
| airfoil | 0.95 ± 0.01 | 0.95 ± 0.01 | 0.95 ± 0.01 | 0.95 ± 0.01 | 0.95 ± 0.02 | 0.95 ± 0.02 | 0.95 ± 0.01 |
| allstate | 0.95 ± 0.01 | 0.95 ± 0.01 | 0.95 ± 0.01 | 0.95 ± 0.01 | 0.95 ± 0.01 | 0.95 ± 0.01 | 0.95 ± 0.01 |
| ca_housing | 0.95 ± 0.01 | 0.95 ± 0.01 | 0.95 ± 0.01 | 0.95 ± 0.01 | 0.95 ± 0.01 | 0.95 ± 0.01 | 0.95 ± 0.01 |
| computer | 0.95 ± 0.01 | 0.97 ± 0.01 | 0.95 ± 0.01 | 0.95 ± 0.01 | 0.96 ± 0.01 | 0.95 ± 0.01 | 0.96 ± 0.01 |
| concrete | 0.95 ± 0.02 | 0.96 ± 0.02 | 0.95 ± 0.02 | 0.95 ± 0.01 | 0.94 ± 0.03 | 0.95 ± 0.01 | 0.95 ± 0.01 |
| elevator | 0.95 ± 0.00 | 0.95 ± 0.01 | 0.95 ± 0.00 | 0.95 ± 0.00 | 0.95 ± 0.01 | 0.95 ± 0.00 | 0.95 ± 0.00 |
| energy_efficiency | 0.95 ± 0.02 | 0.96 ± 0.02 | 0.95 ± 0.02 | 0.96 ± 0.01 | 0.95 ± 0.02 | 0.96 ± 0.01 | 0.96 ± 0.01 |
| insurance | 0.95 ± 0.02 | 0.96 ± 0.01 | 0.96 ± 0.02 | 0.95 ± 0.01 | 0.95 ± 0.02 | 0.96 ± 0.02 | 0.96 ± 0.01 |
| kin8nm | 0.95 ± 0.01 | 0.95 ± 0.01 | 0.95 ± 0.01 | 0.95 ± 0.01 | 0.95 ± 0.01 | 0.95 ± 0.01 | 0.95 ± 0.01 |
| miami_housing | 0.95 ± 0.01 | 0.95 ± 0.00 | 0.95 ± 0.01 | 0.95 ± 0.01 | 0.95 ± 0.01 | 0.95 ± 0.00 | 0.95 ± 0.00 |
| naval_propulsion | 0.95 ± 0.01 | 0.96 ± 0.01 | 0.95 ± 0.01 | 0.95 ± 0.01 | 0.95 ± 0.01 | 0.95 ± 0.01 | 0.95 ± 0.01 |
| parkinsons | 0.95 ± 0.01 | 0.96 ± 0.01 | 0.95 ± 0.01 | 0.95 ± 0.01 | 0.95 ± 0.01 | 0.95 ± 0.01 | 0.95 ± 0.01 |
| powerplant | 0.95 ± 0.01 | 0.95 ± 0.00 | 0.95 ± 0.01 | 0.95 ± 0.01 | 0.95 ± 0.00 | 0.95 ± 0.01 | 0.95 ± 0.01 |
| qsar | 0.95 ± 0.01 | 0.95 ± 0.01 | 0.95 ± 0.01 | 0.95 ± 0.01 | 0.95 ± 0.00 | 0.95 ± 0.01 | 0.95 ± 0.01 |
| sulfur | 0.95 ± 0.01 | 0.95 ± 0.01 | 0.95 ± 0.01 | 0.95 ± 0.00 | 0.95 ± 0.01 | 0.95 ± 0.01 | 0.95 ± 0.01 |
| superconductor | 0.95 ± 0.00 | 0.95 ± 0.00 | 0.95 ± 0.00 | 0.95 ± 0.00 | 0.95 ± 0.01 | 0.95 ± 0.00 | 0.95 ± 0.00 |

Table 15: Variant (a) NIW at 95% prediction intervals, aggregated across 10 seeds. Values $\geq 100$ or $< 0.01$ are presented in scientific notation with 1 decimal place. **Bold** values (desirable) are the minimum for that dataset and metric, while the underlined values indicate the second-best result. Red values are more than 33% worse than the best result.

| Dataset | CLEAR | ALEATORIC | ALEATORIC-R | PCS-EPISTEMIC | Naive | $\gamma_1 = 1$ | $\lambda = 1$ |
|---|---|---|---|---|---|---|---|
| ailerons | 0.193 ± 0.016 | 0.208 ± 0.017 | 0.195 ± 0.017 | 0.215 ± 0.016 | 0.220 ± 0.021 | _0.193 ± 0.016_ | **0.193 ± 0.016** |
| airfoil | 0.207 ± 0.012 | 0.367 ± 0.018 | 0.215 ± 0.018 | 0.215 ± 0.012 | 0.246 ± 0.024 | _0.205 ± 0.012_ | 0.206 ± 0.012 |
| allstate | 0.260 ± 0.037 | 0.266 ± 0.045 | 0.259 ± 0.045 | 0.259 ± 0.044 | 0.327 ± 0.056 | **0.246 ± 0.042** | _0.247 ± 0.042_ |
| ca_housing | 0.339 ± 0.008 | 0.760 ± 0.006 | 0.350 ± 0.006 | 0.354 ± 0.011 | 0.434 ± 0.020 | _0.333 ± 0.006_ | **0.331 ± 0.007** |
| computer | _0.091 ± 0.008_ | 0.099 ± 0.001 | 0.091 ± 0.001 | 18.269 ± 11.847 | 0.115 ± 0.007 | **0.091 ± 0.010** | 0.091 ± 0.007 |
| concrete | **0.249 ± 0.025** | 0.486 ± 0.028 | 0.265 ± 0.029 | _0.249 ± 0.017_ | 0.263 ± 0.028 | 0.251 ± 0.022 | 0.253 ± 0.023 |
| elevator | 0.156 ± 0.009 | 0.144 ± 0.007 | 0.149 ± 0.007 | 0.178 ± 0.012 | **0.144 ± 0.008** | 0.156 ± 0.009 | 0.160 ± 0.008 |
| energy_efficiency | **0.047 ± 0.005** | 0.212 ± 0.015 | _0.050 ± 0.007_ | 0.062 ± 0.006 | 0.062 ± 0.013 | 0.051 ± 0.004 | 0.053 ± 0.004 |
| insurance | _0.309 ± 0.032_ | 0.332 ± 0.049 | 0.329 ± 0.056 | 0.585 ± 0.167 | 0.478 ± 0.070 | 0.331 ± 0.077 | **0.295 ± 0.044** |
| kin8nm | 0.360 ± 0.012 | 0.490 ± 0.020 | 0.374 ± 0.014 | 0.373 ± 0.014 | 0.397 ± 0.017 | **0.359 ± 0.012** | _0.360 ± 0.011_ |
| miami_housing | _0.085 ± 0.002_ | 0.105 ± 0.001 | 0.086 ± 0.004 | 0.098 ± 0.003 | 0.126 ± 0.006 | **0.084 ± 0.002** | 0.085 ± 0.002 |
| naval_propulsion | 6.1e-04 ± 6.3e-06 | 6.2e-04 ± 6.5e-06 | _6.0e-04 ± 6.4e-06_ | 7.2e-04 ± 1.3e-05 | **6.0e-04 ± 7.0e-06** | 6.2e-04 ± 5.7e-06 | 6.1e-04 ± 5.6e-06 |
| parkinsons | _0.227 ± 0.010_ | 0.321 ± 0.013 | 0.249 ± 0.014 | 0.312 ± 0.021 | 0.303 ± 0.022 | 0.228 ± 0.008 | **0.225 ± 0.008** |
| powerplant | **0.170 ± 0.007** | 0.196 ± 0.009 | 0.180 ± 0.009 | 0.178 ± 0.007 | 0.183 ± 0.006 | 0.173 ± 0.007 | _0.172 ± 0.007_ |
| qsar | 0.363 ± 0.121 | 0.486 ± 0.160 | 0.386 ± 0.126 | 0.388 ± 0.121 | 0.420 ± 0.139 | _0.362 ± 0.121_ | **0.362 ± 0.121** |
| sulfur | 0.109 ± 0.010 | 0.112 ± 0.010 | **0.107 ± 0.010** | 0.131 ± 0.017 | 0.122 ± 0.015 | _0.108 ± 0.011_ | 0.108 ± 0.011 |
| superconductor | 0.196 ± 0.022 | 0.233 ± 0.025 | 0.196 ± 0.023 | 0.248 ± 0.028 | 0.329 ± 0.042 | _0.194 ± 0.022_ | **0.194 ± 0.021** |

Table 16: Variant (a) Quantile Loss at 95% prediction intervals, aggregated across 10 seeds. Values $\geq 100$ or $< 0.01$ are presented in scientific notation with 1 decimal place. **Bold** values (desirable) are the minimum for that dataset and metric, while the underlined values indicate the second-best result. Red values are more than 33% worse than the best result.

| Dataset | CLEAR | ALEATORIC | ALEATORIC-R | PCS-EPISTEMIC | Naive | $\gamma_1 = 1$ | $\lambda = 1$ |
|---|---|---|---|---|---|---|---|
| ailerons | 9.2e-06 ± 2.3e-07 | 1.0e-05 ± 2.8e-07 | 9.7e-06 ± 2.8e-07 | 1.0e-05 ± 2.2e-07 | 1.1e-05 ± 3.7e-07 | _9.2e-06 ± 2.3e-07_ | **9.2e-06 ± 2.3e-07** |
| airfoil | **0.109 ± 0.011** | 0.183 ± 0.013 | 0.123 ± 0.016 | 0.117 ± 0.012 | 0.147 ± 0.019 | _0.109 ± 0.011_ | 0.109 ± 0.011 |
| allstate | **1.1e+02 ± 7.835** | 1.3e+02 ± 8.197 | 1.3e+02 ± 7.925 | 1.2e+02 ± 7.770 | 1.7e+02 ± 13.696 | 1.1e+02 ± 7.563 | 1.1e+02 ± 7.631 |
| ca_housing | **2.8e+03 ± 59.700** | 4.8e+03 ± 25.753 | 3.0e+03 ± 73.283 | 3.2e+03 ± 93.396 | 4.0e+03 ± 1.2e+02 | _2.8e+03 ± 66.414_ | 2.8e+03 ± 70.545 |
| computer | _0.147 ± 0.022_ | 0.150 ± 0.012 | 0.160 ± 0.029 | 22.643 ± 14.662 | 0.192 ± 0.010 | **0.146 ± 0.020** | 0.147 ± 0.023 |
| concrete | _0.338 ± 0.040_ | 0.546 ± 0.061 | 0.359 ± 0.053 | **0.327 ± 0.036** | 0.376 ± 0.058 | 0.330 ± 0.037 | 0.331 ± 0.037 |
| elevator | 1.5e-04 ± 2.0e-06 | **1.4e-04 ± 2.9e-06** | _1.5e-04 ± 2.4e-06_ | 1.7e-04 ± 5.6e-06 | 1.6e-04 ± 5.1e-06 | 1.5e-04 ± 2.4e-06 | 1.5e-04 ± 2.0e-06 |
| energy_efficiency | **0.031 ± 0.004** | 0.102 ± 0.008 | 0.033 ± 0.005 | 0.038 ± 0.003 | 0.041 ± 0.005 | 0.033 ± 0.003 | 0.034 ± 0.003 |
| insurance | **3.5e+02 ± 61.616** | 3.6e+02 ± 39.928 | 3.6e+02 ± 54.512 | 5.8e+02 ± 1.0e+02 | 4.4e+02 ± 38.440 | 3.6e+02 ± 68.243 | _3.5e+02 ± 62.851_ |
| kin8nm | 7.2e-03 ± 1.7e-04 | 9.5e-03 ± 1.6e-04 | 7.7e-03 ± 2.7e-04 | 7.6e-03 ± 9.7e-05 | 8.3e-03 ± 2.5e-04 | _7.2e-03 ± 1.6e-04_ | **7.2e-03 ± 1.7e-04** |
| miami_housing | 3.9e+03 ± 1.9e+02 | 5.1e+03 ± 3.1e+02 | 5.2e+03 ± 3.4e+02 | 4.2e+03 ± 1.6e+02 | 8.1e+03 ± 3.6e+02 | **3.9e+03 ± 1.9e+02** | 3.9e+03 ± 1.8e+02 |
| naval_propulsion | _1.5e-05 ± 2.1e-07_ | 1.5e-05 ± 2.6e-07 | 1.5e-05 ± 2.6e-07 | 1.7e-05 ± 2.7e-07 | 1.6e-05 ± 3.5e-07 | 1.5e-05 ± 2.1e-07 | **1.5e-05 ± 2.3e-07** |
| parkinsons | **0.178 ± 0.010** | 0.202 ± 0.007 | 0.215 ± 0.012 | 0.224 ± 0.012 | 0.251 ± 0.014 | 0.185 ± 0.011 | _0.178 ± 0.010_ |
| powerplant | **0.212 ± 0.012** | 0.228 ± 0.011 | 0.220 ± 0.011 | 0.221 ± 0.012 | 0.231 ± 0.010 | 0.213 ± 0.012 | _0.213 ± 0.012_ |
| qsar | _0.049 ± 0.003_ | 0.060 ± 0.003 | 0.052 ± 0.003 | 0.053 ± 0.003 | 0.057 ± 0.003 | **0.049 ± 0.003** | 0.049 ± 0.003 |
| sulfur | 2.0e-03 ± 1.4e-04 | 2.1e-03 ± 1.3e-04 | 2.3e-03 ± 1.1e-04 | 2.3e-03 ± 1.9e-04 | 3.5e-03 ± 2.0e-04 | _2.0e-03 ± 1.4e-04_ | **2.0e-03 ± 1.4e-04** |
| superconductor | _0.490 ± 0.018_ | 0.511 ± 0.014 | 0.538 ± 0.023 | 0.608 ± 0.023 | 0.861 ± 0.036 | **0.490 ± 0.018** | 0.491 ± 0.018 |

Table 17: Variant (a) NCIW at 95% prediction intervals, aggregated across 10 seeds. Values $\geq 100$ or $< 0.01$ are presented in scientific notation with 1 decimal place. **Bold** values (desirable) are the minimum for that dataset and metric, while the underlined values indicate the second-best result. Red values are more than 33% worse than the best result.

| Dataset | CLEAR | ALEATORIC | ALEATORIC-R | PCS-EPISTEMIC | Naive | $\gamma_1 = 1$ | $\lambda = 1$ |
|---|---|---|---|---|---|---|---|
| ailerons | 0.195 ± 0.017 | 0.209 ± 0.019 | 0.196 ± 0.017 | 0.217 ± 0.021 | 0.220 ± 0.020 | 0.195 ± 0.017 | **0.195 ± 0.017** |
| airfoil | 0.203 ± 0.006 | 0.356 ± 0.016 | 0.216 ± 0.012 | 0.211 ± 0.007 | 0.246 ± 0.019 | **0.200 ± 0.007** | 0.200 ± 0.006 |
| allstate | 0.252 ± 0.037 | 0.264 ± 0.042 | 0.262 ± 0.042 | 0.266 ± 0.051 | 0.328 ± 0.051 | **0.245 ± 0.042** | 0.245 ± 0.041 |
| ca_housing | 0.336 ± 0.008 | 0.760 ± 0.009 | 0.348 ± 0.010 | 0.354 ± 0.012 | 0.440 ± 0.016 | 0.331 ± 0.007 | **0.328 ± 0.007** |
| computer | **0.091 ± 0.008** | 0.099 ± 0.001 | 0.091 ± 0.011 | 0.112 ± 0.028 | 0.113 ± 0.008 | 0.092 ± 0.009 | 0.091 ± 0.007 |
| concrete | 0.246 ± 0.033 | 0.421 ± 0.079 | 0.259 ± 0.035 | 0.241 ± 0.025 | 0.264 ± 0.037 | 0.238 ± 0.027 | 0.240 ± 0.026 |
| elevator | 0.155 ± 0.010 | **0.143 ± 0.008** | 0.148 ± 0.008 | 0.177 ± 0.011 | 0.145 ± 0.010 | 0.155 ± 0.010 | 0.159 ± 0.010 |
| energy_efficiency | **0.046 ± 0.005** | 0.196 ± 0.032 | 0.050 ± 0.005 | 0.057 ± 0.004 | 0.060 ± 0.006 | 0.050 ± 0.005 | 0.051 ± 0.004 |
| insurance | 0.303 ± 0.029 | 0.284 ± 0.049 | 0.308 ± 0.043 | 0.530 ± 0.178 | 0.454 ± 0.102 | 0.282 ± 0.049 | 0.271 ± 0.046 |
| kin8nm | **0.354 ± 0.008** | 0.482 ± 0.014 | 0.371 ± 0.009 | 0.371 ± 0.009 | 0.396 ± 0.013 | 0.355 ± 0.008 | 0.355 ± 0.008 |
| miami_housing | 0.084 ± 0.003 | 0.106 ± 0.004 | 0.085 ± 0.003 | 0.098 ± 0.002 | 0.124 ± 0.006 | **0.083 ± 0.002** | 0.084 ± 0.002 |
| naval_propulsion | 6.1e-04 ± 7.9e-06 | 6.1e-04 ± 7.4e-06 | 6.0e-04 ± 5.5e-06 | 7.1e-04 ± 1.5e-05 | **5.9e-04 ± 6.1e-06** | 6.1e-04 ± 9.3e-06 | 6.1e-04 ± 6.4e-06 |
| parkinsons | 0.225 ± 0.012 | 0.321 ± 0.013 | 0.247 ± 0.007 | 0.316 ± 0.032 | 0.302 ± 0.013 | 0.227 ± 0.007 | **0.224 ± 0.008** |
| powerplant | **0.173 ± 0.007** | 0.197 ± 0.009 | 0.182 ± 0.007 | 0.182 ± 0.007 | 0.187 ± 0.008 | 0.176 ± 0.007 | 0.174 ± 0.007 |
| qsar | **0.360 ± 0.119** | 0.479 ± 0.156 | 0.389 ± 0.130 | 0.389 ± 0.134 | 0.421 ± 0.142 | 0.360 ± 0.120 | 0.360 ± 0.120 |
| sulfur | 0.109 ± 0.010 | 0.111 ± 0.008 | **0.106 ± 0.009** | 0.128 ± 0.013 | 0.129 ± 0.011 | 0.108 ± 0.011 | 0.107 ± 0.010 |
| superconductor | 0.195 ± 0.021 | 0.231 ± 0.025 | **0.194 ± 0.021** | 0.247 ± 0.028 | 0.314 ± 0.035 | 0.194 ± 0.021 | 0.194 ± 0.021 |

Table 18: Variant (a) Interval Score Loss at 95% prediction intervals, aggregated across 10 seeds. Values $\geq 100$ or $< 0.01$ are presented in scientific notation with 1 decimal place. **Bold** values (desirable) are the minimum for that dataset and metric, while the underlined values indicate the second-best result. Red values are more than 33% worse than the best result.

| Dataset | CLEAR | ALEATORIC | ALEATORIC-R | PCS-EPISTEMIC | Naive | $\gamma_1 = 1$ | $\lambda = 1$ |
|---|---|---|---|---|---|---|---|
| ailerons | 7.4e-04 ± 1.8e-05 | 8.1e-04 ± 2.2e-05 | 7.8e-04 ± 2.2e-05 | 8.0e-04 ± 1.8e-05 | 9.1e-04 ± 2.9e-05 | 7.4e-04 ± 1.8e-05 | **7.4e-04 ± 1.8e-05** |
| airfoil | **8.717 ± 0.870** | 14.669 ± 1.007 | 9.843 ± 1.249 | 9.397 ± 0.977 | 11.761 ± 1.548 | 8.727 ± 0.902 | 8.730 ± 0.892 |
| allstate | **9.1e+03 ± 6.3e+02** | 1.1e+04 ± 6.6e+02 | 1.0e+04 ± 6.3e+02 | 9.9e+03 ± 6.2e+02 | 1.3e+04 ± 1.1e+03 | 9.2e+03 ± 6.1e+02 | 9.2e+03 ± 6.1e+02 |
| ca_housing | **2.2e+05 ± 4.8e+03** | 3.9e+05 ± 2.1e+03 | 2.4e+05 ± 5.9e+03 | 2.6e+05 ± 7.5e+03 | 3.2e+05 ± 9.7e+03 | 2.2e+05 ± 5.3e+03 | 2.2e+05 ± 5.6e+03 |
| computer | 11.741 ± 1.723 | 12.023 ± 0.924 | 12.761 ± 2.285 | 1.8e+03 ± 1.2e+03 | 15.360 ± 0.785 | **11.667 ± 1.598** | 11.761 ± 1.841 |
| concrete | 27.020 ± 3.202 | 43.690 ± 4.842 | 28.684 ± 4.251 | **26.165 ± 2.857** | 30.054 ± 4.640 | 26.365 ± 2.976 | 26.445 ± 2.996 |
| elevator | 0.012 ± 0.000 | **0.011 ± 0.000** | 0.012 ± 0.000 | 0.014 ± 0.000 | 0.013 ± 0.000 | 0.012 ± 0.000 | 0.012 ± 0.000 |
| energy_efficiency | **2.465 ± 0.347** | 8.199 ± 0.672 | 2.671 ± 0.416 | 3.058 ± 0.217 | 3.253 ± 0.388 | 2.667 ± 0.280 | 2.721 ± 0.270 |
| insurance | **2.8e+04 ± 4.9e+03** | 2.9e+04 ± 3.2e+03 | 2.8e+04 ± 4.4e+03 | 4.7e+04 ± 8.1e+03 | 3.6e+04 ± 3.1e+03 | 2.9e+04 ± 5.5e+03 | 2.8e+04 ± 5.0e+03 |
| kin8nm | 0.577 ± 0.014 | 0.760 ± 0.013 | 0.616 ± 0.021 | 0.607 ± 0.008 | 0.666 ± 0.020 | 0.577 ± 0.013 | **0.577 ± 0.013** |
| miami_housing | 3.1e+05 ± 1.5e+04 | 4.1e+05 ± 2.5e+04 | 4.2e+05 ± 2.7e+04 | 3.3e+05 ± 1.3e+04 | 6.5e+05 ± 2.9e+04 | **3.1e+05 ± 1.5e+04** | 3.1e+05 ± 1.5e+04 |
| naval_propulsion | 1.2e-03 ± 1.7e-05 | 1.2e-03 ± 2.1e-05 | 1.2e-03 ± 2.1e-05 | 1.4e-03 ± 2.2e-05 | 1.3e-03 ± 2.8e-05 | 1.2e-03 ± 1.7e-05 | **1.2e-03 ± 1.8e-05** |
| parkinsons | **14.221 ± 0.792** | 16.142 ± 0.582 | 17.188 ± 0.985 | 17.947 ± 0.957 | 20.088 ± 1.130 | 14.762 ± 0.882 | 14.259 ± 0.786 |
| powerplant | **16.993 ± 0.937** | 18.211 ± 0.897 | 17.577 ± 0.884 | 17.715 ± 0.965 | 18.501 ± 0.815 | 17.064 ± 0.951 | 17.027 ± 0.954 |
| qsar | 3.951 ± 0.230 | 4.795 ± 0.219 | 4.187 ± 0.231 | 4.249 ± 0.217 | 4.558 ± 0.216 | **3.950 ± 0.225** | 3.953 ± 0.224 |
| sulfur | 0.161 ± 0.011 | 0.172 ± 0.011 | 0.187 ± 0.008 | 0.186 ± 0.015 | 0.280 ± 0.016 | 0.161 ± 0.011 | **0.161 ± 0.011** |
| superconductor | 39.217 ± 1.419 | 40.851 ± 1.157 | 43.049 ± 1.848 | 48.630 ± 1.867 | 68.891 ± 2.882 | **39.182 ± 1.438** | 39.296 ± 1.444 |

## F.2 VARIANT (B)

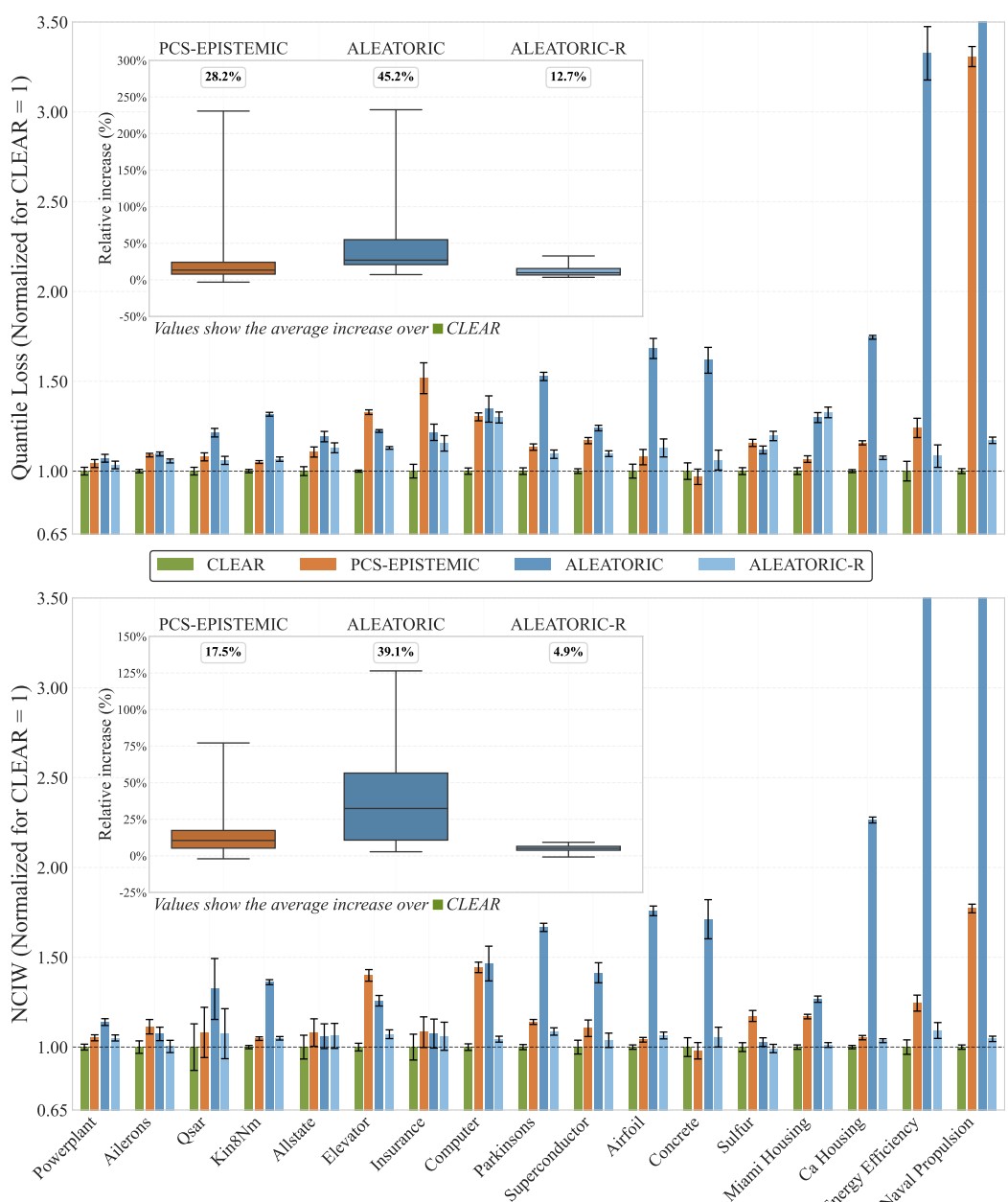

Figure 7: Quantile loss and NCIW performance of different methods (CLEAR, PCS, ALEATORIC, ALEATORIC-R) for variant (b), over 10 seeds normalized relative to CLEAR (baseline = 1.0). Lower values indicate better performance. The inset boxplot shows the % improvement relative to the CLEAR baseline $\pm 1\sigma$. Values inside each subplot represent the mean improvement across all datasets.

Table 19: Variant (b) PICP at 95% prediction intervals, aggregated across 10 seeds.

| Dataset | CLEAR | ALEATORIC | ALEATORIC-R | PCS-EPISTEMIC | Naive | $\gamma_1 = 1$ | $\lambda = 1$ |
|---|---|---|---|---|---|---|---|
| ailerons | 0.95 ± 0.00 | 0.95 ± 0.00 | 0.95 ± 0.00 | 0.95 ± 0.01 | 0.95 ± 0.01 | 0.95 ± 0.00 | 0.95 ± 0.00 |
| airfoil | 0.95 ± 0.01 | 0.95 ± 0.01 | 0.95 ± 0.01 | 0.95 ± 0.01 | 0.95 ± 0.02 | 0.95 ± 0.02 | 0.95 ± 0.01 |
| allstate | 0.95 ± 0.01 | 0.95 ± 0.01 | 0.95 ± 0.01 | 0.95 ± 0.01 | 0.95 ± 0.01 | 0.95 ± 0.01 | 0.95 ± 0.01 |
| ca_housing | 0.95 ± 0.01 | 0.95 ± 0.01 | 0.95 ± 0.01 | 0.95 ± 0.01 | 0.95 ± 0.01 | 0.95 ± 0.01 | 0.95 ± 0.01 |
| computer | 0.95 ± 0.01 | 0.95 ± 0.01 | 0.95 ± 0.01 | 0.95 ± 0.01 | 0.95 ± 0.01 | 0.95 ± 0.01 | 0.95 ± 0.01 |
| concrete | 0.95 ± 0.02 | 0.96 ± 0.02 | 0.95 ± 0.02 | 0.95 ± 0.01 | 0.94 ± 0.03 | 0.95 ± 0.01 | 0.95 ± 0.01 |
| elevator | 0.95 ± 0.00 | 0.95 ± 0.00 | 0.95 ± 0.01 | 0.95 ± 0.00 | 0.95 ± 0.01 | 0.95 ± 0.00 | 0.95 ± 0.00 |
| energy_efficiency | 0.95 ± 0.02 | 0.96 ± 0.02 | 0.95 ± 0.02 | 0.96 ± 0.01 | 0.95 ± 0.02 | 0.96 ± 0.01 | 0.96 ± 0.01 |
| insurance | 0.95 ± 0.01 | 0.96 ± 0.01 | 0.96 ± 0.02 | 0.95 ± 0.01 | 0.95 ± 0.02 | 0.96 ± 0.01 | 0.96 ± 0.01 |
| kin8nm | 0.95 ± 0.01 | 0.95 ± 0.01 | 0.95 ± 0.01 | 0.95 ± 0.01 | 0.95 ± 0.01 | 0.95 ± 0.01 | 0.95 ± 0.01 |
| miami_housing | 0.95 ± 0.01 | 0.95 ± 0.00 | 0.95 ± 0.01 | 0.95 ± 0.01 | 0.95 ± 0.01 | 0.95 ± 0.00 | 0.95 ± 0.00 |
| naval_propulsion | 0.95 ± 0.01 | 0.95 ± 0.00 | 0.95 ± 0.01 | 1.00 ± 0.00 | 0.95 ± 0.00 | 0.95 ± 0.01 | 0.95 ± 0.00 |
| parkinsons | 0.95 ± 0.01 | 0.95 ± 0.01 | 0.95 ± 0.01 | 0.95 ± 0.01 | 0.95 ± 0.01 | 0.95 ± 0.01 | 0.95 ± 0.01 |
| powerplant | 0.95 ± 0.01 | 0.95 ± 0.00 | 0.95 ± 0.01 | 0.95 ± 0.01 | 0.95 ± 0.01 | 0.95 ± 0.01 | 0.95 ± 0.01 |
| qsar | 0.95 ± 0.01 | 0.95 ± 0.01 | 0.95 ± 0.01 | 0.95 ± 0.01 | 0.95 ± 0.00 | 0.95 ± 0.01 | 0.95 ± 0.01 |
| sulfur | 0.95 ± 0.01 | 0.95 ± 0.01 | 0.95 ± 0.01 | 0.95 ± 0.00 | 0.95 ± 0.01 | 0.95 ± 0.01 | 0.95 ± 0.01 |
| superconductor | 0.95 ± 0.00 | 0.95 ± 0.01 | 0.95 ± 0.00 | 0.95 ± 0.00 | 0.95 ± 0.00 | 0.95 ± 0.00 | 0.95 ± 0.00 |

Table 20: Variant (b) NIW at 95% prediction intervals, aggregated across 10 seeds. Values $\geq 100$ or $< 0.01$ are presented in scientific notation with 1 decimal place. **Bold** values (desirable) are the minimum for that dataset and metric, while the underlined values indicate the second-best result. Red values are more than 33% worse than the best result.

| Dataset | CLEAR | ALEATORIC | ALEATORIC-R | PCS-EPISTEMIC | Naive | $\gamma_1 = 1$ | $\lambda = 1$ |
|---|---|---|---|---|---|---|---|
| ailerons | 0.193 ± 0.016 | 0.208 ± 0.017 | 0.195 ± 0.017 | 0.215 ± 0.016 | 0.220 ± 0.021 | 0.193 ± 0.016 | **0.193 ± 0.016** |
| airfoil | 0.207 ± 0.012 | 0.367 ± 0.018 | 0.215 ± 0.018 | 0.215 ± 0.012 | 0.246 ± 0.024 | **0.205 ± 0.012** | 0.206 ± 0.012 |
| allstate | 0.256 ± 0.045 | 0.265 ± 0.047 | 0.260 ± 0.047 | 0.263 ± 0.053 | 0.325 ± 0.055 | **0.243 ± 0.044** | 0.245 ± 0.046 |
| ca_housing | 0.339 ± 0.008 | 0.760 ± 0.006 | 0.350 ± 0.006 | 0.354 ± 0.011 | 0.434 ± 0.020 | 0.333 ± 0.006 | **0.331 ± 0.007** |
| computer | **0.099 ± 0.006** | 0.143 ± 0.029 | 0.104 ± 0.005 | 0.142 ± 0.009 | 0.113 ± 0.006 | 0.101 ± 0.005 | 0.104 ± 0.005 |
| concrete | **0.249 ± 0.025** | 0.486 ± 0.028 | 0.265 ± 0.029 | 0.249 ± 0.017 | 0.263 ± 0.028 | 0.251 ± 0.022 | 0.253 ± 0.023 |
| elevator | 0.137 ± 0.007 | 0.171 ± 0.010 | 0.146 ± 0.009 | 0.193 ± 0.013 | 0.169 ± 0.010 | **0.136 ± 0.008** | 0.137 ± 0.008 |
| energy_efficiency | **0.047 ± 0.005** | 0.212 ± 0.015 | 0.050 ± 0.007 | 0.062 ± 0.006 | 0.062 ± 0.013 | 0.051 ± 0.004 | 0.053 ± 0.004 |
| insurance | 0.329 ± 0.045 | 0.407 ± 0.064 | 0.381 ± 0.055 | 0.434 ± 0.179 | 0.478 ± 0.070 | 0.339 ± 0.104 | **0.306 ± 0.058** |
| kin8nm | 0.360 ± 0.012 | 0.490 ± 0.020 | 0.374 ± 0.014 | 0.373 ± 0.014 | 0.397 ± 0.017 | **0.359 ± 0.012** | 0.360 ± 0.011 |
| miami_housing | 0.085 ± 0.002 | 0.105 ± 0.001 | 0.086 ± 0.004 | 0.098 ± 0.003 | 0.126 ± 0.006 | **0.084 ± 0.002** | 0.085 ± 0.002 |
| naval_propulsion | **1.9e-03 ± 7.3e-05** | 0.222 ± 0.003 | 1.9e-03 ± 9.7e-05 | 8.1e-03 ± 4.1e-04 | 2.4e-03 ± 1.4e-04 | 2.1e-03 ± 8.1e-05 | 2.8e-03 ± 1.3e-04 |
| parkinsons | **0.281 ± 0.009** | 0.483 ± 0.006 | 0.302 ± 0.012 | 0.325 ± 0.011 | 0.345 ± 0.011 | 0.282 ± 0.009 | 0.281 ± 0.009 |
| powerplant | **0.170 ± 0.007** | 0.196 ± 0.009 | 0.180 ± 0.009 | 0.178 ± 0.007 | 0.183 ± 0.006 | 0.173 ± 0.007 | 0.172 ± 0.007 |
| qsar | 0.366 ± 0.123 | 0.486 ± 0.160 | 0.387 ± 0.127 | 0.393 ± 0.134 | 0.421 ± 0.139 | 0.362 ± 0.121 | **0.362 ± 0.121** |
| sulfur | 0.108 ± 0.009 | 0.110 ± 0.008 | 0.107 ± 0.011 | 0.129 ± 0.014 | 0.122 ± 0.015 | **0.106 ± 0.009** | 0.106 ± 0.009 |
| superconductor | 0.219 ± 0.023 | 0.309 ± 0.033 | 0.228 ± 0.026 | 0.243 ± 0.027 | 0.354 ± 0.042 | 0.223 ± 0.025 | **0.216 ± 0.024** |

Table 21: Variant (b) Quantile Loss at 95% prediction intervals, aggregated across 10 seeds. Values $\geq 100$ or $< 0.01$ are presented in scientific notation with 1 decimal place. **Bold** values (desirable) are the minimum for that dataset and metric, while the underlined values indicate the second-best result. Red values are more than 33% worse than the best result.

| Dataset | CLEAR | ALEATORIC | ALEATORIC-R | PCS-EPISTEMIC | Naive | $\gamma_1 = 1$ | $\lambda = 1$ |
|---|---|---|---|---|---|---|---|
| ailerons | 9.2e-06 ± 2.3e-07 | 1.0e-05 ± 2.8e-07 | 9.7e-06 ± 2.8e-07 | 1.0e-05 ± 2.2e-07 | 1.1e-05 ± 3.7e-07 | 9.2e-06 ± 2.3e-07 | **9.2e-06 ± 2.3e-07** |
| airfoil | **0.109 ± 0.011** | 0.183 ± 0.013 | 0.123 ± 0.016 | 0.117 ± 0.012 | 0.147 ± 0.019 | 0.109 ± 0.011 | 0.109 ± 0.011 |
| allstate | **1.1e+02 ± 7.318** | 1.4e+02 ± 8.321 | 1.3e+02 ± 8.046 | 1.3e+02 ± 8.557 | 1.7e+02 ± 10.782 | 1.2e+02 ± 7.533 | 1.2e+02 ± 7.762 |
| ca_housing | **2.8e+03 ± 59.700** | 4.8e+03 ± 25.753 | 3.0e+03 ± 73.283 | 3.2e+03 ± 93.396 | 4.0e+03 ± 1.2e+02 | 2.8e+03 ± 66.414 | 2.8e+03 ± 70.545 |
| computer | 0.159 ± 0.007 | 0.214 ± 0.041 | 0.206 ± 0.015 | 0.207 ± 0.009 | 0.234 ± 0.019 | **0.159 ± 0.007** | 0.161 ± 0.008 |
| concrete | 0.338 ± 0.040 | 0.546 ± 0.061 | 0.359 ± 0.053 | **0.327 ± 0.036** | 0.376 ± 0.058 | 0.330 ± 0.037 | 0.331 ± 0.037 |
| elevator | **1.4e-04 ± 2.0e-06** | 1.8e-04 ± 3.0e-06 | 1.6e-04 ± 3.1e-06 | 1.9e-04 ± 6.2e-06 | 2.1e-04 ± 5.5e-06 | 1.4e-04 ± 2.0e-06 | 1.4e-04 ± 2.1e-06 |
| energy_efficiency | **0.031 ± 0.004** | 0.102 ± 0.008 | 0.033 ± 0.005 | 0.038 ± 0.003 | 0.041 ± 0.005 | 0.033 ± 0.003 | 0.034 ± 0.003 |
| insurance | **3.2e+02 ± 31.593** | 3.9e+02 ± 36.904 | 3.7e+02 ± 35.107 | 4.9e+02 ± 88.647 | 4.4e+02 ± 34.159 | 3.7e+02 ± 62.380 | 3.4e+02 ± 40.487 |
| kin8nm | 7.2e-03 ± 1.7e-04 | 9.5e-03 ± 1.6e-04 | 7.7e-03 ± 2.7e-04 | 7.6e-03 ± 9.7e-05 | 8.3e-03 ± 2.5e-04 | 7.2e-03 ± 1.6e-04 | **7.2e-03 ± 1.7e-04** |
| miami_housing | 3.9e+03 ± 1.9e+02 | 5.1e+03 ± 3.1e+02 | 5.2e+03 ± 3.4e+02 | 4.2e+03 ± 1.6e+02 | 8.1e+03 ± 3.6e+02 | **3.9e+03 ± 1.9e+02** | 3.9e+03 ± 1.8e+02 |
| naval_propulsion | **5.4e-05 ± 1.9e-06** | 4.9e-03 ± 7.0e-05 | 6.4e-05 ± 2.7e-06 | 1.8e-04 ± 9.1e-06 | 8.8e-05 ± 5.9e-06 | 6.0e-05 ± 1.8e-06 | 7.5e-05 ± 2.7e-06 |
| parkinsons | 0.209 ± 0.010 | 0.319 ± 0.009 | 0.228 ± 0.014 | 0.237 ± 0.008 | 0.265 ± 0.013 | 0.209 ± 0.010 | **0.209 ± 0.009** |
| powerplant | **0.212 ± 0.012** | 0.228 ± 0.011 | 0.220 ± 0.011 | 0.221 ± 0.012 | 0.231 ± 0.010 | 0.213 ± 0.012 | 0.213 ± 0.012 |
| qsar | 0.049 ± 0.003 | 0.060 ± 0.003 | 0.052 ± 0.003 | 0.053 ± 0.003 | 0.057 ± 0.003 | 0.049 ± 0.003 | **0.049 ± 0.003** |
| sulfur | **1.9e-03 ± 9.6e-05** | 2.2e-03 ± 1.1e-04 | 2.3e-03 ± 1.4e-04 | 2.2e-03 ± 8.6e-05 | 3.5e-03 ± 2.4e-04 | 1.9e-03 ± 8.2e-05 | 1.9e-03 ± 8.6e-05 |
| superconductor | 0.523 ± 0.018 | 0.648 ± 0.020 | 0.573 ± 0.024 | 0.611 ± 0.025 | 0.892 ± 0.035 | **0.522 ± 0.018** | 0.523 ± 0.020 |

Table 22: Variant (b) NCIW at 95% prediction intervals, aggregated across 10 seeds. Values $\geq 100$ or $< 0.01$ are presented in scientific notation with 1 decimal place. **Bold** values (desirable) are the minimum for that dataset and metric, while the underlined values indicate the second-best result. Red values are more than 33% worse than the best result.

| Dataset | CLEAR | ALEATORIC | ALEATORIC-R | PCS-EPISTEMIC | Naive | $\gamma_1 = 1$ | $\lambda = 1$ |
|---|---|---|---|---|---|---|---|
| ailerons | $0.195 \pm 0.017$ | $0.209 \pm 0.019$ | $0.196 \pm 0.017$ | $0.217 \pm 0.021$ | $0.220 \pm 0.020$ | $0.195 \pm 0.017$ | $\mathbf{0.195 \pm 0.017}$ |
| airfoil | $0.203 \pm 0.006$ | $0.356 \pm 0.016$ | $0.216 \pm 0.012$ | $0.211 \pm 0.007$ | $0.246 \pm 0.019$ | $\underline{\mathbf{0.200 \pm 0.007}}$ | $0.200 \pm 0.006$ |
| allstate | $0.250 \pm 0.043$ | $0.265 \pm 0.043$ | $0.266 \pm 0.044$ | $0.270 \pm 0.052$ | $0.329 \pm 0.053$ | $\underline{0.245 \pm 0.045}$ | $\mathbf{0.244 \pm 0.044}$ |
| ca_housing | $0.336 \pm 0.008$ | $0.760 \pm 0.009$ | $0.348 \pm 0.010$ | $0.354 \pm 0.012$ | $0.440 \pm 0.016$ | $\underline{0.331 \pm 0.007}$ | $\mathbf{0.328 \pm 0.007}$ |
| computer | $\mathbf{0.098 \pm 0.005}$ | $0.144 \pm 0.034$ | $0.103 \pm 0.003$ | $0.142 \pm 0.008$ | $0.111 \pm 0.005$ | $\underline{0.101 \pm 0.004}$ | $0.105 \pm 0.004$ |
| concrete | $0.246 \pm 0.033$ | $0.421 \pm 0.079$ | $0.259 \pm 0.035$ | $0.241 \pm 0.025$ | $0.264 \pm 0.037$ | $\underline{0.238 \pm 0.027}$ | $0.240 \pm 0.026$ |
| elevator | $0.137 \pm 0.008$ | $0.173 \pm 0.011$ | $0.147 \pm 0.009$ | $0.192 \pm 0.012$ | $0.168 \pm 0.012$ | $\underline{0.137 \pm 0.007}$ | $\mathbf{0.137 \pm 0.007}$ |
| energy_efficiency | $\mathbf{0.046 \pm 0.005}$ | $0.196 \pm 0.032$ | $\underline{0.050 \pm 0.005}$ | $0.057 \pm 0.004$ | $0.060 \pm 0.006$ | $0.050 \pm 0.005$ | $0.051 \pm 0.004$ |
| insurance | $0.320 \pm 0.059$ | $0.344 \pm 0.069$ | $0.339 \pm 0.066$ | $0.346 \pm 0.076$ | $0.455 \pm 0.090$ | $\underline{0.292 \pm 0.058}$ | $\mathbf{0.275 \pm 0.054}$ |
| kin8nm | $\mathbf{0.354 \pm 0.008}$ | $0.482 \pm 0.014$ | $0.371 \pm 0.009$ | $0.371 \pm 0.009$ | $0.396 \pm 0.013$ | $\underline{0.355 \pm 0.008}$ | $0.355 \pm 0.008$ |
| miami_housing | $\underline{0.084 \pm 0.003}$ | $0.106 \pm 0.004$ | $0.085 \pm 0.003$ | $0.098 \pm 0.002$ | $0.124 \pm 0.006$ | $\mathbf{0.083 \pm 0.002}$ | $0.084 \pm 0.002$ |
| naval_propulsion | $\mathbf{1.8e\text{-}03 \pm 5.6e\text{-}05}$ | $0.195 \pm 0.011$ | $1.9e\text{-}03 \pm 8.0e\text{-}05$ | $3.2e\text{-}03 \pm 1.2e\text{-}04$ | $2.4e\text{-}03 \pm 1.4e\text{-}04$ | $\underline{2.1e\text{-}03 \pm 7.2e\text{-}05}$ | $2.8e\text{-}03 \pm 1.1e\text{-}04$ |
| parkinsons | $\mathbf{0.284 \pm 0.010}$ | $0.473 \pm 0.016$ | $\underline{0.309 \pm 0.018}$ | $0.324 \pm 0.008$ | $0.349 \pm 0.019$ | $0.285 \pm 0.011$ | $0.285 \pm 0.011$ |
| powerplant | $\mathbf{0.173 \pm 0.007}$ | $0.197 \pm 0.009$ | $0.182 \pm 0.007$ | $0.182 \pm 0.007$ | $0.187 \pm 0.008$ | $\underline{0.176 \pm 0.007}$ | $0.174 \pm 0.007$ |
| qsar | $0.363 \pm 0.121$ | $0.480 \pm 0.157$ | $0.390 \pm 0.130$ | $0.393 \pm 0.131$ | $0.423 \pm 0.142$ | $\underline{0.362 \pm 0.120}$ | $\mathbf{0.362 \pm 0.120}$ |
| sulfur | $0.106 \pm 0.007$ | $0.109 \pm 0.006$ | $0.105 \pm 0.006$ | $0.125 \pm 0.009$ | $0.127 \pm 0.008$ | $\underline{0.105 \pm 0.006}$ | $\mathbf{0.104 \pm 0.006}$ |
| superconductor | $\underline{0.218 \pm 0.022}$ | $0.308 \pm 0.032$ | $0.226 \pm 0.022$ | $0.241 \pm 0.028$ | $0.347 \pm 0.036$ | $0.223 \pm 0.024$ | $\mathbf{0.216 \pm 0.024}$ |

Table 23: Variant (b) Interval Score Loss at 95% prediction intervals, aggregated across 10 seeds. Values $\geq 100$ or $< 0.01$ are presented in scientific notation with 1 decimal place. **Bold** values (desirable) are the minimum for that dataset and metric, while the underlined values indicate the second-best result. Red values are more than 33% worse than the best result.

| Dataset | CLEAR | ALEATORIC | ALEATORIC-R | PCS-EPISTEMIC | Naive | $\gamma_1 = 1$ | $\lambda = 1$ |
|---|---|---|---|---|---|---|---|
| ailerons | $7.4e\text{-}04 \pm 1.8e\text{-}05$ | $8.1e\text{-}04 \pm 2.2e\text{-}05$ | $7.8e\text{-}04 \pm 2.2e\text{-}05$ | $8.0e\text{-}04 \pm 1.8e\text{-}05$ | $9.1e\text{-}04 \pm 2.9e\text{-}05$ | $\underline{7.4e\text{-}04 \pm 1.8e\text{-}05}$ | $\mathbf{7.4e\text{-}04 \pm 1.8e\text{-}05}$ |
| airfoil | $\mathbf{8.717 \pm 0.870}$ | $14.669 \pm 1.007$ | $9.843 \pm 1.249$ | $9.397 \pm 0.977$ | $11.761 \pm 1.548$ | $\underline{8.727 \pm 0.902}$ | $8.730 \pm 0.892$ |
| allstate | $\mathbf{9.2e\text{+}03 \pm 5.9e\text{+}02}$ | $1.1e\text{+}04 \pm 6.7e\text{+}02$ | $1.0e\text{+}04 \pm 6.4e\text{+}02$ | $1.0e\text{+}04 \pm 6.8e\text{+}02$ | $1.3e\text{+}04 \pm 8.6e\text{+}02$ | $\underline{9.3e\text{+}03 \pm 6.0e\text{+}02}$ | $9.3e\text{+}03 \pm 6.2e\text{+}02$ |
| ca_housing | $\mathbf{2.2e\text{+}05 \pm 4.8e\text{+}03}$ | $3.9e\text{+}05 \pm 2.1e\text{+}03$ | $2.4e\text{+}05 \pm 5.9e\text{+}03$ | $2.6e\text{+}05 \pm 7.5e\text{+}03$ | $3.2e\text{+}05 \pm 9.7e\text{+}03$ | $\underline{2.2e\text{+}05 \pm 5.3e\text{+}03}$ | $2.2e\text{+}05 \pm 5.6e\text{+}03$ |
| computer | $12.707 \pm 0.575$ | $17.102 \pm 3.298$ | $16.503 \pm 1.199$ | $16.547 \pm 0.692$ | $18.684 \pm 1.517$ | $\mathbf{12.707 \pm 0.591}$ | $12.897 \pm 0.602$ |
| concrete | $27.020 \pm 3.202$ | $43.690 \pm 4.842$ | $28.684 \pm 4.251$ | $\mathbf{26.165 \pm 2.857}$ | $30.054 \pm 4.640$ | $\underline{26.365 \pm 2.976}$ | $26.445 \pm 2.996$ |
| elevator | $\mathbf{0.011 \pm 0.000}$ | $0.014 \pm 0.000$ | $0.013 \pm 0.000$ | $0.015 \pm 0.000$ | $0.017 \pm 0.000$ | $\underline{0.012 \pm 0.000}$ | $0.012 \pm 0.000$ |
| energy_efficiency | $\mathbf{2.465 \pm 0.347}$ | $8.199 \pm 0.672$ | $2.671 \pm 0.416$ | $3.058 \pm 0.217$ | $3.253 \pm 0.388$ | $\underline{2.667 \pm 0.280}$ | $2.721 \pm 0.270$ |
| insurance | $\underline{2.6e\text{+}04 \pm 2.5e\text{+}03}$ | $3.1e\text{+}04 \pm 3.0e\text{+}03$ | $3.0e\text{+}04 \pm 2.8e\text{+}03$ | $3.9e\text{+}04 \pm 7.1e\text{+}03$ | $3.5e\text{+}04 \pm 2.7e\text{+}03$ | $3.0e\text{+}04 \pm 5.0e\text{+}03$ | $\mathbf{2.8e\text{+}04 \pm 3.2e\text{+}03}$ |
| kin8nm | $0.577 \pm 0.014$ | $0.760 \pm 0.013$ | $0.616 \pm 0.021$ | $0.607 \pm 0.008$ | $0.666 \pm 0.020$ | $\underline{0.577 \pm 0.013}$ | $\mathbf{0.577 \pm 0.013}$ |
| miami_housing | $3.1e\text{+}05 \pm 1.5e\text{+}04$ | $4.1e\text{+}05 \pm 2.5e\text{+}04$ | $4.2e\text{+}05 \pm 2.7e\text{+}04$ | $3.3e\text{+}05 \pm 1.3e\text{+}04$ | $6.5e\text{+}05 \pm 2.9e\text{+}04$ | $\mathbf{3.1e\text{+}05 \pm 1.5e\text{+}04}$ | $3.1e\text{+}05 \pm 1.5e\text{+}04$ |
| naval_propulsion | $\mathbf{4.4e\text{-}03 \pm 1.5e\text{-}04}$ | $0.391 \pm 0.006$ | $5.1e\text{-}03 \pm 2.2e\text{-}04$ | $0.014 \pm 0.001$ | $7.1e\text{-}03 \pm 4.7e\text{-}04$ | $\underline{4.8e\text{-}03 \pm 1.4e\text{-}04}$ | $6.0e\text{-}03 \pm 2.2e\text{-}04$ |
| parkinsons | $16.697 \pm 0.794$ | $25.497 \pm 0.692$ | $18.270 \pm 1.122$ | $18.931 \pm 0.600$ | $21.220 \pm 1.011$ | $\underline{16.688 \pm 0.771}$ | $\mathbf{16.680 \pm 0.750}$ |
| powerplant | $\mathbf{16.993 \pm 0.937}$ | $18.211 \pm 0.897$ | $17.577 \pm 0.884$ | $17.715 \pm 0.965$ | $18.501 \pm 0.815$ | $\underline{17.064 \pm 0.951}$ | $17.027 \pm 0.954$ |
| qsar | $3.956 \pm 0.218$ | $4.802 \pm 0.219$ | $4.193 \pm 0.227$ | $4.270 \pm 0.220$ | $4.562 \pm 0.217$ | $\underline{3.946 \pm 0.222}$ | $\mathbf{3.946 \pm 0.223}$ |
| sulfur | $\mathbf{0.155 \pm 0.008}$ | $0.174 \pm 0.009$ | $0.186 \pm 0.012$ | $0.180 \pm 0.007$ | $0.277 \pm 0.020$ | $0.156 \pm 0.007$ | $0.155 \pm 0.007$ |
| superconductor | $\underline{41.801 \pm 1.407}$ | $51.826 \pm 1.568$ | $45.860 \pm 1.883$ | $48.899 \pm 2.037$ | $71.323 \pm 2.803$ | $\mathbf{41.747 \pm 1.460}$ | $41.821 \pm 1.571$ |

## F.3 VARIANT (C)

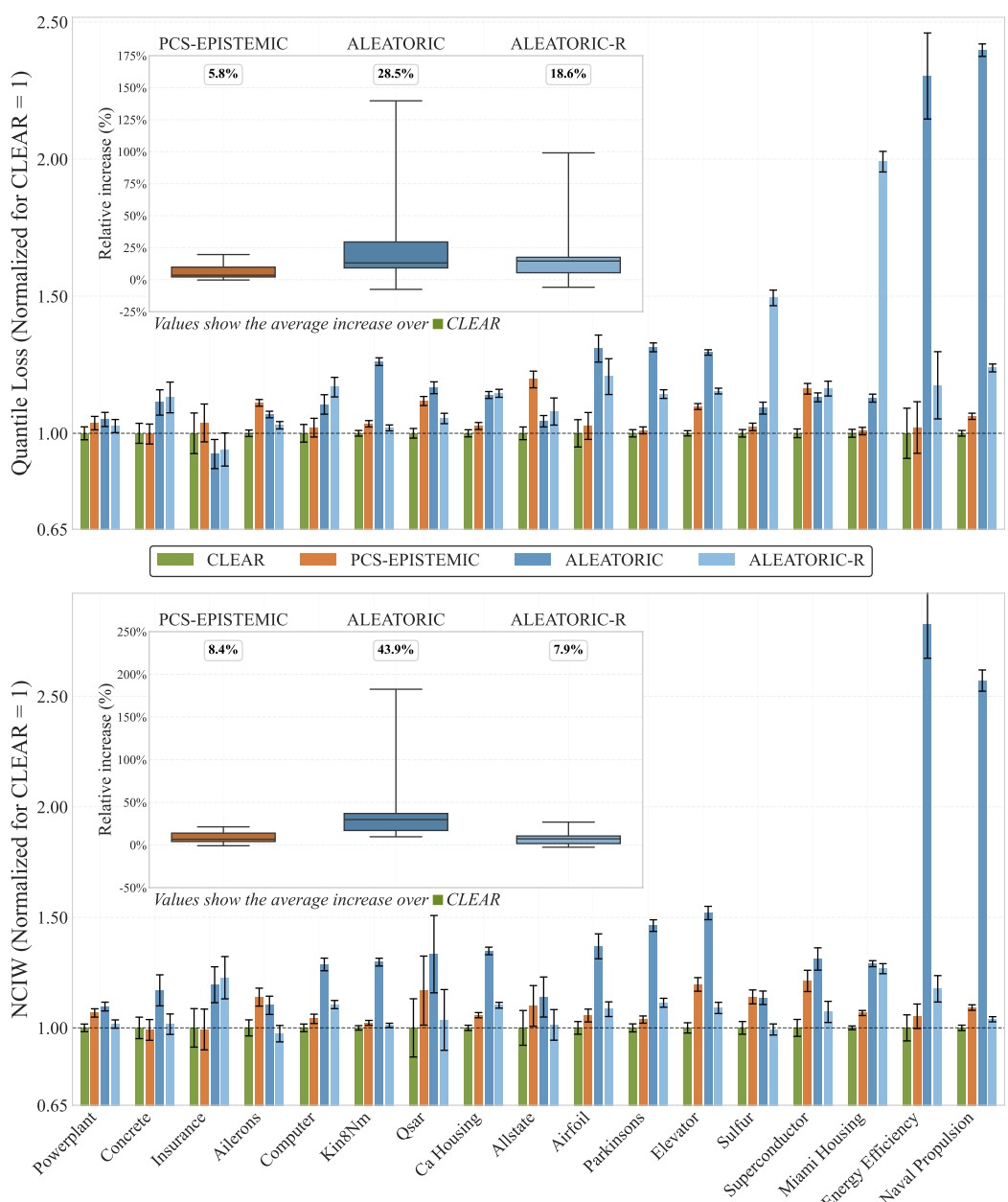

Figure 8: Quantile loss and NCIW performance of different methods (CLEAR, PCS, ALEATORIC, ALEATORIC-R) for variant (c), over 10 seeds normalized relative to CLEAR (baseline = 1.0). Lower values indicate better performance. The inset boxplot shows the % improvement relative to the CLEAR baseline $\pm 1\sigma$. Values inside each subplot represent the mean improvement across all datasets.

Table 24: Variant (c) PICP at 95% prediction intervals, aggregated across 10 seeds.

| Dataset | CLEAR | ALEATORIC | ALEATORIC-R | PCS-EPISTEMIC | Naive | $\gamma_1 = 1$ | $\lambda = 1$ |
|---|---|---|---|---|---|---|---|
| ailerons | 0.95 ± 0.01 | 0.95 ± 0.01 | 0.95 ± 0.01 | 0.95 ± 0.01 | 0.95 ± 0.01 | 0.95 ± 0.01 | 0.95 ± 0.01 |
| airfoil | 0.96 ± 0.01 | 0.95 ± 0.02 | 0.96 ± 0.01 | 0.95 ± 0.02 | 0.95 ± 0.02 | 0.96 ± 0.01 | 0.96 ± 0.01 |
| allstate | 0.95 ± 0.01 | 0.95 ± 0.01 | 0.95 ± 0.01 | 0.95 ± 0.01 | 0.95 ± 0.01 | 0.95 ± 0.01 | 0.95 ± 0.01 |
| ca_housing | 0.95 ± 0.01 | 0.95 ± 0.00 | 0.95 ± 0.01 | 0.95 ± 0.01 | 0.95 ± 0.01 | 0.95 ± 0.01 | 0.95 ± 0.01 |
| computer | 0.95 ± 0.01 | 0.95 ± 0.01 | 0.95 ± 0.01 | 0.95 ± 0.01 | 0.95 ± 0.01 | 0.95 ± 0.01 | 0.95 ± 0.01 |
| concrete | 0.96 ± 0.01 | 0.96 ± 0.02 | 0.96 ± 0.02 | 0.96 ± 0.01 | 0.95 ± 0.02 | 0.96 ± 0.01 | 0.96 ± 0.01 |
| elevator | 0.95 ± 0.01 | 0.95 ± 0.00 | 0.95 ± 0.00 | 0.95 ± 0.00 | 0.95 ± 0.00 | 0.95 ± 0.01 | 0.95 ± 0.01 |
| energy_efficiency | 0.95 ± 0.01 | 0.96 ± 0.01 | 0.96 ± 0.01 | 0.96 ± 0.02 | 0.95 ± 0.01 | 0.97 ± 0.01 | 0.96 ± 0.01 |
| insurance | 0.95 ± 0.01 | 0.95 ± 0.01 | 0.96 ± 0.01 | 0.96 ± 0.01 | 0.95 ± 0.01 | 0.96 ± 0.01 | 0.96 ± 0.01 |
| kin8nm | 0.95 ± 0.01 | 0.95 ± 0.01 | 0.95 ± 0.01 | 0.95 ± 0.01 | 0.95 ± 0.01 | 0.95 ± 0.01 | 0.95 ± 0.01 |
| miami_housing | 0.95 ± 0.01 | 0.95 ± 0.01 | 0.95 ± 0.01 | 0.95 ± 0.00 | 0.95 ± 0.01 | 0.95 ± 0.01 | 0.95 ± 0.01 |
| naval_propulsion | 0.95 ± 0.01 | 0.95 ± 0.01 | 0.95 ± 0.01 | 0.95 ± 0.01 | 0.95 ± 0.00 | 0.95 ± 0.01 | 0.95 ± 0.01 |
| parkinsons | 0.95 ± 0.01 | 0.95 ± 0.01 | 0.95 ± 0.01 | 0.95 ± 0.01 | 0.95 ± 0.01 | 0.95 ± 0.01 | 0.95 ± 0.01 |
| powerplant | 0.95 ± 0.01 | 0.95 ± 0.01 | 0.95 ± 0.01 | 0.95 ± 0.01 | 0.95 ± 0.01 | 0.95 ± 0.01 | 0.95 ± 0.01 |
| qsar | 0.95 ± 0.01 | 0.95 ± 0.01 | 0.95 ± 0.01 | 0.95 ± 0.01 | 0.95 ± 0.01 | 0.95 ± 0.01 | 0.95 ± 0.01 |
| sulfur | 0.95 ± 0.01 | 0.95 ± 0.01 | 0.95 ± 0.01 | 0.95 ± 0.01 | 0.95 ± 0.01 | 0.95 ± 0.01 | 0.95 ± 0.01 |
| superconductor | 0.95 ± 0.00 | 0.95 ± 0.00 | 0.95 ± 0.00 | 0.95 ± 0.00 | 0.95 ± 0.00 | 0.95 ± 0.00 | 0.95 ± 0.00 |

Table 25: Variant (c) NIW at 95% prediction intervals, aggregated across 10 seeds. Values $\geq 100$ or $< 0.01$ are presented in scientific notation with 1 decimal place. **Bold** values (desirable) are the minimum for that dataset and metric, while the underlined values indicate the second-best result. Red values are more than 33% worse than the best result.

| Dataset | CLEAR | ALEATORIC | ALEATORIC-R | PCS-EPISTEMIC | Naive | $\gamma_1 = 1$ | $\lambda = 1$ |
|---|---|---|---|---|---|---|---|
| ailerons | 0.208 ± 0.017 | 0.227 ± 0.017 | **0.202 ± 0.018** | 0.237 ± 0.024 | 0.229 ± 0.020 | 0.208 ± 0.016 | 0.208 ± 0.017 |
| airfoil | **0.199 ± 0.013** | 0.286 ± 0.016 | 0.223 ± 0.024 | 0.204 ± 0.019 | 0.214 ± 0.024 | 0.205 ± 0.018 | 0.202 ± 0.014 |
| allstate | 0.279 ± 0.052 | 0.317 ± 0.065 | 0.277 ± 0.047 | 0.299 ± 0.065 | 0.315 ± 0.059 | 0.274 ± 0.053 | **0.272 ± 0.051** |
| ca_housing | 0.315 ± 0.010 | 0.425 ± 0.005 | 0.345 ± 0.013 | 0.332 ± 0.011 | 0.400 ± 0.016 | 0.312 ± 0.008 | **0.311 ± 0.008** |
| computer | 0.080 ± 0.004 | 0.102 ± 0.001 | 0.089 ± 0.004 | 0.084 ± 0.004 | 0.091 ± 0.006 | 0.079 ± 0.004 | **0.078 ± 0.004** |
| concrete | 0.278 ± 0.029 | 0.315 ± 0.019 | 0.284 ± 0.027 | **0.273 ± 0.031** | 0.276 ± 0.028 | 0.279 ± 0.029 | 0.281 ± 0.029 |
| elevator | 0.127 ± 0.008 | 0.192 ± 0.010 | 0.137 ± 0.008 | 0.152 ± 0.008 | 0.144 ± 0.010 | 0.127 ± 0.007 | **0.127 ± 0.007** |
| energy_efficiency | **0.043 ± 0.007** | 0.138 ± 0.007 | 0.051 ± 0.005 | 0.045 ± 0.007 | 0.049 ± 0.005 | 0.047 ± 0.007 | 0.046 ± 0.007 |
| insurance | 0.397 ± 0.091 | 0.421 ± 0.049 | 0.459 ± 0.050 | **0.395 ± 0.096** | 0.423 ± 0.083 | 0.440 ± 0.092 | 0.475 ± 0.109 |
| kin8nm | **0.325 ± 0.012** | 0.415 ± 0.014 | 0.330 ± 0.014 | 0.329 ± 0.011 | 0.344 ± 0.012 | 0.328 ± 0.014 | 0.326 ± 0.013 |
| miami_housing | 0.088 ± 0.001 | 0.112 ± 0.002 | 0.113 ± 0.008 | 0.093 ± 0.002 | 0.123 ± 0.008 | **0.078 ± 0.002** | 0.078 ± 0.002 |
| naval_propulsion | 1.6e-03 ± 3.2e-05 | 4.1e-03 ± 8.9e-05 | 1.6e-03 ± 4.3e-05 | 1.7e-03 ± 5.4e-05 | 1.8e-03 ± 3.9e-05 | **1.6e-03 ± 1.9e-05** | 1.6e-03 ± 3.0e-05 |
| parkinsons | 0.250 ± 0.007 | 0.371 ± 0.011 | 0.281 ± 0.011 | 0.260 ± 0.008 | 0.285 ± 0.010 | 0.251 ± 0.008 | **0.248 ± 0.007** |
| powerplant | **0.157 ± 0.007** | 0.175 ± 0.008 | 0.161 ± 0.008 | 0.169 ± 0.006 | 0.164 ± 0.007 | 0.158 ± 0.007 | 0.158 ± 0.007 |
| qsar | **0.344 ± 0.117** | 0.464 ± 0.153 | 0.353 ± 0.117 | 0.399 ± 0.138 | 0.386 ± 0.131 | 0.352 ± 0.120 | 0.346 ± 0.118 |
| sulfur | 0.111 ± 0.009 | 0.124 ± 0.010 | 0.109 ± 0.014 | 0.124 ± 0.009 | 0.114 ± 0.013 | 0.105 ± 0.009 | **0.104 ± 0.009** |
| superconductor | 0.195 ± 0.020 | 0.250 ± 0.025 | 0.210 ± 0.029 | 0.235 ± 0.027 | 0.308 ± 0.035 | **0.193 ± 0.020** | 0.193 ± 0.020 |

Table 26: Variant (c) Quantile Loss at 95% prediction intervals, aggregated across 10 seeds. Values $\geq 100$ or $< 0.01$ are presented in scientific notation with 1 decimal place. **Bold** values (desirable) are the minimum for that dataset and metric, while the underlined values indicate the second-best result. Red values are more than 33% worse than the best result.

| Dataset | CLEAR | ALEATORIC | ALEATORIC-R | PCS-EPISTEMIC | Naive | $\gamma_1 = 1$ | $\lambda = 1$ |
|---|---|---|---|---|---|---|---|
| ailerons | **9.6e-06 ± 2.9e-07** | 1.0e-05 ± 2.7e-07 | 9.9e-06 ± 3.6e-07 | 1.1e-05 ± 2.7e-07 | 1.2e-05 ± 3.7e-07 | 9.6e-06 ± 2.9e-07 | 9.6e-06 ± 2.9e-07 |
| airfoil | **0.107 ± 0.014** | 0.140 ± 0.007 | 0.129 ± 0.019 | 0.110 ± 0.013 | 0.130 ± 0.019 | 0.108 ± 0.014 | 0.108 ± 0.014 |
| allstate | **1.2e+02 ± 6.874** | 1.2e+02 ± 4.960 | 1.2e+02 ± 19.401 | 1.4e+02 ± 9.737 | 1.6e+02 ± 10.679 | 1.2e+02 ± 7.379 | 1.2e+02 ± 7.410 |
| ca_housing | **2.8e+03 ± 92.560** | 3.2e+03 ± 71.387 | 3.2e+03 ± 1.0e+02 | 2.9e+03 ± 87.120 | 3.6e+03 ± 1.2e+02 | 2.8e+03 ± 88.458 | 2.8e+03 ± 87.820 |
| computer | **0.137 ± 0.011** | 0.152 ± 0.013 | 0.160 ± 0.012 | 0.140 ± 0.012 | 0.164 ± 0.013 | 0.137 ± 0.011 | 0.138 ± 0.011 |
| concrete | 0.337 ± 0.032 | 0.374 ± 0.044 | 0.381 ± 0.059 | **0.336 ± 0.032** | 0.384 ± 0.056 | 0.336 ± 0.032 | 0.337 ± 0.032 |
| elevator | 1.3e-04 ± 3.2e-06 | 1.7e-04 ± 2.2e-06 | 1.5e-04 ± 3.3e-06 | 1.4e-04 ± 3.7e-06 | 1.6e-04 ± 3.1e-06 | 1.3e-04 ± 2.9e-06 | **1.3e-04 ± 2.9e-06** |
| energy_efficiency | **0.029 ± 0.007** | 0.068 ± 0.005 | 0.034 ± 0.010 | 0.030 ± 0.007 | 0.034 ± 0.010 | 0.030 ± 0.007 | 0.030 ± 0.007 |
| insurance | 4.2e+02 ± 80.269 | **3.9e+02 ± 33.576** | 4.0e+02 ± 53.283 | 4.4e+02 ± 65.348 | 4.1e+02 ± 41.712 | 4.3e+02 ± 77.643 | 4.5e+02 ± 67.994 |
| kin8nm | **6.7e-03 ± 1.8e-04** | 8.4e-03 ± 2.6e-04 | 6.8e-03 ± 1.9e-04 | 6.9e-03 ± 2.2e-04 | 7.4e-03 ± 2.8e-04 | 6.7e-03 ± 1.6e-04 | 6.7e-03 ± 1.7e-04 |
| miami_housing | **3.7e+03 ± 1.4e+02** | 4.2e+03 ± 1.2e+02 | 7.3e+03 ± 4.2e+02 | 3.7e+03 ± 1.2e+02 | 7.7e+03 ± 4.2e+02 | 4.0e+03 ± 2.4e+02 | 4.0e+03 ± 2.3e+02 |
| naval_propulsion | **4.0e-05 ± 1.0e-06** | 9.5e-05 ± 2.1e-06 | 4.9e-05 ± 1.6e-06 | 4.2e-05 ± 1.2e-06 | 5.7e-05 ± 1.2e-06 | 4.1e-05 ± 1.2e-06 | 4.5e-05 ± 1.6e-06 |
| parkinsons | **0.189 ± 0.006** | 0.249 ± 0.007 | 0.216 ± 0.008 | 0.191 ± 0.006 | 0.220 ± 0.008 | 0.189 ± 0.006 | 0.190 ± 0.007 |
| powerplant | 0.203 ± 0.012 | 0.213 ± 0.014 | 0.208 ± 0.012 | 0.211 ± 0.013 | 0.211 ± 0.012 | **0.203 ± 0.012** | 0.203 ± 0.012 |
| qsar | **0.048 ± 0.002** | 0.057 ± 0.003 | 0.051 ± 0.002 | 0.054 ± 0.002 | 0.055 ± 0.003 | 0.049 ± 0.002 | 0.049 ± 0.002 |
| sulfur | **1.8e-03 ± 6.3e-05** | 1.9e-03 ± 1.2e-04 | 2.7e-03 ± 1.6e-04 | 1.8e-03 ± 6.2e-05 | 2.9e-03 ± 1.7e-04 | 1.8e-03 ± 6.8e-05 | 1.8e-03 ± 7.1e-05 |
| superconductor | **0.489 ± 0.020** | 0.553 ± 0.018 | 0.568 ± 0.042 | 0.568 ± 0.025 | 0.792 ± 0.035 | 0.490 ± 0.020 | 0.489 ± 0.021 |

Table 27: Variant (c) NCIW at 95% prediction intervals, aggregated across 10 seeds.Values $\geq 100$ or $< 0.01$ are presented in scientific notation with 1 decimal place. **Bold** values (desirable) are the minimum for that dataset and metric, while the underlined values indicate the second-best result. Red values are more than 33% worse than the best result.

| Dataset | CLEAR | ALEATORIC | ALEATORIC-R | PCS-EPISTEMIC | Naive | $\gamma_1 = 1$ | $\lambda = 1$ |
|---|---|---|---|---|---|---|---|
| ailerons | $0.210 \pm 0.020$ | $0.231 \pm 0.023$ | **$0.204 \pm 0.021$** | $0.239 \pm 0.021$ | $0.230 \pm 0.021$ | $0.209 \pm 0.020$ | $0.210 \pm 0.019$ |
| airfoil | $\underline{0.188 \pm 0.014}$ | $0.258 \pm 0.034$ | $0.204 \pm 0.017$ | $0.199 \pm 0.013$ | $0.204 \pm 0.014$ | $0.190 \pm 0.012$ | **$0.188 \pm 0.010$** |
| allstate | $0.277 \pm 0.056$ | $0.316 \pm 0.065$ | $0.281 \pm 0.041$ | $0.305 \pm 0.070$ | $0.319 \pm 0.051$ | $\underline{0.271 \pm 0.054}$ | **$0.270 \pm 0.053$** |
| ca_housing | $0.311 \pm 0.010$ | $0.419 \pm 0.014$ | $0.343 \pm 0.009$ | $0.329 \pm 0.009$ | $0.401 \pm 0.018$ | **$0.309 \pm 0.009$** | $\underline{0.309 \pm 0.009}$ |
| computer | $0.081 \pm 0.004$ | $0.104 \pm 0.007$ | $0.090 \pm 0.004$ | $0.084 \pm 0.005$ | $0.092 \pm 0.005$ | **$0.079 \pm 0.004$** | $\underline{0.080 \pm 0.004}$ |
| concrete | $0.259 \pm 0.032$ | $0.303 \pm 0.054$ | $0.263 \pm 0.029$ | **$0.257 \pm 0.031$** | $0.265 \pm 0.027$ | $\underline{0.258 \pm 0.033}$ | $0.259 \pm 0.032$ |
| elevator | **$0.126 \pm 0.008$** | $0.192 \pm 0.008$ | $0.138 \pm 0.008$ | $0.151 \pm 0.011$ | $0.144 \pm 0.010$ | $\underline{0.127 \pm 0.007}$ | $0.127 \pm 0.007$ |
| energy_efficiency | **$0.041 \pm 0.006$** | $0.116 \pm 0.015$ | $0.048 \pm 0.005$ | $0.043 \pm 0.005$ | $0.049 \pm 0.005$ | $\underline{0.042 \pm 0.006}$ | $0.042 \pm 0.006$ |
| insurance | $\underline{0.345 \pm 0.078}$ | $0.413 \pm 0.043$ | $0.424 \pm 0.074$ | **$0.343 \pm 0.087$** | $0.385 \pm 0.112$ | $0.367 \pm 0.100$ | $0.407 \pm 0.137$ |
| kin8nm | **$0.323 \pm 0.008$** | $0.420 \pm 0.018$ | $0.327 \pm 0.008$ | $0.331 \pm 0.010$ | $0.341 \pm 0.012$ | $0.325 \pm 0.009$ | $\underline{0.324 \pm 0.009}$ |
| miami_housing | $0.088 \pm 0.002$ | $0.113 \pm 0.004$ | $0.111 \pm 0.007$ | $0.094 \pm 0.003$ | $0.122 \pm 0.007$ | $\underline{0.077 \pm 0.003}$ | **$0.076 \pm 0.003$** |
| naval_propulsion | $1.6e\text{-}03 \pm 5.0e\text{-}05$ | $4.0e\text{-}03 \pm 2.4e\text{-}04$ | $1.6e\text{-}03 \pm 4.1e\text{-}05$ | $1.7e\text{-}03 \pm 4.7e\text{-}05$ | $1.8e\text{-}03 \pm 5.8e\text{-}05$ | **$1.6e\text{-}03 \pm 5.2e\text{-}05$** | $1.6e\text{-}03 \pm 6.2e\text{-}05$ |
| parkinsons | $\underline{0.250 \pm 0.012}$ | $0.366 \pm 0.017$ | $0.279 \pm 0.013$ | $0.259 \pm 0.009$ | $0.284 \pm 0.013$ | $0.251 \pm 0.011$ | **$0.250 \pm 0.011$** |
| powerplant | **$0.161 \pm 0.007$** | $0.176 \pm 0.009$ | $0.163 \pm 0.008$ | $0.171 \pm 0.008$ | $0.166 \pm 0.007$ | $0.161 \pm 0.007$ | $\underline{0.161 \pm 0.007}$ |
| qsar | **$0.348 \pm 0.118$** | $0.465 \pm 0.156$ | $0.361 \pm 0.124$ | $0.407 \pm 0.142$ | $0.395 \pm 0.134$ | $0.359 \pm 0.124$ | $\underline{0.354 \pm 0.122}$ |
| sulfur | $0.109 \pm 0.008$ | $0.124 \pm 0.008$ | $0.108 \pm 0.006$ | $0.124 \pm 0.008$ | $0.115 \pm 0.007$ | $\underline{0.104 \pm 0.007}$ | **$0.102 \pm 0.007$** |
| superconductor | $0.192 \pm 0.019$ | $0.251 \pm 0.025$ | $0.206 \pm 0.027$ | $0.232 \pm 0.025$ | $0.297 \pm 0.029$ | **$0.191 \pm 0.019$** | $\underline{0.191 \pm 0.019}$ |

Table 28: Variant (c) Interval Score Loss at 95% prediction intervals, aggregated across 10 seeds.Values $\geq 100$ or $< 0.01$ are presented in scientific notation with 1 decimal place. **Bold** values (desirable) are the minimum for that dataset and metric, while the underlined values indicate the second-best result. Red values are more than 33% worse than the best result.

| Dataset | CLEAR | ALEATORIC | ALEATORIC-R | PCS-EPISTEMIC | Naive | $\gamma_1 = 1$ | $\lambda = 1$ |
|---|---|---|---|---|---|---|---|
| ailerons | **$7.7e\text{-}04 \pm 2.3e\text{-}05$** | $8.2e\text{-}04 \pm 2.2e\text{-}05$ | $7.9e\text{-}04 \pm 2.9e\text{-}05$ | $8.5e\text{-}04 \pm 2.2e\text{-}05$ | $9.4e\text{-}04 \pm 3.0e\text{-}05$ | $\underline{7.7e\text{-}04 \pm 2.3e\text{-}05}$ | $7.7e\text{-}04 \pm 2.3e\text{-}05$ |
| airfoil | **$8.569 \pm 1.099$** | $11.217 \pm 0.549$ | $10.341 \pm 1.548$ | $8.799 \pm 1.034$ | $10.361 \pm 1.499$ | $\underline{8.675 \pm 1.116}$ | $8.624 \pm 1.105$ |
| allstate | **$9.2e\text{+}03 \pm 5.5e\text{+}02$** | $9.6e\text{+}03 \pm 4.0e\text{+}02$ | $1.0e\text{+}04 \pm 1.6e\text{+}03$ | $1.1e\text{+}04 \pm 7.8e\text{+}02$ | $1.3e\text{+}04 \pm 8.5e\text{+}02$ | $9.2e\text{+}03 \pm 5.9e\text{+}02$ | $9.3e\text{+}03 \pm 5.9e\text{+}02$ |
| ca_housing | **$2.2e\text{+}05 \pm 7.4e\text{+}03$** | $2.5e\text{+}05 \pm 5.7e\text{+}03$ | $2.5e\text{+}05 \pm 8.1e\text{+}03$ | $2.3e\text{+}05 \pm 7.0e\text{+}03$ | $2.9e\text{+}05 \pm 9.7e\text{+}03$ | $\underline{2.2e\text{+}05 \pm 7.1e\text{+}03}$ | $2.2e\text{+}05 \pm 7.0e\text{+}03$ |
| computer | $10.978 \pm 0.911$ | $12.134 \pm 1.001$ | $12.827 \pm 0.945$ | $11.204 \pm 0.996$ | $13.130 \pm 1.051$ | $\underline{10.986 \pm 0.873}$ | $11.030 \pm 0.866$ |
| concrete | $26.928 \pm 2.529$ | $29.955 \pm 3.544$ | $30.449 \pm 4.693$ | **$26.855 \pm 2.535$** | $30.720 \pm 4.505$ | $\underline{26.882 \pm 2.568}$ | $26.962 \pm 2.581$ |
| elevator | $0.010 \pm 0.000$ | $0.014 \pm 0.000$ | $0.012 \pm 0.000$ | $0.011 \pm 0.000$ | $0.013 \pm 0.000$ | $\underline{0.010 \pm 0.000}$ | **$0.010 \pm 0.000$** |
| energy_efficiency | **$2.348 \pm 0.554$** | $5.407 \pm 0.418$ | $2.759 \pm 0.822$ | $2.397 \pm 0.573$ | $2.756 \pm 0.822$ | $\underline{2.392 \pm 0.564}$ | $2.395 \pm 0.588$ |
| insurance | $3.4e\text{+}04 \pm 6.4e\text{+}03$ | **$3.1e\text{+}04 \pm 2.7e\text{+}03$** | $\underline{3.2e\text{+}04 \pm 4.3e\text{+}03}$ | $3.5e\text{+}04 \pm 5.2e\text{+}03$ | $3.3e\text{+}04 \pm 3.3e\text{+}03$ | $3.4e\text{+}04 \pm 6.2e\text{+}03$ | $3.6e\text{+}04 \pm 5.4e\text{+}03$ |
| kin8nm | **$0.534 \pm 0.014$** | $0.673 \pm 0.021$ | $0.544 \pm 0.015$ | $0.552 \pm 0.017$ | $0.591 \pm 0.022$ | $0.536 \pm 0.013$ | $\underline{0.535 \pm 0.013}$ |
| miami_housing | **$2.9e\text{+}05 \pm 1.1e\text{+}04$** | $3.3e\text{+}05 \pm 9.5e\text{+}03$ | $5.9e\text{+}05 \pm 3.4e\text{+}04$ | $3.0e\text{+}05 \pm 9.6e\text{+}03$ | $6.1e\text{+}05 \pm 3.4e\text{+}04$ | $3.2e\text{+}05 \pm 1.9e\text{+}04$ | $3.2e\text{+}05 \pm 1.9e\text{+}04$ |
| naval_propulsion | **$3.2e\text{-}03 \pm 8.4e\text{-}05$** | $7.6e\text{-}03 \pm 1.7e\text{-}04$ | $3.9e\text{-}03 \pm 1.3e\text{-}04$ | $3.4e\text{-}03 \pm 9.6e\text{-}05$ | $4.6e\text{-}03 \pm 9.6e\text{-}05$ | $\underline{3.2e\text{-}03 \pm 9.3e\text{-}05}$ | $3.6e\text{-}03 \pm 1.3e\text{-}04$ |
| parkinsons | $15.149 \pm 0.518$ | $19.899 \pm 0.576$ | $17.316 \pm 0.634$ | $15.312 \pm 0.459$ | $17.596 \pm 0.621$ | $\underline{15.150 \pm 0.499}$ | $15.161 \pm 0.537$ |
| powerplant | $16.243 \pm 0.985$ | $17.062 \pm 1.128$ | $16.673 \pm 0.987$ | $16.846 \pm 1.029$ | $16.874 \pm 0.975$ | **$16.231 \pm 0.984$** | $\underline{16.231 \pm 0.983}$ |
| qsar | **$3.877 \pm 0.176$** | $4.521 \pm 0.225$ | $4.086 \pm 0.193$ | $4.334 \pm 0.125$ | $4.390 \pm 0.205$ | $3.927 \pm 0.146$ | $\underline{3.892 \pm 0.151}$ |
| sulfur | **$0.143 \pm 0.005$** | $0.156 \pm 0.010$ | $0.213 \pm 0.013$ | $0.146 \pm 0.005$ | $0.228 \pm 0.013$ | $\underline{0.143 \pm 0.005}$ | $0.145 \pm 0.006$ |
| superconductor | **$39.104 \pm 1.604$** | $44.230 \pm 1.428$ | $45.479 \pm 3.358$ | $45.456 \pm 2.039$ | $63.347 \pm 2.824$ | $39.224 \pm 1.613$ | $\underline{39.119 \pm 1.642}$ |

Table 29: Variant (c) CLEAR calibration parameters $\lambda$ and $\gamma_1$ for 95% prediction intervals across 10 seeds. Using all available variables. Showing median [min:max] values.

| Dataset | $\lambda$ | $\gamma_1$ |
|---|---|---|
| ailerons | 0.89 [0.17:4.47] | 0.98 [0.87:1.58] |
| airfoil | 0.76 [0.23:5.25] | 1.70 [0.25:4.30] |
| allstate | 0.21 [0.00:14.06] | 1.14 [0.16:1.93] |
| ca_housing | 1.90 [1.14:5.08] | 0.70 [0.33:0.93] |
| computer | 2.21 [1.63:8.87] | 0.74 [0.21:0.96] |
| concrete | 1.33 [0.06:100.00] | 1.65 [0.01:16.73] |
| elevator | 0.94 [0.67:1.43] | 1.04 [0.82:1.22] |
| energy_efficiency | 0.12 [0.01:100.00] | 7.96 [0.02:24.25] |
| insurance | 5.07 [0.14:100.00] | 0.38 [0.02:27.45] |
| kin8nm | 1.87 [1.64:3.82] | 0.76 [0.58:0.83] |
| miami_housing | 7.06 [4.84:19.45] | 0.23 [0.09:0.33] |
| naval_propulsion | 16.16 [12.48:21.17] | 0.63 [0.53:0.76] |
| parkinsons | 1.16 [0.72:8.42] | 1.26 [0.22:1.84] |
| powerplant | 1.05 [0.73:3.26] | 1.05 [0.51:1.36] |
| qsar | 0.51 [0.32:1.70] | 1.45 [0.96:2.37] |
| sulfur | 2.41 [1.53:7.16] | 0.71 [0.29:0.91] |
| superconductor | 0.91 [0.34:1.39] | 1.07 [0.80:1.34] |

## F.4    COMPARING VARIANTS OF CLEAR

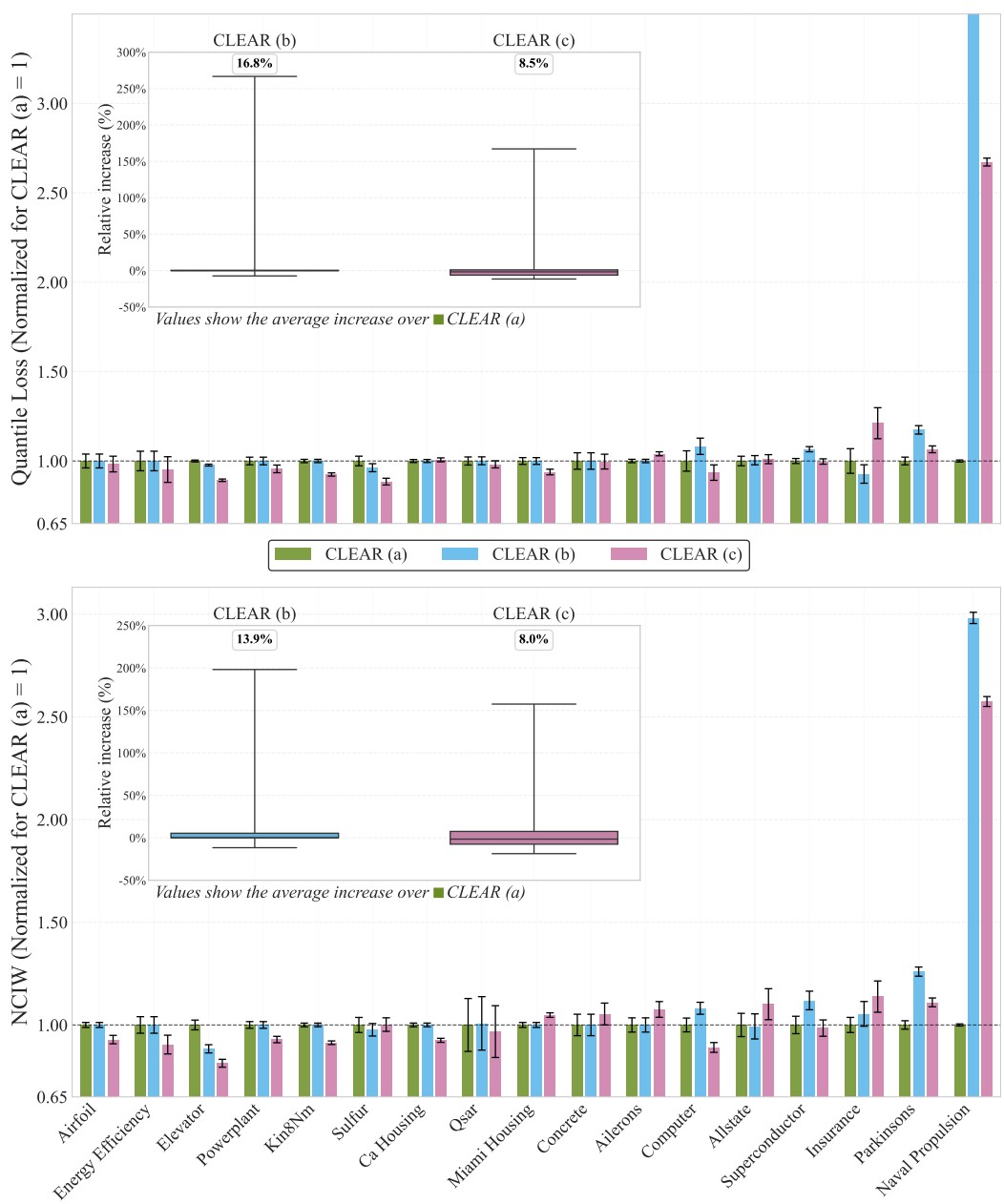

Figure 9: Quantile loss and NCIW performance of different variants of CLEAR (a, b and c) over 10 seeds normalized relative to CLEAR (a) (baseline = 1.0). Lower values indicate better performance. The inset boxplot shows the % improvement relative to the CLEAR (a) baseline $\pm 1\sigma$. Values inside each subplot represent the mean improvement across all datasets.

## F.5    RUNTIME

The grid search for finding the optimal parameter is extremely fast and negligible compared to the baselines, despite using a grid with 4000 points, which is finer and larger than necessary. In Tables 30 to 32, we provide the average training time for each variant on a given real-world dataset, computed on a machine with an Intel® Core™ i9-13900KF CPU (with maximum 20+ threads used in parallel). For experiments involving SQR and DEs, we had access to an NVIDIA® GeForce RTX™ 4090 GPU.

All results are computed using 32 GB of memory. The times are provided in seconds and have been averaged over 10 seeds. For CLEAR, the required computation is the calibration time (computation of $\lambda$ and $\gamma_1$, denoted as *Grid Search* in the tables). Only the runtimes for the methods included in the paper are provided. The experiments were run exactly as described in the paper (particularly for the 100 bootstraps).

As stated previously, CLEAR is highly modular, allowing the modules and their components (such as the choice of base models within the PCS module) to be replaced or modified to adhere to computational budget limitations. For example, if the dataset is very large, base models that scale well with the training data size can be used (such as deep learning models, trained via stochastic gradient descent). Alternatively, if training each model is expensive, one might want to use fewer than 100 different bootstraps per dataset, or maybe even use techniques (such as Monte-Carlo Dropout) to obtain an ensemble from a single trained model (Kendall & Gal, 2017; Gal & Ghahramani, 2016; Wen et al., 2020; Havasi et al., 2021; Rossellini et al., 2024; Chan et al., 2025; Agarwal et al., 2025).

Note that ALEATORIC-R cannot be computed without first computing PCS, as it utilizes the residuals from PCS. In practice, the total computational costs of ALEATORIC-R would be equal to the costs of PCS plus the additional costs of the residual approach. Thus, the minimal total computational costs necessary to obtain CLEAR results are equal to the minimal total computational costs required to obtain ALEATORIC-R results (up to a few seconds of the grid search).

*Remark* F.1 (Parallelization and Scalability). Every step in the entire CLEAR pipeline is parallelizable and distributable. The most expensive part is fitting multiple models to multiple bootstraps of the data for PCS. However, these models can be trained perfectly in parallel on different distributed nodes, as they do not have to communicate with each other. Similar parallelization (distributed) is also possible for ALEATORIC-R, where multiple models are fitted independently. The computational costs of CLEAR's calibration step (*Grid-Search*), albeit negligible, could be further reduced by distributing the exploration of the grid across multiple servers. The computations necessary for each grid point are fully vectorized and are highly suited for GPUs (in the case of a large calibration dataset). In other words, regardless of whether the user has access to many weak CPU servers, a powerful GPU, or multiple powerful GPU servers, the calibration step can efficiently utilize all these different infrastructures (when tuning the implementation accordingly). For all these reasons, CLEAR is highly scalable.

◇

Table 30: Variant (a) average runtime (over 10 seeds) in seconds for base components and the CLEAR's grid search. The grid search includes any other overhead for CLEAR.

| Dataset | PCS | ALEATORIC-R | Grid-Search | Total |
|---|---|---|---|---|
| ailerons | 13.46 | 6.80 | 0.27 | 20.53 |
| airfoil | 3.10 | 1.92 | 0.17 | 5.19 |
| allstate | 1124.47 | 22.93 | 0.19 | 1147.59 |
| ca_housing | 8.51 | 7.79 | 0.54 | 16.84 |
| computer | 61.92 | 76.09 | 0.16 | 138.16 |
| concrete | 1.10 | 2.16 | 0.17 | 3.43 |
| elevator | 18.79 | 122.16 | 0.60 | 141.54 |
| energy_efficiency | 0.81 | 1.65 | 0.16 | 2.62 |
| insurance | 3.48 | 17.23 | 0.11 | 20.82 |
| kin8nm | 3.25 | 5.43 | 0.21 | 8.89 |
| miami_housing | 7.31 | 7.66 | 0.72 | 15.68 |
| naval_propulsion | 19.83 | 81.32 | 0.29 | 101.44 |
| parkinsons | 39.53 | 60.99 | 0.14 | 100.66 |
| powerplant | 3.08 | 4.92 | 0.21 | 8.22 |
| qsar | 348.59 | 10.00 | 0.20 | 358.78 |
| sulfur | 21.55 | 16.56 | 0.21 | 38.32 |
| superconductor | 561.84 | 795.45 | 12.70 | 1369.99 |
| **Total** | **2240.60** | **1241.06** | **17.06** | **3498.71** |

Table 31: Variant (b) average runtime (over 10 seeds) in seconds for base components and the CLEAR's grid search. The grid search includes any other overhead for CLEAR.

| Dataset | PCS | ALEATORIC-R | Grid-Search | Total |
|---|---|---|---|---|
| ailerons | 4.51 | 6.89 | 0.26 | 11.66 |
| airfoil | 0.78 | 1.64 | 0.16 | 2.59 |
| allstate | 27.73 | 14.22 | 0.20 | 42.15 |
| ca_housing | 5.42 | 7.69 | 0.39 | 13.50 |
| computer | 3.46 | 5.97 | 0.21 | 9.65 |
| concrete | 0.86 | 1.82 | 0.17 | 2.85 |
| elevator | 4.43 | 6.71 | 0.79 | 11.93 |
| energy_efficiency | 0.68 | 1.40 | 0.17 | 2.25 |
| insurance | 0.80 | 1.65 | 0.17 | 2.62 |
| kin8nm | 2.61 | 4.96 | 0.21 | 7.78 |
| miami_housing | 4.61 | 7.70 | 0.70 | 13.01 |
| naval_propulsion | 3.72 | 6.38 | 0.55 | 10.64 |
| parkinsons | 2.71 | 5.37 | 0.20 | 8.28 |
| powerplant | 2.73 | 4.32 | 0.22 | 7.26 |
| qsar | 5.00 | 9.67 | 0.20 | 14.86 |
| sulfur | 2.89 | 4.64 | 0.23 | 7.76 |
| superconductor | 13.81 | 22.74 | 1.00 | 37.55 |
| **Total** | **86.77** | **113.76** | **5.81** | **206.34** |

Table 32: Variant (c) average runtime (over 10 seeds) in seconds for base components and the CLEAR's grid search. The grid search includes any other overhead for CLEAR.

| Dataset | PCS | ALEATORIC-R | Grid-Search | Total |
|---|---|---|---|---|
| ailerons | 10.20 | 104.32 | 0.65 | 115.17 |
| airfoil | 0.88 | 17.82 | 0.11 | 18.81 |
| allstate | 651.02 | 125.77 | 0.14 | 776.93 |
| ca_housing | 3.63 | 130.87 | 4.16 | 138.66 |
| computer | 4.55 | 65.28 | 0.16 | 69.98 |
| concrete | 1.16 | 17.49 | 0.11 | 18.76 |
| elevator | 3.25 | 98.73 | 1.97 | 103.95 |
| energy_efficiency | 0.95 | 15.42 | 0.10 | 16.47 |
| insurance | 0.90 | 19.80 | 0.11 | 20.81 |
| kin8nm | 16.89 | 69.17 | 0.16 | 86.22 |
| miami_housing | 5.34 | 123.32 | 0.92 | 129.58 |
| naval_propulsion | 1.14 | 115.41 | 0.74 | 117.28 |
| parkinsons | 3.71 | 53.43 | 0.14 | 57.27 |
| powerplant | 2.37 | 60.99 | 0.17 | 63.52 |
| qsar | 21.48 | 93.84 | 0.14 | 115.46 |
| sulfur | 2.43 | 67.41 | 0.17 | 70.01 |
| superconductor | 60.63 | 644.13 | 8.10 | 712.86 |
| **Total** | **790.52** | **1823.18** | **18.05** | **2631.75** |

# G  CONFORMALIZED CLEAR WITH PCS AND CQR: RESULTS ON REAL-WORLD DATA

This section presents results from our conformalized experimental configuration, where we split the 20% validation set into separate 10% validation and 10% calibration sets. This approach provides stronger finite-sample distribution-free marginal coverage guarantees following conformal prediction principles, as the calibration set remains completely unseen during model selection and hyperparameter optimization. While the conformalized approach may sacrifice some performance due to reduced data availability for validation, it offers theoretical rigor by ensuring the conformal calibration step operates on the held-out data. Similar to the standard results, CLEAR adapts to this more stringent experimental setting while maintaining its advantages over baseline methods across all three variants (a), (b), and (c).

## G.1  VARIANT (A) CONFORMALIZED

Table 33: Conformalized Variant (a) PICP at 95% prediction intervals, aggregated across 10 seeds. Methods with suffix '-c' denote conformalized variants obtained using the validation set divided into two parts, one for validation and one for calibration.

| Dataset | CLEAR-c | PCS-EPISTEMIC-c | ALEATORIC-R-c | CLEAR |
|---|---|---|---|---|
| ailerons | $0.95 \pm 0.01$ | $0.95 \pm 0.01$ | $0.95 \pm 0.01$ | $0.95 \pm 0.00$ |
| airfoil | $0.96 \pm 0.01$ | $0.95 \pm 0.02$ | $0.95 \pm 0.02$ | $0.95 \pm 0.01$ |
| allstate | $0.96 \pm 0.01$ | $0.94 \pm 0.01$ | $0.95 \pm 0.01$ | $0.95 \pm 0.01$ |
| ca_housing | $0.95 \pm 0.01$ | $0.95 \pm 0.01$ | $0.95 \pm 0.01$ | $0.95 \pm 0.01$ |
| computer | $0.95 \pm 0.01$ | $0.95 \pm 0.01$ | $0.96 \pm 0.01$ | $0.95 \pm 0.01$ |
| concrete | $0.95 \pm 0.02$ | $0.95 \pm 0.02$ | $0.94 \pm 0.02$ | $0.95 \pm 0.02$ |
| elevator | $0.95 \pm 0.00$ | $0.95 \pm 0.01$ | $0.95 \pm 0.01$ | $0.95 \pm 0.00$ |
| energy_efficiency | $0.97 \pm 0.02$ | $0.95 \pm 0.02$ | $0.96 \pm 0.02$ | $0.95 \pm 0.02$ |
| insurance | $0.96 \pm 0.02$ | $0.95 \pm 0.02$ | $0.96 \pm 0.02$ | $0.95 \pm 0.02$ |
| kin8nm | $0.96 \pm 0.01$ | $0.95 \pm 0.01$ | $0.95 \pm 0.01$ | $0.95 \pm 0.01$ |
| miami_housing | $0.95 \pm 0.01$ | $0.95 \pm 0.01$ | $0.95 \pm 0.01$ | $0.95 \pm 0.01$ |
| naval_propulsion | $0.96 \pm 0.01$ | $0.95 \pm 0.01$ | $0.95 \pm 0.01$ | $0.95 \pm 0.01$ |
| parkinsons | $0.95 \pm 0.01$ | $0.95 \pm 0.01$ | $0.95 \pm 0.01$ | $0.95 \pm 0.01$ |
| powerplant | $0.95 \pm 0.01$ | $0.95 \pm 0.01$ | $0.95 \pm 0.01$ | $0.95 \pm 0.01$ |
| qsar | $0.95 \pm 0.01$ | $0.95 \pm 0.01$ | $0.95 \pm 0.01$ | $0.95 \pm 0.01$ |
| sulfur | $0.95 \pm 0.01$ | $0.96 \pm 0.01$ | $0.95 \pm 0.01$ | $0.95 \pm 0.01$ |
| superconductor | $0.95 \pm 0.01$ | $0.95 \pm 0.01$ | $0.95 \pm 0.01$ | $0.95 \pm 0.00$ |

Table 34: Conformalized Variant (a) NIW at 95% prediction intervals, aggregated across 10 seeds. Methods with suffix '-c' denote conformalized variants obtained using the validation set divided into two parts, one for validation and one for calibration. Values $\geq 100$ or $< 0.01$ are presented in scientific notation with 1 decimal place. **Bold** values (desirable) are the minimum for that dataset and metric, while the underlined values indicate the second-best result. Red values are more than 33% worse than the best result.

| Dataset | CLEAR-c | PCS-EPISTEMIC-c | ALEATORIC-R-c | CLEAR |
|---|---|---|---|---|
| ailerons | **0.193 ± 0.016** | 0.217 ± 0.017 | 0.195 ± 0.018 | 0.193 ± 0.016 |
| airfoil | 0.215 ± 0.019 | 0.217 ± 0.022 | 0.227 ± 0.031 | **0.207 ± 0.012** |
| allstate | 0.267 ± 0.041 | 0.263 ± 0.060 | **0.259 ± 0.046** | 0.260 ± 0.037 |
| ca_housing | 0.339 ± 0.006 | 0.355 ± 0.012 | 0.349 ± 0.008 | **0.339 ± 0.008** |
| computer | **0.091 ± 0.006** | 8.020 ± 11.958 | 0.094 ± 0.016 | 0.091 ± 0.008 |
| concrete | 0.272 ± 0.030 | 0.260 ± 0.050 | 0.266 ± 0.038 | **0.249 ± 0.025** |
| elevator | 0.157 ± 0.009 | 0.177 ± 0.014 | **0.149 ± 0.008** | 0.156 ± 0.009 |
| energy_efficiency | 0.053 ± 0.009 | 0.058 ± 0.009 | 0.053 ± 0.009 | **0.047 ± 0.005** |
| insurance | 0.338 ± 0.063 | 0.535 ± 0.315 | 0.379 ± 0.089 | **0.309 ± 0.032** |
| kin8nm | 0.362 ± 0.014 | 0.373 ± 0.013 | 0.370 ± 0.011 | **0.360 ± 0.012** |
| miami_housing | 0.085 ± 0.003 | 0.097 ± 0.002 | 0.087 ± 0.005 | **0.085 ± 0.002** |
| naval_propulsion | 6.2e-04 ± 5.9e-06 | 7.1e-04 ± 1.5e-05 | **6.0e-04 ± 9.6e-06** | 6.1e-04 ± 6.3e-06 |
| parkinsons | **0.227 ± 0.017** | 0.315 ± 0.034 | 0.253 ± 0.020 | 0.227 ± 0.010 |
| powerplant | 0.171 ± 0.010 | 0.179 ± 0.008 | 0.183 ± 0.008 | **0.170 ± 0.007** |
| qsar | 0.369 ± 0.123 | 0.388 ± 0.132 | 0.381 ± 0.125 | **0.363 ± 0.121** |
| sulfur | 0.112 ± 0.009 | 0.136 ± 0.014 | **0.108 ± 0.010** | 0.109 ± 0.010 |
| superconductor | 0.197 ± 0.022 | 0.249 ± 0.032 | **0.195 ± 0.023** | 0.196 ± 0.022 |

Table 35: Conformalized Variant (a) Quantile Loss at 95% prediction intervals, aggregated across 10 seeds. Methods with suffix '-c' denote conformalized variants obtained using the validation set divided into two parts, one for validation and one for calibration. Values $\geq 100$ or $< 0.01$ are presented in scientific notation with 1 decimal place. **Bold** values (desirable) are the minimum for that dataset and metric, while the underlined values indicate the second-best result. Red values are more than 33% worse than the best result.

| Dataset | CLEAR-c | PCS-EPISTEMIC-c | ALEATORIC-R-c | CLEAR |
|---|---|---|---|---|
| ailerons | 9.2e-06 ± 2.3e-07 | 1.0e-05 ± 2.1e-07 | 9.7e-06 ± 2.7e-07 | **9.2e-06 ± 2.3e-07** |
| airfoil | 0.110 ± 0.013 | 0.119 ± 0.012 | 0.126 ± 0.015 | **0.109 ± 0.011** |
| allstate | 1.1e+02 ± 8.530 | 1.3e+02 ± 8.381 | 1.3e+02 ± 7.926 | **1.1e+02 ± 7.835** |
| ca_housing | **2.8e+03 ± 56.920** | 3.2e+03 ± 91.466 | 3.0e+03 ± 73.272 | 2.8e+03 ± 59.700 |
| computer | **0.142 ± 0.013** | 9.961 ± 14.797 | 0.157 ± 0.021 | 0.147 ± 0.022 |
| concrete | 0.342 ± 0.039 | **0.337 ± 0.047** | 0.355 ± 0.047 | 0.338 ± 0.040 |
| elevator | 1.5e-04 ± 2.0e-06 | 1.7e-04 ± 6.3e-06 | **1.5e-04 ± 2.5e-06** | 1.5e-04 ± 2.0e-06 |
| energy_efficiency | 0.033 ± 0.005 | 0.038 ± 0.003 | 0.034 ± 0.005 | **0.031 ± 0.004** |
| insurance | 3.5e+02 ± 55.800 | 5.8e+02 ± 1.7e+02 | 3.8e+02 ± 49.983 | **3.5e+02 ± 61.616** |
| kin8nm | 7.2e-03 ± 1.7e-04 | 7.6e-03 ± 1.1e-04 | 7.7e-03 ± 2.7e-04 | **7.2e-03 ± 1.7e-04** |
| miami_housing | 3.9e+03 ± 2.1e+02 | 4.2e+03 ± 1.7e+02 | 5.2e+03 ± 3.4e+02 | **3.9e+03 ± 1.9e+02** |
| naval_propulsion | 1.5e-05 ± 1.9e-07 | 1.7e-05 ± 2.9e-07 | 1.5e-05 ± 2.7e-07 | **1.5e-05 ± 2.1e-07** |
| parkinsons | 0.179 ± 0.008 | 0.226 ± 0.016 | 0.216 ± 0.012 | **0.178 ± 0.010** |
| powerplant | 0.213 ± 0.011 | 0.222 ± 0.011 | 0.220 ± 0.011 | **0.212 ± 0.012** |
| qsar | 0.050 ± 0.003 | 0.053 ± 0.003 | 0.052 ± 0.003 | **0.049 ± 0.003** |
| sulfur | 2.0e-03 ± 1.7e-04 | 2.4e-03 ± 2.2e-04 | 2.3e-03 ± 1.6e-04 | **2.0e-03 ± 1.4e-04** |
| superconductor | 0.490 ± 0.018 | 0.609 ± 0.021 | 0.538 ± 0.023 | **0.490 ± 0.018** |

Table 36: Conformalized Variant (a) NCIW at 95% prediction intervals, aggregated across 10 seeds. Methods with suffix '-c' denote conformalized variants obtained using the validation set divided into two parts, one for validation and one for calibration. Values $\geq 100$ or $< 0.01$ are presented in scientific notation with 1 decimal place. **Bold** values (desirable) are the minimum for that dataset and metric, while the underlined values indicate the second-best result. Red values are more than 33% worse than the best result.

| Dataset | CLEAR-c | PCS-EPISTEMIC-c | ALEATORIC-R-c | CLEAR |
|---|---|---|---|---|
| ailerons | 0.195 ± 0.018 | 0.217 ± 0.021 | 0.196 ± 0.017 | **0.195 ± 0.017** |
| airfoil | 0.203 ± 0.007 | 0.211 ± 0.007 | 0.216 ± 0.011 | **0.203 ± 0.006** |
| allstate | 0.255 ± 0.040 | 0.272 ± 0.057 | 0.263 ± 0.043 | **0.252 ± 0.037** |
| ca_housing | 0.336 ± 0.006 | 0.354 ± 0.012 | 0.348 ± 0.011 | **0.336 ± 0.008** |
| computer | 0.091 ± 0.008 | 0.108 ± 0.016 | **0.090 ± 0.008** | 0.091 ± 0.008 |
| concrete | 0.262 ± 0.038 | 0.259 ± 0.047 | 0.264 ± 0.040 | **0.246 ± 0.033** |
| elevator | 0.155 ± 0.010 | 0.177 ± 0.011 | **0.148 ± 0.008** | 0.155 ± 0.010 |
| energy_efficiency | 0.046 ± 0.006 | 0.057 ± 0.004 | 0.050 ± 0.005 | **0.046 ± 0.005** |
| insurance | **0.299 ± 0.037** | 0.480 ± 0.181 | 0.312 ± 0.036 | 0.303 ± 0.029 |
| kin8nm | 0.355 ± 0.008 | 0.371 ± 0.009 | 0.371 ± 0.009 | **0.354 ± 0.008** |
| miami_housing | 0.084 ± 0.004 | 0.098 ± 0.002 | 0.085 ± 0.003 | **0.084 ± 0.003** |
| naval_propulsion | 6.1e-04 ± 7.3e-06 | 7.1e-04 ± 1.5e-05 | **6.0e-04 ± 5.4e-06** | 6.1e-04 ± 7.9e-06 |
| parkinsons | 0.226 ± 0.012 | 0.316 ± 0.032 | 0.248 ± 0.007 | **0.225 ± 0.012** |
| powerplant | 0.173 ± 0.008 | 0.182 ± 0.007 | 0.182 ± 0.007 | **0.173 ± 0.007** |
| qsar | 0.361 ± 0.120 | 0.389 ± 0.134 | 0.389 ± 0.131 | **0.360 ± 0.119** |
| sulfur | 0.110 ± 0.007 | 0.127 ± 0.008 | **0.105 ± 0.005** | 0.109 ± 0.010 |
| superconductor | 0.195 ± 0.021 | 0.247 ± 0.028 | **0.194 ± 0.021** | 0.195 ± 0.021 |

Table 37: Conformalized Variant (a) Interval Score Loss at 95% prediction intervals, aggregated across 10 seeds. Methods with suffix '-c' denote conformalized variants obtained using the validation set divided into two parts, one for validation and one for calibration. Values $\geq 100$ or $< 0.01$ are presented in scientific notation with 1 decimal place. **Bold** values (desirable) are the minimum for that dataset and metric, while the underlined values indicate the second-best result. Red values are more than 33% worse than the best result.

| Dataset | CLEAR-c | PCS-EPISTEMIC-c | ALEATORIC-R-c | CLEAR |
|---|---|---|---|---|
| ailerons | 7.4e-04 ± 1.8e-05 | 8.1e-04 ± 1.7e-05 | 7.8e-04 ± 2.2e-05 | **7.4e-04 ± 1.8e-05** |
| airfoil | 8.794 ± 1.047 | 9.523 ± 0.959 | 10.075 ± 1.227 | **8.717 ± 0.870** |
| allstate | 9.2e+03 ± 6.8e+02 | 1.0e+04 ± 6.7e+02 | 1.0e+04 ± 6.3e+02 | **9.1e+03 ± 6.3e+02** |
| ca_housing | **2.2e+05 ± 4.6e+03** | 2.6e+05 ± 7.3e+03 | 2.4e+05 ± 5.9e+03 | 2.2e+05 ± 4.8e+03 |
| computer | **11.388 ± 1.051** | 8.0e+02 ± 1.2e+03 | 12.535 ± 1.698 | 11.741 ± 1.723 |
| concrete | 27.347 ± 3.094 | **26.963 ± 3.739** | 28.402 ± 3.793 | 27.020 ± 3.202 |
| elevator | 0.012 ± 0.000 | 0.014 ± 0.001 | **0.012 ± 0.000** | 0.012 ± 0.000 |
| energy_efficiency | 2.609 ± 0.417 | 3.019 ± 0.237 | 2.681 ± 0.431 | **2.465 ± 0.347** |
| insurance | 2.8e+04 ± 4.5e+03 | 4.6e+04 ± 1.3e+04 | 3.0e+04 ± 4.0e+03 | **2.8e+04 ± 4.9e+03** |
| kin8nm | 0.578 ± 0.014 | 0.607 ± 0.009 | 0.615 ± 0.022 | **0.577 ± 0.014** |
| miami_housing | 3.2e+05 ± 1.7e+04 | 3.3e+05 ± 1.4e+04 | 4.2e+05 ± 2.7e+04 | **3.1e+05 ± 1.5e+04** |
| naval_propulsion | 1.2e-03 ± 1.6e-05 | 1.4e-03 ± 2.3e-05 | 1.2e-03 ± 2.1e-05 | **1.2e-03 ± 1.7e-05** |
| parkinsons | 14.332 ± 0.644 | 18.091 ± 1.263 | 17.266 ± 0.988 | **14.221 ± 0.792** |
| powerplant | 17.066 ± 0.919 | 17.724 ± 0.919 | 17.567 ± 0.899 | **16.993 ± 0.937** |
| qsar | 3.967 ± 0.222 | 4.251 ± 0.212 | 4.194 ± 0.234 | **3.951 ± 0.230** |
| sulfur | 0.161 ± 0.013 | 0.189 ± 0.018 | 0.182 ± 0.013 | **0.161 ± 0.011** |
| superconductor | 39.234 ± 1.418 | 48.702 ± 1.695 | 43.062 ± 1.835 | **39.217 ± 1.419** |

## G.2 VARIANT (B) CONFORMALIZED

Table 38: Conformalized Variant (b) PICP at 95% prediction intervals, aggregated across 10 seeds. Methods with suffix '-c' denote conformalized variants obtained using the validation set divided into two parts, one for validation and one for calibration.

| Dataset | CLEAR-c | PCS-EPISTEMIC-c | ALEATORIC-R-c | CLEAR |
|---|---|---|---|---|
| ailerons | 0.95 ± 0.01 | 0.95 ± 0.01 | 0.95 ± 0.01 | 0.95 ± 0.00 |
| airfoil | 0.96 ± 0.01 | 0.95 ± 0.02 | 0.95 ± 0.02 | 0.95 ± 0.01 |
| allstate | 0.95 ± 0.01 | 0.94 ± 0.01 | 0.95 ± 0.01 | 0.95 ± 0.01 |
| ca_housing | 0.95 ± 0.01 | 0.95 ± 0.01 | 0.95 ± 0.01 | 0.95 ± 0.01 |
| computer | 0.95 ± 0.01 | 0.95 ± 0.01 | 0.95 ± 0.01 | 0.95 ± 0.01 |
| concrete | 0.95 ± 0.02 | 0.95 ± 0.02 | 0.95 ± 0.02 | 0.95 ± 0.02 |
| elevator | 0.95 ± 0.01 | 0.95 ± 0.01 | 0.95 ± 0.01 | 0.95 ± 0.00 |
| energy_efficiency | 0.97 ± 0.02 | 0.95 ± 0.02 | 0.96 ± 0.02 | 0.95 ± 0.02 |
| insurance | 0.96 ± 0.02 | 0.95 ± 0.02 | 0.96 ± 0.02 | 0.95 ± 0.01 |
| kin8nm | 0.96 ± 0.01 | 0.95 ± 0.01 | 0.95 ± 0.01 | 0.95 ± 0.01 |
| miami_housing | 0.95 ± 0.01 | 0.95 ± 0.01 | 0.95 ± 0.01 | 0.95 ± 0.01 |
| naval_propulsion | 0.95 ± 0.01 | 1.00 ± 0.00 | 0.95 ± 0.01 | 0.95 ± 0.01 |
| parkinsons | 0.95 ± 0.01 | 0.95 ± 0.01 | 0.95 ± 0.01 | 0.95 ± 0.01 |
| powerplant | 0.95 ± 0.01 | 0.95 ± 0.01 | 0.95 ± 0.01 | 0.95 ± 0.01 |
| qsar | 0.95 ± 0.01 | 0.95 ± 0.01 | 0.95 ± 0.01 | 0.95 ± 0.01 |
| sulfur | 0.95 ± 0.01 | 0.95 ± 0.01 | 0.95 ± 0.01 | 0.95 ± 0.01 |
| superconductor | 0.95 ± 0.01 | 0.95 ± 0.00 | 0.95 ± 0.00 | 0.95 ± 0.00 |

Table 39: Conformalized Variant (b) NIW at 95% prediction intervals, aggregated across 10 seeds. Methods with suffix '-c' denote conformalized variants obtained using the validation set divided into two parts, one for validation and one for calibration. Values $\geq 100$ or $< 0.01$ are presented in scientific notation with 1 decimal place. **Bold** values (desirable) are the minimum for that dataset and metric, while the underlined values indicate the second-best result. Red values are more than 33% worse than the best result.

| Dataset | CLEAR-c | PCS-EPISTEMIC-c | ALEATORIC-R-c | CLEAR |
|---|---|---|---|---|
| ailerons | **0.193 ± 0.016** | 0.217 ± 0.017 | 0.195 ± 0.018 | 0.193 ± 0.016 |
| airfoil | 0.215 ± 0.019 | 0.217 ± 0.022 | 0.227 ± 0.031 | **0.207 ± 0.012** |
| allstate | 0.257 ± 0.044 | 0.263 ± 0.057 | 0.258 ± 0.046 | **0.256 ± 0.045** |
| ca_housing | 0.339 ± 0.006 | 0.355 ± 0.012 | 0.349 ± 0.008 | **0.339 ± 0.008** |
| computer | **0.099 ± 0.006** | 0.142 ± 0.010 | 0.104 ± 0.005 | 0.099 ± 0.006 |
| concrete | 0.268 ± 0.023 | **0.247 ± 0.040** | 0.271 ± 0.044 | 0.249 ± 0.025 |
| elevator | **0.137 ± 0.009** | 0.194 ± 0.012 | 0.145 ± 0.008 | 0.137 ± 0.007 |
| energy_efficiency | 0.053 ± 0.009 | 0.058 ± 0.009 | 0.053 ± 0.009 | **0.047 ± 0.005** |
| insurance | 0.357 ± 0.070 | 0.461 ± 0.292 | 0.408 ± 0.102 | **0.329 ± 0.045** |
| kin8nm | 0.362 ± 0.014 | 0.373 ± 0.013 | 0.370 ± 0.011 | **0.360 ± 0.012** |
| miami_housing | 0.085 ± 0.003 | 0.097 ± 0.002 | 0.087 ± 0.005 | **0.085 ± 0.002** |
| naval_propulsion | 1.9e-03 ± 9.1e-05 | 8.1e-03 ± 4.1e-04 | 1.9e-03 ± 1.0e-04 | **1.9e-03 ± 7.3e-05** |
| parkinsons | **0.279 ± 0.014** | 0.325 ± 0.016 | 0.305 ± 0.012 | 0.281 ± 0.009 |
| powerplant | 0.171 ± 0.010 | 0.179 ± 0.008 | 0.183 ± 0.008 | **0.170 ± 0.007** |
| qsar | 0.371 ± 0.126 | 0.395 ± 0.134 | 0.386 ± 0.127 | **0.366 ± 0.123** |
| sulfur | 0.111 ± 0.010 | 0.129 ± 0.016 | **0.106 ± 0.011** | 0.108 ± 0.009 |
| superconductor | 0.220 ± 0.025 | 0.240 ± 0.027 | 0.227 ± 0.025 | **0.219 ± 0.023** |

Table 40: Conformalized Variant (b) Quantile Loss at 95% prediction intervals, aggregated across 10 seeds. Methods with suffix '-c' denote conformalized variants obtained using the validation set divided into two parts, one for validation and one for calibration.Values $\geq 100$ or $< 0.01$ are presented in scientific notation with 1 decimal place. **Bold** values (desirable) are the minimum for that dataset and metric, while the underlined values indicate the second-best result. Red values are more than 33% worse than the best result.

| Dataset | CLEAR-c | PCS-EPISTEMIC-c | ALEATORIC-R-c | CLEAR |
|---|---|---|---|---|
| ailerons | 9.2e-06 ± 2.3e-07 | 1.0e-05 ± 2.1e-07 | 9.7e-06 ± 2.7e-07 | **9.2e-06 ± 2.3e-07** |
| airfoil | 0.110 ± 0.013 | 0.119 ± 0.012 | 0.126 ± 0.015 | **0.109 ± 0.011** |
| allstate | 1.2e+02 ± 7.662 | 1.3e+02 ± 8.290 | 1.3e+02 ± 7.774 | **1.1e+02 ± 7.318** |
| ca_housing | **2.8e+03 ± 56.920** | 3.2e+03 ± 91.466 | 3.0e+03 ± 73.272 | 2.8e+03 ± 59.700 |
| computer | **0.159 ± 0.007** | 0.207 ± 0.009 | 0.206 ± 0.015 | 0.159 ± 0.007 |
| concrete | 0.340 ± 0.037 | **0.327 ± 0.039** | 0.356 ± 0.048 | 0.338 ± 0.040 |
| elevator | 1.4e-04 ± 2.4e-06 | 1.9e-04 ± 6.7e-06 | 1.6e-04 ± 3.0e-06 | **1.4e-04 ± 2.0e-06** |
| energy_efficiency | 0.033 ± 0.005 | 0.038 ± 0.003 | 0.034 ± 0.005 | **0.031 ± 0.004** |
| insurance | 3.3e+02 ± 28.437 | 5.2e+02 ± 1.4e+02 | 3.8e+02 ± 43.883 | **3.2e+02 ± 31.593** |
| kin8nm | 7.2e-03 ± 1.7e-04 | 7.6e-03 ± 1.1e-04 | 7.7e-03 ± 2.7e-04 | **7.2e-03 ± 1.7e-04** |
| miami_housing | 3.9e+03 ± 2.1e+02 | 4.2e+03 ± 1.7e+02 | 5.2e+03 ± 3.4e+02 | **3.9e+03 ± 1.9e+02** |
| naval_propulsion | 5.5e-05 ± 1.9e-06 | 1.8e-04 ± 9.1e-06 | 6.4e-05 ± 2.7e-06 | **5.4e-05 ± 1.9e-06** |
| parkinsons | 0.209 ± 0.008 | 0.237 ± 0.008 | 0.228 ± 0.013 | **0.209 ± 0.010** |
| powerplant | 0.213 ± 0.011 | 0.222 ± 0.011 | 0.220 ± 0.011 | **0.212 ± 0.012** |
| qsar | 0.050 ± 0.003 | 0.053 ± 0.003 | 0.052 ± 0.003 | **0.049 ± 0.003** |
| sulfur | 2.0e-03 ± 1.1e-04 | 2.3e-03 ± 8.5e-05 | 2.3e-03 ± 1.5e-04 | **1.9e-03 ± 9.6e-05** |
| superconductor | 0.523 ± 0.018 | 0.611 ± 0.026 | 0.573 ± 0.024 | **0.523 ± 0.018** |

Table 41: Conformalized Variant (b) NCIW at 95% prediction intervals, aggregated across 10 seeds. Methods with suffix '-c' denote conformalized variants obtained using the validation set divided into two parts, one for validation and one for calibration.Values $\geq 100$ or $< 0.01$ are presented in scientific notation with 1 decimal place. **Bold** values (desirable) are the minimum for that dataset and metric, while the underlined values indicate the second-best result. Red values are more than 33% worse than the best result.

| Dataset | CLEAR-c | PCS-EPISTEMIC-c | ALEATORIC-R-c | CLEAR |
|---|---|---|---|---|
| ailerons | 0.195 ± 0.018 | 0.217 ± 0.021 | 0.196 ± 0.017 | **0.195 ± 0.017** |
| airfoil | 0.203 ± 0.007 | 0.211 ± 0.007 | 0.216 ± 0.011 | **0.203 ± 0.006** |
| allstate | **0.248 ± 0.041** | 0.270 ± 0.052 | 0.266 ± 0.044 | 0.250 ± 0.043 |
| ca_housing | 0.336 ± 0.006 | 0.354 ± 0.012 | 0.348 ± 0.011 | **0.336 ± 0.008** |
| computer | 0.098 ± 0.005 | 0.142 ± 0.008 | 0.103 ± 0.003 | **0.098 ± 0.005** |
| concrete | 0.248 ± 0.032 | **0.241 ± 0.025** | 0.261 ± 0.039 | 0.246 ± 0.033 |
| elevator | **0.137 ± 0.008** | 0.192 ± 0.012 | 0.147 ± 0.009 | 0.137 ± 0.008 |
| energy_efficiency | 0.046 ± 0.006 | 0.057 ± 0.004 | 0.050 ± 0.005 | **0.046 ± 0.005** |
| insurance | **0.314 ± 0.052** | 0.346 ± 0.076 | 0.335 ± 0.059 | 0.320 ± 0.059 |
| kin8nm | 0.355 ± 0.008 | 0.371 ± 0.009 | 0.371 ± 0.009 | **0.354 ± 0.008** |
| miami_housing | 0.084 ± 0.004 | 0.098 ± 0.002 | 0.085 ± 0.003 | **0.084 ± 0.003** |
| naval_propulsion | 1.8e-03 ± 4.7e-05 | 3.2e-03 ± 1.2e-04 | 1.9e-03 ± 8.4e-05 | **1.8e-03 ± 5.6e-05** |
| parkinsons | **0.284 ± 0.011** | 0.324 ± 0.008 | 0.309 ± 0.018 | 0.284 ± 0.010 |
| powerplant | 0.173 ± 0.008 | 0.182 ± 0.007 | 0.182 ± 0.007 | **0.173 ± 0.007** |
| qsar | 0.363 ± 0.122 | 0.393 ± 0.131 | 0.390 ± 0.131 | **0.363 ± 0.121** |
| sulfur | 0.108 ± 0.008 | 0.125 ± 0.009 | **0.105 ± 0.006** | 0.106 ± 0.007 |
| superconductor | 0.218 ± 0.023 | 0.241 ± 0.028 | 0.226 ± 0.022 | **0.218 ± 0.022** |

Table 42: Conformalized Variant (b) Interval Score Loss at 95% prediction intervals, aggregated across 10 seeds. Methods with suffix '-c' denote conformalized variants obtained using the validation set divided into two parts, one for validation and one for calibration.Values $\geq 100$ or $< 0.01$ are presented in scientific notation with 1 decimal place. **Bold** values (desirable) are the minimum for that dataset and metric, while the underlined values indicate the second-best result. Red values are more than 33% worse than the best result.

| Dataset | CLEAR-c | PCS-EPISTEMIC-c | ALEATORIC-R-c | CLEAR |
|---|---|---|---|---|
| ailerons | 7.4e-04 ± 1.8e-05 | 8.1e-04 ± 1.7e-05 | 7.8e-04 ± 2.2e-05 | **7.4e-04 ± 1.8e-05** |
| airfoil | 8.794 ± 1.047 | 9.523 ± 0.959 | 10.075 ± 1.227 | **8.717 ± 0.870** |
| allstate | 9.2e+03 ± 6.1e+02 | 1.0e+04 ± 6.6e+02 | 1.0e+04 ± 6.2e+02 | **9.2e+03 ± 5.9e+02** |
| ca_housing | **2.2e+05 ± 4.6e+03** | 2.6e+05 ± 7.3e+03 | 2.4e+05 ± 5.9e+03 | 2.2e+05 ± 4.8e+03 |
| computer | **12.704 ± 0.571** | 16.543 ± 0.706 | 16.505 ± 1.190 | 12.707 ± 0.575 |
| concrete | 27.168 ± 2.969 | **26.195 ± 3.142** | 28.509 ± 3.874 | 27.020 ± 3.202 |
| elevator | 0.011 ± 0.000 | 0.015 ± 0.001 | 0.013 ± 0.000 | **0.011 ± 0.000** |
| energy_efficiency | 2.609 ± 0.417 | 3.019 ± 0.237 | 2.681 ± 0.431 | **2.465 ± 0.347** |
| insurance | 2.6e+04 ± 2.3e+03 | 4.2e+04 ± 1.1e+04 | 3.1e+04 ± 3.5e+03 | **2.6e+04 ± 2.5e+03** |
| kin8nm | 0.578 ± 0.014 | 0.607 ± 0.009 | 0.615 ± 0.022 | **0.577 ± 0.014** |
| miami_housing | 3.2e+05 ± 1.7e+04 | 3.3e+05 ± 1.4e+04 | 4.2e+05 ± 2.7e+04 | **3.1e+05 ± 1.5e+04** |
| naval_propulsion | 4.4e-03 ± 1.5e-04 | 0.014 ± 0.001 | 5.1e-03 ± 2.2e-04 | **4.4e-03 ± 1.5e-04** |
| parkinsons | 16.712 ± 0.669 | 18.967 ± 0.645 | 18.250 ± 1.030 | **16.697 ± 0.794** |
| powerplant | 17.066 ± 0.919 | 17.724 ± 0.919 | 17.567 ± 0.899 | **16.993 ± 0.937** |
| qsar | 3.960 ± 0.207 | 4.277 ± 0.215 | 4.195 ± 0.229 | **3.956 ± 0.218** |
| sulfur | 0.157 ± 0.008 | 0.180 ± 0.007 | 0.186 ± 0.012 | **0.155 ± 0.008** |
| superconductor | 41.818 ± 1.464 | 48.902 ± 2.063 | 45.854 ± 1.910 | **41.801 ± 1.407** |

## G.3 VARIANT (C) CONFORMALIZED

Table 43: Conformalized Variant (c) PICP at 95% prediction intervals, aggregated across 10 seeds. Methods with suffix '-c' denote conformalized variants obtained using the validation set divided into two parts, one for validation and one for calibration.

| Dataset | CLEAR-c | PCS-EPISTEMIC-c | ALEATORIC-R-c | CLEAR | UACQR-P | UACQR-S |
|---|---|---|---|---|---|---|
| ailerons | 0.95 ± 0.01 | 0.95 ± 0.01 | 0.95 ± 0.01 | 0.95 ± 0.01 | 0.95 ± 0.01 | 0.97 ± 0.00 |
| airfoil | 0.96 ± 0.02 | 0.96 ± 0.02 | 0.96 ± 0.02 | 0.96 ± 0.01 | 0.95 ± 0.01 | 0.96 ± 0.02 |
| allstate | 0.95 ± 0.01 | 0.94 ± 0.01 | 0.95 ± 0.01 | 0.95 ± 0.01 | 0.95 ± 0.01 | 0.96 ± 0.01 |
| ca_housing | 0.95 ± 0.01 | 0.95 ± 0.01 | 0.95 ± 0.01 | 0.95 ± 0.01 | 0.95 ± 0.00 | 0.96 ± 0.00 |
| computer | 0.95 ± 0.01 | 0.95 ± 0.01 | 0.95 ± 0.01 | 0.95 ± 0.01 | 0.95 ± 0.01 | 0.97 ± 0.00 |
| concrete | 0.96 ± 0.02 | 0.96 ± 0.01 | 0.96 ± 0.02 | 0.96 ± 0.01 | 0.96 ± 0.02 | 0.97 ± 0.02 |
| elevator | 0.95 ± 0.01 | 0.95 ± 0.01 | 0.95 ± 0.01 | 0.95 ± 0.01 | 0.95 ± 0.01 | 0.97 ± 0.00 |
| energy_efficiency | 0.97 ± 0.01 | 0.95 ± 0.03 | 0.96 ± 0.01 | 0.95 ± 0.01 | 0.97 ± 0.02 | 0.96 ± 0.01 |
| insurance | 0.96 ± 0.02 | 0.95 ± 0.02 | 0.96 ± 0.02 | 0.95 ± 0.01 | 0.96 ± 0.01 | 0.96 ± 0.01 |
| kin8nm | 0.95 ± 0.01 | 0.95 ± 0.01 | 0.95 ± 0.01 | 0.95 ± 0.01 | 0.95 ± 0.01 | 0.95 ± 0.01 |
| miami_housing | 0.95 ± 0.01 | 0.95 ± 0.01 | 0.95 ± 0.01 | 0.95 ± 0.01 | 0.95 ± 0.01 | 0.95 ± 0.01 |
| naval_propulsion | 0.95 ± 0.01 | 0.95 ± 0.01 | 0.95 ± 0.01 | 0.95 ± 0.01 | 0.95 ± 0.00 | 0.99 ± 0.00 |
| parkinsons | 0.95 ± 0.01 | 0.95 ± 0.01 | 0.95 ± 0.02 | 0.95 ± 0.01 | 0.95 ± 0.01 | 0.97 ± 0.01 |
| powerplant | 0.95 ± 0.01 | 0.95 ± 0.00 | 0.95 ± 0.01 | 0.95 ± 0.01 | 0.95 ± 0.01 | 0.95 ± 0.01 |
| qsar | 0.95 ± 0.01 | 0.94 ± 0.01 | 0.94 ± 0.01 | 0.95 ± 0.01 | 0.95 ± 0.01 | 0.96 ± 0.01 |
| sulfur | 0.95 ± 0.01 | 0.95 ± 0.01 | 0.95 ± 0.01 | 0.95 ± 0.01 | 0.95 ± 0.01 | 0.95 ± 0.01 |
| superconductor | 0.95 ± 0.01 | 0.95 ± 0.00 | 0.95 ± 0.00 | 0.95 ± 0.00 | 0.95 ± 0.00 | 0.95 ± 0.00 |

Table 44: Conformalized Variant (c) NIW at 95% prediction intervals, aggregated across 10 seeds. Methods with suffix '-c' denote conformalized variants obtained using the validation set divided into two parts, one for validation and one for calibration. Values $\geq 100$ or $< 0.01$ are presented in scientific notation with 1 decimal place. **Bold** values (desirable) are the minimum for that dataset and metric, while the underlined values indicate the second-best result. Red values are more than 33% worse than the best result. $+\infty_v^{a/10}$ denotes that UACQR-P produced infinitely wide intervals for $a$ out of the 10 seeds, and the mean NIW of the remaining $10 - a$ seeds was $v$.

| Dataset | CLEAR-c | PCS-EPISTEMIC-c | ALEATORIC-R-c | CLEAR | UACQR-P | UACQR-S |
|---|---|---|---|---|---|---|
| ailerons | 0.207 ± 0.017 | 0.239 ± 0.017 | 0.202 ± 0.019 | 0.208 ± 0.017 | **0.175 ± 0.014** | 0.207 ± 0.018 |
| airfoil | 0.210 ± 0.021 | 0.212 ± 0.020 | 0.230 ± 0.040 | **0.199 ± 0.013** | 0.367 ± 0.022 | 0.395 ± 0.030 |
| allstate | 0.282 ± 0.052 | 0.292 ± 0.061 | **0.279 ± 0.037** | 0.279 ± 0.052 | 0.284 ± 0.050 | 0.306 ± 0.053 |
| ca_housing | 0.317 ± 0.009 | 0.330 ± 0.013 | 0.344 ± 0.010 | **0.315 ± 0.010** | 0.365 ± 0.006 | 0.405 ± 0.006 |
| computer | **0.079 ± 0.003** | 0.083 ± 0.003 | 0.088 ± 0.004 | 0.080 ± 0.004 | 0.083 ± 0.003 | 0.102 ± 0.001 |
| concrete | 0.290 ± 0.037 | **0.273 ± 0.032** | 0.287 ± 0.037 | 0.278 ± 0.029 | 0.378 ± 0.020 | 0.405 ± 0.022 |
| elevator | **0.126 ± 0.008** | 0.153 ± 0.008 | 0.136 ± 0.008 | 0.127 ± 0.008 | 0.158 ± 0.011 | 0.199 ± 0.010 |
| energy_efficiency | 0.052 ± 0.007 | 0.043 ± 0.008 | 0.053 ± 0.007 | **0.043 ± 0.007** | $+\infty_{0.154}^{1/10}$ | 0.157 ± 0.006 |
| insurance | 0.414 ± 0.094 | 0.376 ± 0.134 | 0.434 ± 0.086 | 0.397 ± 0.091 | **0.303 ± 0.042** | 0.312 ± 0.041 |
| kin8nm | 0.328 ± 0.015 | 0.327 ± 0.012 | 0.327 ± 0.015 | **0.325 ± 0.012** | 0.444 ± 0.017 | 0.452 ± 0.015 |
| miami_housing | 0.088 ± 0.003 | 0.096 ± 0.005 | 0.112 ± 0.019 | **0.088 ± 0.001** | 0.104 ± 0.003 | 0.114 ± 0.003 |
| naval_propulsion | 1.6e-03 ± 3.6e-05 | 1.7e-03 ± 6.5e-05 | 1.6e-03 ± 4.9e-05 | **1.6e-03 ± 3.2e-05** | 1.7e-03 ± 4.4e-05 | 3.5e-03 ± 8.1e-05 |
| parkinsons | **0.250 ± 0.015** | 0.262 ± 0.009 | 0.281 ± 0.015 | 0.250 ± 0.007 | 0.271 ± 0.020 | 0.314 ± 0.019 |
| powerplant | 0.158 ± 0.011 | 0.170 ± 0.007 | 0.162 ± 0.008 | **0.157 ± 0.007** | 0.193 ± 0.008 | 0.203 ± 0.009 |
| qsar | 0.354 ± 0.120 | 0.402 ± 0.143 | 0.349 ± 0.115 | **0.344 ± 0.117** | 0.409 ± 0.135 | 0.450 ± 0.154 |
| sulfur | 0.112 ± 0.011 | 0.126 ± 0.012 | **0.107 ± 0.016** | 0.111 ± 0.009 | 0.115 ± 0.012 | 0.126 ± 0.013 |
| superconductor | 0.196 ± 0.022 | 0.234 ± 0.026 | 0.208 ± 0.025 | **0.195 ± 0.020** | 0.219 ± 0.022 | 0.234 ± 0.025 |

Table 45: Conformalized Variant (c) Quantile Loss at 95% prediction intervals, aggregated across 10 seeds. Methods with suffix '-c' denote conformalized variants obtained using the validation set divided into two parts, one for validation and one for calibration. Values $\geq 100$ or $< 0.01$ are presented in scientific notation with 1 decimal place. **Bold** values (desirable) are the minimum for that dataset and metric, while the underlined values indicate the second-best result. Red values are more than 33% worse than the best result. $+\infty_v^{a/10}$ denotes that UACQR-P produced infinitely wide intervals for $a$ out of the 10 seeds, and the mean Quantile Loss of the remaining $10 - a$ seeds was $v$.

| Dataset | CLEAR-c | PCS-EPISTEMIC-c | ALEATORIC-R-c | CLEAR | UACQR-P | UACQR-S |
|---|---|---|---|---|---|---|
| ailerons | **9.6e-06 ± 2.9e-07** | 1.1e-05 ± 2.9e-07 | 9.8e-06 ± 3.2e-07 | 9.6e-06 ± 2.9e-07 | 9.9e-06 ± 3.8e-07 | 9.9e-06 ± 3.1e-07 |
| airfoil | 0.110 ± 0.015 | 0.111 ± 0.013 | 0.131 ± 0.019 | **0.107 ± 0.014** | 0.189 ± 0.012 | 0.193 ± 0.011 |
| allstate | 1.2e+02 ± 6.810 | 1.4e+02 ± 11.278 | 1.3e+02 ± 22.506 | 1.2e+02 ± 6.874 | **1.1e+02 ± 7.290** | 1.2e+02 ± 6.653 |
| ca_housing | 2.8e+03 ± 91.022 | 2.9e+03 ± 88.973 | 3.2e+03 ± 1.1e+02 | **2.8e+03 ± 92.560** | 2.9e+03 ± 53.918 | 3.0e+03 ± 49.727 |
| computer | 0.138 ± 0.012 | 0.140 ± 0.012 | 0.163 ± 0.013 | **0.137 ± 0.011** | 0.147 ± 0.006 | 0.152 ± 0.004 |
| concrete | 0.343 ± 0.033 | **0.334 ± 0.031** | 0.381 ± 0.058 | 0.337 ± 0.032 | 0.411 ± 0.029 | 0.426 ± 0.035 |
| elevator | 1.3e-04 ± 3.3e-06 | 1.4e-04 ± 3.9e-06 | 1.5e-04 ± 3.5e-06 | **1.3e-04 ± 3.2e-06** | 1.8e-04 ± 4.2e-06 | 1.8e-04 ± 4.1e-06 |
| energy_efficiency | 0.031 ± 0.006 | 0.030 ± 0.007 | 0.035 ± 0.010 | **0.029 ± 0.007** | $+\infty_{0.078}^{1/10}$ | 0.078 ± 0.005 |
| insurance | 4.1e+02 ± 73.015 | 4.3e+02 ± 61.342 | 3.9e+02 ± 53.648 | 4.2e+02 ± 80.269 | **3.4e+02 ± 44.920** | 3.5e+02 ± 49.833 |
| kin8nm | 6.7e-03 ± 1.7e-04 | 6.9e-03 ± 2.2e-04 | 6.8e-03 ± 2.0e-04 | **6.7e-03 ± 1.8e-04** | 8.8e-03 ± 1.7e-04 | 9.1e-03 ± 1.8e-04 |
| miami_housing | 3.8e+03 ± 1.9e+02 | 3.8e+03 ± 2.2e+02 | 6.8e+03 ± 1.2e+03 | **3.7e+03 ± 1.4e+02** | 4.3e+03 ± 1.3e+02 | 4.6e+03 ± 1.5e+02 |
| naval_propulsion | 4.0e-05 ± 1.0e-06 | 4.2e-05 ± 1.3e-06 | 4.9e-05 ± 1.6e-06 | **4.0e-05 ± 1.0e-06** | 9.7e-05 ± 1.0e-05 | 8.3e-05 ± 1.7e-06 |
| parkinsons | 0.190 ± 0.007 | 0.192 ± 0.006 | 0.217 ± 0.008 | 0.189 ± 0.006 | **0.181 ± 0.009** | 0.200 ± 0.010 |
| powerplant | 0.204 ± 0.012 | 0.210 ± 0.013 | 0.209 ± 0.013 | **0.203 ± 0.012** | 0.226 ± 0.011 | 0.235 ± 0.011 |
| qsar | 0.049 ± 0.002 | 0.054 ± 0.001 | 0.051 ± 0.003 | **0.048 ± 0.002** | 0.051 ± 0.003 | 0.054 ± 0.003 |
| sulfur | 1.8e-03 ± 8.0e-05 | 1.9e-03 ± 8.7e-05 | 2.6e-03 ± 2.6e-04 | **1.8e-03 ± 6.3e-05** | 1.9e-03 ± 1.2e-04 | 2.0e-03 ± 1.3e-04 |
| superconductor | **0.488 ± 0.020** | 0.573 ± 0.026 | 0.563 ± 0.047 | 0.489 ± 0.020 | 0.508 ± 0.018 | 0.522 ± 0.013 |

Table 46: Conformalized Variant (c) NCIW at 95% prediction intervals, aggregated across 10 seeds. Methods with suffix '-c' denote conformalized variants obtained using the validation set divided into two parts, one for validation and one for calibration. Values $\geq 100$ or $< 0.01$ are presented in scientific notation with 1 decimal place. **Bold** values (desirable) are the minimum for that dataset and metric, while the underlined values indicate the second-best result. Red values are more than 33% worse than the best result. $+\infty_v^{a/10}$ denotes that UACQR-P produced infinitely wide intervals for $a$ out of the 10 seeds, and the mean NCIW of the remaining $10 - a$ seeds was $v$.

| Dataset | CLEAR-c | PCS-EPISTEMIC-c | ALEATORIC-R-c | CLEAR | UACQR-P | UACQR-S |
|---|---|---|---|---|---|---|
| ailerons | 0.209 ± 0.021 | 0.242 ± 0.021 | 0.203 ± 0.021 | 0.210 ± 0.020 | **0.192 ± 0.030** | 0.207 ± 0.018 |
| airfoil | 0.189 ± 0.016 | 0.199 ± 0.013 | 0.204 ± 0.017 | **0.188 ± 0.014** | 0.317 ± 0.025 | 0.315 ± 0.038 |
| allstate | **0.274 ± 0.046** | 0.296 ± 0.058 | 0.285 ± 0.029 | 0.277 ± 0.056 | 0.278 ± 0.047 | 0.297 ± 0.052 |
| ca_housing | 0.312 ± 0.010 | 0.329 ± 0.009 | 0.342 ± 0.011 | **0.311 ± 0.010** | 0.366 ± 0.005 | 0.405 ± 0.006 |
| computer | **0.080 ± 0.003** | 0.083 ± 0.003 | 0.090 ± 0.004 | 0.081 ± 0.004 | 0.087 ± 0.004 | 0.102 ± 0.001 |
| concrete | **0.255 ± 0.032** | 0.257 ± 0.031 | 0.263 ± 0.028 | 0.259 ± 0.032 | 0.332 ± 0.051 | 0.329 ± 0.044 |
| elevator | **0.126 ± 0.007** | 0.151 ± 0.011 | 0.138 ± 0.008 | 0.126 ± 0.008 | 0.167 ± 0.013 | 0.199 ± 0.010 |
| energy_efficiency | 0.043 ± 0.005 | 0.043 ± 0.005 | 0.048 ± 0.005 | **0.041 ± 0.006** | $+\infty_{0.138}^{1/10}$ | 0.135 ± 0.015 |
| insurance | 0.356 ± 0.071 | 0.358 ± 0.078 | 0.400 ± 0.091 | 0.345 ± 0.078 | **0.291 ± 0.044** | 0.299 ± 0.040 |
| kin8nm | **0.323 ± 0.008** | 0.331 ± 0.010 | 0.327 ± 0.008 | 0.323 ± 0.008 | 0.436 ± 0.014 | 0.444 ± 0.011 |
| miami_housing | **0.087 ± 0.003** | 0.096 ± 0.004 | 0.107 ± 0.013 | 0.088 ± 0.002 | 0.101 ± 0.004 | 0.109 ± 0.004 |
| naval_propulsion | **1.6e-03 ± 5.0e-05** | 1.7e-03 ± 4.7e-05 | 1.6e-03 ± 4.1e-05 | 1.6e-03 ± 5.0e-05 | 1.7e-03 ± 4.4e-05 | 3.5e-03 ± 8.1e-05 |
| parkinsons | **0.250 ± 0.010** | 0.259 ± 0.009 | 0.279 ± 0.013 | 0.250 ± 0.012 | 0.274 ± 0.019 | 0.314 ± 0.019 |
| powerplant | 0.162 ± 0.007 | 0.171 ± 0.008 | 0.163 ± 0.008 | **0.161 ± 0.007** | 0.190 ± 0.009 | 0.200 ± 0.011 |
| qsar | 0.350 ± 0.121 | 0.412 ± 0.144 | 0.360 ± 0.120 | **0.348 ± 0.118** | 0.410 ± 0.135 | 0.444 ± 0.150 |
| sulfur | 0.111 ± 0.010 | 0.124 ± 0.008 | **0.107 ± 0.005** | 0.109 ± 0.008 | 0.115 ± 0.010 | 0.126 ± 0.010 |
| superconductor | 0.194 ± 0.021 | 0.234 ± 0.025 | 0.204 ± 0.024 | **0.192 ± 0.019** | 0.219 ± 0.023 | 0.233 ± 0.025 |

Table 47: Conformalized Variant (c) Interval Score Loss at 95% prediction intervals, aggregated across 10 seeds. Methods with suffix '-c' denote conformalized variants obtained using the validation set divided into two parts, one for validation and one for calibration. Values $\geq 100$ or $< 0.01$ are presented in scientific notation with 1 decimal place. **Bold** values (desirable) are the minimum for that dataset and metric, while the underlined values indicate the second-best result. Red values are more than 33% worse than the best result. $+\infty_v^{a/10}$ denotes that UACQR-P produced infinitely wide intervals for $a$ out of the 10 seeds, and the mean AISL of the remaining $10 - a$ seeds was $v$.

| Dataset | CLEAR-c | PCS-EPISTEMIC-c | ALEATORIC-R-c | CLEAR | UACQR-P | UACQR-S |
|---|---|---|---|---|---|---|
| ailerons | **7.7e-04 ± 2.3e-05** | 8.6e-04 ± 2.3e-05 | 7.9e-04 ± 2.6e-05 | 7.7e-04 ± 2.3e-05 | 7.9e-04 ± 3.0e-05 | 7.9e-04 ± 2.5e-05 |
| airfoil | 8.780 ± 1.202 | 8.857 ± 1.035 | 10.504 ± 1.503 | **8.569 ± 1.099** | 15.102 ± 0.941 | 15.451 ± 0.852 |
| allstate | 9.4e+03 ± 5.4e+02 | 1.1e+04 ± 9.0e+02 | 1.0e+04 ± 1.8e+03 | 9.2e+03 ± 5.5e+02 | **8.9e+03 ± 5.8e+02** | 9.3e+03 ± 5.3e+02 |
| ca_housing | 2.2e+05 ± 7.3e+03 | 2.3e+05 ± 7.1e+03 | 2.5e+05 ± 8.6e+03 | **2.2e+05 ± 7.4e+03** | 2.4e+05 ± 4.3e+03 | 2.4e+05 ± 4.0e+03 |
| computer | 11.040 ± 0.933 | 11.164 ± 0.938 | 13.042 ± 1.009 | **10.978 ± 0.911** | 11.750 ± 0.449 | 12.130 ± 0.358 |
| concrete | 27.403 ± 2.641 | **26.754 ± 2.444** | 30.441 ± 4.624 | 26.928 ± 2.529 | 32.895 ± 2.347 | 34.089 ± 2.796 |
| elevator | 0.011 ± 0.000 | 0.012 ± 0.000 | 0.012 ± 0.000 | **0.010 ± 0.000** | 0.014 ± 0.000 | 0.015 ± 0.000 |
| energy_efficiency | 2.463 ± 0.488 | 2.422 ± 0.596 | 2.796 ± 0.804 | **2.348 ± 0.554** | $+\infty_{6.207}^{1/10}$ | 6.238 ± 0.413 |
| insurance | 3.3e+04 ± 5.8e+03 | 3.4e+04 ± 4.9e+03 | 3.1e+04 ± 4.3e+03 | 3.4e+04 ± 6.4e+03 | **2.7e+04 ± 3.6e+03** | 2.8e+04 ± 4.0e+03 |
| kin8nm | 0.536 ± 0.014 | 0.552 ± 0.018 | 0.545 ± 0.016 | **0.534 ± 0.014** | 0.704 ± 0.013 | 0.729 ± 0.014 |
| miami_housing | 3.0e+05 ± 1.5e+04 | 3.1e+05 ± 1.7e+04 | 5.5e+05 ± 9.6e+04 | **2.9e+05 ± 1.1e+04** | 3.5e+05 ± 1.0e+04 | 3.7e+05 ± 1.2e+04 |
| naval_propulsion | 3.2e-03 ± 8.0e-05 | 3.4e-03 ± 1.0e-04 | 3.9e-03 ± 1.3e-04 | **3.2e-03 ± 8.4e-05** | 7.8e-03 ± 8.3e-04 | 6.6e-03 ± 1.4e-04 |
| parkinsons | 15.231 ± 0.540 | 15.327 ± 0.476 | 17.375 ± 0.656 | 15.149 ± 0.518 | **14.486 ± 0.724** | 15.967 ± 0.795 |
| powerplant | 16.287 ± 0.959 | 16.839 ± 1.025 | 16.687 ± 1.010 | **16.243 ± 0.985** | 18.062 ± 0.917 | 18.771 ± 0.840 |
| qsar | 3.903 ± 0.190 | 4.360 ± 0.105 | 4.094 ± 0.213 | **3.877 ± 0.176** | 4.101 ± 0.232 | 4.324 ± 0.208 |
| sulfur | 0.145 ± 0.006 | 0.148 ± 0.007 | 0.209 ± 0.021 | **0.143 ± 0.005** | 0.154 ± 0.010 | 0.161 ± 0.011 |
| superconductor | **39.077 ± 1.615** | 45.819 ± 2.070 | 45.024 ± 3.738 | 39.104 ± 1.604 | 40.649 ± 1.469 | 41.797 ± 1.060 |

Table 48: Conformalized Variant (c) CLEAR calibration parameters $\lambda$ and $\gamma_1$ for 95% prediction intervals across 10 seeds. Using all available variables. Showing median [min:max] values.

| Dataset | $\lambda$ | $\gamma_1$ |
|---|---|---|
| ailerons | 1.01 [0.26:1.75] | 1.01 [0.87:1.32] |
| airfoil | 1.04 [0.10:100.00] | 1.45 [0.02:6.72] |
| allstate | 0.72 [0.00:100.00] | 1.10 [0.02:1.43] |
| ca_housing | 1.93 [1.11:13.99] | 0.67 [0.14:0.95] |
| computer | 2.85 [1.39:10.47] | 0.60 [0.18:0.95] |
| concrete | 1.18 [0.25:100.00] | 1.79 [0.02:5.83] |
| elevator | 0.73 [0.45:1.29] | 1.13 [0.86:1.43] |
| energy_efficiency | 0.22 [0.02:100.00] | 6.87 [0.02:19.92] |
| insurance | 4.77 [0.37:100.00] | 0.55 [0.03:5.83] |
| kin8nm | 2.21 [0.39:3.27] | 0.74 [0.61:1.02] |
| miami_housing | 4.61 [1.23:100.00] | 0.36 [0.02:1.09] |
| naval_propulsion | 14.44 [10.01:22.37] | 0.69 [0.52:0.85] |
| parkinsons | 1.11 [0.53:15.09] | 1.34 [0.12:2.31] |
| powerplant | 0.96 [0.57:3.30] | 1.12 [0.51:1.41] |
| qsar | 0.59 [0.22:2.58] | 1.45 [0.78:2.51] |
| sulfur | 2.98 [0.90:100.00] | 0.63 [0.02:1.25] |
| superconductor | 0.73 [0.22:1.38] | 1.17 [0.85:1.33] |

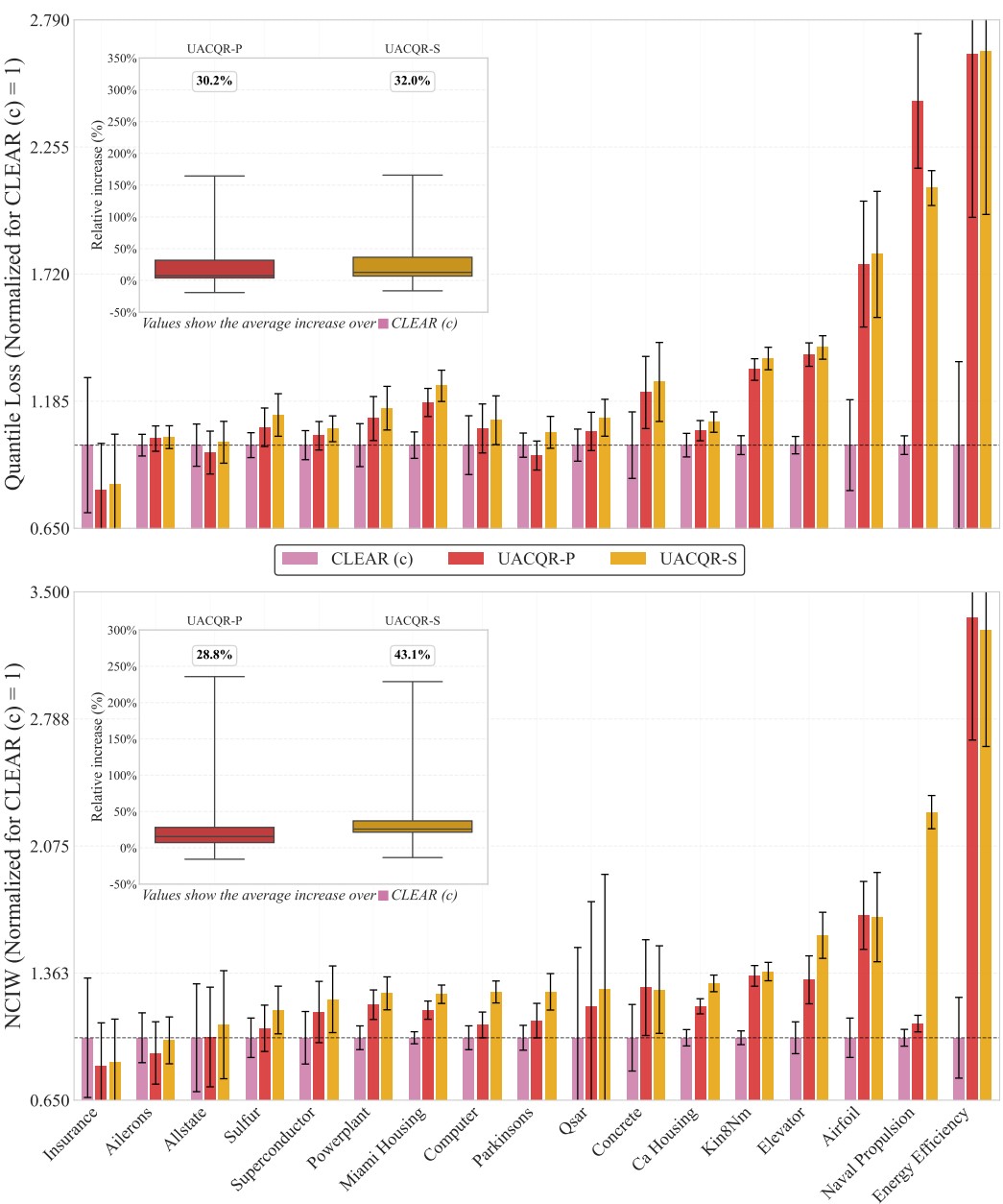

Figure 10: Quantile loss and NCIW performance of UACQR-P and UACQR-S over 10 seeds normalized relative to conformalized CLEAR (c) (baseline = 1.0). Higher values indicate worse performance. The inset boxplot shows the % increase relative to the CLEAR (c) baseline $\pm 1\sigma$. Values inside each subplot represent the mean increase across all datasets.

# H  CASE STUDY: HOUSE PRICE PREDICTION WITH VARYING NUMBER OF PREDICTORS

We illustrate our method using the Ames Housing dataset, which contains data on 2,930 residential properties sold in Ames, Iowa, between 2006 and 2010. The target variable is the sale price of a house, and the full dataset includes around 80 predictor variables describing various aspects such as square footage, neighborhood, and building type. This dataset, originally collected by the Ames City Assessor's Office and curated by Cock (2011), was explored in Chapter 13 of Yu & Barter (2024).

The PCS framework (Yu & Barter, 2024) involves quantifying all sources of extended epistemic uncertainty stemming from the entire data-science cycle, such as uncertainty stemming from data processing. A pipeline of PCS uncertainty quantification applied to the Ames Housing data can be found in Chapter 13 in (Yu & Barter, 2024). We follow the same steps[6] for estimating $\hat{f}$ and estimating epistemic uncertainty bounds $\hat{q}_{0.05}^{\text{epi}}, \hat{q}_{0.95}^{\text{epi}}$.

To investigate how the amount of available information affects predictive uncertainty, we vary both the feature set and the sample size. Starting from approximately 80 features, we construct a reduced version using only the top two predictors; this setup naturally increases aleatoric uncertainty (due to limited information) while decreasing epistemic uncertainty (due to reduced model complexity). We also subsample the training data to 50% and 20% to isolate the effect of training sample size on epistemic uncertainty. In each case, we applied the CLEAR procedure as described in Section 2.4. This involved: (1) data cleaning and preprocessing (excluding irregular sales, imputing missing values, encoding categorical variables), resulting in $N_1 = 438$ cleaned datasets; (2) fitting a predictive model $\hat{f}$ and estimating epistemic uncertainty bounds $\hat{q}_{0.05}^{\text{epi}}, \hat{q}_{0.95}^{\text{epi}}$; (3) estimating aleatoric uncertainty bounds $\hat{q}_{0.05}^{\text{ale}}, \hat{q}_{0.95}^{\text{ale}}$ via quantile regression, as described in Section 2.3; (4) estimating $\lambda$ and calibrating prediction intervals on the validation set.

Table 49 reveals that while PCS performs competitively with all features, CQR is stronger with fewer features but struggles severely with limited training data (20% subsample). In contrast, CLEAR adapts effectively to all scenarios through its calibrated uncertainty combination (calibration parameters from Table 4). CLEAR estimates $\lambda = 0.64$ in the 2-variable case (prioritizing aleatoric uncertainty) versus $\lambda = 14.45$ in the full-feature case (emphasizing epistemic uncertainty), with corresponding calibrated epistemic-to-aleatoric ratios of 0.03 and 7.72. As training data is reduced to 50% and 20%, CLEAR progressively increases $\lambda$ to 17.97 and finally to 100, driving the epistemic-to-aleatoric ratio to 14.09 and 250.5, respectively. This demonstrates the model's ability to accurately quantify the increased epistemic uncertainty arising from limited training samples. Consequently, both in terms of NCIW and Quantile Loss, CLEAR is never outperformed by more than 1%, but outperforms PCS by $\sim 20\%$ for reduced features, and outperforms CQR by $\sim 20\%$ for sub-sampled data.

Table 49: Ames Housing results for all four scenarios (90% coverage target).

| Experiment | Method | NCIW | Quantile Loss | Average Width ($) | Coverage |
|---|---|---|---|---|---|
| 2 features | PCS | 0.214 | 3,818 | 107,880 | 0.87 |
| | CQR | 0.186 | 3,448 | 104,741 | **0.90** |
| | CLEAR | **0.171** | **3,131** | **95,177** | 0.89 |
| All features | PCS | 0.105 | **1,922** | 57,594 | **0.89** |
| | CQR | 0.117 | 2,194 | 62,398 | 0.88 |
| | CLEAR | **0.103** | 1,923 | **55,910** | 0.88 |
| All features 50% data | PCS | **0.108** | 1,978 | **56,716** | 0.87 |
| | CQR | 0.124 | 2,438 | 68,855 | **0.90** |
| | CLEAR | **0.108** | **1,975** | 57,125 | 0.88 |
| All features 20% data | PCS | 0.110 | 2,026 | **58,574** | 0.87 |
| | CQR | 0.130 | 2,645 | 72,884 | **0.90** |
| | CLEAR | **0.110** | **2,024** | 59,290 | 0.88 |

---

[6]Minor differences arise due to: (a) implementation differences in base models between R and Python, and (b) manual calibration of PCS intervals, which was not part of the original implementation. Additionally, both CQR and CLEAR were trained using only linear quantile regressors as the model selection step of PCS selected a linear model based on the RMSE on the validation dataset.

# I ON THE ROLE OF RELATIVE AND ABSOLUTE UNCERTAINTY IN COVERAGE GUARANTEES

In predictive inference, a fundamental distinction arises between *conditional* and *marginal* coverage—closely related to what has been termed "relative vs. absolute uncertainty" (Heiss et al., 2022b) or "adaptive vs. calibrated uncertainty." Conditional coverage requires that prediction intervals achieve the target coverage level at *every individual input*, thereby capturing *local* or *relative* uncertainty. Formally, a prediction interval $C(X_{n+1})$ satisfies conditional coverage at level $1 - \alpha$ if

$$\forall x \in \text{supp}(X): \quad \mathbb{P}\left[Y_{n+1} \in C(X_{n+1}) \mid X_{n+1} = x\right] \geq 1 - \alpha.$$

By contrast, *marginal coverage* only guarantees coverage *on average* over the distribution of inputs:

$$\mathbb{P}\left[Y_{n+1} \in C(X_{n+1})\right] \geq 1 - \alpha.$$

While conditional coverage implies marginal coverage, the reverse does not hold. This distinction is especially important in settings with *heteroskedasticity*, where the variability of $Y \mid X$ changes across the input space, and under *distribution shift*, where the test distribution of $X$ differs from the training distribution. Distribution shift—such as covariate shift or domain adaptation—can render marginal guarantees unreliable since they depend on the marginal $\mathbb{P}_X$. In contrast, conditional coverage ensures that prediction intervals remain valid even when $\mathbb{P}_X$ changes, provided the conditional distribution $\mathbb{P}(Y \mid X)$ remains stable. In what follows, we explore the implications of these distinctions and how they shape both the evaluation and design of uncertainty quantification methods.

## I.1 METRICS: RELEVANCE

To assess the quality of predictive uncertainty, various metrics capture different aspects of coverage and adaptivity:

- **Quantile Loss**, **CRPS**, and **AISL** combine both relative and absolute components incentivizing conditional coverage. These metrics penalize both poor ranking and miscalibration, rewarding methods that adapt well to heteroskedasticity.

- **NCIW** is invariant to the overall scale of the uncertainty but evaluates the ranking—whether a method assigns wider intervals to more uncertain points. In practice, a low NCIW often correlates with good relative uncertainty. However, a minimal NCIW theoretically encourages suboptimal conditional coverage, as it can lead to under-coverage in high-uncertainty regions and over-coverage in low-uncertainty regions. Consequently, good conditional coverage typically requires a slightly higher NCIW than the minimum. Similar to NCIW, Minimum Negative Log-Likelihood ($\text{NLL}_{\min}$) (Heiss et al., 2022b) also assesses relative uncertainty by focusing on whether a method correctly ranks more versus less uncertain inputs, independent of the predicted scale.

- **PICP** and **NIW** each provide only partial information: PICP measures calibration but is blind to adaptivity, while NIW captures the average scale of uncertainty.

These metrics reflect different priorities in uncertainty quantification. Choosing among them (or combining them) depends on whether the primary goal is calibration, adaptivity, or both.

## I.2 APPLICATIONS

Understanding whether a method captures relative or absolute uncertainty has practical implications across a range of applications. In **active learning**, the primary objective is to identify inputs $x$ for which the model is most uncertain, guiding efficient data acquisition. Here, only the ranking of uncertainty matters—selecting the point with the highest epistemic uncertainty. Methods that preserve good relative uncertainty, even if miscalibrated, often suffice. In **Bayesian optimization**, many acquisition functions (such as upper confidence bound or entropy search) depend more on relative than absolute uncertainty (De Ath et al., 2021; Weissteiner et al., 2023). Using upper bounds of the form $\hat{f}(x) + c$ with constant $c$ across all $x$ does not improve over exploiting $\hat{f}(x)$ alone, highlighting the centrality of uncertainty ranking over calibration in this setting. In **human-in-the-loop automation**, relative uncertainty can guide prioritization—for instance, flagging uncertain cases for expert review. While calibrated intervals may not always be necessary, correct ordering of confidence can improve decision efficiency and safety.

## I.3 METHODS

Different uncertainty quantification methods prioritize and estimate relative and absolute uncertainty to varying degrees:

- **NOMU** (Heiss et al., 2022b) is explicitly designed to estimate only relative uncertainty. It does not attempt to calibrate the absolute scale, making it suitable for applications where ranking matters but calibrated intervals are unnecessary.

- **Deep ensembles** (Lakshminarayanan et al., 2017) typically yield strong performance in capturing relative uncertainty, particularly through diversity in predictions across ensemble members. However, they often suffer from miscalibration in the absolute scale of uncertainty unless explicitly corrected.

- **CLEAR** separately estimates relative epistemic and aleatoric uncertainty and combines them using a tunable parameter $\lambda$, which also refines the ranking of uncertainty—offering an alternative to standard calibration techniques. The absolute scale is then calibrated using a second parameter, $\gamma_1$, allowing for flexible control over both adaptivity and calibration.

These methods highlight the spectrum of approaches to uncertainty quantification, from ranking-only models to fully calibrated systems.

## I.4 CALIBRATION TECHNIQUES

The absolute scale of uncertainty is critical in applications where the expected risk or cost over a population matters (e.g., in probabilistic risk management, climate forecasting, or decision-making under uncertainty). However, as emphasized by pathological cases that achieve perfect PICP with no adaptivity, relative uncertainty remains essential for practicality. Several approaches can be taken to adjust the scale of uncertainty estimates while preserving different structures:

- **Multiplicative calibration** scales all predictions by a constant factor, preserving the multiplicative structure of uncertainty. This is appropriate when the model's ranking is reliable.

- **Additive calibration** shifts all intervals uniformly, preserving additive differences but potentially distorting proportional uncertainty.

- **Isotonic calibration** applies a nonparametric monotonic transformation that preserves the ranking of uncertainty, suitable when only order is trusted. E.g., (Kuleshov et al., 2018) uses isotonic calibration for distributional uncertainty quantification.

- **CLEAR** calibrates aleatoric and epistemic uncertainty separately, allowing independent yet coherent control of both components. This facilitates calibrated estimates of total predictive uncertainty while preserving the relative structure.

## I.5 ACHIEVING CONDITIONAL COVERAGE

Achieving conditional coverage requires addressing both components of predictive uncertainty:

1. Estimating relative uncertainty accurately—capturing how uncertainty varies across inputs.
2. Calibrating the absolute scale to ensure the desired coverage level holds at each input.

Even when conditional coverage is not required, improving relative uncertainty tends to reduce average interval width and improve marginal calibration under distribution shifts. Thus, adaptivity and calibration are not mutually exclusive but can reinforce one another. **CLEAR** is explicitly designed to target both forms of uncertainty—modeling and calibrating relative and absolute epistemic, aleatoric, and total predictive uncertainty. As such, it provides a flexible and principled framework for applications demanding both adaptivity and reliability.

