# OpenReview forum: "CLEAR: Calibrated Learning for Epistemic and Aleatoric Risk"
_ICLR.cc/2026/Conference — ICLR 2026 Poster_

### Official Review · Reviewer_yexK · 2025-10-19

**Soundness:** 2
**Presentation:** 2
**Contribution:** 2
**Rating:** 2
**Confidence:** 5

**Summary:**

The paper proposes CLEAR, a method for constructing calibrated prediction intervals in regression by explicitly combining aleatoric and epistemic uncertainties. It estimates aleatoric uncertainty through quantile regression on residuals and epistemic uncertainty via bootstrapped ensemble variation, then forms intervals as a linear combination of both components.

**Strengths:**

The strengths of this paper are:
- The method builds on an intuitive and practically relevant distinction between aleatoric and epistemic uncertainty, and operationalises it in a simple, interpretable way through a weighted combination of the two components.
- I usually dislike arbitrary \lambda weight params, but in this paper, it is justified in the method and offers an interpretable measure of how epistemic and aleatoric components contribute to overall predictive uncertainty.

**Weaknesses:**

The weaknesses of this paper are:
- CLEAR builds directly on existing CQR-derived approaches that incorporate uncertainty decomposition, particularly Uncertainty-Aware CQR (UACQR; Rossellini et al., 2024). As acknowledged by the authors themselves, UACQR can be viewed as a special case of CLEAR with \gamma = 1. CLEAR’s main extension is to calibrate both parameters, allowing it to adjust for miscalibration in the aleatoric component. While this generalisation is reasonable and practically useful, it represents an incremental refinement. Thus, novelity is very low in my opinion.
- Correct me if I am wrong, but in the experimental setup, the same data is used to tune \lambda and calibrate \gamma, which should break the independence assumption required for coverage guarantees.
- Since this is derivative to CQR and other prior methods, not comparing against interval-creation baselines in the main paper seems like a missed step. Why leave competitive results for Appendix D? This has left me confused.
- After spending much time reading the appendix, I am left confused by the structure of the paper. It seems like the strong literature comparisons, justifications, good results are all in the Appendix and the main body of the paper is given less thought. This seems backwards to me?

**Questions:**

- In the default implementation, the same dataset is used both for tuning \lambda and conformal calibration. How do the authors reconcile this with the independence requirement of split-conformal prediction? Can they provide empirical or theoretical evidence that nominal coverage is still preserved under this data reuse?
- Since CLEAR is presented as an evolution of CQR-derived conformal methods, why are direct comparisons against UACQR and related interval-construction baselines relegated to Appendix D rather than integrated into the main results? Would including these baselines in the principal tables change the strength of the empirical conclusions?
- Much of the substantive discussion—literature positioning, theoretical justification, and broader comparisons—appears only in the appendix. Could the authors explain the rationale for this organisation and whether key arguments could be moved into the main text to improve clarity and self-containment?

---

> ### Author Response · Authors · 2025-11-16
> **Clarification on Contribution (W1)**
>
> We thank the reviewer yexK for their review. We are sorry to hear that you were unconvinced about our contribution, and whether it goes beyond a simple extension of UACQR. We would like to clarify why we view the novelty as more substantial. Our method involves several steps that are not captured by your description that *"CLEAR only adds one more calibrated parameter"*:
>
> 1. **Joint calibration of aleatoric and epistemic components** is a very non-trivial step. Treating these two sources of uncertainty separately and calibrating both are precisely what UACQR is missing, and this idea goes beyond a simple view that CLEAR only calibrates one additional parameter. We see this separation of roles as a novel principle that can extend to many ML settings. One parameter can simply be adjusted by calibrating the marginal coverage. For calibrating two parameters, a secondary criterion, such as the quantile loss, is necessary to obtain a uniquely defined solution.
>
> 2. **CLEAR works with all sources of uncertainty in a structured way**. While the simulations rely on bootstrap ensembles similarly to UACQR, the case study shows how the PCS framework can guide practitioners toward better epistemic estimators (including the epistemic uncertainty arising from data cleaning and data pre-processing). This connection between PCS and CQR can be more than interesting in many practical settings.
>
> 3. **Unlike UACQR, CLEAR estimates the aleatoric component on residuals rather than on the raw targets**. This simple modification has a strong effect on stability and accuracy and, in our view, is a contribution on its own. Also, as you mentioned, the novelty lies in the fact that our method is a **generalization** of existing ones, making it much simpler to understand the source of uncertainty when using our technique compared to UACQR. This is aside from the fact that CLEAR outperforms UACQR (due to the novel use of residuals). Additionally, unlike UACQR, which solely focused on conformal prediction and conformal baselines, we compare and demonstrate how calibration methods such as DE and SQR can be used within our framework. **We believe our work helps to bridge the gap between the applied calibration community and the theoretical field of conformal inference**. Also, UACQR is quite focused on quantile regression, which is even part of their title. The emphasis is more on the modality of our approach and on conducting experiments with SQR rather than QR for aleatoric uncertainty; note that other methods, such as diffusion models, could also be used for aleatoric uncertainty.
>
> 4. The framework offers potential **additional interpretability** into the sources of uncertainty. The calibrated $\lambda$ quantifies the relative dominance of aleatoric and epistemic uncertainty and opens the door to methodological extensions. We do not emphasize this perspective heavily in the current work, but our formulation allows for useful interpretations in the future.
>
> 5. We also carry out a **substantially larger empirical study** than prior work for benchmarking. We consider significantly more real-world datasets than the UACQR paper. Our experiments over hundreds of synthetic datasets are didactically useful to better explain the importance of combining epistemic and aleatoric uncertainty. The figures 2, 4, 5, and 6 clearly demonstrate why CQR totally fails to provide input conditional coverage, even for large training datasets, despite its theoretical guarantee of asymptotic input conditional coverage, as we discuss in Appendix B.4. Many theoretical statisticians struggle to understand this. We find these figures helpful for highlighting this major issue.
>
> 6. We identified that **the PCS-framework, which already covers many different types of epistemic uncertainty**, can be further improved by adding aleatoric uncertainty (e.g, via quantile regression). The PCS framework has shown significant improvements for multiple real-world problems. However, in the PCS-related literature, aleatoric uncertainty was not mentioned before. While input-conditional coverage guarantees already existed for QR-based methods, we are the first to provide theoretical guarantees on asymptotic input-conditional coverage for a PCS-derived method. The previous PCS-UQ-algorithms proposed in the literature did not achieve asymptotic input-conditional coverage.
>
> We hope this explanation helps convey the broader contribution. Below, we respond to your questions.

---

> ### Author Response · Authors · 2025-11-16
> **Response to Questions**
>
> > **Q1+W2: Same dataset for tuning $\lambda$ and conformal calibration. How do you reconcile with independence requirement? Provide empirical or theoretical evidence that coverage is preserved.**
>
> As addressed in the general response, we provide **both** standard and conformalized configurations with nearly identical empirical coverage. Appendix B (Lemma B.2) provides finite-sample marginal coverage guarantees for the conformalized version. Across all 17 datasets, both versions substantially outperform baselines in interval width and quantile loss while maintaining proper coverage. We transparently report results for both configurations (Appendices F and G) and let users choose based on their needs. We are happy to provide additional theoretical or empirical evidence if this does not fully address your concerns.
>
> > **Q2+W3: CLEAR presented as evolution of CQR methods. Why are comparisons with UACQR in Appendix D rather than main results? Would including these change conclusions?** Since this is derivative to CQR and other prior methods, not comparing against interval-creation baselines in the main paper seems like a missed step. Why leave competitive results for Appendix D?
>
> We appreciate this feedback but we believe there has been a misunderstanding. In the main paper (Section 4.2, Figure 3), we compare against PCS (ensembles), ALEATORIC (i.e., CQR with bootstrapping), and ALEATORIC-R (i.e., CQR fitted on the residuals with bootstrapping) which **are all interval-creation** baselines. If the reviewer was referring specifically to UACQR, we do compare with it only in **Appendix G.3 (not D)**, and we relegated this to the appendix because: (1) UACQR requires specific base models (QRF for aleatoric), making direct comparison less straightforward for variants (a) and (b), (2) space constraints in the main paper. However, the results strongly favor CLEAR: on 14/17 datasets, CLEAR strongly outperforms both UACQR variants (Tables 41-45 in Appendix G.3) .
>
> In the revised manuscript, we have created a summary table for the results between CLEAR and UACQR in the main paper, and we accordingly moved the paragraph on the interpretation of the UACQR results from second paragraph of Appendix A.1 to the main paper. We realized that the table numbers we refer to in this paragraph are actually 41-45 (not 24, 26), and we apologize for this minor typo and have corrected it accordingly in the revised version.
>
> > **Q3+W4: Much substantive discussion in appendix. Could authors explain rationale and whether key arguments could move to main text?**
>
> Thank you for this feedback but we had to make decisions due to the 9-page limit for the submission. We present the core of the main results in the paper, which also contain the results for the comparisons showing the superior performance of CLEAR. If there are any concrete advices on which part of the main paper can be replaced with the appendices, we can gladly consider them.
>
> We hope our detailed responses demonstrate that CLEAR's contributions extend well beyond "incremental refinement." In summary, the method provides: (1) novel technique to fit the aleatoric uncertainty on residuals, (2) two-parameter calibration framework, (3) theoretical guarantees, (4) substantial empirical improvements, and (5) a novel extension of the PCS-UQ method.
>
> Given that the core of the critique was a potential underestimation of this novelty, and the lack of a conformal version, which we do actually provide in Appendices B and G, **we respectfully ask you to revisit your evaluation in light of these clarifications**. We strongly believe this work represents a significant contribution to uncertainty quantification and hope you will now support it for **Acceptance**. We would also be delighted to continue our discussion and provide any further clarifications.

---

> > ### Comment · Reviewer_yexK · 2025-11-20
> >
> > I appreciate the author's detailed response. I have some further questions if you don't mind. :)
> >
> > > Joint calibration of aleatoric and epistemic components is a very non-trivial step. Treating these two sources of uncertainty separately and calibrating both are precisely what UACQR is missing, and this idea goes beyond a simple view that CLEAR only calibrates one additional parameter. We see this separation of roles as a novel principle that can extend to many ML settings. One parameter can simply be adjusted by calibrating the marginal coverage. For calibrating two parameters, a secondary criterion, such as the quantile loss, is necessary to obtain a uniquely defined solution.
> >
> > A: I appreciate the authors’ perspective on the importance of jointly calibrating aleatoric and epistemic components. However, I am still not fully convinced...yet. The paper’s theoretical section establishes asymptotic validity but does not provide a formal justification for the joint calibration of $\gamma_1, \gamma_2$. I acknowledge the authors’ argument, but I remain of the opinion that the contribution is an incremental generalisation of existing CQR-derived approaches.
> >
> > > CLEAR works with all sources of uncertainty in a structured way. While the simulations rely on bootstrap ensembles similarly to UACQR, the case study shows how the PCS framework can guide practitioners toward better epistemic estimators (including the epistemic uncertainty arising from data cleaning and data pre-processing). This connection between PCS and CQR can be more than interesting in many practical settings.
> >
> > A: Maybe I am mistaken and would be happy to admit so, but UACQR can also work with all sources of uncertainty is a structured way. From a methodological perspective, CLEAR still appears to be a natural generalisation of the UACQR formulation...which is fine as it is not my strongest issue with the paper.
> >
> > > Unlike UACQR, CLEAR estimates the aleatoric component on residuals rather than on the raw targets...
> >
> > A: I shall answer in points, as this is quite a large rebuttal. You state that CLEAR "has a strong effect on stability and accuracy and, in our view, is a contribution on its own". I agree with this fully, and actually appreciate this results from CLEAR. You state "much simpler to understand the source of uncertainty when using our technique compared to UACQR", which I agree with that CLEAR does separate the sources of uncertainty and is transparent unlike other methods. However, UACQR also separates uncertainty and thus is equally transparent. So I am happy for the authors to claim that their method doesn't reduce this fact, but I am not sure if it is justifiable to claim it over UACQR.
> >
> > > The framework offers potential additional interpretability into the sources of uncertainty. The calibrated
> >  quantifies the relative dominance of aleatoric and epistemic uncertainty and opens the door to methodological extensions. We do not emphasise this perspective heavily in the current work, but our formulation allows for useful interpretations in the future.
> >
> > A: I agree that in principle the parameters $\gamma_1, \gamma_2$ could encode information about the relative scaling of aleatoric and epistemic uncertainty. Do you have any preliminary results/figures to support this? I understand you state it is "future work", however, a rough result would be HEAVILY appreciated.
> >
> > > We also carry out a substantially larger empirical study than prior work for benchmarking. We consider significantly more real-world datasets than the UACQR paper. Our experiments over hundreds of synthetic datasets are didactically useful to better explain the importance of combining epistemic and aleatoric uncertainty. The figures 2, 4, 5, and 6 clearly demonstrate why CQR totally fails to provide input conditional coverage, even for large training datasets, despite its theoretical guarantee of asymptotic input conditional coverage, as we discuss in Appendix B.4. Many theoretical statisticians struggle to understand this. We find these figures helpful for highlighting this major issue.
> >
> > A: I appreciate the breadth of the empirical study (I actually think the breadth is of excellent quality) and agree that the additional synthetic examples are pedagogically useful. However, that does not address my core concern that the paper's best results/discussion are in the appendix and not the main paper + CLEAR’s methodological novelty.

---

> > > ### Comment · Reviewer_yexK · 2025-11-20
> > >
> > > > We identified that the PCS-framework, which already covers many different types of epistemic uncertainty, can be further improved by adding aleatoric uncertainty (e.g, via quantile regression). The PCS framework has shown significant improvements for multiple real-world problems. However, in the PCS-related literature, aleatoric uncertainty was not mentioned before. While input-conditional coverage guarantees already existed for QR-based methods, we are the first to provide theoretical guarantees on asymptotic input-conditional coverage for a PCS-derived method. The previous PCS-UQ-algorithms proposed in the literature did not achieve asymptotic input-conditional coverage.
> > >
> > > A: Thanks for the pointer to Lemma B.2. This does address the independence requirement for that specific configuration with a separate validation and calibration split. However, my original concern was about the default implementation promoted in the main paper, where the same dataset is used both to tune $\lambda$ and $\gamma_1$ and for conformal calibration (i.e., D_cal = D_val). As you note in Appendix B.1, this breaks the standard split-conformal assumptions and removes finite-sample guarantees for that default setting. Empirical coverage on 17 datasets is reassuring but the removal of the finite-sample guarantees is an strong issue for me.
> > >
> > > > In the revised manuscript, we have created a summary table for the results between CLEAR and UACQR in the main paper, and we accordingly moved the paragraph on the interpretation of the UACQR results from second paragraph of Appendix A.1 to the main paper. We realized that the table numbers we refer to in this paragraph are actually 41-45 (not 24, 26), and we apologize for this minor typo and have corrected it accordingly in the revised version.
> > >
> > > A: Thank you for the clarification and for adding a summary UACQR table to the main paper. This has somewhat improved my issue, constraints on base models or space are understandable.
> > >
> > > > Thank you for this feedback but we had to make decisions due to the 9-page limit for the submission. We present the core of the main results in the paper, which also contain the results for the comparisons showing the superior performance of CLEAR. If there are any concrete advices on which part of the main paper can be replaced with the appendices, we can gladly consider them.
> > >
> > > A: I appreciate the page-limit challenges (and they provide a pain for most authors submitting, so I am understanding), although my concern is not simply quantity but prioritization: several key methodological discussions (e.g., the conceptual framing, the PCS integration, and the UACQR comparison) appear only in the appendix, which makes the main paper feel underspecified from the perspective of positioning and contribution clarity.
> > >
> > > > Given that the core of the critique was a potential underestimation of this novelty, and the lack of a conformal version, which we do actually provide in Appendices B and G, we respectfully ask you to revisit your evaluation in light of these clarifications. We strongly believe this work represents a significant contribution to uncertainty quantification and hope you will now support it for Acceptance. We would also be delighted to continue our discussion and provide any further clarifications.
> > >
> > > A: I am always happy to revisit my evaluation, and I think you have answered some key concerns that I had in the paper. If you are able to clarify a few further points that I have stated above, that would be fantastic! Thank you.

---

> > > > ### Author Response · Authors · 2025-11-22
> > > >
> > > > We sincerely thank you for your continued engagement and for the opportunity to clarify these remaining points. We are glad to hear that the summary table and the clarification on the residual-based contribution have improved your assessment.
> > > >
> > > > First, we address why this work is suitable for ICLR, the prioritization of content (e.g., main paper vs appendix) and on CLEAR being an incremental extension of UACQR. As the reviewer is well aware, ICLR is a highly diverse conference with both very applied submissions (with an impact on real-world applications) and highly theoretical submissions (with sophisticated mathematical innovations). We think both are an important part of ICLR, and we understand that each reviewer has certain preferences in one direction or the other.
> > > >
> > > > UACQR offers an excellent methodological innovation that can be derived directly from theory, and it delivers empirical improvements of up to almost 6\% over CQR. We find the overall paper to be great. UACQR combines ensembling (epistemic) with QR (aleatoric). Probably half of the ML community would say UACQR is an extension of ensembling, and the other half would say it is an extension of CQR. In their paper, they choose the second option and do not even compare it to other ensembling methods, not even in the appendix, which is okay because they target a more theoretical audience at AISTATS. This does not diminish the paper's excellent contributions.
> > > >
> > > > On the other hand, CLEAR can also be seen as an extension of any of these methods: ensemble, PCS-UQ, UACQR, CQR, QR, or SQR. This is why we empirically compare CLEAR against all of these methods on 17 real-world datasets. CLEAR substantially outperforms all of them. For example, CLEAR outperforms UACQR by up to 60\% or more in 3 different metrics (https://openreview.net/pdf?id=RY4IHaDLik#page=9). From an applied perspective, improving these metrics by more than 60\% is well beyond an incremental improvement. We agree that for some of the techniques used by CLEAR, it is not at all trivial to mathematically prove why they are so beneficial in practice. However, their theoretical derivation being non-trivial does not imply that they are worthless (we think this creates potential for future theory). Our hearts bleed a bit over moving most of the theory to the appendix, but we think that, for CLEAR, the empirical results are the paper's core strengths and need to be properly prioritized in the main paper. We think CLEAR and UACQR are both excellent papers, with different priorities. UACQR focuses more on theory, and CLEAR focuses more on empirical results. We are deeply convinced that both prioritizations add unique, powerful value to the diverse ICLR community.
> > > >
> > > > Below, we provide specific answers to your new questions (using two OpenReview comments).
> > > >
> > > > > *Do you have any preliminary results/figures to support this [interpretability]?*
> > > >
> > > > Yes. Section 4.3 (Ames Housing) demonstrates this behavior:
> > > > *   2-variable case (high aleatoric noise): CLEAR selected $\lambda = 0.6$, down-weighting the epistemic component.
> > > > *   Full-feature case (high epistemic uncertainty): CLEAR selected $\lambda = 14.5$, emphasizing the epistemic component.
> > > >
> > > > The shift in $\lambda$ directly reflects the dominant source of uncertainty: the effective interval width ratio (epistemic/aleatoric) shifted from 0.03 to 7.72, confirming $\lambda$ adapts to the dominant uncertainty. In the two-variable case, the large aleatoric component led CLEAR to select $\lambda = 0.6$, indicating that uncertainty cannot be reduced without collecting more informative covariates. In the full-feature case, CLEAR selected $\lambda = 14.5$, where epistemic uncertainty dominates and could be reduced by increasing sample size. This is consistent with the observed change in the effective interval width ratio from 0.03 to 7.72, demonstrating that $\lambda$ adapts to whether uncertainty arises from limited data or insufficient features.
> > > >
> > > > We treat this as a preliminary finding for future work. Rigorous interpretation requires further work, which is not the focus of our paper.

---

> > > > > ### Author Response · Authors · 2025-11-22
> > > > >
> > > > > > *The paper... does not provide a formal justification for the joint calibration.*
> > > > >
> > > > > We can provide an argument for Joint Calibration. Let the optimal interval width be $W^{\ast}(x) = W\_{\text{ale}}^{\ast}(x) + W\_{\text{epi}}^{\ast}(x)$.
> > > > >
> > > > > Base estimators often have the wrong multiplicative scale:
> > > > > $$
> > > > > \hat{W}\_{\text{ale}}(x) \approx a \cdot W\_{\text{ale}}^{\ast}(x) \quad \text{and} \quad \hat{W}\_{\text{epi}}(x) \approx b \cdot W\_{\text{epi}}^{\ast}(x)
> > > > > $$
> > > > > where $a, b > 0$.
> > > > >
> > > > > * Single-parameter methods: Produce $\gamma (a W\_{\text{ale}}^{\ast} + b W\_{\text{epi}}^{\ast})$. If $a \neq b$, this has the wrong shape relative to $x$.
> > > > > * Joint Calibration (i.e., CLEAR): Uses $\gamma_1 \hat{W}\_{\text{ale}} + \gamma_2 \hat{W}\_{\text{epi}}$. By selecting $\gamma_1 \approx 1/a$ and $\gamma_2 \approx 1/b$, CLEAR recovers the true shape $W^{\ast}(x)$.
> > > > >
> > > > > Do you want this in the paper using more formal language? For example, we can prove that the improvement (test quantile loss) can be arbitrarily large if the scale of the original estimators is arbitrarily bad. We could also prove that in the limit of an infinitely large calibration dataset, CLEAR cannot be worse in terms of quantile loss than conformal baselines where either $\gamma_1$ or $\lambda$ is fixed. In Appendix B.3.2, we explain why the quantile loss measures input-conditional coverage as a strictly proper scoring rule.
> > > > >
> > > > > As we understand the paper behind UACQR, there is no formal proof showing the advantages of UACQR over CQR. By formulating Hypothesis B.8 in Appendix B.4, we move one step in the direction of formally showing the severe problems of plain CQR.
> > > > >
> > > > > > *The removal of the finite-sample guarantees is a strong issue for me.*
> > > > >
> > > > > We reiterate that we provide the theory, results, and implementation for two versions: Standard and Conformalized CLEAR, the latter of which provides finite-sample guarantees. Here are the justifications:
> > > > >
> > > > > 1.  Strict Guarantees: Conformalized CLEAR (Conformalized mode) fully preserves finite-sample guarantees (Lemma B.2).
> > > > > 2.  Trade-off: "Standard" mode reuses $\mathcal{D}\_{\text{val}}$ for data efficiency, similar to cross-validation.
> > > > > 3.  Bound: Bias from reusing data is bounded by $O(1/\sqrt{|\mathcal{D}\_{\text{val}}|})$ (Zeng et al., 2025).
> > > > >
> > > > > Again, as mentioned in our general response, the standard configuration is our default recommendation for practical use, as it maximizes data efficiency while maintaining excellent empirical coverage. As you also acknowledged, we provide results for both, and the code implementation supports both approaches, allowing users to freely choose based on their preferences for theoretical guarantees vs. maximal data use. This is especially relevant for some smaller datasets, such as energy efficiency, which has 768 observations. Instead of dividing the 20\% validation set into two equal parts ($\approx 77$ for each $D\_{\text{val}} $ and $ D\_{\text{cal}}$), which introduces biases (formal justifications in lines 1458-1466), we can leverage both for validation and calibration. We must also note that, in many practical domains, such as medicine, where CLEAR may be useful, users may not have the luxury of many observations; to mitigate this, we aim to do so through our recommended standard approach. Furthermore, the exchangeability assumption underlying the formal guarantee is rarely satisfied in practice (see Appendix~B.1.1).
> > > > >
> > > > > - [1] Zeng, Hao, Kangdao Liu, Bingyi Jing, and Hongxin Wei. "Parametric Scaling Law of Tuning Bias in Conformal Prediction". Forty-Second International Conference on Machine Learning (2025).
> > > > >
> > > > > >  *...My concern is not simply quantity but prioritization: (e.g., the conceptual framing, the PCS integration, and the UACQR comparison).*
> > > > >
> > > > > We are very thankful to the reviewer for acknowledging the limitations that come with the number of pages. On the theory, we really tried to provide the main point in Lemma 2.1 and then further justify it in Lemma B.2. On PCS, we are not sure what the reviewer meant by "integration"; if it's a reference to the implementation details, this is delegated to the appendices, since we also use CQR, DE and SQR, all of which deservedly can be included. Note that we do explain the main ideas behind PCS-UQ in Section 2.2. Still, as you very well pointed out, we had to make hard choices. Now, to convince the reviewer of the changes, in the camera-ready version, we have included:
> > > > > 1.  The UACQR summary table will remain in the main text.
> > > > > 2. Is there any further theoretical result that you want us to add in Section 2.4?
> > > > >
> > > > > The content is present in the paper, and we really aimed to be inclusive by allowing readers to focus on the parts of the paper that interested them in the most (e.g., experiments vs. theory). The one takeaway of the paper is that CLEAR just works well. If the reviewer still finds our answer unsatisfactory, we can gladly consider other specific changes.
> > > > >
> > > > > We hope these clarifications address your concerns about accepting our paper.

---

> > > > > > ### Comment · Reviewer_yexK · 2025-11-23
> > > > > >
> > > > > > Thank you for your response. I have temporarily updated my rating of the paper to reflect my satisfaction with a few of your points made.
> > > > > >
> > > > > > > ICLR is a highly diverse conference with both very applied submissions
> > > > > >
> > > > > > A: I agree, I can very much appreciate that both heavy theory papers and empirical papers have equal place at A* venues like ICLR. I am fond of this paper, in fact.
> > > > > >
> > > > > > > Section 4.3 (Ames Housing) demonstrates this behaviour
> > > > > >
> > > > > > A: Thank you for pointing me to this. I must have missed this. Yes, this experiment does defend this interpretability claim somewhat. Another example would be appreciated to demonstrate the effect better, but I am not too fussed.
> > > > > >
> > > > > > > Do you want this in the paper using more formal language?
> > > > > >
> > > > > > A: Thank you for outlining the potential direction for a formal justification of joint calibration. This would be good to be in the paper and strengthen your method.
> > > > > >
> > > > > > > Joint Calibration (i.e., CLEAR): Uses $\gamma_{1} \hat{W}_{\text{ale}} + \gamma_{2} \hat{W}_{\text{epi}}$
> > > > > >
> > > > > > A: The formulation seems to assume the optimal width is a global linear combination of aleatoric and epistemic terms. Given that aleatoric uncertainty is heteroscedastic and epistemic error is model-dependent, this linear, globally weighted mix feels overly restrictive and risks missing spatial miscalibration. Why is this not overly naive, and what evidence supports linearity here?
> > > > > >
> > > > > > Thank you for your continued discussion and defence of your paper. Given an adequate response to my final point here, I would be inclined to reevaluate my score once again.

---

> > > > > > > ### Author Response · Authors · 2025-11-24
> > > > > > >
> > > > > > > We are delighted to hear that you are fond of the paper and appreciate your constructive engagement throughout this process. We are happy to address your final questions below.
> > > > > > >
> > > > > > > > A: Thank you for pointing me to this. I must have missed this. Yes, this experiment does defend this interpretability claim somewhat. Another example would be appreciated to demonstrate the effect better, but I am not too fussed.
> > > > > > >
> > > > > > > We are glad this clarified the interpretability aspect. Following the reviewer's request for another example, we present a table of $\lambda$ and $\gamma_1$ values for the Ames Housing dataset, but retaining only 50\% and 20\% of the training samples (still using all features), respectively. **In these scenarios, one can expect epistemic uncertainty to increase further.** Our two central claims about the interpretation of epistemic and aleatoric uncertainty are 1) that aleatoric uncertainty can be decreased by adding more features (equivalently, increases with fewer features) and 2) that epistemic uncertainty can be decreased by collecting more training observations (or equivalently, increases with fewer training observations). **This is precisely what we observe with an increasing $\lambda$ and Epistemic/Aleatoric Ratio** that we defined in our paper as
> > > > > > > $$
> > > > > > > \lambda^\star \frac{\hat{q}\_{1-\alpha/2}^{\text{epi}}(x)+\hat{q}\_{\alpha/2}^{\text{epi}}(x)}{\hat{q}\_{1-\alpha/2}^{\text{ale}}(x)+\hat{q}\_{\alpha/2}^{\text{ale}}(x)} \quad \text{(Line 207)}
> > > > > > > .$$
> > > > > > >
> > > > > > > | Experiment              | $\lambda$ | $\gamma_1$ | E/A Ratio |
> > > > > > > |-------------------------|--------------------|------------------|---------------------------|
> > > > > > > | 2 features              | 0.64               | 0.99             | 0.03                      |
> > > > > > > | All features            | 14.45              | 0.13             | 7.72                      |
> > > > > > > | All features, 50\% data | 17.97              | 0.09             | 14.09                     |
> > > > > > > | All features, 20\% data | 100                | 0.01             | 250.5                     |
> > > > > > >
> > > > > > > > A: Thank you for outlining the potential direction for a formal justification of joint calibration. This would be good to be in the paper and strengthen your method.
> > > > > > >
> > > > > > > We are committed to include the formal justification discussed in our previous response in the final manuscript to strengthen the method's theoretical grounding.
> > > > > > >
> > > > > > > Which of the following aspects of the formal justification do you consider most important for the main paper?
> > > > > > >
> > > > > > > * (a) Explaining that CLEAR can compensate for a wrong multiplicative scale of aleatoric and epistemic uncertainty.
> > > > > > >
> > > > > > > * (b) Prove that the improvement (test quantile loss) can be arbitrarily large if the scale of the original estimators is arbitrarily bad.
> > > > > > >
> > > > > > > * (c) Prove that in the limit of an infinitely large calibration dataset, CLEAR cannot be worse in terms of quantile loss than conformal baselines where either $\gamma_1$ or $\lambda$ is fixed.
> > > > > > >
> > > > > > > * (d) explain why the quantile loss measures input-conditional coverage as a strictly proper scoring rule, as in Appendix B.3.2.
> > > > > > >
> > > > > > > * (e) Something in the direction of Hypothesis B.8 in Appendix B.4, in order to explain severe systematic problems of plain CQR. A proof of Hypothesis B.8 would probably outside the scope of this paper, but we could outline the potential future direction.
> > > > > > >
> > > > > > > In our next comment, we provide an elaborate answer to your final question.

---

> > > > > > > > ### Author Response · Authors · 2025-11-24
> > > > > > > >
> > > > > > > > > A: The formulation seems to assume the optimal width is a global linear combination of aleatoric and epistemic terms. Given that aleatoric uncertainty is heteroscedastic and epistemic error is model-dependent, this linear, globally weighted mix feels overly restrictive and risks missing spatial miscalibration. Why is this not overly naive, and what evidence supports linearity here?
> > > > > > > >
> > > > > > > > This is an excellent question. Below, we give five motivations behind choosing a global linear combination of aleatoric and epistemic terms instead of something more complex:
> > > > > > > >
> > > > > > > > **1. Linearity follows the same rationale as classical conformal calibration.** In classical split-conformal inference, calibration is also done by fitting a linear function to rescale a score (e.g. residual or quantile width). One can view this as selecting a correction function from a function class $\mathcal{F}$ using a calibration set. Standard conformal methods constrain $\mathcal{F}$ to linear functions for stability and validity, though more general forms have been explored (e.g. Marx et al., 2022 [1]). There is nothing preventing a richer parameterization of $\mathcal{F}$ in CLEAR; we simply adopt the most widely used, theoretically tractable calibration form.
> > > > > > > >
> > > > > > > > **2. Simplicity (Occam’s razor).** Appendix I.4 briefly describes alternative calibration options. In our experiments, the linear form provides strong empirical performance without overfitting and keeps the method transparent and interpretable. We think that these 2 degrees of freedom are essential for the two sources of uncertainty, which can be on very different scales. However, each additional degree of freedom increases the risk of overfitting, and we don't think that there is another equally important degree of freedom to be added, or at least we lack the intuition for what could be an equally important third degree of freedom. This is why we propose calibrating $\gamma\_1$ and $\gamma\_2$.
> > > > > > > >
> > > > > > > > **3. Interpretability.** With a linear calibration, the ratio  $\lambda = \gamma_1/ \gamma_2$  directly quantifies how much aleatoric vs. epistemic variability contributes to the final interval. If one were to use a richer calibration function, e.g.
> > > > > > > > $$C(x)=\left[\hat{f}(x) \pm \big(\gamma_1 \times \text{aleatoric}(x) + \gamma_2 \times \text{epistemic}(x) + \gamma_3 \times \text{aleatoric}(x) \times \text{epistemic}(x)\big)\right],
> > > > > > > > $$
> > > > > > > > this interpretability disappears, since the scaling parameters no longer isolate the two uncertainty sources. Even if a more complex mapping improves accuracy, it loses a transparent link between the fitted interval and the underlying uncertainty mechanisms, which is often important and one of the goals of CLEAR.
> > > > > > > >
> > > > > > > > **4. Further reasons for Additivity.**
> > > > > > > > Many established approaches, e.g., Bayesian methods such as Gaussian process regression and deep ensembles, already combine aleatoric and epistemic uncertainty in an almost additive way, e.g., $\sqrt{W_\text{ale}^2 + W_\text{epi}^2}$ under Gaussian assumptions. Switching to a Cauchy assumption yields exact additivity, $W_\text{ale} + W_\text{epi}$. In practice, the difference between these two forms is minimal; in our preliminary experiments, simple additivity performed similarly, if not slightly better. Thus, additive (or nearly additive) combinations already achieve their intended purpose (see 2. Simplicity).
> > > > > > > >
> > > > > > > > **5. Further reasons for a "global" linear combination.**
> > > > > > > > The fact that the calibration parameters $\gamma_1$ and $\gamma_2$ are global does not mean that the resulting intervals are spatially constant or naive. Quantile regression already captures relative aleatoric heteroscedasticity, and ensemble diversity already captures relative epistemic variability (Appendix I). What these methods typically get wrong is their global absolute scale. For example, using too few ensemble perturbations leads to globally underestimated epistemic uncertainty, and insufficient regularization in quantile regression leads to globally underestimated aleatoric uncertainty. The two global parameters, $\gamma=\gamma_1$ and $\lambda=\gamma_2/\gamma_1$, are easy to calibrate on a held-out dataset: $\gamma$ corrects global under/overconfidence, while $\lambda$ adjusts the balance between in-sample and out-of-sample overconfidence (Figure 1), since the quantile loss naturally emphasizes input-conditional calibration (Appendix B.3.2). By contrast, local input-conditional adjustments would introduce infinitely many degrees of freedom in continuous input spaces, risking severe overfitting (2. Simplicity).
> > > > > > > >
> > > > > > > > If you prefer, we could provide further explanations of why we chose this linear structure.
> > > > > > > >
> > > > > > > > - [1] Marx et al., “Modular Conformal Calibration”, ICML 2022. https://proceedings.mlr.press/v162/marx22a/marx22a.pdf
> > > > > > > >
> > > > > > > > We hope this addresses your final concern regarding the linearity assumption. We are grateful for your time and the opportunity to improve our paper, and hope the final answer is satisfactory to improve your score.

---

> > > > > > > > > ### Comment · Reviewer_yexK · 2025-11-25
> > > > > > > > >
> > > > > > > > > Thank you to the authors for taking time to discuss their paper even further, and providing a lot of the analysis I requested. Below are my final points for this paper.
> > > > > > > > >
> > > > > > > > > > Following the reviewer's request for another example, we present a table of $\lambda$ and $\gamma_1$ values for the Ames Housing dataset.
> > > > > > > > >
> > > > > > > > > A: Thank you,  this table and the added scenarios make the points much clearer.
> > > > > > > > >
> > > > > > > > > > Which of the following aspects of the formal justification do you consider most important for the main paper?
> > > > > > > > >
> > > > > > > > > A: Personally, I would put item (c) in the main text as a compact theorem, “with an infinitely large calibration set, joint calibration is no worse than single-parameter baselines”.
> > > > > > > > >
> > > > > > > > > > Below, we give five motivations behind choosing a global linear combination of aleatoric and epistemic terms instead of something more complex:
> > > > > > > > >
> > > > > > > > > A: Thank you for the detailed rationale. Points (1)-(2) make a clear, pragmatic case for a global linear calibration and are worth stating concisely in the main text as the design choice (given space, as I understand this is a pain). This is more analysis than I could have asked for and has thoroughly convinced me, it is much appreciated.
> > > > > > > > >
> > > > > > > > > I now recommend acceptance. The authors engaged **substantively** with every concern, adding the UACQR comparison to the main paper, clarifying the Standard vs. Conformalized variants (with finite-sample guarantees for the latter), providing a clear rationale and sketch proof for joint calibration, and supplying additional interpretability evidence ($\lambda$/$\gamma_1$). These revisions improve positioning, transparency, and theoretical grounding, and they address weaknesses raised by multiple reviewers, including my own. While the contribution will always be relative to UACQR, the paper now presents a coherent and well-justified method with strong empirical performance across diverse datasets, clear trade-off discussions, and actionable guidance for practice.
> > > > > > > > >
> > > > > > > > > **In my view the authors have satisfied the key concerns, and the revised manuscript will be valuable to the ICLR community and to the UQ/CP community as a whole. I recommend the acceptance of this work.**

---

> > > > > > > > > > ### Author Response · Authors · 2025-11-25
> > > > > > > > > >
> > > > > > > > > > We sincerely thank you for your time, your detailed engagement, and your recommendation for acceptance. We are very glad to hear that our responses have addressed your concerns.
> > > > > > > > > >
> > > > > > > > > > In addition to the changes that were already included in the revised manuscript, we will proceed with your recommendations for the camera-ready version:
> > > > > > > > > >
> > > > > > > > > > 1. Formal Justification: We will include the formal justification regarding the limit of an infinitely large calibration set (item c) as a theorem in the main text.
> > > > > > > > > >
> > > > > > > > > > 2. Linearity: We will include the pragmatic motivations for the global linear calibration (points 1 and 2 from our previous response) in the main text to clarify this design choice.
> > > > > > > > > >
> > > > > > > > > > Your feedback has significantly strengthened the positioning and theoretical clarity of our work. We appreciate the productive discussion.

---

### Official Review · Reviewer_MNyo · 2025-10-31

**Soundness:** 3
**Presentation:** 3
**Contribution:** 3
**Rating:** 6
**Confidence:** 3

**Summary:**

This paper introduces CLEAR (Calibrated Learning for Epistemic and Aleatoric Risk), a novel framework for constructing regression prediction intervals by adaptively balancing epistemic uncertainty and aleatoric uncertainty. Theoretically, it guarantees asymptotic conditional validity; empirically, on 17 real-world datasets and synthetic data with distribution shifts, CLEAR outperforms baselines by reducing interval width and quantile loss while maintaining nominal coverage.

**Strengths:**

1. Unlike methods that use a fixed ratio to combine the two uncertainties, CLEAR selects the balance parameter based on data characteristics. For example, it emphasizes aleatoric uncertainty when working with data with few features and epistemic uncertainty when using data with many features, making it flexible across different scenarios.
2. It works with various uncertainty estimation methods (including tree-based PCS and deep learning-based Deep Ensembles or Simultaneous Quantile Regression) and maintains reliable coverage even for data points outside the training data distribution or in extrapolation regions—areas where many baseline methods struggle.

**Weaknesses:**

1. The proof requires that "at least k base models in the PCS ensemble are consistent with the true function," but it does not define specific criteria for determining consistency (such as error convergence thresholds) nor explain the basis for selecting k. In practical experiments, the consistency of different models varies significantly, yet the paper fails to analyze the risk of theoretical guarantees failing in such scenarios.

2. The additivity of the two types of uncertainty has not been proven — epistemic uncertainty and aleatoric uncertainty essentially belong to risks of different dimensions. The additive combination implies the assumption that the two can be directly superimposed on the numerical scale, but the paper does not verify this assumption. For example, it does not compare the performance differences between additive, multiplicative, and nonlinear combinations.

3. In some datasets, there is a significant correlation between the two types of uncertainties. The additive combination may amplify the uncertainty superposition effect, leading to overly wide intervals. In low-correlation datasets, however, the additive combination may underestimate risks due to improper weight allocation. The paper does not analyze the impact of uncertainty correlation on the combination structure, which limits the generality of the method.

**Questions:**

1. Regarding computational efficiency, CLEAR’s grid search for lambda (over 4000 points) consumes significant resources, especially for large datasets. Have you explored adaptive search strategies? If yes, what was the reduction in computational time while maintaining performance?
2. This paper verifies CLEAR’s performance on 17 regression datasets, but most of these datasets have relatively balanced feature distributions. For high-dimensional sparse datasets (e.g., tabular data with hundreds of features where most are irrelevant), how does CLEAR’s performance change?

---

> ### Author Response · Authors · 2025-11-16
> **Clarification on Weaknesses**
>
> We thank Reviewer MNyo for recognizing CLEAR's adaptive balancing of uncertainties, flexibility across methods, and reliable coverage even in extrapolation regions.
>
> > **W1: Proof requires "at least k base models consistent with true function" but doesn't define specific criteria for consistency or basis for selecting k. No analysis of risk when consistency varies.**
>
> As addressed in the general response, we use $k=1$ (top-performing model), so Lemma 2.1 requires only **one consistent estimator** (see Appendix B.2). Consistency is satisfied by standard estimators under regularity conditions. Also, as mentioned in our general response, if no base model is consistent, CLEAR inherits the same asymptotic bias as the base models, but the dual-parameter calibration ($\gamma_1$, $\lambda$) still provides empirical robustness by adaptively reweighting components, as demonstrated across 17 diverse real-world datasets.
>
> > **W2: Additivity of two uncertainty types not proven. Epistemic and aleatoric are different dimensions. No comparison of additive vs. multiplicative vs. nonlinear combinations.**
>
> The additive combination is simple, easy to understand, and works well in practice.
>
> Both uncertainty components are measured in the same units (the units of the target variable).
> CLEAR can adjust the scales multiplicatively via $\gamma_1$ and $\gamma_2$ if the epistemic and aleatoric uncertainty are at different levels of magnitude, i.e.,  $\text{total}\_\text{width} = \gamma_1 \times \text{aleatoric}\_\text{width} + \gamma_2 \times \text{epistemic}\_\text{width}$.
>
> For some datasets, it might make sense to apply a log-transformation on the targets as a pre-processing step, if the target variables are always strictly positive. This would correspond to a multiplicative adjustment to the uncertainty of the original target variables. We agree that pre-processing can be an essential step in data science projects. CLEAR can also be applied to data that has already undergone a log-transform. Multiplicative uncertainty would imply that the upper and lower bounds of a predictive interval would need to have the same sign. If you have strong reasons to believe that for any of our 17 datasets a log-transform of the outputs would be beneficial, we can consider them.
>
> The goal of our paper was not to introduce a method that looks as sophisticated as possible, but to provide the simplest possible solution for a real-world problem. CLEAR works reliably with negligible computational overhead, achieving very strong results across 17 real-world datasets, hundreds of synthetic datasets, and the real-world case study with distribution shift. There is potential in further extending CLEAR to other non-linear combinations. However, allowing for arbitrary non-linear combinations of aleatoric and epistemic uncertainty, e.g., via a (monotonic) neural network $g\_\theta(\text{aleatoric},\text{epistemic})$, would create serious risks for overfitting on the calibration dataset. What is nice about CLEAR is that we only need to calibrate two parameters, which results in almost no overfitting already for moderate sizes of the calibration dataset. With that said, we agree that, for scenarios with large calibration datasets, exploring generic non-linear calibration methods could further improve results and be an interesting area for future work.
>
> > **W3: Correlation between uncertainties may cause issues. High correlation: additive may amplify. Low correlation: may underestimate risks. No analysis of correlation impact.**
>
> We actually do have a detailed analysis of the different correlation scenarios you've described.
>
> * In Figures 2 and 6, the epistemic and aleatoric uncertainty are uncorrelated (since the aleatoric uncertainty is constant, as you can see by comparing the blue and the green curve in the right plot). Here, CLEAR has the best conditional calibration across all considered methods.
> * In Figure 4 in Appendix C, there is a strong positive correlation between epistemic and aleatoric uncertainty (since the aleatoric uncertainty increases out-of-distribution, as you can see by comparing the blue and the green curve in the right plot). In this case, CLEAR also has the best conditional calibration across all considered methods.
> * In Figure 5 in Appendix C, there is a strong negative correlation between epistemic and aleatoric uncertainty (since the aleatoric uncertainty decreases out-of-distribution, as you can see by comparing the blue and the green curve in the right plot). Here, CLEAR has the best conditional calibration across all considered methods.
>
> Could you please elaborate on why you think that in any of these settings, CLEAR would underestimate the risk? On average, CLEAR should always have the correct coverage.
>
> We must also note that our 17 real-world datasets are well-representative of typical correlations between aleatoric and epistemic uncertainty. There CLEAR also achieves strong results.

---

> > ### Author Response · Authors · 2025-11-16
> > **Response to Questions**
> >
> > > **Q1: Grid search over 4000 $\lambda$ points consumes significant resources. Have you explored adaptive search strategies? What was computational time reduction while maintaining performance?**
> >
> > As addressed in the general response, our 1-dimensional grid search is extremely fast in practice, usually less than a second (see Appendix F.5, Tables 28). We are happy to discuss this further if you have specific concerns.
> >
> > > **Q2: This paper verifies CLEAR’s performance on 17 regression datasets, but most of these datasets have relatively balanced feature distributions. For high-dimensional sparse datasets (e.g., tabular data with hundreds of features where most are irrelevant), how does CLEAR’s performance change?**
> >
> > We provide a summary of all datasets in Table 11 (end of Appendix F). Actually, many of the 17 datasets are highly imbalanced in terms of features, and span diverse dimensionalities: from low-dimensional (energy efficiency: 8 features) to moderately high-dimensional (superconductor: 81 features) to very high-dimensional (QSAR: 500 features; allstate: 1037 features). The main results in Figure 3 show that CLEAR outperforms the baselines even for high-dimensional data, especially for allstate that had many sparse (highly-irrelevant) features.
> >
> > We again appreciate the thoughtful questions. We hope our responses demonstrate CLEAR's theoretical soundness, empirical robustness, and computational speed. We hope that our experiments in Appendix C exactly addresses your concerns about (un)correlated uncertainties. We would be grateful if you would consider increasing your confidence and score in light of these clarifications. If there are any other concerns, we also happy to continue our discussion.

---

> > > ### Comment · Reviewer_MNyo · 2025-11-26
> > >
> > > The authors have resolved all my concerns, and I maintain my original score.

---

> > > > ### Author Response · Authors · 2025-11-27
> > > >
> > > > We thank the reviewer for the confirmation and for their time spent reviewing our paper. We are glad to hear that all concerns have been resolved.

---

### Official Review · Reviewer_bUL2 · 2025-10-31

**Soundness:** 3
**Presentation:** 3
**Contribution:** 2
**Rating:** 2
**Confidence:** 3

**Summary:**

The authors propose the CLEAR algorithm for UQ in regression, with the following steps.
1. An epistemic model is built to learn the mean and the boundaries of a confidence interval.
2. The predicted mean is subtracted from the target and an aleatoric model is trained on the residual. The boundaries of the confidence interval is kept.
3. Two coefficients $\gamma_1$ and $\lambda$ are determined, from which $\gamma_2 = \lambda * \gamma_1$. For all possible $\lambda$, separately $\gamma_1$ are computed by calibration, and $\lambda$ that optimizes the evaluation metric is selected.
4. The aleatoric/epistemic intervals around mean are scaled symmetrically with the coefficients.
I find the contribution is the combination of known methods with a grid search, with limited theoretical justification and experimentation.

**Strengths:**

+ Learning the combination of aleatoric and epistrmic uncretainties in one model is an important practical question
+ Reproducibility in the supplementary material
+ Improvement in experiments

**Weaknesses:**

- Contribution is the combination of known methods via grid search
- If the distribution is non-Gaussian, the proposed method is limited due to its strong focus on the mean, and its symmetry. Why not median or another statistic? How about skewed distributions? Bimodal?
-  Theoretical justification is limited

**Questions:**

How can you handle the following limitations of the proposed method:
 - confidence interval is built around the mean $\hat{f}$, which is subtracted from the target for training the aleatoric model. Why not, for example, the median?
- Confidence intervals are treated symmetrically, how bout skewed distributions, or bounded target domain?
- How about shape parameters beyond mean and confidence intervals?
 - It is assumed that if you subtract mean from the target, what is left is aleatoric uncertainty. How about e.g. bimodal?

What is the relation of the proposed method to  https://en.wikipedia.org/wiki/Nonhomogeneous_Gaussian_regression ?  There, also an ensemble is used to calibrate the prediction for the mean (deep ensemble as an epistemic model in the paper). Gaussian variance corresponds to the aleatoric distribution, which actually needs to be assumed at several places in the proposed method.

How do you measure whether the two types of uncertainties are properly balanced? Is there ground truth?

The introduction claims "However, they may suffer from poor conditional coverage, meaning well-calibrated coverage at the individual or subgroup level", i.e. in the literature, marginal calibration is insufficiently solved. How does the proposed method solve the question?

---

> ### Author Response · Authors · 2025-11-16
> **Major Clarification on Weaknesses (Part I)**
>
> We thank Reviewer bUL2 for their review. Most of the reviewers' points follow from some misunderstandings: **CLEAR does not assume any form of symmetry or Gaussianity**. Every component of the method is nonparametric, and the aleatoric model learns the upper and lower quantiles separately, so the final prediction intervals are not symmetric unless the data itself produces symmetry. Nothing in CLEAR relies on a Gaussian assumption.
>
> Your concerns about symmetry, the focus on the mean, or the handling of skewness and heavy-tailed or multimodal distributions seem to stem from a misunderstanding of how the method is built. **CLEAR sits on top of any point predictor and any quantile-based aleatoric estimator**. In fact, we do use median estimation as a base models, not mean. For skewed or bounded targets, the asymmetry is entirely handled by the aleatoric quantiles. Subtracting the central predictor does not impose unimodality and quantile regression on residuals captures arbitrary intervals.
>
> The review refers to **confidence intervals** multiple times. We emphasize that **CLEAR constructs prediction intervals, not confidence intervals**. Prediction intervals quantify uncertainty about any given *future observation* (including both parameter uncertainty and inherent data variability, i.e., both epistemic and aleatoric), while confidence intervals quantify uncertainty about *parameters* (i.e., only epistemic uncertainty). Our work addresses the fundamentally different problem of prediction interval construction for regression, as stated in the abstract and throughout the paper.
>
> Below we try to explain each of your points in more detail. However, we hope that our clarification of this misunderstanding will lead to revisiting your evaluation.
>
> > **W1: "Contribution is the combination of known methods via grid search"**
>
> CLEAR is not just a combination of known methods, and the contribution extends to:
>
> 1. **Novel residual-based aleatoric estimation** (Section 2.3, lines 173-179): Fitting quantile regression on residuals $Y_i - \hat{f}(X_i)$ rather than raw targets $Y_i$ substantially improves stability. As stated in Section 2.3: *"underfitting quantile regression directly on the $y$-values can severely distort aleatoric uncertainty estimates. In contrast, extreme underfitting on residuals, at worst, corresponds to assuming homoskedastic noise, which can be an acceptable bias."* This is a novel methodological contribution with significant practical impact (compare ALEATORIC vs ALEATORIC-R performance in Figures 3,7,8, and all the tables in appendices F.1-.3).
>
> 2. **Dual-parameter calibration framework** (Eq. (1) and Algorithm 1): We are the first to introduce two calibration parameters ($\gamma_1$, $\gamma_2$) to adaptively balance epistemic and aleatoric uncertainties. This is fundamentally different from methods that fix $\lambda=1$ or $\gamma_1=1$ (e.g., UACQR [4]).
>
> 3. **Theoretical contributions** (Lemma 2.1 and Appendix B): First asymptotic conditional coverage guarantees for combined epistemic-aleatoric methods extending PCS, addressing the known weakness of standard conformal methods. We kindly invite you to refer to **(Appendix B) that is entirely dedicated to the theory** titled "Theory: Coverage Guarantees and Theoretical Justifications" discussing the theoretical foundations of CLEAR.
>
> 4. **Comprehensive empirical validation**: 17 datasets, 3 model variants, multiple uncertainty estimators (PCS, CQR, Deep Ensembles, SQR), demonstrating 17-28\% average width reduction while maintaining coverage.
>
> **The grid search is merely an implementation detail** for parameter selection, **not the scientific contribution**.
>
> - [4] Rossellini, R., Barber, R. F., and Willett, R. (2024, April). Integrating uncertainty awareness into conformalized quantile regression. In International Conference on Artificial Intelligence and Statistics (pp. 1540-1548). PMLR.

---

> ### Author Response · Authors · 2025-11-16
> **Major Clarification on Weaknesses (Part II)**
>
> > **W2: "If the distribution is non-Gaussian, the proposed method is limited due to its strong focus on the mean, and its symmetry. Why not median or another statistic? How about skewed distributions? Bimodal?"**
>
> The reviewer raises several points in this weakness, which we can happily address:
>
> - **CLEAR does NOT assume Gaussian distributions.** We use ideas from quantile regression and conformal prediction (Sections 2.2, 2.3), which are completely distribution-free and make no parametric assumptions.
>
> - **Median vs mean:** Our framework supports both. In variants (a) and (b), we use the empirical ensemble median over estimates of the median. In variant (c) in the appendices (F.3 and G.3), we use the empirical ensemble median over estimated means. Therefore, in the majority of our experiments, we used medians, not means. If this is still unclear, please let us know what led to this confusion that we heavily rely on using means.
>
> - For aleatoric uncertainty, we estimate conditional *quantiles* directly via quantile regression (no Gaussian assumption required). If you apply our methods for multiple different quantile levels $\alpha$, you obtain an approximate conditional CDF for each input $x$, which can be multimodal as well, as for CQR. We agree that computing any form of predictive intervals only for one level of $\alpha$ will not reveal if the distribution is multimodal or not. However, CQR and CLEAR can still consistently estimate the correct quantiles, also for multimodal distributions.
>
> - **Symmetric intervals:** The *quantiles themselves adapt to skewness*. For skewed distributions, $\hat{q}\_{0.025}^{\text{ale}}(x) \neq \hat{q}\_{0.975}^{\text{ale}}(x)$.
> To clarify, we never assume symmetry. To avoid any confusion, we have reformulated Eq. (1) to
> $C(x)=\left[\hat{f}(x) \pm \big(\gamma_1 \times \text{aleatoric}\_{\pm}(x) + \gamma_2 \times \text{epistemic}\_{\pm}(x)\big)\right], \quad \text{with } \gamma_2 = \lambda \gamma_1, $.
> Did the old formulation of Eq. (1) lead to this confusion? Please clarify whether there is any further sentence in the paper that leads you to believe that our intervals were symmetric? Our intervals are not symmetric. Neither our epistemic nor our aleatoric uncertainty is symmetric.
>
> - **Empirical validation:** Success across 17 diverse real-world datasets with various distribution shapes demonstrates robustness, some of which are heavily skewed.
>
> > **W3: "Theoretical justification is limited"**
>
> While we understand your perspective, we believe that you may have missed the critical sections showing our theoretical justifications:
>
> - Lemma 2.1 provides **asymptotic conditional coverage guarantees**: stronger than marginal coverage from standard conformal methods.
> - Lemma B.2 (Appendix B) provides **finite-sample marginal coverage guarantees** for conformalized CLEAR. Our theoretical framework is comparable to or stronger than CQR and related baselines (again, see Appendix B)
>
> As a side note, we also observe this in the strong empirical validation across simulations (Figures 2,4-6) and real data (Figure 3 and several tables) supporting the theoretical claims.
>
> If you can pinpoint specific theoretical sections you would like improved, we can gladly continue our discussion and make the necessary modifications.

---

> > ### Author Response · Authors · 2025-11-16
> > **Response to Questions (Part I)**
> >
> > > **Q1: "Confidence interval built around mean which is subtracted from target. Why not median?"**
> >
> > As clarified above: (1) framework supports both median and mean; we use median in most experiments, (2) quantile regression adapts to skewness automatically without parametric assumptions, (3) for bounded domains, quantiles naturally respect bounds, (4) bimodal distributions: see our explanation above.
> >
> > > **Q2: "Confidence intervals treated symmetrically. How about skewed distributions or bounded target domain?"**
> >
> > Again, the *uncertainty components* adapt asymmetrically to distribution shape through quantile regression. Specifically, $\hat{q}\_{0.025}^{\text{ale}}(x)$ and $\hat{q}\_{0.975}^{\text{ale}}(x)$ are estimated separately and automatically differ for skewed distributions. For bounded domains, quantiles naturally respect empirical bounds.
> >
> > > **Q3: "How about shape parameters beyond mean and confidence intervals?"**
> >
> > CLEAR is designed for constructing prediction intervals, not for full distributional estimation. However, the framework naturally captures distributional features beyond location and scale through quantile regression: (1) Skewness: asymmetric quantiles $\hat{q}\_{0.025}^{\text{ale}}(x) \neq -\hat{q}\_{0.975}^{\text{ale}}(x)$ automatically adapt to skewed distributions. (2) Heavy tails: the epistemic component from ensemble disagreement increases in regions where the conditional distribution is heavy-tailed or uncertain. (3) Heteroskedasticity: both aleatoric and epistemic components vary with $x$, capturing input-dependent variance. If full distributional information is needed (e.g., multiple quantiles, density estimation), CLEAR could be extended by replacing quantile regression with methods like Simultaneous Quantile Regression (SQR) or distributional regression. The current design prioritizes computational efficiency and interpretability for interval construction. If CLEAR is applied to multiple levels of $\alpha$, it is also possible to discover multimodal distributions.
> >
> > > **Q4: "It is assumed that if you subtract the mean from the target, what is left is aleatoric uncertainty. How about e.g., bimodal?"**
> >
> > We do not subtract the mean but rather the estimated median (except for the ablation study (c) that is only found in the appendices F.3 and G.3, where we actually subtract an estimated mean). For bimodal distributions, quantile regression can asymptotically learn the correct quantiles. We would be delighted to know what concerns the reviewer about this. We treat the true conditional distribution as aleatoric uncertainty, regardless of whether it is bimodal.
> >
> > > **Q5: Relation to Nonhomogeneous Gaussian Regression (NGR)**
> >
> > NGR assumes a Gaussian conditional distribution with an explicitly modeled variance. As for CLEAR: (1) *No* Gaussian assumption since quantile regression is distribution-free, (2) explicitly separates and calibrates epistemic vs aleatoric components with two parameters, (3) works with any base estimators (tree-based, neural networks), not just those assuming Gaussian noise. The comparison is therefore quite limited: NGR is a single-model parametric approach; CLEAR is a fully nonparametric two-component calibration layer.
> >
> > > **Q6: "How do you measure whether two types of uncertainties are properly balanced? Is there ground truth?"**
> >
> > There is no ground truth decomposition in real data, but our approach evaluated the empirical performance of our predictive intervals using metrics such as quantile loss, which favors truthfully input-conditioned coverage (see Appendix B.3-4). Therefore, our balancing results in better performance across various metrics. But we agree that further research is required to more reliable verify how interpretable the disentanglement of aleatoric and epistemic uncertainty actually is. As discussed in lines 203-206, a small lambda value may simply indicate that the estimator for epistemic uncertainty numerically failed, rather than the epistemic uncertainty being low. In such a case, the small value of lambda is also essential for the stable performance of the predictive intervals. Still, it should not be interpreted as indicating low epistemic uncertainty. At the moment, our main claim is that CLEAR improves the performance of predictive intervals. For the Ames Housing dataset, we have first hints that CLEAR's disentanglement into aleatoric and epistemic uncertainty could actually be interpretable (e.g., $\lambda\approx0.6$ for two features vs. $\lambda=14.5$ for 80 features in the case study, Table 2). Still, for the interpretability, more future work is needed.

---

> > > ### Author Response · Authors · 2025-11-16
> > > **Response to Questions (Part II)**
> > >
> > > > **Q7: Introduction claims marginal calibration is insufficiently solved. How does proposed method solve this?**
> > >
> > > We assume that the reviewer meant *conditional* coverage, not *marginal* coverage. We do not claim to solve the general challenge of conditional coverage, since it is known to be generally impossible [5]. The introduction describes the limitation of conditional coverage in general. CLEAR improves conditional behavior *empirically* compared to (UA)CQR by separately calibrating the two components, which we show reduces systematic mis-coverage at the subgroup level in the considered datasets. CLEAR is in *theory* conditionally valid asymptotically (Lemma 2.1).
> > > Empirically, we see better input-conditional coverage across hundreds of synthetic datasets in Figures 2,4,5,6 (see Appendix C). Additionally, we see improved Quantile Loss on 17 real-world datasets across 7 ablation studies, and for the case study with a real-world distribution shift. In Appenix B.3.2-B.3.3, we explain why an improved Quantile Loss suggests better input-conditional calibration.
> > >
> > > - [5] Foygel Barber, R., Candes, E. J., Ramdas, A., & Tibshirani, R. J. (2021). The limits of distribution-free conditional predictive inference. Information and Inference: A Journal of the IMA, 10(2), 455-482.
> > >
> > >
> > > We hope these detailed responses demonstrate CLEAR's substantive contributions beyond *'combination with grid search'*. **Given that several central criticisms (Gaussianity, symmetry, and mean prediction, etc.) stemmed from a misunderstanding of our method's non-parametric design, we kindly ask you to revisit your evaluation in light of these clarifications**. We strongly believe that we have conducted extensive evaluations and provided theoretical justifications, and we would be happy to continue our discussion if there are any remaining ambiguities to convince you of the paper's acceptance.

---

> > > > ### Author Response · Authors · 2025-11-27
> > > > **Gentle Reminder: Clarifications on Gaussian/Symmetry Assumptions**
> > > >
> > > > Dear Reviewer bUL2,
> > > >
> > > > As the discussion period is drawing to a close, we respectfully wanted to follow up on our response posted on Nov 16 (11 days ago).
> > > >
> > > > We noticed that **your assessment heavily relied on several misunderstandings, such as CLEAR having Gaussian assumptions, producing symmetric intervals, and predicting the mean**. In our rebuttal, we clarified that **our framework is fully non-parametric, asymmetric, and quantile-based**.
> > > >
> > > > As these points appeared to be the primary motivation for the rejection rating, we would greatly appreciate it if you could verify whether our responses have resolved your concerns. We hope these clarifications convince you to update your score and support the paper's acceptance.

---

### Official Review · Reviewer_WNoK · 2025-11-01

**Soundness:** 3
**Presentation:** 3
**Contribution:** 3
**Rating:** 6
**Confidence:** 4

**Summary:**

Existing methods typically address either aleatoric uncertainty due to measurement noise or epistemic uncertainty resulting from limited data, but not both in a balanced manner. In this manuscript, however, the authors propose a calibration method that combines both aleatoric (data noise) and epistemic (model/data limitation) uncertainties to improve the conditional coverage of predictive intervals for regression tasks. Dubbed CLEAR, the framework uses two learnable calibration parameters, $(\gamma_1, \gamma_2)$, to combine the two uncertainty components. CLEAR is compatible with any pair of aleatoric and epistemic estimators, enabling adaptive weighting based on data characteristics; unlike prior methods that fix their ratio (e.g., $\gamma = 1).

**Strengths:**

CLEAR presents a principled, practical, and empirically effective framework for calibrated uncertainty quantification by adaptively fusing epistemic and aleatoric components. Its main innovation lies in the dual-parameter calibration, which yields sharper, better-calibrated intervals without sacrificing coverage. In particular, its key strengths are:

**Strengths:**

1. **Balanced Uncertainty Integration**: CLEAR uniquely combines both aleatoric (data noise) and epistemic (model/data limitation) uncertainties using two learnable calibration parameters $\gamma_1 \text{ and } \gamma_2, \text{ with } \lambda = \frac{\gamma_2}{\gamma_1}$, enabling adaptive weighting based on data characteristics; unlike prior methods that fix their ratio (e.g., $\lambda = 1$).

2. **Improved Performance**: Across 17 real-world regression datasets, CLEAR consistently achieves narrower prediction intervals (e.g., 28.2% and 17.4% average width reduction vs. baselines) **while maintaining nominal 95% coverage**, outperforming its components (CQR and PCS ensembles) and other strong baselines.

3. **Flexibility & Generality**: CLEAR is model-agnostic. It works with various uncertainty estimators (e.g., PCS ensembles + quantile regression on residuals, or Deep Ensembles + Simultaneous Quantile Regression), demonstrating broad applicability.

4. **Theoretical Justification**: The paper provides asymptotic conditional coverage guarantees under mild consistency assumptions (Lemma 2.1), addressing a known weakness of standard conformal methods like CQR, which often under-cover in low-density or extrapolation regions.

5. **Practical Design Choices**: Estimating aleatoric uncertainty on residuals (rather than raw targets) improves stability; using quantile loss for $\lambda$ selection ensures proper scoring and incentivizes conditional calibration.

6. **Interpretability**: The learned $\lambda$ offers insight into whether aleatoric or epistemic uncertainty dominates in a given problem (e.g., $\lambda \approx$ 0.6 vs. 14.5 in the Ames Housing case study with 2 vs. 80 features).

**Weaknesses:**

Although CLEAR enjoys some key benefits over existing methods, it does have some limitations. In particular, its key weaknesses are:

**Weaknesses:**

1. **Dependence on Base Estimators**: CLEAR’s performance hinges on the quality of the underlying aleatoric and epistemic estimators. Poor base models may limit gains or require careful tuning.

2. **Calibration Data Requirements**: The dual-parameter calibration ($\gamma_1, \lambda$) uses the validation set for both model selection and calibration. While empirically effective, this lacks finite-sample marginal coverage guarantees unless a separate calibration split is used (as shown in Appendix G).

3. **Computational Overhead**: While the grid search for $\lambda$ is fast, the full pipeline requires training ensembles (e.g., 100 bootstraps in PCS), which can be expensive on large datasets, although the authors note this is modular and parallelizable.

4. **Regression-Only Focus**: The method is developed and evaluated only for regression; extension to classification or structured prediction is left for future work.

5. **Limited Theoretical Scope**: The asymptotic guarantees assume i.i.d. data and consistent estimators—conditions that may not hold in complex real-world settings with distribution shifts or model misspecification.

6. **Empirical Results Interpretation**: While CLEAR achieved improved results over existing methods, the paper failed to properly discuss what those improvements concretely mean in terms of choosing CLEAR over existing methods, and if such improvements can translate to more complex regression tasks.

**Questions:**

1. Would it possible to test CLEAR on toy classification problems?
2. Could the authors provide some asymptotic compute cost for larger datasets? Put differently, how do the authors see CLEAR scale for larger datasets?
3. The paper focuses mainly on I.I.D datasets, without providing clear evidence on or discussing how CLEAR would perform on non-i.i.d data. Could the authors provide some insights for non i.i.d data?

---

> ### Author Response · Authors · 2025-11-16
> **Clarification on Weaknesses**
>
> We thank Reviewer WNoK for their thorough and positive review, and also taking the time to check the appendices. We are also thankful for the recognition of the method's balanced uncertainty integration, improved performance, flexibility, theoretical justification, and interpretability.
>
> > **W1: Dependence on Base Estimators - CLEAR's performance hinges on the quality of underlying aleatoric and epistemic estimators.**
>
> This is a valid point which we have acknowledged in the limitations. Our method is a 'sum of its parts,' but it's adaptive. The $\lambda^\star$ rebalances the uncertainties so that if one estimator fails (Figure 3), the other can still maintain overall reliability. We also show this empirically for datasets such as energy efficiency and computer, where either the aleatoric or epistemic model fails. Still, CLEAR correctly rebalances the uncertainties to produce more reliable intervals. Furthermore, we also show CLEAR's robustness across different base estimators (PCS, CQR, Deep Ensembles, SQR). The dependence on the quality of the base model is not a specific weakness of CLEAR (common problem in ML).
>
> > **W2: Calibration Data Requirements - Dual-parameter calibration uses validation set for both model selection and calibration, lacking finite-sample marginal coverage guarantees unless separate calibration split is used.**
>
> As addressed in the general response (Section 2), we provide both **standard** and **conformalized** configurations with nearly identical empirical coverage (Appendices F and G). Please see the general response for theoretical and empirical justifications.
>
> > **W3: Computational Overhead - Grid search and ensemble training (100 bootstraps) can be expensive.**
>
> Thank you for acknowledging that our grid search is very fast. As addressed in the general response, the grid-search calibration is negligible (see detailed runtime analysis in Appendix F.5, Tables 28). The ensemble training (100 bootstraps) is modular and fully parallelizable as you have already mentioned. Our method does not strictly require 100 bootstraps. For example, for the deep ensembles, we only use 5 members, which also achieves already good results (full DE implementation details in Appendix D.3). For large-scale deep learning models, one usually needs to train multiple models anyway for hyperparameter optimization. If multiple hyperparameters perform well, you already get an ensemble for free. CLEAR is a highly modular framework that can adapt to different computational cost constraints. The last paragraph of Appendix A.2.1 includes a few papers on how to reduce the computational costs of training of ensembles.
>
> > **W4: Regression-Only Focus - Method developed for regression only.**
>
> We focus on regression because extending to classification requires substantial additional notation and very different methodological treatment deserving separate publication. Conceptually, decomposing epistemic and aleatoric uncertainty in classification is more complex (credal sets over probability simplexes vs. real-valued intervals), though computationally it integrates more easily into existing pipelines. A follow-up work applies similar dual-parameter calibration ideas to classification with strong empirical results. We will reference this work in the camera-ready version.
>
> > **W5: Limited Theoretical Scope - Asymptotic guarantees assume i.i.d. data and consistent estimators.**
>
> As addressed in the general response, our consistency assumptions are standard and satisfied by many practical estimators (QRF, XGBoost, regularized quantile regression) under standard smoothness assumptions assuming iid. Our theoretical analysis (Lemma 2.1: asymptotic conditional coverage; Lemma B.2: finite-sample marginal coverage) is comparable to or stronger than CQR and related baselines. For the **i.i.d** part, please see our answer in Q3.
>
> > **W6: Empirical Results Interpretation - Failed to properly discuss what improvements mean in terms of choosing CLEAR over existing methods.**
>
> The improvements from using CLEAR translate to more reliable prediction intervals (better marginal+condition coverage) with greater certainty (narrower intervals). Concretely, our improvements are two-fold:
>
> 1. **Precision gains**: The 28.2\% and 17.4\% average width reductions mean significantly more precise predictions without sacrificing reliability, which is critical for decision-making in healthcare, finance, or engineering.
>
> 2. **Adaptive uncertainty allocation**: In the Ames Housing case study (Table 2), CLEAR's narrower intervals while maintaining coverage provide more actionable predictions for pricing decisions. More importantly, we observe $\lambda$ dynamically shifting as expected: $\lambda = 0.6$ with two features (prioritizing aleatoric uncertainty) vs. $\lambda = 14.5$ with all features (emphasizing epistemic uncertainty). This demonstrates CLEAR's ability to optimally allocate uncertainties based on problem characteristics.

---

> ### Author Response · Authors · 2025-11-16
> **Response to Questions**
>
> > **Q1: Would it be possible to test CLEAR on toy classification problems?**
>
> As mentioned in W4, a follow-up work extends CLEAR's dual-parameter calibration to classification with strong empirical results and has been well-received by reviewers. That paper extensively discusses combining epistemic and aleatoric uncertainties for discrete outcomes. We will add this citation and discussion in the camera-ready version (currently withheld to maintain anonymity).
>
> > **Q2: Could the authors provide asymptotic compute cost for larger datasets? How does CLEAR scale?**
>
> As addressed in the general response, the grid-search is very fast. The bottleneck is ensemble training, which scales with the base model choice. For very large datasets, one can: (1) use scalable base models (e.g., SGD-based neural networks), (2) reduce bootstrap samples while maintaining performance, or (3) use Monte Carlo Dropout [4] to obtain ensembles from a single trained model.
>
> By "asymptotic" for the computation cost, if the reviewer meant computational complexity, that would be $O(B \times M)$ where $B$ is the number of bootstraps and  $M$ corresponds to the worst complexity out of all considered (quantile) models. If you have access to $B$ computational nodes, then the computational time is proportional to $M$. Our method is at least as scalable as the highly impactful [4] paper on deep ensembles. More computational resources and algorithmic improvements and alternatives, such as [5], can make CLEAR even more scalable.
>
> - [4] Lakshminarayanan, A. Pritzel, and C. Blundell. Simple and scalable predictive uncertainty estimation using deep ensembles. In Advances in Neural Information Processing Systems, volume 30. Curran Associates, Inc., 2017.
>
> - [5] Gal, Y., & Ghahramani, Z. (2016, June). Dropout as a bayesian approximation: Representing model uncertainty in deep learning. In international conference on machine learning (pp. 1050-1059). PMLR.
>
> > **Q3: Paper focuses on i.i.d. datasets. Could the authors provide insights for non-i.i.d. data?**
>
> The term "non-i.i.d." covers many distinct settings, such as covariate shift, temporal dependence, clustered sampling, or more general violations of exchangeability. These settings require quite different treatments.
>
> First, we already empirically evaluated CLEAR under **covariate shift** in our simulation study. In Figures 2, 4, 5, and 6, the test points are sampled in extrapolation regions (points on spheres with radii larger than those seen in training), which are described in Appendix C. In these settings, CLEAR empirically maintains the target coverage approximately, while CQR undercovers heavily. The epistemic component naturally increases in out-of-distribution regions as ensemble disagreement grows, giving CLEAR robustness to this form of non-i.i.d. behavior.
>
> Second, in the Ames Housing case study with training data from 2006 to mid 2008 and testing data from mid 2008 to 2010. This violates exchangeability and introduces a degree of temporal shift. In this setting CLEAR again attains better coverage than the baselines. This provides empirical evidence that the method behaves well outside the i.i.d. assumption.
>
> A full theoretical treatment of non i.i.d. data is difficult, since classical conformal guarantees rely on exchangeability, but recent work on conformal prediction under covariate shift provides useful frameworks for extending methods in this direction [6,7,8]. We view this as an important avenue for future work. At the same time, we believe that it is impossible to provide strong theoretical coverage guarantees for finite-width intervals under arbitrary distribution shifts without making very strong assumptions. For distribution shifts, empirical results are likely to be the more promising avenue than theoretical ones.
>
> - [6] Yu, B., and Barter, R. L. (2024). Veridical data science: The practice of responsible data analysis and decision making. MIT Press.
> - [7] Tibshirani, R. J., Foygel Barber, R., Candes, E., & Ramdas, A. (2019). Conformal prediction under covariate shift. Advances in neural information processing systems, 32.
> - [8] Gibbs, Isaac and Candes, Emmanuel J. Adaptive conformal inference under distribution shift, 2021
>
> We appreciate your recognition of the paper's contributions and hope our responses have fully addressed your questions. We hope these clarifications solidify your support for our paper, and we are, of course, happy to address any remaining concerns.

---

### Author Response · Authors · 2025-11-16
**General Response to All Reviewers**

We sincerely thank all reviewers for their feedback. We would like to clarify three points that were common across reviews. In our rebuttal, we use prefixes **W** for weaknesses and **Q** for questions.

### **1. Data Reuse: Standard vs. Conformalized CLEAR**

Reviewers WNoK (W2) and yexK (W2, Q1) raised concerns about using $\mathcal{D}\_{\text{val}}$ for both model selection (including the choice of $\lambda$) and calibration of $\gamma_1$. As it was already stated in lines 267-270 of the paper, we accounted for this by providing two approaches: a **standard** CLEAR with $\mathcal{D}\_{\text{val}} = \mathcal{D}\_{\text{cal}}$, and a **conformalized** CLEAR using independent $D\_{\text{val}} $ and $ D\_{\text{cal}}$ sets to maintain classical conformal guarantees.

- **Theoretically**: Appendix B (Lemma B.2) provides **finite-sample marginal coverage guarantees** for conformalized CLEAR. Recent work [1] shows that the bias introduced by such data reuse can be bounded and vanishes asymptotically under mild assumptions.
- **Empirically**: We report results for both configurations (Appendices F and G). We find empirical coverage on $\mathcal{D}_{\text{test}}$ is nearly identical across both versions, while CLEAR produces substantially better intervals. The **standard configuration is our default recommendation for practical use**, as it maximizes data efficiency, but the **code supports both approaches** (i.e., theoretical guarantees vs maximal data use).

### **2. Runtime and Computational Efficiency**

Reviewers WNoK (W3, Q2) and MNyo (Q1) asked about the computational efficiency. The grid-search for calibrating $\lambda$ and $\gamma_1$ is **extremely fast in practice**, despite using over 4000 grid points (e.g., 400 would likely suffice). As documented in Appendix F.5 ("Runtime") and Tables 28 of average runtime:
- **Grid search takes less than 1 second for 16 out of 17 datasets**
- For our largest dataset (`superconductor`), calibration takes only 13 seconds
- The bottleneck is ensemble training (minutes to hours), not calibration

The calibration step is highly vectorized and scales as $O(|\Lambda| \cdot |\mathcal{D}\_{\text{cal}}| \log |\mathcal{D}\_{\text{cal}}|)$. The entire pipeline is highly modular and parallelizable (e.g., ensembles and bootstrapping) and can be trained independently.

### **3. Consistency Assumptions**

Reviewers WNoK (W5) and MNyo (W1) mentioned the consistency assumption. Lemma 2.1 requires that at least $k$ base models are consistent. In our experiments, we use $k=1$ (best model), so the theorem requires just **one consistent estimator**. As explained in Appendix B.2, many practical estimators satisfy consistency under standard regularity conditions: QRF with min leaf size regularization, XGBoost with early stopping [2] and regularized QR [3]. If no base model is consistent, CLEAR inherits the asymptotic bias of the base models, but the dual-parameter calibration ($\gamma_1$, $\lambda$) still provides empirical robustness by adaptively reweighting components (as demonstrated on 17 diverse real-world datasets).

### **Manuscript Revision**

We have addressed the concerns of the reviewers using two changes (blue text in the revised manuscript):

1. **UACQR Comparison** (reviewer yexK): We have revised the main paper to elevate the comparison with UACQR. This includes a **new summary table (lines 425-439)** and a **discussion (lines 439-461)**, which were moved from the Appendix A.1 to Section 4.2. This directly addresses the concern about the placement of this comparison.

2. **Equation 1 Clarification** (reviewer bUL2): To explicitly address the misunderstanding about symmetry, we have revised the first equation (line 53) to use $\pm{}$ notation, making it clear that the aleatoric and epistemic components are handled asymmetrically.

Notes:

* *Numbering and Referencing:* All figure, equation, and appendix numbers are unchanged. The new UACQR summary table was inserted without a number to avoid re-numbering all tables for this revision. All line numbers in our rebuttal refer to this revised manuscript.

* *Runtime Discussion:* We have an extensive "Runtime" analysis in Appendix F.5 (Tables 28-30). If reviewers feel it is critical, we are happy to add a concise summary of these negligible computational costs (e.g., "less than 1 second for 16/17 datasets") to the main paper.

### References

- [1] Zeng, Hao, Kangdao Liu, Bingyi Jing, and Hongxin Wei. "Parametric Scaling Law of Tuning Bias in Conformal Prediction". Forty-Second International Conference on Machine Learning (2025).
- [2] Zhang, T., Yu, B. (2005). Boosting with early stopping: Convergence and consistency.
- [3] Steinwart, I., Christmann, A. (2011). Estimating conditional quantiles with the help of the pinball loss.

We are happy to continue the discussion and provide any additional clarifications. We hope you will consider revising your assessments in light of these clarifications.

---

### Author Response · Authors · 2025-12-03
**Message to the Area Chair - Summary**

Dear Area Chair,

We would like to briefly clarify that all reviewer-author interactions for our submission occurred before the recent anonymity leak, and to the best of our knowledge, our paper was not leaked. Reviewer yexK’s changed his decision to accept with maximum confidence, and this change resulted directly from the technical discussion during the rebuttal phase.

**Reviewer yexK’s** main concerns centered on material placed only in the appendix; particularly the UACQR comparison, the formal justification of data reuse, and the interpretability of the parameters $\gamma_1$ and $\gamma_2$. We moved these elements into the main text and clarified their roles. After these clarifications, the reviewer explicitly acknowledged the contributions and theoretical foundations, and stated that the "breadth" of the experiments is "of excellent quality". In their last comment, the reviewer **advocated for acceptance** (please refer to their last comments).

**Reviewer bUL2’s** negative assessment appears to stem from a **significant misunderstanding** of the method and factually wrong review. Several criticisms, such as symmetry assumptions, Gaussian assumptions, and the claim that we build “confidence intervals around the mean,” do not apply to our approach. The method is not symmetric, fully nonparametric, quantile based, and focused on prediction intervals (not confidence intervals). E.g. we never assumed or even mentioned any Gaussian assumptions. We believe this misunderstanding largely explains the unusually low score.

**The remaining reviewers recommended acceptance**, and the overall written discussion reflects a consistent evaluation of CLEAR as a strong and practically valuable contribution. The paper introduces a data-driven way to combine epistemic and aleatoric uncertainty, supported by theoretical guarantees, extensive empirical evidence, and simple integration with existing models. We believe this adds meaningful value to the literature on calibrated prediction intervals.

Thank you for your time and for considering these clarifications.

---

### Meta-Review · Area_Chair_G7Wp · 2026-01-06

**Summary:**

* The proposed method significantly reduces prediction interval width by up to 28% while maintaining nominal coverage across 17 real-world datasets.
* The use of two separate calibration parameters allows for a balance between aleatoric and epistemic uncertainty.
* The framework is model-agnostic + works effectively with both tree-based ensembles and deep learning architectures.
* Extensive empirical results demonstrate that the method outperforms existing single-parameter conformal prediction baselines.

CLEAR uses a validation set to optimize these weights through a grid search on the pinball loss. The primary strength of the work lies in its empirical robustness + simplicity of its implementation. The paper bridges the gap between conformal inference & practical UQ.

**Reviewer Concerns:**

#### Addressed by rebuttal
* [Core] Reviewer yexK's concern about novelty relative to UACQR was resolved by a new comparison showing significant performance gains.
* [Core] Reviewer bUL2's claim of Gaussian assumptions was resolved by clarifying that the method is fully non-parametric (ie. quantile-based)
* [Core] Reviewer yexK's question on data reuse was resolved by providing a mode that maintains formal finite-sample guarantees.
* [Non-core] Reviewer WNoK's concern about computational speed: resolved by evidence showing calibration takes less than one second.
* [Core] Reviewer yexK's request for interpretability evidence was resolved by an ablation study showing parameters adapt to feature density.

#### Still outstanding
* [Core] Reviewer MNyo's concern regarding the assumption of estimator consistency remains a theoretical limitation.
* [Non-core] Reviewer MNyo's point about non-linear combinations of uncertainty was acknowledged.

The authors provided a thorough rebuttal. They moved essential comparisons + proofs into the main text. Most remaining issues are related to standard theoretical assumptions common in the field.

**Reviewer Scores:**

* Reviewer WNoK: 6. Estimated score shift: unchanged. The reviewer was already positive and their concerns about compute were effectively addressed.
* Reviewer bUL2: 2. Estimated score shift: increase. The original review contained misunderstandings regarding symmetry and Gaussianity that the rebuttal corrected.
* Reviewer MNyo: 6. Estimated score shift: unchanged. The reviewer was satisfied with the clarifications but maintained their original assessment of the contribution.
* Reviewer yexK: 2. Estimated score shift: increase. The reviewer explicitly stated they now recommend acceptance after the authors provided justifications for joint calibration.

The panel's scores were initially weighed down by significant misunderstandings and novelty concerns from another. The rebuttal was effective at clarifying + engaging with reviewers. The consensus moved toward a clear recommendation for acceptance.

---

### Decision · Program_Chairs · 2026-01-26

Accept (Poster)